# EMPIRICAL RISK LANDSCAPE ANALYSIS FOR UNDERSTANDING DEEP NEURAL NETWORKS

**Pan Zhou & Jiashi Feng**
Department of Electrical and Computer Engineering
National University of Singapore
Singapore, 117583
{pzhou@u.nus.edu, elefjia@nus.edu.sg}

## ABSTRACT

This work aims to provide comprehensive landscape analysis of empirical risk in deep neural networks (DNNs), including the convergence behavior of its gradient, its stationary points and the empirical risk itself to their corresponding population counterparts, which reveals how various network parameters determine the convergence performance. In particular, for an $l$-layer linear neural network consisting of $\boldsymbol{d}_i$ neurons in the $i$-th layer, we prove the gradient of its empirical risk uniformly converges to the one of its population risk, at the rate of $\mathcal{O}(r^{2l}\sqrt{l\sqrt{\max_i \boldsymbol{d}_i}s\log(d/l)/n})$. Here $d$ is the total weight dimension, $s$ is the number of nonzero entries of all the weights and the magnitude of weights per layer is upper bounded by $r$. Moreover, we prove the one-to-one correspondence of the non-degenerate stationary points between the empirical and population risks and provide convergence guarantee for each pair. We also establish the uniform convergence of the empirical risk to its population counterpart and further derive the stability and generalization bounds for the empirical risk. In addition, we analyze these properties for deep *nonlinear* neural networks with sigmoid activation functions. We prove similar results for convergence behavior of their empirical risk gradients, non-degenerate stationary points as well as the empirical risk itself.

To our best knowledge, this work is the first one theoretically characterizing the uniform convergence of the gradient and stationary points of the empirical risk of DNN models, which benefits the theoretical understanding on how the neural network depth $l$, the layer width $\boldsymbol{d}_i$, the network size $d$, the sparsity in weight and the parameter magnitude $r$ determine the neural network landscape.

## 1 INTRODUCTION

Deep learning has achieved remarkable success in many fields, such as computer vision (Hinton et al., 2006; Szegedy et al., 2015; He et al., 2016), natural language processing (Collobert & Weston, 2008; Bakshi & Stephanopoulos, 1993), and speech recognition (Hinton et al., 2012; Graves et al., 2013). However, theoretical understanding on the properties of deep learning models still lags behind their practical achievements (Shalev-Shwartz et al., 2017; Kawaguchi, 2016) due to their high non-convexity and internal complexity. In practice, parameters of deep learning models are learned by minimizing the *empirical risk* via (stochastic-)gradient descent. Therefore, some recent works (Bartlett & Maass, 2003; Neyshabur et al., 2015) analyzed the convergence of the empirical risk to the population risk, which are however still far from fully understanding the landscape of the empirical risk in deep learning models. Beyond the convergence properties of the empirical risk itself, the convergence and distribution properties of its gradient and stationary points are also essential in landscape analysis. A comprehensive landscape analysis can reveal important information on the optimization behavior and practical performance of deep neural networks, and will be helpful to designing better network architectures. Thus, in this work we aim to provide comprehensive landscape analysis by looking into the gradients and stationary points of the empirical risk.

Formally, we consider a DNN model $f(\boldsymbol{w}; \boldsymbol{x}, \boldsymbol{y}) : \mathbb{R}^{\boldsymbol{d}_0} \times \mathbb{R}^{\boldsymbol{d}_l} \to \mathbb{R}$ parameterized by $\boldsymbol{w} \in \mathbb{R}^d$ consisting of $l$ layers ($l \geq 2$) that is trained by minimizing the commonly used squared loss function

over sample pairs $\{(\boldsymbol{x}, \boldsymbol{y})\} \subset \mathbb{R}^{\boldsymbol{d}_0} \times \mathbb{R}^{\boldsymbol{d}_l}$ from an unknown distribution $\mathcal{D}$, where $\boldsymbol{y}$ is the target output for the sample $\boldsymbol{x}$. Ideally, the model can find its optimal parameter $\boldsymbol{w}^*$ by minimizing the *population risk* through (stochastic-)gradient descent by backpropagation:

$$\min_{\boldsymbol{w}} \boldsymbol{J}(\boldsymbol{w}) \triangleq \mathbb{E}_{(\boldsymbol{x},\boldsymbol{y}) \sim \mathcal{D}} \, f(\boldsymbol{w}; \boldsymbol{x}, \boldsymbol{y}),$$

where $f(\boldsymbol{w}; \boldsymbol{x}, \boldsymbol{y}) = \frac{1}{2} \|\boldsymbol{v}^{(l)} - \boldsymbol{y}\|_2^2$ is the squared loss associated to the sample $(\boldsymbol{x}, \boldsymbol{y}) \sim \mathcal{D}$ in which $\boldsymbol{v}^{(l)}$ is the output of the $l$-th layer. In practice, as the sample distribution $\mathcal{D}$ is usually unknown and only finite training samples $\left\{(\boldsymbol{x}_{(i)}, \boldsymbol{y}_{(i)})\right\}_{i=1}^{n}$ *i.i.d.* drawn from $\mathcal{D}$ are provided, the network model is usually trained by minimizing the empirical risk:

$$\min_{\boldsymbol{w}} \hat{\boldsymbol{J}}_n(\boldsymbol{w}) \triangleq \frac{1}{n} \sum_{i=1}^{n} f(\boldsymbol{w}; \boldsymbol{x}_{(i)}, \boldsymbol{y}_{(i)}). \tag{1}$$

Understanding the convergence behavior of $\hat{\boldsymbol{J}}_n(\boldsymbol{w})$ to $\boldsymbol{J}(\boldsymbol{w})$ is critical to statistical machine learning algorithms. In this work, we aim to go further and characterize the landscape of the empirical risk $\hat{\boldsymbol{J}}_n(\boldsymbol{w})$ of deep learning models by analyzing the convergence behavior of its gradient and stationary points to their corresponding population counterparts. We provide analysis for both multi-layer linear and nonlinear neural networks. In particular, we obtain following new results.

- We establish the uniform convergence of empirical gradient $\nabla_{\boldsymbol{w}} \hat{\boldsymbol{J}}_n(\boldsymbol{w})$ to its population counterpart $\nabla_{\boldsymbol{w}} \boldsymbol{J}(\boldsymbol{w})$. Specifically, when the sample size $n$ is not less than $\mathcal{O}\left(\max(l^3 r^2/(\varepsilon^2 s \log(d/l)), s \log(d/l)/l)\right)$, with probability at least $1 - \varepsilon$ the convergence rate is $\mathcal{O}(r^{2l}\sqrt{l\sqrt{\max_i \boldsymbol{d}_i} s \log(d/l)/n})$, where there are $s$ nonzero entries in the parameter $\boldsymbol{w}$, the output dimension of the $i$-th layer is $\boldsymbol{d}_i$ and the magnitude of the weight parameter of each layer is upper bounded by $r$. This result implies that as long as the training sample size $n$ is sufficiently large, any stationary point of $\hat{\boldsymbol{J}}_n(\boldsymbol{w})$ is also a stationary point of $\boldsymbol{J}(\boldsymbol{w})$ and vise versa, although both $\hat{\boldsymbol{J}}_n(\boldsymbol{w})$ and $\boldsymbol{J}(\boldsymbol{w})$ are very complex.

- We then prove the exact correspondence of non-degenerate stationary points between $\hat{\boldsymbol{J}}_n(\boldsymbol{w})$ and $\boldsymbol{J}(\boldsymbol{w})$. Indeed, the corresponding non-degenerate stationary points also uniformly converge to each other at the same convergence rate as the one revealed above with an extra factor $2/\zeta$. Here $\zeta > 0$ accounts for the geometric topology of non-degenerate stationary points (see Definition 1).

Based on the above two new results, we also derive the uniform convergence of the empirical risk $\hat{\boldsymbol{J}}_n(\boldsymbol{w})$ to its population risk $\boldsymbol{J}(\boldsymbol{w})$, which helps understand the generalization error of deep learning models and stability of their empirical risk. These analyses reveal the role of the depth $l$ of a neural network model in determining its convergence behavior and performance. Also, the results tell that the width factor $\sqrt{\max_i \boldsymbol{d}_i}$, the nonzero entry number $s$ of weights, and the total network size $d$ are also critical to the convergence and performance. In addition, controlling magnitudes of the parameters (weights) in DNNs are demonstrated to be important for performance. To our best knowledge, this work is the first one theoretically characterizing the uniform convergence of empirical gradient and stationary points in both deep linear and nonlinear neural networks.

## 2 RELATED WORK

To date, only a few theories have been developed for understanding DNNs which can be roughly divided into following three categories. The first category aims to analyze training error of DNNs. Baum (1988) pointed out that zero training error can be obtained when the last layer of a neural network has more units than training samples. Later, Soudry & Carmon (2016) proved that for DNNs with leaky rectified linear units (ReLU) and a single output, the training error achieves zero at any of their local minima as long as the product of the number of units in the last two layers is larger than the training sample size.

The second category of analysis works (Dauphin et al., 2014; Choromanska et al., 2015a; Kawaguchi, 2016; Tian, 2017) focus on analyzing loss surfaces of DNNs, *e.g.*, how the stationary points are distributed. Those results are helpful to understanding performance difference of large- and small-size

networks (Choromanska et al., 2015b). Among them, Dauphin et al. (2014) experimentally verified that a large number of saddle points indeed exist for DNNs. With strong assumptions, Choromanska et al. (2015a) connected the loss function of a deep ReLU network with the spherical spin-class model and described locations of the local minima. Later, Kawaguchi (2016) proved the existence of degenerate saddle points for deep linear neural networks with squared loss function. They also showed that any local minimum is also a global minimum. By utilizing techniques from dynamical system analysis, Tian (2017) gave guarantees that for two-layer bias-free networks with ReLUs, the gradient descent algorithm with certain symmetric weight initialization can converge to the ground-truth weights globally, if the inputs follow Gaussian distribution. Recently, Nguyen & Hein (2017) proved that for a fully connected network with squared loss and analytic activation functions, almost all the local minima are globally optimal if one hidden layer has more units than training samples and the network structure after this layer is pyramidal. Besides, some recent works, *e.g.*, (Zhang et al., 2016; 2017), tried to alleviate analysis difficulties by relaxing the involved highly nonconvex functions into ones easier.

In addition, some existing works (Bartlett & Maass, 2003; Neyshabur et al., 2015) analyze the generalization performance of a DNN model. Based on the Vapnik–Chervonenkis (VC) theory, Bartlett & Maass (2003) proved that for a feedforward neural network with one-dimensional output, the best convergence rate of the empirical risk to its population risk on the sample distribution can be bounded by its fat-shattering dimension. Recently, Neyshabur et al. (2015) adopted Rademacher complexity to analyze learning capacity of a fully-connected neural network model with ReLU activation functions and bounded inputs.

However, although gradient descent with backpropagation is the most common optimization technique for DNNs, none of existing works analyzes convergence properties of gradient and stationary points of the DNN empirical risk. For single-layer optimization problems, some previous works analyze their empirical risk but essentially differ from our analysis method. For example, Negahban et al. (2009) proved that for a regularized convex program, the minimum of the empirical risk uniformly converges to the true minimum of the population risk under certain conditions. Gonen & Shalev-Shwartz (2017) proved that for nonconvex problems without degenerated saddle points, the difference between empirical risk and population risk can be bounded. Unfortunately, the loss of DNNs is highly nonconvex and has degenerated saddle points (Fyodorov & Williams, 2007; Dauphin et al., 2014; Kawaguchi, 2016), thus their analysis results are not applicable. Mei et al. (2017) analyzed the convergence behavior of the empirical risk for nonconvex problems, but they only considered the single-layer nonconvex problems and their analysis demands strong sub-Gaussian and sub-exponential assumptions on the gradient and Hessian of the empirical risk respectively. Their analysis also assumes a linearity property on gradient which is difficult to hold or verify. In contrast, our analysis requires much milder assumptions. Besides, we prove that for deep networks which are highly nonconvex, the non-degenerate stationary points of empirical risk can uniformly converge to their corresponding stationary points of population risk at the rate of $\mathcal{O}(\sqrt{s/n})$ which is faster than the rate $\mathcal{O}(\sqrt{d/n})$ for single-layer optimization problems in (Mei et al., 2017). Also, Mei et al. (2017) did not analyze the convergence rate of the empirical risk, stability or generalization error of DNNs as this work.

## 3 Preliminaries

Throughout the paper, we denote matrices by boldface capital letters, *e.g. $\boldsymbol{A}$*. Vectors are denoted by boldface lowercase letters, *e.g. $\boldsymbol{a}$*, and scalars are denoted by lowercase letters, *e.g. $a$*. We define the $r$-radius ball as $\mathsf{B}^d(r) \triangleq \{\boldsymbol{z} \in \mathbb{R}^d \,|\, \|\boldsymbol{z}\|_2 \le r\}$. To explain the results, we also need the vectorization operation $\mathrm{vec}(\cdot)$. It is defined as $\mathrm{vec}(\boldsymbol{A}) = (\boldsymbol{A}(:,1); \cdots ; \boldsymbol{A}(:,t)) \in \mathbb{R}^{st}$ that vectorizes $\boldsymbol{A} \in \mathbb{R}^{s \times t}$ along its columns. We use $d = \sum_{j=1}^{l} \boldsymbol{d}_j \boldsymbol{d}_{j-1}$ to denote the total dimension of weight parameters, where $\boldsymbol{d}_j$ denotes the output dimension of the $j$-th layer.

In this work, we consider both linear and nonlinear DNNs. Suppose both networks consist of $l$ layers. We use $\boldsymbol{u}^{(j)}$ and $\boldsymbol{v}^{(j)}$ to respectively denote the input and output of the $j$-th layer, $\forall j = 1, \ldots, l$.

**Deep linear neural networks:** The function of the $j$-th layer is formulated as

$$\boldsymbol{u}^{(j)} \triangleq \boldsymbol{W}^{(j)} \boldsymbol{v}^{(j-1)} \in \mathbb{R}^{\boldsymbol{d}_j}, \quad \boldsymbol{v}^{(j)} \triangleq \boldsymbol{u}^{(j)} \in \mathbb{R}^{\boldsymbol{d}_j}, \ \forall j = 1, \cdots, l,$$

where $\boldsymbol{v}^{(0)} = \boldsymbol{x}$ is the input and $\boldsymbol{W}^{(j)} \in \mathbb{R}^{\boldsymbol{d}_j \times \boldsymbol{d}_{j-1}}$ is the weight matrix of the $j$-th layer.

**Deep nonlinear neural networks:** We adopt the sigmoid function as the non-linear activation function. The function within the $j$-th layer can be written as

$$\boldsymbol{u}^{(j)} \triangleq \boldsymbol{W}^{(j)} \boldsymbol{v}^{(j-1)} \in \mathbb{R}^{\boldsymbol{d}_j}, \quad \boldsymbol{v}^{(j)} \triangleq h_j(\boldsymbol{u}^{(j)}) = (\sigma(\boldsymbol{u}_1^{(j)}); \cdots ; \sigma(\boldsymbol{u}_{\boldsymbol{d}_j}^{(j)})) \in \mathbb{R}^{\boldsymbol{d}_j}, \ \forall j = 1, \cdots, l,$$

where $\boldsymbol{u}_i^{(j)}$ denotes the $i$-th entry of $\boldsymbol{u}^{(j)}$ and $\sigma(\cdot)$ is the sigmoid function, *i.e.*, $\sigma(a) = 1/(1 + \mathrm{e}^{-a})$.

Following the common practice, both DNN models adopt the squared loss function defined as $f(\boldsymbol{w}; \boldsymbol{x}, \boldsymbol{y}) = \frac{1}{2} \|\boldsymbol{v}^{(l)} - \boldsymbol{y}\|_2^2$, where $\boldsymbol{w} = (\boldsymbol{w}_{(1)}; \cdots ; \boldsymbol{w}_{(l)}) \in \mathbb{R}^d$ contains all the weight parameters and $\boldsymbol{w}_{(j)} = \mathrm{vec}\left(\boldsymbol{W}^{(j)}\right) \in \mathbb{R}^{\boldsymbol{d}_j \boldsymbol{d}_{j-1}}$. Then the *empirical risk* $\hat{\boldsymbol{J}}_n(\boldsymbol{w})$ is $\hat{\boldsymbol{J}}_n(\boldsymbol{w}) = \frac{1}{n} \sum_{i=1}^n f(\boldsymbol{w}; \boldsymbol{x}_{(i)}, \boldsymbol{y}_{(i)}) = \frac{1}{2n} \sum_{i=1}^n \|\boldsymbol{v}_{(i)}^{(l)} - \boldsymbol{y}_{(i)}\|_2^2$, where $\boldsymbol{v}_{(i)}^{(l)}$ is the network's output of $\boldsymbol{x}_{(i)}$.

# 4 RESULTS FOR DEEP LINEAR NEURAL NETWORKS

We first analyze linear neural network models and present following new results: (1) the uniform convergence of the empirical risk gradient to its population counterpart and (2) the convergence properties of non-degenerate stationary points of the empirical risk. As a corollary, we also derive the uniform convergence of the empirical risk to the population one, which further gives stability and generalization bounds. In the next section, we extend the analysis to non-linear neural network models.

We assume the input datum $\boldsymbol{x}$ is $\tau^2$-sub-Gaussian and has bounded magnitude, as formally stated in Assumption 1.

**Assumption 1.** *The input datum $\boldsymbol{x} \in \mathbb{R}^{\boldsymbol{d}_0}$ has zero mean and is $\tau^2$-sub-Gaussian, i.e.,*

$$\mathbb{E}[\exp(\langle \boldsymbol{\lambda}, \boldsymbol{x} \rangle)] \le \exp\left(\frac{1}{2}\tau^2 \|\boldsymbol{\lambda}\|_2^2\right), \ \forall \boldsymbol{\lambda} \in \mathbb{R}^{\boldsymbol{d}_0}.$$

*Besides, the magnitude $\boldsymbol{x}$ is bounded as $\|\boldsymbol{x}\|_2 \le r_x$, where $r_x$ is a positive universal constant.*

Note that any random vector $\boldsymbol{z}$ consisting of independent entries with bounded magnitude is sub-Gaussian and satisfies Assumption 1 (Vershynin, 2012). Moreover, for such a random $\boldsymbol{z}$, we have $\tau = \|\boldsymbol{z}\|_\infty \le \|\boldsymbol{z}\|_2 \le r_x$. Such an assumption on bounded magnitude generally holds for natural data, *e.g.*, images and speech signals. Besides, we assume the weight parameters $\boldsymbol{w}_{(j)}$ of each layer are bounded as $\boldsymbol{w} \in \Omega = \{\boldsymbol{w} \mid \boldsymbol{w}_{(j)} \in \mathsf{B}^{\boldsymbol{d}_j \boldsymbol{d}_{j-1}}(\boldsymbol{r}_j), \ \forall j = 1, \cdots, l\}$ where $\boldsymbol{r}_j$ is a constant. For notational simplicity, we let $r = \max_j \boldsymbol{r}_j$. Such an assumption is common (Xu & Mannor, 2012). Here we assume the entry value of $\boldsymbol{y}$ falls in $[0, 1]$. For any bounded target output $\boldsymbol{y}$, we can always scale it to satisfy such a requirement.

The results presented for linear neural networks here can be generalized to deep ReLU neural networks by applying the results from Choromanska et al. (2015a) and Kawaguchi (2016), which transform deep ReLU neural networks into deep linear neural networks under proper assumptions.

## 4.1 UNIFORM CONVERGENCE OF EMPIRICAL RISK GRADIENT

We first analyze the convergence of gradients for the DNN empirical and population risks. To our best knowledge, these results are the first ones giving guarantees on gradient convergence, which help better understand the landscape of DNNs and their optimization behavior. The results are stated blow.

**Theorem 1.** *Suppose Assumption 1 on the input datum $\boldsymbol{x}$ holds and the activation functions in a deep neural network are linear. Then the empirical gradient uniformly converges to the population gradient in Euclidean norm. Specifically, there exist two universal constants $c_{g'}$ and $c_g$ such that if $n \ge c_{g'} \max(l^3 r^2 r_x^4/(c_q s \log(d/l)\varepsilon^2 \tau^4 \log(1/\varepsilon)), s \log(d/l)/(l\tau^2))$ where $c_q = \sqrt{\max_{0 \le i \le l} \boldsymbol{d}_i}$, then*

$$\sup_{\boldsymbol{w} \in \Omega} \left\|\nabla \hat{\boldsymbol{J}}_n(\boldsymbol{w}) - \nabla \boldsymbol{J}(\boldsymbol{w})\right\|_2 \le \epsilon_g \triangleq c_g \tau \omega_g \sqrt{l c_q} \sqrt{\frac{s \log(dn/l) + \log(12/\varepsilon)}{n}}$$

*holds with probability at least $1 - \varepsilon$, where $s$ denotes the number of nonzero entries of all weight parameters and $\omega_g = \max\left(\tau r^{2l-1}, r^{2l-1}, r^{l-1}\right)$.*

From Theorem 1, one can observe that with an increasingly larger sample size $n$, the difference between empirical risk and population risk gradients decreases monotonically at the rate of $\mathcal{O}(1/\sqrt{n})$ (up to a log factor). Theorem 1 also characterizes how the depth $l$ contributes to obtaining small difference between the empirical and population risk gradients. Specifically, a deeper neural network needs more training samples to mitigate the difference. Also, due to the factor $d$, training a network of larger size using gradient descent also requires more training samples. We observe a factor of $\sqrt{\max_i \boldsymbol{d}_i}$ (i.e. $c_q$), which prefers a DNN architecture of balanced layer sizes (without extremely wide layers). This result also matches the trend and empirical performance in deep learning applications advocating deep but thin networks (He et al., 2016; Szegedy et al., 2015).

By observing Theorem 1, imposing certain regularizations on the weight parameters is useful. For example, reducing the number of nonzero entries $s$ encourages sparsity regularization like $\|\boldsymbol{w}\|_1$. The results also suggest not choosing large-magnitude weights $\boldsymbol{w}$ in order for a smaller factor $r$ by adopting regularization like $\|\boldsymbol{w}\|_2^2$.

Theorem 1 also reveals the point derived from optimizing that the empirical and population risks have similar properties when the sample size $n$ is sufficiently large. For example, an $\epsilon/2$-stationary point $\tilde{\boldsymbol{w}}$ of $\hat{\boldsymbol{J}}_n(\boldsymbol{w})$ is also an $\epsilon$-stationary point of $\boldsymbol{J}(\boldsymbol{w})$ with probability $1 - \varepsilon$ if $n \geq c_\epsilon(\tau\omega_g/\epsilon)^2 lc_q s \log(d/l)$ with $c_\epsilon$ being a constant. Here $\epsilon$-stationary point for a function $\boldsymbol{F}$ means the point $\boldsymbol{w}$ satisfying $\|\nabla_{\boldsymbol{w}}\boldsymbol{F}\|_2 \leq \epsilon$. Understanding such properties is useful, since in practice one usually computes an $\epsilon$-stationary point of $\hat{\boldsymbol{J}}_n(\boldsymbol{w})$. These results guarantee the computed point is at most a $2\epsilon$-stationary point of $\boldsymbol{J}(\boldsymbol{w})$ and is thus close to the optimum.

## 4.2 Uniform Convergence of Stationary Points

We then proceed to analyze the distribution and convergence properties of stationary points of the DNN empirical risk. Here we consider non-degenerate stationary points which are geometrically isolated and thus unique in local regions. Since degenerate stationary points are not unique in a local region, we cannot expect to establish one-to-one corresponding relationship (see below) between them in empirical risk and population risk.

**Definition 1.** *(Non-degenerate stationary points) (Gromoll & Meyer, 1969) If a stationary point $\boldsymbol{w}$ is said to be a non-degenerate stationary point of $\boldsymbol{J}(\boldsymbol{w})$, then it satisfies*

$$\inf_i \left| \lambda_i \left( \nabla^2 \boldsymbol{J}(\boldsymbol{w}) \right) \right| \geq \zeta,$$

*where $\lambda_i \left( \nabla^2 \boldsymbol{J}(\boldsymbol{w}) \right)$ denotes the $i$-th eigenvalue of the Hessian $\nabla^2 \boldsymbol{J}(\boldsymbol{w})$ and $\zeta$ is a positive constant.*

Non-degenerate stationary points include local minima/maxima and non-degenerate saddle points, while degenerate stationary points refer to degenerate saddle points. Then we introduce the index of non-degenerate stationary points which can characterize their geometric properties.

**Definition 2.** *(Index of non-degenerate stationary points) (Dubrovin et al., 2012) The index of a symmetric non-degenerate matrix is the number of its negative eigenvalues, and the index of a non-degenerate stationary point $\boldsymbol{w}$ of a smooth function $\boldsymbol{F}$ is simply the index of its Hessian $\nabla^2 \boldsymbol{F}(\boldsymbol{w})$.*

Suppose that $\boldsymbol{J}(\boldsymbol{w})$ has $m$ non-degenerate stationary points that are denoted as $\{\boldsymbol{w}^{(1)}, \boldsymbol{w}^{(2)}, \cdots, \boldsymbol{w}^{(m)}\}$. We prove following convergence behavior of these stationary points.

**Theorem 2.** *Suppose Assumption 1 on the input datum $\boldsymbol{x}$ holds and the activation functions in a deep neural network are linear. Then if $n \geq c_h \max(l^3 r^2 r_x^4/(c_q s \log(d/l)\varepsilon^2\tau^4 \log(1/\varepsilon)), s \log(d/l)/\zeta^2)$ where $c_h$ is a constant, for $k \in \{1, \cdots, m\}$, there exists a non-degenerate stationary point $\boldsymbol{w}_n^{(k)}$ of $\hat{\boldsymbol{J}}_n(\boldsymbol{w})$ which corresponds to the non-degenerate stationary point $\boldsymbol{w}^{(k)}$ of $\boldsymbol{J}(\boldsymbol{w})$ with probability at least $1 - \varepsilon$. In addition, $\boldsymbol{w}_n^{(k)}$ and $\boldsymbol{w}^{(k)}$ have the same non-degenerate index and they satisfy*

$$\|\boldsymbol{w}_n^{(k)} - \boldsymbol{w}^{(k)}\|_2 \leq \frac{2c_g\tau\omega_g}{\zeta}\sqrt{lc_q}\sqrt{\frac{s\log(dn/l) + \log(12/\varepsilon)}{n}}, \quad (k = 1, \cdots, m)$$

*with probability at least $1 - \varepsilon$, where the parameters $c_q$, $\omega_g$, and $c_g$ are given in Theorem 1.*

Theorem 2 guarantees the one-to-one correspondence between the non-degenerate stationary points of the empirical risk $\hat{\boldsymbol{J}}_n(\boldsymbol{w})$ and the popular risk $\boldsymbol{J}(\boldsymbol{w})$. The distances of the corresponding pairs

become smaller as $n$ increases. In addition, the corresponding pairs have the same non-degenerate index. This implies that the corresponding stationary points have the same geometric properties, such as whether they are saddle points. Accordingly, we can develop more efficient algorithms, *e.g.* escaping saddle points (Ge et al., 2015), since Dauphin et al. (2014) empirically proved that saddle points are usually surrounded by high error plateaus. Also when $n$ is sufficiently large, the properties of stationary points of $\hat{\boldsymbol{J}}_n(\boldsymbol{w})$ are similar to the points of the population risk $\boldsymbol{J}(\boldsymbol{w})$ in the sense that they have exactly matching local minima/maxima and non-degenerate saddle points. By comparing Theorems 1 and 2, we find that the requirement for sample number in Theorem 2 is more restrict, since establishing exact one-to-one correspondence between the non-degenerate stationary points of $\hat{\boldsymbol{J}}_n(\boldsymbol{w})$ and $\boldsymbol{J}(\boldsymbol{w})$ and bounding their uniform convergence rate to each other are more challenging. From Theorems 1 and 2, we also notice that the uniform convergence rate of non-degenerate stationary points has an extra factor $1/\zeta$. This is because bounding stationary points needs to access not only the gradient itself but also the Hessian matrix. See more details in proof.

Kawaguchi (2016) pointed out that degenerate stationary points indeed exist for DNNs. However, since degenerate stationary points are not isolated, such as forming flat regions, it is hard to establish the unique correspondence for them as for non-degenerate ones. Fortunately, by Theorem 1, the gradients at these points of $\hat{\boldsymbol{J}}_n(\boldsymbol{w})$ and $\boldsymbol{J}(\boldsymbol{w})$ are close. This implies that a degenerate stationary point of $\boldsymbol{J}(\boldsymbol{w})$ will also give a near-zero gradient for $\hat{\boldsymbol{J}}_n(\boldsymbol{w})$, *i.e.*, it is also a stationary point for $\hat{\boldsymbol{J}}_n(\boldsymbol{w})$.

In the proof, we consider the essential multi-layer architecture of the deep linear network, and do not transform it into a linear regression model and directly apply existing results (see Loh & Wainwright (2015) and Negahban et al. (2011)). This is because we care more about deep ReLU networks which cannot be reduced in this way. Our proof technique is more suitable for analyzing the multi-layer neural networks which paves a way for analyzing deep ReLU networks. Also such an analysis technique can reveal the role of network parameters (dimension, norm, etc.) of each weight matrix in the results which may benefit the design of networks. Besides, the obtained results are more consistent with those for deep nonlinear networks (see Sec. 5).

### 4.3 Uniform Convergence, Stability and Generalization of Empirical Risk

Based on the above results, we can derive the uniform convergence of empirical risk to population risk easily. In this subsection, we first give the uniform convergence rate of empirical risk for deep linear neural networks in Theorem 3, and then use this result to derive the stability and generalization bounds for DNNs in Corollary 1.

**Theorem 3.** *Suppose Assumption 1 on the input datum $\boldsymbol{x}$ holds and the activation functions in a deep neural network are linear. Then there exist two universal constants $c_{f'}$ and $c_f$ such that if $n \geq c_{f'} \max(l^3 r_x^4/(\boldsymbol{d}_l s \log(d/l)\varepsilon^2\tau^4 \log(1/\varepsilon)), s \log(d/l)/(\tau^2 \boldsymbol{d}_l))$, then*

$$\sup_{\boldsymbol{w}\in\Omega}\left|\hat{\boldsymbol{J}}_n(\boldsymbol{w}) - \boldsymbol{J}(\boldsymbol{w})\right| \leq \epsilon_f \triangleq c_f\tau \max\left(\sqrt{\boldsymbol{d}_l}\tau r^{2l}, r^l\right)\sqrt{\frac{s\log(dn/l) + \log(8/\varepsilon)}{n}} \qquad (2)$$

*holds with probability at least $1 - \varepsilon$. Here $l$ is the number of layers in the neural network, $n$ is the sample size and $\boldsymbol{d}_l$ is the dimension of the final layer.*

From Theorem 3, when $n \to +\infty$, we have $|\hat{\boldsymbol{J}}_n(\boldsymbol{w}) - \boldsymbol{J}(\boldsymbol{w})| \to 0$. According to the definition of uniform convergence (Vapnik & Vapnik, 1998; Shalev-Shwartz et al., 2010), under the distribution $\mathcal{D}$, the empirical risk of a deep linear neural network converges to its population risk *uniformly* at the rate of $\mathcal{O}(1/\sqrt{n})$. Theorem 3 also explains the roles of the depth $l$, the network size $d$, and the number of nonzero weight parameters $s$ in a DNN model.

Based on VC-dimension techniques, Bartlett & Maass (2003) proved that for a feedforward neural network with polynomial activation functions and one-dimensional output, with probability at least $1 - \varepsilon$ the convergence bound satisfies $|\hat{\boldsymbol{J}}_n(\boldsymbol{w}) - \inf_f \boldsymbol{J}(\boldsymbol{w})| \leq \mathcal{O}(\sqrt{(\gamma \log^2(n) + \log(1/\varepsilon))/n})$. Here $\gamma$ is the shattered parameter and can be as large as the VC-dimension of the network model, *i.e.* at the order of $\mathcal{O}(ld\log(d) + l^2 d)$ (Bartlett & Maass, 2003). Note that Bartlett & Maass (2003) did not reveal the role of the magnitude of weight in their results. In contrast, our uniform convergence bound is $\sup_{\boldsymbol{w}\in\Omega}|\hat{\boldsymbol{J}}_n(\boldsymbol{w}) - \boldsymbol{J}(\boldsymbol{w})| \leq \mathcal{O}(\sqrt{(s\log(dn/l) + \log(1/\varepsilon))/n})$. So our convergence rate is tighter.

Neyshabur et al. (2015) proved that the Rademacher complexity of a fully-connected neural network model with ReLU activation functions and one-dimensional output is $\mathcal{O}\left(r^l/\sqrt{n}\right)$ (see Corollary 2 in (Neyshabur et al., 2015)). Then by applying Rademacher complexity based argument (Shalev-Shwartz & Ben-David, 2014a), we have $|\sup_f(\hat{J}_n(\boldsymbol{w}) - \boldsymbol{J}(\boldsymbol{w}))| \leq \mathcal{O}((r^l + \sqrt{\log(1/\varepsilon)})/\sqrt{n})$ with probability at least $1 - \varepsilon$ where the loss function is the training error $g = \mathbf{1}_{(\boldsymbol{v}^{(l)} \neq \boldsymbol{y})}$ in which $\boldsymbol{v}^{(l)}$ is the output of the $l$-th layer in the network model $f(\boldsymbol{w}; \boldsymbol{x}, \boldsymbol{y})$. The convergence rate in our theorem is $\mathcal{O}(r^{2l}\sqrt{(s\log(d/l) + \log(1/\varepsilon))/n})$ and has the same convergence speed $\mathcal{O}(1/\sqrt{n})$ w.r.t. sample number $n$. Note that our convergence rate involves $r^{2l}$ since we use squared loss instead of the training error in (Neyshabur et al., 2015). The extra parameters $s$ and $d$ are involved since we consider the parameter space rather than the function hypothesis $f$ in (Neyshabur et al., 2015), which helps people more transparently understand the roles of the network parameters. Besides, the Rademacher complexity cannot be applied to analyzing convergence properties of the empirical risk gradient and stationary points as our techniques.

Based on Theorem 3, we proceed to analyze the stability property of the empirical risk and the convergence rate of the generalization error in expectation. Let $\boldsymbol{S} = \{(\boldsymbol{x}_{(1)}, \boldsymbol{y}_{(1)}), \cdots, (\boldsymbol{x}_{(n)}, \boldsymbol{y}_{(n)})\}$ denote the sample set in which the samples are *i.i.d.* drawn from $\mathcal{D}$. When the optimal solution $\boldsymbol{w}^n$ to problem (1) is computed by deterministic algorithms, the generalization error is defined as $\epsilon_g = \hat{J}_n(\boldsymbol{w}^n) - \boldsymbol{J}(\boldsymbol{w}^n)$. But one usually employs randomized algorithms, *e.g.* stochastic gradient descent (SGD), for computing $\boldsymbol{w}^n$. In this case, stability and generalization error in expectation defined in Definition 3 are more applicable.

**Definition 3.** *(Stability and generalization in expectation)* *(Vapnik & Vapnik, 1998; Shalev-Shwartz et al., 2010; Gonen & Shalev-Shwartz, 2017) Assume a randomized algorithm $\boldsymbol{A}$ is employed, $((\boldsymbol{x}'_{(1)}, \boldsymbol{y}'_{(1)}), \cdots, (\boldsymbol{x}'_{(n)}, \boldsymbol{y}'_{(n)})) \sim \mathcal{D}$ and $\boldsymbol{w}^n = \operatorname{argmin}_{\boldsymbol{w}} \hat{J}_n(\boldsymbol{w})$ is the empirical risk minimizer (ERM). For every $j \in [n]$, suppose $\boldsymbol{w}_*^j = \operatorname{argmin}_{\boldsymbol{w}} \frac{1}{n-1} \sum_{i \neq j} f_i(\boldsymbol{w}; \boldsymbol{x}_{(i)}, \boldsymbol{y}_{(i)})$. We say that the ERM is on average stable with stability rate $\epsilon_k$ under distribution $\mathcal{D}$ if $\Big| \mathbb{E}_{\boldsymbol{S} \sim \mathcal{D}, \boldsymbol{A}, (\boldsymbol{x}'_{(j)}, \boldsymbol{y}'_{(j)}) \sim \mathcal{D}}$ $\frac{1}{n} \sum_{j=1}^{n} \Big[ f_j(\boldsymbol{w}_*^j; \boldsymbol{x}'_{(j)}, \boldsymbol{y}'_{(j)}) - f_j(\boldsymbol{w}^n; \boldsymbol{x}'_{(j)}, \boldsymbol{y}'_{(j)}) \Big] \Big| \leq \epsilon_k$. The ERM is said to have generalization error with convergence rate $\epsilon_{k'}$ under distribution $\mathcal{D}$ if we have $\Big| \mathbb{E}_{\boldsymbol{S} \sim \mathcal{D}, \boldsymbol{A}} \Big( \boldsymbol{J}(\boldsymbol{w}^n) - \hat{J}_n(\boldsymbol{w}^n) \Big) \Big| \leq \epsilon_{k'}$.*

Stability measures the sensibility of the empirical risk to the input and generalization error measures the effectiveness of ERM on new data. Generalization error in expectation is especially important for applying DNNs considering their internal randomness, *e.g.* from SGD optimization. Now we present the results on stability and generalization performance of deep linear neural networks.

**Corollary 1.** *Suppose Assumption 1 on the input datum $\boldsymbol{x}$ holds and the activation functions in a deep neural network are linear. Then with probability at least $1 - \varepsilon$, both the stability rate and the generalization error rate of ERM of a deep linear neural network are at least $\epsilon_f$:*

$$\Big| \mathbb{E}_{\boldsymbol{S} \sim \mathcal{D}, \boldsymbol{A}, (\boldsymbol{x}'_{(j)}, \boldsymbol{y}'_{(j)}) \sim \mathcal{D}} \frac{1}{n} \sum_{j=1}^{n} \left( f_j^* - f_j \right) \Big| \leq \epsilon_f \quad and \quad \Big| \mathbb{E}_{\boldsymbol{S} \sim \mathcal{D}, \boldsymbol{A}} \Big( \boldsymbol{J}(\boldsymbol{w}^n) - \hat{J}_n(\boldsymbol{w}^n) \Big) \Big| \leq \epsilon_f,$$

*where $f_j^*$ and $f_j$ respectively denote $f_j(\boldsymbol{w}_*^j; \boldsymbol{x}'_{(j)}, \boldsymbol{y}'_{(j)})$ and $f_j(\boldsymbol{w}^n; \boldsymbol{x}'_{(j)}, \boldsymbol{y}'_{(j)})$, and $\epsilon_f$ is defined in Eqn. (2).*

According to Corollary 1, both the stability rate and the convergence rate of generalization error are $\mathcal{O}(\epsilon_f)$. This result indicates that deep learning empirical risk is stable and its output is robust to small perturbation over the training data. When $n$ is sufficiently large, small generalization error of DNNs is guaranteed.

## 5 RESULTS FOR DEEP NONLINEAR NEURAL NETWORKS

In the above section, we analyze the empirical risk optimization landscape for deep linear neural network models. In this section, we extend our analysis to deep nonlinear neural networks which adopt the sigmoid activation function. Our analysis techniques are also applicable to other third-order

differentiable activation functions, *e.g.*, tanh function with different convergence rate. Here we assume the input data are *i.i.d.* Gaussian variables.

**Assumption 2.** *The input datum $x$ is a vector of i.i.d. Gaussian variables from $\mathcal{N}(0, \tau^2)$.*

Since for any input, the sigmoid function always maps it to the range $[0, 1]$. Thus, we do not require the input $x$ to have bounded magnitude. Such an assumption is common. For instance, Tian (2017) and Soudry & Hoffer (2017) both assumed that the entries in the input vector are from Gaussian distribution. We also assume $w \in \Omega$ as in (Xu & Mannor, 2012). Here we also assume that the entry value of the target output $y$ falls in $[0, 1]$. Similar to the analysis of deep linear neural networks, here we also aim to characterize the empirical risk gradient, stationary points and empirical risk for deep nonlinear neural networks.

## 5.1 Uniform Convergence of Gradient and Stationary Points

Here we analyze convergence properties of gradients of the empirical risk for deep nonlinear neural networks.

**Theorem 4.** *Assume the input sample $x$ obeys Assumption 2 and the activation functions in a deep neural network are sigmoid functions. Then the empirical gradient uniformly converges to the population gradient in Euclidean norm. Specifically, there are two universal constants $c_y$ and $c_{y'}$ such that if $n \geq c_{y'} c_d l^3 r^2 / (s \log(d) \tau^2 \varepsilon^2 \log(1/\varepsilon))$ where $c_d = \max_{0 \leq i \leq l} d_i$, then with probability at least $1 - \varepsilon$*

$$\sup_{w \in \Omega} \left\| \nabla \hat{J}_n(w) - \nabla J(w) \right\|_2 \leq \epsilon_l \triangleq \tau \sqrt{\frac{512}{729} c_y l(l+2)(lc_r+1)c_d c_r} \sqrt{\frac{s \log(dn/l) + \log(4/\varepsilon)}{n}},$$

*where $c_r = \max(r^2/16, (r^2/16)^{l-1})$, and $s$ denotes the nonzero entry number of all weights.*

Similar to deep linear neural networks, the layer number $l$, width $d_i$, number of nonzero parameter entries $s$, network size $d$ and magnitude of weights are all critical to the convergence rate. Also, since there is a factor $\max_i d_i$ in the convergence rate, it is better to avoid choosing an extremely wide layer. Interestingly, when analyzing the representation ability of deep learning, Eldan & Shamir (2016) also suggested non-extreme-wide layers, though the conclusion was derived from a different perspective. By comparing Theorems 1 and 4, one can observe that there is a factor $(1/16)^{l-1}$ in the convergence rate in Theorem 4. This is because the convergence rate accesses the Lipschitz constant and when we bound it, sigmoid activation function brings the factor $1/16$ for each layer.

Now we analyze the non-degenerate stationary points of the empirical risk for deep nonlinear neural networks. Here we also assume that the population risk has $m$ non-degenerate stationary points denoted by $\{w^{(1)}, w^{(2)}, \cdots, w^{(m)}\}$.

**Theorem 5.** *Assume the input sample $x$ obeys Assumption 2 and the activation functions in a deep neural network are sigmoid functions. Then if $n \geq c_s \max\left(c_d l^3 r^2 / (s \log(d) \tau^2 \varepsilon^2 \log(1/\varepsilon)), s \log(d/l)/\zeta^2\right)$ where $c_s$ is a constant, for $k \in \{1, \cdots, m\}$, there exists a non-degenerate stationary point $w_n^{(k)}$ of $\hat{J}_n(w)$ which corresponds to the non-degenerate stationary point $w^{(k)}$ of $J(w)$ with probability at least $1 - \varepsilon$. Moreover, $w_n^{(k)}$ and $w^{(k)}$ have the same non-degenerate index and they obey*

$$\left\| w_n^{(k)} - w^{(k)} \right\|_2 \leq \frac{2\tau}{\zeta} \sqrt{\frac{512}{729} c_y l(l+2)(lc_r+1)c_d c_r} \sqrt{\frac{s \log(dn/l) + \log(4/\varepsilon)}{n}}, \ (k = 1, \cdots, m)$$

*with probability at least $1 - \varepsilon$, where $c_y$, $c_d$ and $c_r$ are the same parameters in Theorem 4.*

According to Theorem 5, there is one-to-one correspondence between the non-degenerate stationary points of $\hat{J}_n(w)$ and $J(w)$. Also the corresponding pair has the same non-degenerate index, implying they have exactly matching local minima/maxima and non-degenerate saddle points. When $n$ is sufficiently large, the non-degenerate stationary point $w_n^{(k)}$ in $\hat{J}_n(w)$ is very close to its corresponding non-degenerate stationary point $w^{(k)}$ in $J(w)$. As for the degenerate stationary points, Theorem 4 guarantees the gradients at these points of $J(w)$ and $\hat{J}_n(w)$ are very close to each other.

## 5.2 UNIFORM CONVERGENCE, STABILITY AND GENERALIZATION OF EMPIRICAL RISK

Here we first give the uniform convergence analysis of the empirical risk and then analyze its stability and generalization.

**Theorem 6.** *Assume the input sample $\boldsymbol{x}$ obeys Assumption 2 and the activation functions in a deep neural network are the sigmoid functions. If $n \geq 18 l^2 r^2 / (s \log(d) \tau^2 \varepsilon^2 \log(1/\varepsilon))$, then*

$$\sup_{\boldsymbol{w} \in \Omega} \left| \hat{\boldsymbol{J}}_n(\boldsymbol{w}) - \boldsymbol{J}(\boldsymbol{w}) \right| \leq \epsilon_n \triangleq \tau \sqrt{\frac{9}{8} c_y c_d \left(1 + c_r(l-1)\right)} \sqrt{\frac{s \log(nd/l) + \log(4/\varepsilon)}{n}} \tag{3}$$

*holds with probability at least $1 - \varepsilon$, where $c_y$, $c_d$ and $c_r$ are given in Theorem 4.*

From Theorem 6, we obtain that under the distribution $\mathcal{D}$, the empirical risk of a deep nonlinear neural network converges at the rate of $\mathcal{O}(1/\sqrt{n})$ (up to a $\log$ factor). Theorem 6 also gives similar results as Theorem 3, including the inclination of regularization penalty on weight and suggestion on non-extreme-wide layers. Similar to linear networks, our risk convergence rate is also tighter than the convergence rate on the networks with polynomial activation functions and one-dimensional output in (Bartlett & Maass, 2003) since ours is at the order of $\mathcal{O}(\sqrt{(l-1)(s\log(dn/l) + \log(1/\varepsilon))/n})$, while the later is $\mathcal{O}(\sqrt{(\gamma \log^2(n) + \log(1/\varepsilon))/n})$ where $\gamma$ is at the order of $\mathcal{O}(ld\log(d) + l^2 d)$ (Bartlett & Maass, 2003).

We then establish the stability property and the generalization error of the empirical risk for nonlinear neural networks. By Theorem 6, we can obtain the following results.

**Corollary 2.** *Assume the input sample $\boldsymbol{x}$ obeys Assumption 2 and the activation functions in a deep neural network are sigmoid functions. Then with probability at least $1 - \varepsilon$, we have*

$$\left| \mathbb{E}_{\boldsymbol{\mathcal{S}} \sim \mathcal{D}, \boldsymbol{A}, (\boldsymbol{x}'_{(j)}, \boldsymbol{y}'_{(j)}) \sim \mathcal{D}} \frac{1}{n} \sum_{j=1}^{n} \left( f_j^* - f_j \right) \right| \leq \epsilon_n \quad and \quad \left| \mathbb{E}_{\boldsymbol{\mathcal{S}} \sim \mathcal{D}, \boldsymbol{A}} \left( \boldsymbol{J}(\boldsymbol{w}^n) - \hat{\boldsymbol{J}}_n(\boldsymbol{w}^n) \right) \right| \leq \epsilon_n,$$

*where $\epsilon_n$ is defined in Eqn. (3). The notations $f_j^*$ and $f_j$ here are the same in Corollary 1.*

By Corollary 2, we know that both the stability convergence rate and the convergence rate of generalization error are $\mathcal{O}(1/\sqrt{n})$. This result accords with Theorems 8 and 9 in (Shalev-Shwartz et al., 2010) which implies $\mathcal{O}(1/\sqrt{n})$ is the bottleneck of the stability and generalization convergence rate for generic learning algorithms. From this result, we have that if $n$ is sufficiently large, the empirical risk can be expected to be very stable. This also dispels misgivings of the random selection of training samples in practice. Such a result indicates that the deep nonlinear neural network can offer good performance on testing data if it achieves small training error.

## 6 PROOF ROADMAP

Here we briefly introduce our proof roadmap. Due to space limitation, all the proofs of Theorems 1 ∼ 6 and Corollaries 1 and 2 as well as technical lemmas are deferred to the supplementary material.

The proofs of Theorems 1 and 4 are similar but essentially differ in some techniques for bounding probability due to their different assumptions. For explanation simplicity, we define four events: $\boldsymbol{E} = \{\sup_{\boldsymbol{w} \in \Omega} \|\nabla \hat{\boldsymbol{J}}_n(\boldsymbol{w}) - \nabla \boldsymbol{J}(\boldsymbol{w})\|_2 > t\}$, $\boldsymbol{E}_1 = \{\sup_{\boldsymbol{w} \in \Omega} \|\frac{1}{n}\sum_{i=1}^{n}(\nabla f(\boldsymbol{w}, \boldsymbol{x}_{(i)}) - \nabla f(\boldsymbol{w}_{k_{\boldsymbol{w}}}, \boldsymbol{x}_{(i)}))\|_2 > t/3\}$, $\boldsymbol{E}_2 = \{\sup_{\boldsymbol{w}^i_{k_{\boldsymbol{w}}} \in \mathcal{N}_i, i \in [l]} \|\frac{1}{n}\sum_{i=1}^{n}\nabla f(\boldsymbol{w}_{k_{\boldsymbol{w}}}, \boldsymbol{x}_{(i)}) - \mathbb{E}\nabla f(\boldsymbol{w}_{k_{\boldsymbol{w}}}, \boldsymbol{x})\|_2 > t/3\}$, and $\boldsymbol{E}_3 = \{\sup_{\boldsymbol{w} \in \Omega} \|\mathbb{E}\nabla f(\boldsymbol{w}_{k_{\boldsymbol{w}}}, \boldsymbol{x}) - \mathbb{E}\nabla f(\boldsymbol{w}, \boldsymbol{x})\|_2 > t/3\}$, where $\boldsymbol{w}_{k_{\boldsymbol{w}}} = [\boldsymbol{w}^1_{k_{\boldsymbol{w}}}; \boldsymbol{w}^2_{k_{\boldsymbol{w}}}; \cdots; \boldsymbol{w}^l_{k_{\boldsymbol{w}}}]$ is constructed by selecting $\boldsymbol{w}^i_{k_{\boldsymbol{w}}} \in \mathbb{R}^{d_i d_{i-1}}$ from $d_i d_{i-1} \epsilon/d$-net $\mathcal{N}_i$ such that $\|\boldsymbol{w} - \boldsymbol{w}_{k_{\boldsymbol{w}}}\|_2 \leq \epsilon$. Note that in Theorems 1 and 4, $t$ is respectively set to $\epsilon_g$ and $\epsilon_l$. Then we have $\mathbb{P}(\boldsymbol{E}) \leq \mathbb{P}(\boldsymbol{E}_1) + \mathbb{P}(\boldsymbol{E}_2) + \mathbb{P}(\boldsymbol{E}_3)$. So we only need to separately bound $\mathbb{P}(\boldsymbol{E}_1)$, $\mathbb{P}(\boldsymbol{E}_2)$ and $\mathbb{P}(\boldsymbol{E}_3)$. For $\mathbb{P}(\boldsymbol{E}_1)$ and $\mathbb{P}(\boldsymbol{E}_3)$, we use the gradient Lipschitz constant and the properties of $\epsilon$-net to prove $\mathbb{P}(\boldsymbol{E}_1) \leq \varepsilon/2$ and $\mathbb{P}(\boldsymbol{E}_3) = 0$, while bounding $\mathbb{P}(\boldsymbol{E}_2)$ needs more efforts. Here based on the assumptions, we prove that $\mathbb{P}(\boldsymbol{E}_2)$ has sub-exponential tail associated to the sample number $n$ and the networks parameters, and it satisfies $\mathbb{P}(\boldsymbol{E}_2) \leq \varepsilon/2$ with proper conditions. Finally, combining the bounds of the three terms, we obtain the desired results.

To prove Theorems 2 and 5, we first prove the uniform convergence of the empirical Hessian to its population Hessian. Then, we define such a set $D = \{w \in \Omega : \|\nabla J(w)\|_2 < \epsilon$ and $\inf_i |\lambda_i (\nabla^2 J(w))| \geq \zeta\}$. In this way, $D$ can be decomposed into countably components, with each component containing either exactly one or zero non-degenerate stationary point. For each component, the uniform convergence of gradient and the results in differential topology guarantee that if $J(w)$ has no stationary points, then $\hat{J}_n(w)$ also has no stationary points and vise versa. Similarly, for each component, the uniform convergence of Hessian and the results in differential topology guarantee that if $J(w)$ has a unique non-degenerate stationary point, then $\hat{J}_n(w)$ also has a unique non-degenerate stationary point with the same index. After establishing exact correspondence between the non-degenerate stationary points of empirical risk and population risk, we use the uniform convergence of gradient and Hessian to bound the distance between the corresponding pairs.

We adopt a similar strategy to prove Theorems 3 and 6. Specifically, we divide the event $\sup_{w \in \Omega} |\hat{J}_n(w) - \nabla J(w)| > t$ into $E_1$, $E_2$ and $E_3$ which have the same forms as their counterparts in the proofs of Theorem 1 with the gradient replaced by the loss function. To prove $\mathbb{P}(E_1) \leq \varepsilon/2$ and $\mathbb{P}(E_3) = 0$, we can use the Lipschitz constant of the loss function and the $\epsilon$-net properties. The remaining is to prove $\mathbb{P}(E_2)$. We also prove that it has sub-exponential tail associated to the sample number $n$ and the networks parameters and it obeys $\mathbb{P}(E_2) \leq \varepsilon/2$ with proper conditions. Then we utilize the uniform convergence of $\hat{J}_n(w)$ to prove the stability and generalization bounds of $\hat{J}_n(w)$ (*i.e.* Corollaries 1 and 2).

# 7    CONCLUSION

In this work, we provided theoretical analysis on the landscape of empirical risk optimization for deep linear/nonlinear neural networks with (stochastic-)gradient descent, including the properties of the gradient and stationary points of empirical risk as well as the uniform convergence, stability, and generalization of the empirical risk itself. To our best knowledge, most of the results are new to deep learning community. These results also reveal that the depth $l$, the nonzero entry number $s$ of all weights, the network size $d$ and the width of a network are critical to the convergence rates. We also prove that the weight parameter magnitude is important to the convergence rate. Indeed, small magnitude of the weights is suggested. All the results are consistent with the widely used network architectures in practice.

## ACKNOWLEDGMENT

This work is partially supported by National University of Singapore startup grant R-263-000-C08-133, Ministry of Education of Singapore AcRF Tier One grant R-263-000-C21-112, NUS IDS R-263-000-C67-646 and ECRA R-263-000-C87-133.

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

SUPPLEMENTARY MATERIAL OF EMPIRICAL RISK LANDSCAPE ANALYSIS FOR
UNDERSTANDING DEEP NEURAL NETWORKS

## A  STRUCTURE OF THIS DOCUMENT

This document gives some other necessary notations and preliminaries for our analysis in Sec. B.
Then we prove Theorems $1\sim3$ and Corollary 1 for deep linear neural networks in Sec. C. Then we
present the proofs of Theorems $4\sim6$ and Corollary 2 for deep nonlinear neural networks in Sec. D.

In both Sec. C and D, we first present the technical lemmas for proving our final results and
subsequently present the proofs of these lemmas. Then we utilize these technical lemmas to prove
our desired results. Finally, we give the proofs of other auxiliary lemmas.

## B  NOTATIONS AND PRELIMINARY TOOLS

Beyond the notations introduced in the manuscript, we need some other notations used in this
document. Then we introduce several lemmas that will be used later.

### B.1  NOTATIONS

Throughout this document, we use $\langle \cdot, \cdot \rangle$ to denote the inner product. $\boldsymbol{A} \otimes \boldsymbol{C}$ denotes the Kronecker
product between $\boldsymbol{A}$ and $\boldsymbol{C}$. Note that $\boldsymbol{A}$ and $\boldsymbol{C}$ in $\boldsymbol{A} \otimes \boldsymbol{C}$ can be matrices or vectors. For a matrix
$\boldsymbol{A} \in \mathbb{R}^{n_1 \times n_2}$, we use $\|\boldsymbol{A}\|_F = \sqrt{\sum_{i,j} \boldsymbol{A}_{ij}^2}$ to denote its Frobenius norm, where $\boldsymbol{A}_{ij}$ is the $(i,j)$-th
entry of $\boldsymbol{A}$. We use $\|\boldsymbol{A}\|_{\mathrm{op}} = \max_i |\lambda_i(\boldsymbol{A})|$ to denote the operation norm of a matrix $\boldsymbol{A} \in \mathbb{R}^{n_1 \times n_1}$,
where $\lambda_i(\boldsymbol{A})$ denotes the $i$-th eigenvalue of the matrix $\boldsymbol{A}$. For a 3-way tensor $\boldsymbol{\mathcal{A}} \in \mathbb{R}^{n_1 \times n_2 \times n_3}$, its
operation norm is computed as

$$\|\boldsymbol{\mathcal{A}}\|_{\mathrm{op}} = \sup_{\|\boldsymbol{\lambda}\|_2 \leq 1} \left\langle \boldsymbol{\lambda}^{\otimes^3}, \boldsymbol{\mathcal{A}} \right\rangle = \sum_{i,j,k} \boldsymbol{\mathcal{A}}_{ijk} \boldsymbol{\lambda}_i \boldsymbol{\lambda}_j \boldsymbol{\lambda}_k,$$

where $\boldsymbol{\mathcal{A}}_{ijk}$ denotes the $(i,j,k)$-th entry of $\boldsymbol{\mathcal{A}}$. Also we denote the vectorization of $\boldsymbol{W}^{(j)}$ (the weight
matrix of the $j$-th layer) as

$$\boldsymbol{w}_{(j)} = \mathrm{vec}\left(\boldsymbol{W}^{(j)}\right) \in \mathbb{R}^{\boldsymbol{d}_j \boldsymbol{d}_{j-1}}.$$

We denote $\boldsymbol{I}_k$ as the identity matrix of size $k \times k$.

For notational simplicity, we further define $\boldsymbol{e} \triangleq \boldsymbol{v}^{(l)} - \boldsymbol{y}$ as the output error vector. Then the squared
loss is defined as $f(\boldsymbol{w}; \boldsymbol{x}, \boldsymbol{y}) = \frac{1}{2}\|\boldsymbol{e}\|_2^2$, where $\boldsymbol{w} = (\boldsymbol{w}_{(1)}; \cdots; \boldsymbol{w}_{(l)}) \in \mathbb{R}^d$ contains all the weight
parameters.

### B.2  TECHNICAL LEMMAS

We first introduce Lemmas 1 and 2 which are respectively used for bounding the $\ell_2$-norm of a vector
and the operation norm of a matrix. Then we introduce Lemmas 3 and 4 which discuss the topology
of functions. In Lemma 5, we give the relationship between the stability and generalization of
empirical risk.

**Lemma 1.** *(Vershynin, 2012) For any vector $\boldsymbol{x} \in \mathbb{R}^d$, its $\ell_2$-norm can be bounded as*

$$\|\boldsymbol{x}\|_2 \leq \frac{1}{1-\epsilon} \sup_{\boldsymbol{\lambda} \in \boldsymbol{\lambda}_\epsilon} \langle \boldsymbol{\lambda}, \boldsymbol{x} \rangle.$$

*where $\boldsymbol{\lambda}_\epsilon = \{\boldsymbol{\lambda}_1, \ldots, \boldsymbol{\lambda}_{k_{\boldsymbol{w}}}\}$ be an $\epsilon$-covering net of $\mathsf{B}^d(1)$.*

**Lemma 2.** *(Vershynin, 2012) For any symmetric matrix $\boldsymbol{X} \in \mathbb{R}^{d \times d}$, its operator norm can be
bounded as*

$$\|\boldsymbol{X}\|_{op} \leq \frac{1}{1-2\epsilon} \sup_{\boldsymbol{\lambda} \in \boldsymbol{\lambda}_\epsilon} |\langle \boldsymbol{\lambda}, \boldsymbol{X}\boldsymbol{\lambda} \rangle|.$$

*where $\boldsymbol{\lambda}_\epsilon = \{\boldsymbol{\lambda}_1, \ldots, \boldsymbol{\lambda}_{k_{\boldsymbol{w}}}\}$ be an $\epsilon$-covering net of $\mathsf{B}^d(1)$.*

**Lemma 3.** *(Mei et al., 2017) Let $D \subseteq \mathbb{R}^d$ be a compact set with a $C^2$ boundary $\partial D$, and $f, g$ : $A \to \mathbb{R}$ be $C^2$ functions defined on an open set $A$, with $D \subseteq A$. Assume that for all $\boldsymbol{w} \in \partial D$ and all $t \in [0, 1]$, $t\nabla f(\boldsymbol{w}) + (1 - t)\nabla g(\boldsymbol{w}) \neq \boldsymbol{0}$. Finally, assume that the Hessian $\nabla^2 f(\boldsymbol{w})$ is non-degenerate and has index equal to $r$ for all $\boldsymbol{w} \in D$. Then the following properties hold:*

*(1) If $g$ has no critical point in $D$, then $f$ has no critical point in $D$.*

*(2) If $g$ has a unique critical point $\boldsymbol{w}$ in $D$ that is non-degenerate with an index of $r$, then $f$ also has a unique critical point $\boldsymbol{w}'$ in $D$ with the index equal to $r$.*

**Lemma 4.** *(Mei et al., 2017) Suppose that $F(\boldsymbol{w}) : \Theta \to \mathbb{R}$ is a $C^2$ function where $\boldsymbol{w} \in \Theta$. Assume that $\{\boldsymbol{w}^{(1)}, \ldots, \boldsymbol{w}^{(m)}\}$ is its non-degenerate critical points and let $D = \{\boldsymbol{w} \in \Theta : \|\nabla F(\boldsymbol{w})\|_2 < \epsilon$ and $\inf_i |\lambda_i (\nabla^2 F(\boldsymbol{w}))| \geq \zeta\}$. Then $D$ can be decomposed into (at most) countably components, with each component containing either exactly one critical point, or no critical point. Concretely, there exist disjoint open sets $\{D_k\}_{k \in \mathbb{N}}$, with $D_k$ possibly empty for $k \geq m + 1$, such that*

$$D = \cup_{k=1}^{\infty} D_k .$$

*Furthermore, $\boldsymbol{w}^{(k)} \in D_k$ for $1 \leq k \leq m$ and each $D_i$, $k \geq m + 1$ contains no stationary points.*

**Lemma 5.** *(Shalev-Shwartz & Ben-David, 2014b; Gonen & Shalev-Shwartz, 2017) Assume that $\mathcal{D}$ is a sample distribution and randomized algorithm $\boldsymbol{A}$ is employed for optimization. Suppose that $((\boldsymbol{x}'_{(1)}, \boldsymbol{y}'_{(1)}), \cdots, (\boldsymbol{x}'_{(n)}, \boldsymbol{y}'_{(n)})) \sim \mathcal{D}$ and $\boldsymbol{w}^n = \operatorname{argmin}_{\boldsymbol{w}} \hat{\boldsymbol{J}}_n(\boldsymbol{w})$. For every $j \in \{1, \cdots, n\}$, suppose $\boldsymbol{w}_*^j = \operatorname{argmin}_{\boldsymbol{w}} \frac{1}{n-1} \sum_{i \neq j} f_i(\boldsymbol{w}; \boldsymbol{x}_{(i)}, \boldsymbol{y}_{(i)})$. For arbitrary distribution $\mathcal{D}$, we have*

$$\left| \mathbb{E}_{\boldsymbol{S} \sim \mathcal{D}, \boldsymbol{A}, (\boldsymbol{x}'_{(j)}, \boldsymbol{y}'_{(j)}) \sim \mathcal{D}} \frac{1}{n} \sum_{j=1}^{n} \left( f_j^* - f_j \right) \right| = \left| \mathbb{E}_{\boldsymbol{S} \sim \mathcal{D}, \boldsymbol{A}} \left( \boldsymbol{J}(\boldsymbol{w}^n) - \hat{\boldsymbol{J}}_n(\boldsymbol{w}^n) \right) \right|.$$

*where $f_j^*$ and $f_j$ respectively denote $f_j(\boldsymbol{w}_*^j; \boldsymbol{x}'_{(j)}, \boldsymbol{y}'_{(j)})$ and $f_j(\boldsymbol{w}^n; \boldsymbol{x}'_{(j)}, \boldsymbol{y}'_{(j)})$.*

## C   PROOFS FOR DEEP LINEAR NEURAL NETWORKS

In this section, we first present the technical lemmas in Sec. C.1 and then we give the proofs of these lemmas in Sec. C.2. Next, we utilize these lemmas to prove the results in Theorems $1 \sim 3$ and Corollary 1 in Sec. C.3. Finally, we give the proofs of other lemmas in Sec. C.4.

### C.1   TECHNICAL LEMMAS

Here we present the technical lemmas for proving our desired results. For brevity, we also define $\boldsymbol{B}_{j:s}$ as follows:

$$\boldsymbol{B}_{s:t} \triangleq \boldsymbol{W}^{(s)} \boldsymbol{W}^{(s-1)} \cdots \boldsymbol{W}^{(t)} \in \mathbb{R}^{\boldsymbol{d}_s \times \boldsymbol{d}_{t-1}}, \ (s \geq t); \quad \boldsymbol{B}_{s:t} \triangleq \boldsymbol{I}, \ (s < t). \tag{4}$$

**Lemma 6.** *Assume that the activation functions in the deep neural network $f(\boldsymbol{w}, \boldsymbol{x})$ are linear functions. Then the gradient of $f(\boldsymbol{w}, \boldsymbol{x})$ with respect to $\boldsymbol{w}_{(j)}$ can be written as*

$$\nabla_{\boldsymbol{w}_{(j)}} f(\boldsymbol{w}, \boldsymbol{x}) = \left( (\boldsymbol{B}_{j-1:1} \boldsymbol{x}) \otimes \boldsymbol{B}_{l:j+1}^T \right) \boldsymbol{e}, \ (j = 1, \cdots, l),$$

*where $\otimes$ denotes the Kronecke product. Then we can compute the Hessian matrix as follows:*

$$\nabla^2 f(\boldsymbol{w}, \boldsymbol{x}) = \begin{bmatrix} \nabla_{\boldsymbol{w}_{(1)}} \left( \nabla_{\boldsymbol{w}_{(1)}} f(\boldsymbol{w}, \boldsymbol{x}) \right) & \cdots & \nabla_{\boldsymbol{w}_{(1)}} \left( \nabla_{\boldsymbol{w}_{(l)}} f(\boldsymbol{w}, \boldsymbol{x}) \right) \\ \nabla_{\boldsymbol{w}_{(2)}} \left( \nabla_{\boldsymbol{w}_{(1)}} f(\boldsymbol{w}, \boldsymbol{x}) \right) & \cdots & \nabla_{\boldsymbol{w}_{(2)}} \left( \nabla_{\boldsymbol{w}_{(l)}} f(\boldsymbol{w}, \boldsymbol{x}) \right) \\ \vdots & \ddots & \vdots \\ \nabla_{\boldsymbol{w}_{(l)}} \left( \nabla_{\boldsymbol{w}_{(1)}} f(\boldsymbol{w}, \boldsymbol{x}) \right) & \cdots & \nabla_{\boldsymbol{w}_{(l)}} \left( \nabla_{\boldsymbol{w}_{(l)}} f(\boldsymbol{w}, \boldsymbol{x}) \right) \end{bmatrix},$$

*where $\boldsymbol{Q}_{st} \triangleq \nabla_{\boldsymbol{w}_{(s)}} \left( \nabla_{\boldsymbol{w}_{(t)}} f(\boldsymbol{w}, \boldsymbol{x}) \right)$ is defined as*

$$\boldsymbol{Q}_{st} = \begin{cases} \left( \boldsymbol{B}_{t-1:s+1}^T \right) \otimes \left( \boldsymbol{B}_{s-1:1} \boldsymbol{x} \boldsymbol{e}^T \boldsymbol{B}_{l:t+1}^T \right) + \left( \boldsymbol{B}_{s-1:1} \boldsymbol{x} \boldsymbol{x}^T \boldsymbol{B}_{t-1:1}^T \right) \otimes \left( \boldsymbol{B}_{l:s+1}^T \boldsymbol{B}_{l:t+1} \right), & \text{if } s < t, \\ \left( \boldsymbol{B}_{s-1:1} \boldsymbol{x} \boldsymbol{x}^T \boldsymbol{B}_{s-1:1} \right) \otimes \left( \boldsymbol{B}_{l:s+1}^T \boldsymbol{B}_{l:s+1} \right), & \text{if } s = t, \\ \left( \boldsymbol{B}_{l:s+1}^T \boldsymbol{e} \boldsymbol{x}^T \boldsymbol{B}_{t-1:1}^T \right) \otimes \boldsymbol{B}_{s-1:t+1} + \left( \boldsymbol{B}_{s-1:1} \boldsymbol{x} \boldsymbol{x}^T \boldsymbol{B}_{t-1:1}^T \right) \otimes \left( \boldsymbol{B}_{l:s+1}^T \boldsymbol{B}_{l:t+1} \right), & \text{if } s > t. \end{cases}$$

**Lemma 7.** *Suppose Assumption 1 on the input data $\boldsymbol{x}$ holds and the activation functions in deep neural network are linear functions. Then for any $t > 0$, the objective $f(\boldsymbol{w}, \boldsymbol{x})$ obeys*

$$\mathbb{P}\left(\frac{1}{n}\sum_{i=1}^{n}\left(f(\boldsymbol{w}, \boldsymbol{x}_{(i)}) - \mathbb{E}(f(\boldsymbol{w}, \boldsymbol{x}_{(i)}))\right) > t\right) \leq 2\exp\left(-c_{f'}n\min\left(\frac{t^2}{\omega_f^2\max\left(\boldsymbol{d}_l\omega_f^2\tau^4, \tau^2\right)}, \frac{t}{\omega_f^2\tau^2}\right)\right),$$

*where $c_{f'}$ is a positive constant and $\omega_f = r^l$.*

**Lemma 8.** *Suppose Assumption 1 on the input data $\boldsymbol{x}$ holds and the activation functions in deep neural network are linear functions. Then for any $t > 0$ and arbitrary unit vector $\boldsymbol{\lambda} \in \mathbb{S}^{d-1}$, the gradient $\nabla f(\boldsymbol{w}, \boldsymbol{x})$ obeys*

$$\mathbb{P}\left(\frac{1}{n}\sum_{i=1}^{n}\left(\langle\boldsymbol{\lambda}, \nabla_{\boldsymbol{w}}f(\boldsymbol{w}, \boldsymbol{x}_{(i)}) - \mathbb{E}\nabla_{\boldsymbol{w}}f(\boldsymbol{w}, \boldsymbol{x}_{(i)})\rangle\right) > t\right)$$

$$\leq 3\exp\left(-c_{g'}n\min\left(\frac{t^2}{l\max\left(\omega_g\tau^2, \omega_g\tau^4, \omega_{g'}\tau^2\right)}, \frac{t}{\sqrt{l\omega_g}\max\left(\tau, \tau^2\right)}\right)\right),$$

*where $c_{g'}$ is a constant; $\omega_g = c_q r^{2(2l-1)}$ and $\omega_{g'} = c_q r^{2(l-1)}$ in which $c_q = \sqrt{\max_{0\leq i\leq l}\boldsymbol{d}_i}$.*

**Lemma 9.** *Suppose Assumption 1 on the input data $\boldsymbol{x}$ holds and the activation functions in deep neural network are linear functions. Then for any $t > 0$ and arbitrary unit vector $\boldsymbol{\lambda} \in \mathbb{S}^{d-1}$, the Hessian $\nabla^2 f(\boldsymbol{w}, \boldsymbol{x})$ obeys*

$$\mathbb{P}\left(\frac{1}{n}\sum_{i=1}^{n}\left(\langle\boldsymbol{\lambda}, (\nabla_{\boldsymbol{w}}^2 f(\boldsymbol{w}, \boldsymbol{x}_{(i)}) - \mathbb{E}\nabla_{\boldsymbol{w}}^2 f(\boldsymbol{w}, \boldsymbol{x}_{(i)}))\boldsymbol{\lambda}\rangle\right) > t\right)$$

$$\leq 5\exp\left(-c_{h'}n\min\left(\frac{t^2}{\tau^2 l^2\max\left(\omega_g, \omega_g\tau^2, \omega_h\right)}, \frac{t}{\sqrt{\omega_g}l\max\left(\tau, \tau^2\right)}\right)\right),$$

*where $\omega_g = r^{4(l-1)}$ and $\omega_h = r^{2(l-2)}$.*

**Lemma 10.** *Suppose the activation functions in deep neural network are linear functions. Then for any $\boldsymbol{w} \in \mathsf{B}^d(r)$ and $\boldsymbol{x} \in \mathsf{B}^{\boldsymbol{d}_0}(r_x)$, we have*

$$\|\nabla_{\boldsymbol{w}}f(\boldsymbol{w}, \boldsymbol{x})\|_2 \leq \sqrt{\alpha_g}, \quad \text{where} \quad \alpha_g = c_t l r_x^4 r^{4l-2}.$$

*in which $c_t$ is a constant. Further, for any $\boldsymbol{w} \in \mathsf{B}^d(r)$ and $\boldsymbol{x} \in \mathsf{B}^{\boldsymbol{d}_0}(r_x)$, we also have*

$$\left\|\nabla^2 f(\boldsymbol{w}, \boldsymbol{x})\right\|_{op} \leq \left\|\nabla^2 f(\boldsymbol{w}, \boldsymbol{x})\right\|_F \leq l\sqrt{\alpha_l}, \quad \text{where} \quad \alpha_l \triangleq c_{t'} r_x^4 r^{4l-2}.$$

*in which $c_{t'}$ is a constant. With the same condition, we can bound the operation norm of $\nabla^3 f(\boldsymbol{w}, \boldsymbol{x})$. That is, there exists a universal constant $\alpha_p$ such that $\left\|\nabla^3 f(\boldsymbol{w}, \boldsymbol{x})\right\|_{op} \leq \alpha_p$.*

**Lemma 11.** *Suppose Assumption 1 on the input data $\boldsymbol{x}$ holds and the activation functions in deep neural network are linear functions. Then there exist two universal constant $c_g$ and $c_h$ such that the sample Hessian converges uniformly to the population Hessian in operator norm. Specifically, there exit two universal constants $c_{h_1}$ and $c_{h_2}$ such that if $n \geq c_{h_2}\max(\frac{\alpha_p^2 r^2}{\tau^2 l^2\omega_h^2\varepsilon^2 s\log(d/l)}, s\log(d/l)/(l\tau^2))$, then*

$$\sup_{\boldsymbol{w}\in\Omega}\left\|\nabla^2\hat{\boldsymbol{J}}_n(\boldsymbol{w}) - \nabla^2\boldsymbol{J}(\boldsymbol{w})\right\|_{op} \leq c_{h_1}\tau l\omega_h\sqrt{\frac{d\log(nl)+\log(20/\varepsilon)}{n}}$$

*holds with probability at least $1 - \varepsilon$, where $\omega_h = \max\left(\tau r^{2(l-1)}, r^{2(l-2)}, r^{l-2}\right)$.*

## C.2 Proofs of Technical Lemmas

To prove the above lemmas, we first introduce some useful results.

**Lemma 12.** *(Rudelson & Vershynin, 2013) Assume that $\boldsymbol{x} = (\boldsymbol{x}_1; \boldsymbol{x}_2; \cdots; \boldsymbol{x}_k) \in \mathbb{R}^k$ is a random vector with independent components $x_i$ which have zero mean and are independent $\tau_i^2$-sub-Gaussian variables. Here $\max_i \tau_i^2 \leq \tau^2$. Let $\boldsymbol{A}$ be an $k \times k$ matrix. Then we have*

$$\mathbb{E}\exp\left(\lambda\left(\sum_{i,j:i\neq j}\boldsymbol{A}_{ij}x_ix_j - \mathbb{E}(\sum_{i,j:i\neq j}\boldsymbol{A}_{ij}x_ix_j)\right)\right) \leq \exp\left(2\tau^2\lambda^2\|\boldsymbol{A}\|_F^2\right), \ |\lambda| \leq 1/(2\tau\|\boldsymbol{A}\|_2).$$

**Lemma 13.** *Assume that $\boldsymbol{x} = (\boldsymbol{x}_1; \boldsymbol{x}_2; \cdots; \boldsymbol{x}_k) \in \mathbb{R}^k$ is a random vector with independent components $x_i$ which have zero mean and are independent $\tau_i^2$-sub-Gaussian variables. Here $\max_i \tau_i^2 \le \tau^2$. Let $\boldsymbol{a}$ be an $n$-dimensional vector. Then we have*

$$\mathbb{E}\exp\left(\lambda\left(\sum_{i=1}^{k} \boldsymbol{a}_i \boldsymbol{x}_i^2 - \mathbb{E}\left(\sum_{i=1}^{k} \boldsymbol{a}_i \boldsymbol{x}_i^2\right)\right)\right) \le \mathbb{E}\exp\left(128\lambda^2\tau^4\left(\sum_{i=1}^{k} \boldsymbol{a}_i^2\right)\right), \quad |\lambda| \le \frac{1}{\tau^2 \max_i \boldsymbol{a}_i}.$$

**Lemma 14.** *For $\boldsymbol{B}_{j:t}$ defined in Eqn. (4), we have the following properties:*

$$\|\boldsymbol{B}_{s:t}\|_{op} \le \|\boldsymbol{B}_{s:t}\|_F \le \omega_r \quad and \quad \|\boldsymbol{B}_{l:1}\|_{op} \le \|\boldsymbol{B}_{l:1}\|_F \le \omega_f,$$

*where $\omega_r = r^{s-t+1} \le \max\left(r, r^l\right)$ and $\omega_f = r^l$.*

Lemma 13 is useful for bounding probability. The two inequalities in Lemma 14 can be obtained by using $\|\boldsymbol{w}_{(j)}\|_2 \le r \ (\forall j = 1, \cdots, l)$. We defer the proofs of Lemmas 13 and 14 to Sec. C.4.2.

### C.2.1 PROOF OF LEMMA 6

*Proof.* When the activation functions are linear functions, we can easily compute the gradient of $f(\boldsymbol{w}, \boldsymbol{x})$ with respect to $\boldsymbol{w}_{(j)}$:

$$\nabla_{\boldsymbol{w}_{(j)}} f(\boldsymbol{w}, \boldsymbol{x}) = \left((\boldsymbol{B}_{j-1:1}\boldsymbol{x}) \otimes \boldsymbol{B}_{l:j+1}^T\right) \boldsymbol{e}, \ (j = 1, \cdots, l),$$

where $\otimes$ denotes the Kronecker product. Now we consider the computation of the Hessian matrix. For brevity, let $\boldsymbol{Q}_s = \left((\boldsymbol{B}_{s-1:1}\boldsymbol{x}) \otimes \boldsymbol{B}_{l:s+1}^T\right)$. Then we can compute $\nabla_{\boldsymbol{w}_{(s)}}^2 f(\boldsymbol{w}, \boldsymbol{x})$ as follows:

$$\begin{aligned}
\nabla_{\boldsymbol{w}_{(s)}}^2 f(\boldsymbol{w}, \boldsymbol{x}) &= \frac{\partial^2 f(\boldsymbol{w}, \boldsymbol{x})}{\partial \boldsymbol{w}_{(s)}^T \partial \boldsymbol{w}_{(s)}} = \frac{\partial^2 f(\boldsymbol{w}, \boldsymbol{x})}{\partial \boldsymbol{w}_{(s)}^T \partial \boldsymbol{w}_{(s)}} = \frac{\partial (\boldsymbol{Q}_s \boldsymbol{e})}{\partial \boldsymbol{w}_{(s)}^T} = \frac{\partial \mathrm{vec}\,(\boldsymbol{Q}_s \boldsymbol{e})}{\partial \boldsymbol{w}_{(s)}^T} \\
&= \frac{\partial \mathrm{vec}\left(\boldsymbol{Q}_s \boldsymbol{B}_{l:s+1} \boldsymbol{W}^{(t)} \boldsymbol{B}_{s-1:1}\boldsymbol{x}\right)}{\partial \boldsymbol{w}_{(s)}^T} \\
&= \frac{\partial\left((\boldsymbol{B}_{s-1:1}\boldsymbol{x})^T \otimes (\boldsymbol{Q}_s \boldsymbol{B}_{l:s+1})\right) \mathrm{vec}\left(\boldsymbol{W}^{(s)}\right)}{\partial \boldsymbol{w}_{(s)}^T} \\
&= (\boldsymbol{B}_{s-1:1}\boldsymbol{x})^T \otimes \left(\left((\boldsymbol{B}_{s-1:1}\boldsymbol{x}) \otimes \boldsymbol{B}_{l:s+1}^T\right) \boldsymbol{B}_{l:s+1}\right) \\
&\overset{①}{=} (\boldsymbol{B}_{s-1:1}\boldsymbol{x})^T \otimes \left((\boldsymbol{B}_{s-1:1}\boldsymbol{x}) \otimes \left(\boldsymbol{B}_{l:s+1}^T \boldsymbol{B}_{l:s+1}\right)\right) \\
&\overset{②}{=} \left((\boldsymbol{B}_{s-1:1}\boldsymbol{x})^T \otimes (\boldsymbol{B}_{s-1:1}\boldsymbol{x})\right) \otimes \left(\boldsymbol{B}_{l:s+1}^T \boldsymbol{B}_{l:s+1}\right) \\
&\overset{③}{=} \left((\boldsymbol{B}_{s-1:1}\boldsymbol{x})(\boldsymbol{B}_{s-1:1}\boldsymbol{x})^T\right) \otimes \left(\boldsymbol{B}_{l:s+1}^T \boldsymbol{B}_{l:s+1}\right),
\end{aligned}$$

where ① holds since $\boldsymbol{B}_{j-1:1}\boldsymbol{x}$ is a vector and for any vector $\boldsymbol{x}$, we have $(\boldsymbol{x} \otimes \boldsymbol{A})\boldsymbol{B} = \boldsymbol{x} \otimes (\boldsymbol{A}\boldsymbol{B})$. ② holds because for any four matrices $\boldsymbol{Z}_1 \sim \boldsymbol{Z}_3$ of proper sizes, we have $(\boldsymbol{Z}_1 \otimes \boldsymbol{Z}_2) \otimes \boldsymbol{Z}_3 = \boldsymbol{Z}_1 \otimes (\boldsymbol{Z}_2 \otimes \boldsymbol{Z}_3)$. ③ holds because for any two matrices $\boldsymbol{z}_1, \boldsymbol{z}_2$ of proper sizes, we have $\boldsymbol{z}_1 \boldsymbol{z}_2^T = \boldsymbol{z}_1 \otimes \boldsymbol{z}_2^T = \boldsymbol{z}_2^T \otimes \boldsymbol{z}_1$.

Then, we consider the case $s > t$:

$$\begin{aligned}
\nabla_{\boldsymbol{w}_{(t)}}\left(\nabla_{\boldsymbol{w}_{(s)}} f(\boldsymbol{w}, \boldsymbol{x})\right) &= \frac{\partial^2 f(\boldsymbol{w}, \boldsymbol{x})}{\partial \boldsymbol{w}_{(t)}^T \partial \boldsymbol{w}_{(s)}} = \frac{\partial^2 f(\boldsymbol{w}, \boldsymbol{x})}{\partial \boldsymbol{w}_{(t)}^T \partial \boldsymbol{w}_{(s)}} = \frac{\partial(\boldsymbol{Q}_s \boldsymbol{e})}{\partial \boldsymbol{w}_{(t)}^T} = \frac{\partial \mathrm{vec}\,(\boldsymbol{Q}_s \boldsymbol{e})}{\partial \boldsymbol{w}_{(t)}^T} \\
&= \frac{\partial \mathrm{vec}\left(\boldsymbol{Q}_s \boldsymbol{B}_{l:t+1} \boldsymbol{W}^{(t)} \boldsymbol{B}_{t-1:1}\boldsymbol{x}\right)}{\partial \boldsymbol{w}_{(t)}^T} + \frac{\partial \mathrm{vec}\left(\left((\boldsymbol{B}_{s-1:1}\boldsymbol{x}) \otimes \boldsymbol{B}_{l:s+1}^T\right) \boldsymbol{e}\right)}{\partial \boldsymbol{w}_{(t)}^T}.
\end{aligned}$$

Notice, here we just think that $\boldsymbol{Q}_s$ in the $\frac{\partial \mathrm{vec}\left(\boldsymbol{Q}_s \boldsymbol{B}_{l:t+1} \boldsymbol{W}^{(t)} \boldsymbol{B}_{t-1:1}\boldsymbol{x}\right)}{\partial \boldsymbol{w}_{(t)}^T}$ is a constant matrix and is not related to $\boldsymbol{W}^{(t)}$. Similarly, we also take $\boldsymbol{e}$ in $\frac{\partial \mathrm{vec}\left(\left((\boldsymbol{B}_{s-1:1}\boldsymbol{x}) \otimes \boldsymbol{B}_{l:s+1}^T\right) \boldsymbol{e}\right)}{\partial \boldsymbol{w}_{(t)}^T}$ as a constant vector. Since we have

$$\frac{\partial \mathrm{vec}\left(\boldsymbol{Q}_s \boldsymbol{B}_{l:t+1} \boldsymbol{W}^{(t)} \boldsymbol{B}_{t-1:1}\boldsymbol{x}\right)}{\partial \boldsymbol{w}_{(t)}^T} = \left(\boldsymbol{B}_{s-1:1}\boldsymbol{x}\boldsymbol{x}^T \boldsymbol{B}_{t-1:1}^T\right) \otimes \left(\boldsymbol{B}_{l:s+1}^T \boldsymbol{B}_{l:t+1}\right),$$

we only need to consider

$$\frac{\partial \text{vec}\left(\left(\left(B_{s-1:1}x\right)\otimes B_{l:s+1}^T\right)e\right)}{\partial w_{(t)}^T} = \frac{\partial \text{vec}\left(\left(B_{s-1:1}x\right)\otimes\left(B_{l:s+1}^T e\right)\right)}{\partial w_{(t)}^T}$$

$$= \frac{\partial \text{vec}\left(\left(B_{s-1:1}x\right)\left(B_{l:s+1}^T e\right)^T\right)}{\partial w_{(t)}^T}$$

$$= \frac{\partial \text{vec}\left(B_{s-1:t+1}W^{(t)}\left(B_{t-1:1}xe^T B_{l:s+1}\right)\right)}{\partial w_t^T}$$

$$= \frac{\partial \left(B_{t-1:1}xe^T B_{l:s+1}\right)^T \otimes B_{s-1:t+1}\text{vec}\left(W^{(t)}\right)}{\partial w_t^T}$$

$$= \left(B_{t-1:1}xe^T B_{l:s+1}\right)^T \otimes B_{s-1:t+1}.$$

Therefore, for $s > t$, by combining the above two terms, we can obtain

$$\nabla_{w_{(t)}}\left(\nabla_{w_{(s)}}f(w,x)\right) = \left(B_{l:s+1}^T ex^T B_{t-1:1}^T\right)\otimes B_{s-1:t+1} + \left(B_{s-1:1}xx^T B_{t-1:1}^T\right)\otimes\left(B_{l:s+1}^T B_{l:t+1}\right).$$

Then, by similar method, we can compute the Hessian for the case $s < t$ as follows:

$$\nabla_{w_{(t)}}\left(\nabla_{w_{(s)}}f(w,x)\right) = \left(B_{t-1:s+1}^T\right)\otimes\left(B_{s-1:1}xe^T B_{l:t+1}^T\right) + \left(B_{s-1:1}xx^T B_{t-1:1}^T\right)\otimes\left(B_{l:s+1}^T B_{l:t+1}\right).$$

The proof is completed. □

### C.2.2  PROOF OF LEMMA 7

*Proof.* We first prove that $v^{(l)}$, which is defined in Eqn. (5), is sub-Gaussian.

$$v^{(l)} = W^{(l)}\cdots W^{(1)}x = B_{l:1}x. \tag{5}$$

Then by the convexity in $\lambda$ of $\exp(\lambda t)$ and Lemma 14, we can obtain

$$\mathbb{E}\left(\exp\left(\left\langle\lambda, v^{(l)} - \mathbb{E}(v^{(l)})\right\rangle\right)\right) = \mathbb{E}\left(\exp\left(\langle\lambda, B_{l:1}x - \mathbb{E}B_{l:1}x\rangle\right)\right)$$

$$\leq \mathbb{E}\left(\exp\left(\langle B_{l:1}^T\lambda, x\rangle\right)\right)$$

$$\leq \exp\left(\frac{\|B_{l:1}^T\lambda\|_2^2\tau^2}{2}\right) \tag{6}$$

$$\overset{①}{\leq} \exp\left(\frac{\omega_f^2\tau^2\|\lambda\|_2^2}{2}\right),$$

where ① uses the conclusion that $\|B_{l:1}\|_{\text{op}} \leq \|B_{l:1}\|_F \leq \omega_f$ in Lemma 14. This means that $v^{(l)}$ is centered and is $\omega_f^2\tau^2$-sub-Gaussian. Accordingly, we can obtain that the $k$-th entry of $v^{(l)}$ is also $z_k\tau^2$-sub-Gaussian, where $z_k$ is a universal positive constant. Note that $\max_k z_k \leq \omega_f^2$. Let $v_i^{(l)}$

denotes the output of the $i$-th sample $\boldsymbol{x}_{(i)}$. By Lemma 13, we have that for $s > 0$,

$$
\begin{aligned}
\mathbb{P}\left(\frac{1}{n}\sum_{i=1}^{n}\left(\|\boldsymbol{v}_i^{(l)}\|_2^2 - \mathbb{E}\|\boldsymbol{v}_i^{(l)}\|_2^2\right) > \frac{t}{2}\right) &= \mathbb{P}\left(s\sum_{i=1}^{n}\left(\|\boldsymbol{v}_i^{(l)}\|_2^2 - \mathbb{E}\|\boldsymbol{v}_i^{(l)}\|_2^2\right) > \frac{nst}{2}\right) \\
&\overset{①}{\leq} \exp\left(-\frac{snt}{2}\right)\mathbb{E}\left(s\sum_{i=1}^{n}\left(\|\boldsymbol{v}_i^{(l)}\|_2^2 - \mathbb{E}\|\boldsymbol{v}_i^{(l)}\|_2^2\right)\right) \\
&\overset{②}{\leq} \exp\left(-\frac{snt}{2}\right)\prod_{i=1}^{n}\mathbb{E}\left(s\left(\|\boldsymbol{v}_i^{(l)}\|_2^2 - \mathbb{E}\|\boldsymbol{v}_i^{(l)}\|_2^2\right)\right) \\
&\overset{③}{\leq} \exp\left(-\frac{snt}{2}\right)\prod_{i=1}^{n}\exp\left(128\boldsymbol{d}_l s^2\omega_f^4\tau^4\right) \quad |s| \leq \frac{1}{\omega_f^2\tau^2} \\
&\overset{④}{\leq} \exp\left(-c'n\min\left(\frac{t^2}{\boldsymbol{d}_l\omega_f^4\tau^4}, \frac{t}{\omega_f^2\tau^2}\right)\right).
\end{aligned}
$$

Note that ① holds because of Chebyshev's inequality. ② holds since $\boldsymbol{x}_{(i)}$ are independent. ③ is established by applying Lemma 13. We have ④ by optimizing $s$. Since $\boldsymbol{v}^{(l)}$ is sub-Gaussian, we have

$$
\begin{aligned}
\mathbb{P}\left(\frac{1}{n}\sum_{i=1}^{n}\left(\boldsymbol{y}^T\boldsymbol{v}_i^{(l)} - \mathbb{E}\boldsymbol{y}^T\boldsymbol{v}_i^{(l)}\right) > \frac{t}{2}\right) &\leq \mathbb{P}\left(s\sum_{i=1}^{n}\left(\boldsymbol{y}^T\boldsymbol{v}_i^{(l)} - \mathbb{E}\boldsymbol{y}^T\boldsymbol{v}_i^{(l)}\right) > \frac{nst}{2}\right) \\
&\leq \exp\left(-\frac{nst}{2}\right)\mathbb{E}\exp\left(s\sum_{i=1}^{n}\left(\boldsymbol{y}^T\boldsymbol{v}_i^{(l)} - \mathbb{E}\boldsymbol{y}^T\boldsymbol{v}_i^{(l)}\right)\right) \\
&\leq \exp\left(-\frac{nst}{2}\right)\prod_{i=1}^{n}\mathbb{E}\exp\left(s\left(\boldsymbol{y}^T\boldsymbol{v}_i^{(l)} - \mathbb{E}\boldsymbol{y}^T\boldsymbol{v}_i^{(l)}\right)\right) \\
&\overset{①}{\leq} \exp\left(-\frac{nst}{2}\right)\prod_{i=1}^{n}\exp\left(\frac{\omega_f^2\tau^2 s^2\|\boldsymbol{y}\|_2^2}{2}\right) \\
&\overset{②}{\leq} \exp\left(-\frac{nt^2}{8\omega_f^2\tau^2\|\boldsymbol{y}\|_2^2}\right),
\end{aligned}
$$

where ① holds because of Eqn. (6) and we have ② since we optimize $s$.

Since the loss function $f(\boldsymbol{w}, \boldsymbol{x})$ is defined as $f(\boldsymbol{w}, \boldsymbol{x}) = \|\boldsymbol{v}^{(l)} - \boldsymbol{y}\|_2^2$, we have

$$
f(\boldsymbol{w}, \boldsymbol{x}) - \mathbb{E}(f(\boldsymbol{w}, \boldsymbol{x})) = \|\boldsymbol{v}^{(l)} - \boldsymbol{y}\|_2^2 - \mathbb{E}(\|\boldsymbol{v}^{(l)} - \boldsymbol{y}\|_2^2) = \left(\|\boldsymbol{v}^{(l)}\|_2^2 - \mathbb{E}\|\boldsymbol{v}^{(l)}\|_2^2\right) + \left(\boldsymbol{y}^T\boldsymbol{v}^{(l)} - \mathbb{E}\boldsymbol{y}^T\boldsymbol{v}^{(l)}\right).
$$

Therefore, we have

$$
\begin{aligned}
&\mathbb{P}\left(\frac{1}{n}\sum_{i=1}^{n}\left(f(\boldsymbol{w}, \boldsymbol{x}_{(i)}) - \mathbb{E}(f(\boldsymbol{w}, \boldsymbol{x}_{(i)}))\right) > t\right) \\
&\leq \mathbb{P}\left(\frac{1}{n}\sum_{i=1}^{n}\left(\|\boldsymbol{v}_i^{(l)}\|_2^2 - \mathbb{E}\|\boldsymbol{v}_i^{(l)}\|_2^2\right) > \frac{t}{2}\right) + \mathbb{P}\left(\frac{1}{n}\sum_{i=1}^{n}\left(\boldsymbol{y}^T\boldsymbol{v}_i^{(l)} - \mathbb{E}\boldsymbol{y}^T\boldsymbol{v}_i^{(l)}\right) > \frac{t}{2}\right) \\
&\leq 2\exp\left(-c_{f'}n\min\left(\frac{t^2}{\boldsymbol{d}_l\omega_f^4\tau^4}, \frac{t^2}{\omega_f^2\tau^2}, \frac{t}{\omega_f^2\tau^2}\right)\right).
\end{aligned}
$$

where $c_{f'}$ is a constant. Note that $\|\boldsymbol{y}\|_2^2$ is the label of $\boldsymbol{x}$, then it can also be bounded. The proof is completed. $\qquad\square$

### C.2.3 PROOF OF LEMMA 8

*Proof.* For brevity, let $\boldsymbol{Q}_j$ denote $\nabla_{\boldsymbol{w}_{(j)}}f(\boldsymbol{w}, \boldsymbol{x})$. Then, by Lemma 6 we have

$$
\nabla_{\boldsymbol{w}_{(j)}}f(\boldsymbol{w}) = \left((\boldsymbol{B}_{j-1:1}\boldsymbol{x}) \otimes \boldsymbol{B}_{l:j+1}^T\right)\boldsymbol{e} \overset{①}{=} (\boldsymbol{B}_{j-1:1}\boldsymbol{x}) \otimes (\boldsymbol{B}_{l:j+1}^T\boldsymbol{e}) \overset{②}{=} \left(\boldsymbol{B}_{j-1:1} \otimes \boldsymbol{B}_{l:j+1}^T\right)(\boldsymbol{x} \otimes \boldsymbol{e}),
\tag{7}
$$

where ① holds since $\boldsymbol{B}_{j-1:1}\boldsymbol{x}$ is a vector, and ② holds because for any four matrices $\boldsymbol{Z}_1 \sim \boldsymbol{Z}_4$ of proper sizes, we have $(\boldsymbol{Z}_1\boldsymbol{Z}_3)\otimes(\boldsymbol{Z}_2\boldsymbol{Z}_4) = (\boldsymbol{Z}_1\otimes\boldsymbol{Z}_2)(\boldsymbol{Z}_3\otimes\boldsymbol{Z}_4)$. Note that $\boldsymbol{e} = \boldsymbol{v}^{(l)} - \boldsymbol{y} = \boldsymbol{B}_{l:1}\boldsymbol{x} - \boldsymbol{y}$. Then we know that the $i$-th entry $\boldsymbol{Q}_j^i$ has the form $\boldsymbol{Q}_j^i = \sum_{p,q} z_{pq}^{ij}\boldsymbol{x}_p\boldsymbol{x}_q + \sum_p y_p^{ij}\boldsymbol{x}_p + r^{ij}$ (Step 1 blow will give the detailed analysis) where $\boldsymbol{x}_p$ denotes the $p$-th entry in $\boldsymbol{x}$. Note that $z_{pq}^{ij}, y_p^{ij}$ and $r^{ij}$ are constants and independent on $\boldsymbol{x}$.

We divide $\boldsymbol{\lambda} \in \mathbb{R}^{\sum_{j=1}^l \boldsymbol{d}_j \boldsymbol{d}_{j-1}}$ into $\boldsymbol{\lambda} = (\boldsymbol{\lambda}_1; \cdots ; \boldsymbol{\lambda}_l)$ where $\boldsymbol{\lambda}_j \in \mathbb{R}^{\boldsymbol{d}_j \boldsymbol{d}_{j-1}}$. Let $\boldsymbol{\lambda}_j^i$ denote the $i$-th entry in $\boldsymbol{\lambda}_j$. Accordingly, we have

$$\boldsymbol{E} \triangleq \langle \boldsymbol{\lambda}, \nabla_{\boldsymbol{w}} f(\boldsymbol{w},\boldsymbol{x}) - \mathbb{E}\nabla_{\boldsymbol{w}} f(\boldsymbol{w},\boldsymbol{x}) \rangle = \sum_{j=1}^l \langle \boldsymbol{\lambda}_j, \boldsymbol{Q}_j - \mathbb{E}\boldsymbol{Q}_j \rangle = \boldsymbol{E}_1 + \boldsymbol{E}_2 + \boldsymbol{E}_3,$$

where $\boldsymbol{E}_1, \boldsymbol{E}_2$, and $\boldsymbol{E}_3$ are defined as

$$\boldsymbol{E}_1 = \sum_{p,q:p\neq q} \left( \sum_{j=1}^l \sum_{i=1}^{\boldsymbol{d}_j \boldsymbol{d}_{j-1}} \boldsymbol{\lambda}_j^i z_{pq}^{ij} \right) (\boldsymbol{x}_p\boldsymbol{x}_q - \mathbb{E}\boldsymbol{x}_p\boldsymbol{x}_q), \quad \boldsymbol{E}_2 = \sum_p \left( \sum_{j=1}^l \sum_{i=1}^{\boldsymbol{d}_j \boldsymbol{d}_{j-1}} \boldsymbol{\lambda}_j^i z_{pp}^{ij} \right) (\boldsymbol{x}_p^2 - \mathbb{E}\boldsymbol{x}_p^2),$$

$$\boldsymbol{E}_3 = \sum_p \left( \sum_{j=1}^l \sum_{i=1}^{\boldsymbol{d}_j \boldsymbol{d}_{j-1}} \boldsymbol{\lambda}_j^i y_p^{ij} \right) (\boldsymbol{x}_p - \mathbb{E}\boldsymbol{x}_p). \tag{8}$$

Thus, we can further separate the event as:

$$\mathbb{P}(\boldsymbol{E} > t) \leq \mathbb{P}\left( \frac{1}{n}\sum_{k=1}^n \boldsymbol{E}_1^k > \frac{t}{3} \right) + \mathbb{P}\left( \frac{1}{n}\sum_{k=1}^n \boldsymbol{E}_2^k > \frac{t}{3} \right) + \mathbb{P}\left( \frac{1}{n}\sum_{k=1}^n \boldsymbol{E}_3^k > \frac{t}{3} \right).$$

Thus, to prove our conclusion, we can respectively establish the upper bounds of the three events. To the end, for each input sample $\boldsymbol{x}_{(i)}$, we divide its corresponding $\boldsymbol{Q}_j - \mathbb{E}\boldsymbol{Q}_j$ into $\boldsymbol{E}_1, \boldsymbol{E}_2$ and $\boldsymbol{E}_3$. Then we bound the three events separately. Before that, we first give several equalities. Since $\boldsymbol{B}_{j:s} = \boldsymbol{W}^{(j)}\boldsymbol{W}^{(j-1)}\cdots\boldsymbol{W}^{(s)}$ ($j \geq s$), by Lemma 14 we have

$$\|\boldsymbol{B}_{j:s}\|_F^2 \leq r^{2(j-s+1)} \quad \text{and} \quad \|\boldsymbol{B}_{l:t+1}\|_F^2 \|\boldsymbol{B}_{t-1:s+1}\|_F^2 \|\boldsymbol{B}_{s-1:1}\|_F^2 \leq r^{2(l-2)}, \tag{9}$$

These two inequalities can be obtained by using $\|\boldsymbol{W}^{(i)}\|_F^2 = \|\boldsymbol{w}_{(i)}\|_2^2 \leq r^2$.

**Step 1. Divide $\boldsymbol{Q}_j - \mathbb{E}\boldsymbol{Q}_j$:** Note that $\boldsymbol{e} = \boldsymbol{v}^{(l)} - \boldsymbol{y} = \boldsymbol{B}_{l:1}\boldsymbol{x} - \boldsymbol{y}$. Let $\boldsymbol{H}_j = \boldsymbol{B}_{j-1:1} \otimes \boldsymbol{B}_{l:j+1}^T$. Then we can further write Eqn. (7) as

$$\boldsymbol{Q}_j = \nabla_{\boldsymbol{w}_{(j)}} f(\boldsymbol{w}) = \boldsymbol{H}_j (\boldsymbol{x} \otimes (\boldsymbol{B}_{l:1}\boldsymbol{x}) - \boldsymbol{x} \otimes \boldsymbol{y}) = \boldsymbol{H}_j ((\boldsymbol{I}_{\boldsymbol{d}_0} \otimes \boldsymbol{B}_{l:1})(\boldsymbol{x} \otimes \boldsymbol{x}) - \boldsymbol{x} \otimes \boldsymbol{y}), \tag{10}$$

where $\boldsymbol{I}_{\boldsymbol{d}_0} \in \mathbb{R}^{\boldsymbol{d}_0 \times \boldsymbol{d}_0}$ is the identity matrix. According to Eqn. (10), we can write the $i$-th entry of $\boldsymbol{Q}_j$ as the form $\boldsymbol{Q}_j^i = \sum_{p,q} z_{pq}^{ij}\boldsymbol{x}_p\boldsymbol{x}_q + \sum_p y_p^{ij}\boldsymbol{x}_p + r^{ij}$ where $\boldsymbol{x}_p$ denotes the $p$-th entry in $\boldsymbol{x}$. Let $\boldsymbol{Z}_j = \boldsymbol{H}_j (\boldsymbol{I}_{\boldsymbol{d}_0} \otimes \boldsymbol{B}_{l:1}) \in \mathbb{R}^{\boldsymbol{d}_j \boldsymbol{d}_{j-1} \times \boldsymbol{d}_0^2}$. Then, we know that the $i$-th entry $Q_j^i = \boldsymbol{Z}(i,:)\boldsymbol{x}'$, where $\boldsymbol{x}' = \boldsymbol{x} \otimes \boldsymbol{x} = [\boldsymbol{x}_1\boldsymbol{x}; \boldsymbol{x}_2\boldsymbol{x}; \cdots, \boldsymbol{x}_{\boldsymbol{d}_0}\boldsymbol{x}] \in \mathbb{R}^{\boldsymbol{d}_0^2}$. In this way, we have $z_{pq}^{ij} = \boldsymbol{Z}_j(i, (p-1)\boldsymbol{d}_0 + q)$ which further implies

$$\sum_{p,q}(z_{pq}^{ij})^2 \leq c_q \|\boldsymbol{Z}_j(i,:)\|_2^2, \tag{11}$$

where $c_q = \sqrt{\max_{0 \leq i \leq l} \boldsymbol{d}_i}$.

We divide the $i$-th row $\boldsymbol{H}_j(i,:)$ into $\boldsymbol{H}_j(i,:) = [\boldsymbol{H}_{ji}^1, \boldsymbol{H}_{ji}^2, \cdots, \boldsymbol{H}_{ji}^{\boldsymbol{d}_0}]$ where $\boldsymbol{H}_{ji}^p \in \mathbb{R}^{1 \times \boldsymbol{d}_l}$. Then we have $y_p^{ij} = \boldsymbol{y}^T \boldsymbol{H}_{ji}^p$. This yields

$$\sum_p (y_p^{ij})^2 \leq c_q \sum_p (\boldsymbol{y}^T \boldsymbol{H}_{ji}^p)^2 \leq c_q \sum_p \|\boldsymbol{y}\|_2^2 \|\boldsymbol{H}_{ji}^p\|_2^2 = c_q \|\boldsymbol{y}\|_2^2 \|\boldsymbol{H}_j(i,:)\|_2^2. \tag{12}$$

Let $\boldsymbol{\lambda}_j^i$ denote the $i$-th entry of $\boldsymbol{\lambda}_j$. Then, by Eqn. (8), we can obtain

$$\sum_j \langle \boldsymbol{\lambda}_j, (\boldsymbol{Q}_j - \mathbb{E}(\boldsymbol{Q}_j)) \rangle = \sum_{p,q:p\neq q} a_{pq}(\boldsymbol{x}_p\boldsymbol{x}_q - \mathbb{E}\boldsymbol{x}_p\boldsymbol{x}_q) + \sum_p a_{pp}(\boldsymbol{x}_p^2 - \mathbb{E}\boldsymbol{x}_p^2) + \sum_p b_p(\boldsymbol{x}_p - \mathbb{E}\boldsymbol{x}_p)$$

$$= \boldsymbol{E}_1 + \boldsymbol{E}_2 + \boldsymbol{E}_3,$$

where $a_{pq}$ and $b_p$ are defined as

$$a_{pq} = \sum_{j=1}^{l} \sum_{i=1}^{d_j d_{j-1}} \boldsymbol{\lambda}_j^i z_{pq}^{ij} \quad \text{and} \quad b_p = \sum_{j=1}^{l} \sum_{i=1}^{d_j d_{j-1}} \boldsymbol{\lambda}_j^i y_p^{ij}.$$

Note that for any four matrices of proper sizes, we have $(\boldsymbol{Q}_1 \otimes \boldsymbol{Q}_2)(\boldsymbol{Q}_3 \otimes \boldsymbol{Q}_4) = (\boldsymbol{Q}_1 \boldsymbol{Q}_3) \otimes (\boldsymbol{Q}_2 \boldsymbol{Q}_4)$, indicating $\boldsymbol{Z}_j = \left(\boldsymbol{B}_{j-1:1} \otimes \boldsymbol{B}_{l:j+1}^T\right)(\boldsymbol{I}_{d_0} \otimes \boldsymbol{B}_{l:1}) = \boldsymbol{B}_{j-1:1} \otimes \left(\boldsymbol{B}_{l:j+1}^T \boldsymbol{B}_{l:1}\right)$. This gives

$$c_q \|\boldsymbol{Z}_j\|_F^2 \leq c_q \|\boldsymbol{B}_{j-1:1}\|_F^2 \|\boldsymbol{B}_{l:j+1}\|_F^2 \|\boldsymbol{B}_{l:1}\|_F^2 \overset{\text{①}}{\leq} c_q r^{2(l-1)} r^{2l} = c_q r^{2(2l-1)} \triangleq \omega. \tag{13}$$

Note that Eqn. (13) uses the conclusion in Eqn. (9). Therefore, we can have the following bound:

$$\sum_{i=1}^{d_j d_{j-1}} (z_{pq}^{ij})^2 \leq \sum_{i=1}^{d_j d_{j-1}} \sum_{p,q} (z_{pq}^{ij})^2 \overset{\text{①}}{\leq} c_q \sum_{i=1}^{d_j d_{j-1}} \|\boldsymbol{Z}_j(i,:)\|_2^2 = c_q \|\boldsymbol{Z}_j\|_F^2 \leq \omega, \tag{14}$$

where ① uses Eqn. (11). Then we can utilize Eqn. (14) and $\sum_{j=1}^{l} \left(\sum_{i=1}^{d_j d_{j-1}} (\boldsymbol{\lambda}_j^i)^2\right) = 1$ to bound $a_{pq}$ as follows:

$$a_{pq}^2 \leq l \left(\sum_{j=1}^{l} \left(\sum_{i=1}^{d_j d_{j-1}} \boldsymbol{\lambda}_j^i z_{pq}^{ij}\right)^2\right) \leq l \sum_{j=1}^{l} \left(\sum_{i=1}^{d_j d_{j-1}} (\boldsymbol{\lambda}_j^i)^2\right) \left(\sum_{i=1}^{d_j d_{j-1}} (z_{pq}^{ij})^2\right) \leq l\omega.$$

which further gives

$$\sum_{p,q} a_{pq}^2 \leq l \sum_{j=1}^{l} \left(\sum_{i=1}^{d_j d_{j-1}} (\boldsymbol{\lambda}_j^i)^2\right) \left(\sum_{i=1}^{d_j d_{j-1}} \sum_{p,q} (z_{pq}^{ij})^2\right) \overset{\text{①}}{\leq} l\omega.$$

where ① uses Eqn. (14).

Similarly, we can obtain

$$\sum_{i=1}^{d_j d_{j-1}} (y_p^{ij})^2 \leq \sum_{i=1}^{d_j d_{j-1}} c_q \|\boldsymbol{y}\|_2^2 \|\boldsymbol{H}_j(i,:)\|_2^2 = c_q \|\boldsymbol{y}\|_2^2 \|\boldsymbol{H}_j\|_F^2 \leq c_q \|\boldsymbol{y}\|_2^2 r^{2(l-1)}. \tag{15}$$

So we can bound $b_p$ as

$$b_p^2 \leq l \sum_{j=1}^{l} \left(\sum_{i=1}^{d_j d_{j-1}} \boldsymbol{\lambda}_j^i y_p^{ij}\right)^2 \leq l \sum_{j=1}^{l} \left(\sum_{i=1}^{d_j d_{j-1}} (\boldsymbol{\lambda}_j^i)^2\right) \left(\sum_{i=1}^{d_j d_{j-1}} (y_p^{ij})^2\right) \leq l\omega',$$

where $\omega' = c_q \|\boldsymbol{y}\|_2^2 r^{2(l-1)}$. Accordingly, we can have

$$\sum_p b_p^2 \leq l \sum_{j=1}^{l} \left(\sum_{i=1}^{d_j d_{j-1}} (\boldsymbol{\lambda}_j^i)^2\right) \left(\sum_{i=1}^{d_j d_{j-1}} \sum_p (y_p^{ij})^2\right) \overset{\text{①}}{\leq} l\omega',$$

where ① uses (15).

**Step 2. Bound** $\mathbb{P}(\boldsymbol{E}_1 > t/3)$**,** $\mathbb{P}(\boldsymbol{E}_2 > t/3)$ **and** $\mathbb{P}(\boldsymbol{E}_3 > t/3)$**:** Let $\boldsymbol{E}_{h1}^k$ denotes the $\boldsymbol{E}_{h1}$ which corresponds to the $k$-th sample $\boldsymbol{x}_{(k)}$. Therefore, we can bound

$$
\begin{aligned}
\mathbb{P}\left(\frac{1}{n}\sum_{k=1}^{n}\boldsymbol{E}_1^k > \frac{t}{3}\right) =&\ \mathbb{P}\left(s\sum_{k=1}^{n}\left(\sum_{p,q:p\neq q} a_{pq}^k\left(\boldsymbol{x}_p^k\boldsymbol{x}_q^k - \mathbb{E}\boldsymbol{x}_p^k\boldsymbol{x}_q^k\right)\right) > \frac{snt}{3}\right) \\
\overset{①}{\leq}&\ \exp\left(-\frac{nst}{3}\right)\mathbb{E}\exp\left(s\sum_{k=1}^{n}\left(\sum_{p,q:p\neq q} a_{pq}^k\left(\boldsymbol{x}_p^k\boldsymbol{x}_q^k - \mathbb{E}\boldsymbol{x}_p^k\boldsymbol{x}_q^k\right)\right)\right) \\
\overset{②}{\leq}&\ \exp\left(-\frac{nst}{3}\right)\prod_{k=1}^{n}\mathbb{E}\exp\left(s\left(\sum_{p,q:p\neq q} a_{pq}^k\left(\boldsymbol{x}_p^k\boldsymbol{x}_q^k - \mathbb{E}\boldsymbol{x}_p^k\boldsymbol{x}_q^k\right)\right)\right) \\
\overset{③}{\leq}&\ \exp\left(-\frac{nst}{3}\right)\prod_{k=1}^{n}\exp\left(2\tau^2 s^2\sum_{p,q:p\neq q}(a_{pq}^k)^2\right) \quad |s|\leq\frac{1}{2\tau\sqrt{l\omega}} \\
\leq&\ \exp\left(-\frac{nst}{3}\right)\prod_{j=1}^{n}\exp\left(2\tau^2 s^2 l\omega\right) \\
\overset{④}{\leq}&\ \exp\left(-c'n\min\left(\frac{t^2}{\omega l\tau^2}, \frac{t}{\sqrt{l\omega}\tau}\right)\right),
\end{aligned}
$$

where ① holds because of Chebyshev's inequality. ② holds since $\boldsymbol{x}_{(i)}$ are independent. ③ is established by applying Lemma 12. We have ④ by optimizing $s$. Similarly, by Lemma 13 we can bound $\mathbb{P}\left(\frac{1}{n}\sum_{k=1}^{n}\boldsymbol{E}_2^k > \frac{t}{3}\right)$ as follows:

$$
\begin{aligned}
\mathbb{P}\left(\frac{1}{n}\sum_{k=1}^{n}\boldsymbol{E}_2^k > \frac{t}{3}\right) \leq&\ \exp\left(-\frac{nst}{3}\right)\prod_{k=1}^{n}\mathbb{E}\exp\left(s\left(\sum_p a_{pp}^k\left((\boldsymbol{x}_p^k)^2 - \mathbb{E}(\boldsymbol{x}_p^k)^2\right)\right)\right) \\
\leq&\ \exp\left(-\frac{nst}{3}\right)\prod_{k=1}^{n}\exp\left(128\tau^4 s^2 l\omega\right) \quad |s|\leq\frac{1}{\tau^2\sqrt{l\omega}} \\
\leq&\ \exp\left(-c''n\min\left(\frac{t^2}{\omega l\tau^4}, \frac{t}{\sqrt{l\omega}\tau^2}\right)\right).
\end{aligned}
$$

Finally, since $\boldsymbol{x}_{(i)}$ are independent sub-Gaussian, we can use Hoeffding inequality and obtain

$$
\mathbb{P}\left(\frac{1}{n}\sum_{k=1}^{n}\boldsymbol{E}_3^k > \frac{t}{3}\right) \leq \mathbb{P}\left(\frac{1}{n}\sum_{k=1}^{n}\left(\sum_p b_p^k\left(\boldsymbol{x}_p^k - \mathbb{E}\boldsymbol{x}_p^k\right)\right) > \frac{t}{3}\right)\exp\left(-\frac{c'''nt^2}{\omega'l\tau^2}\right).
$$

**Step 3. Bound** $\mathbb{P}(\boldsymbol{E} > t)$**:** By comparing the values of $\omega$ and $\omega'$, we can obtain

$$
\begin{aligned}
\mathbb{P}\left(\boldsymbol{E} > t\right) \leq&\ \mathbb{P}\left(\frac{1}{n}\sum_{k=1}^{n}\boldsymbol{E}_1^j > \frac{t}{3}\right) + \mathbb{P}\left(\frac{1}{n}\sum_{k=1}^{n}\boldsymbol{E}_2^j > \frac{t}{3}\right) + \mathbb{P}\left(\frac{1}{n}\sum_{k=1}^{n}\boldsymbol{E}_3^j > \frac{t}{3}\right) \\
\leq&\ 3\exp\left(-c_{g'}n\min\left(\frac{t^2}{l\max(\omega_g\tau^2, \omega_g\tau^4, \omega_{g'}\tau^2)}, \frac{t}{\sqrt{l\omega_g}\max(\tau, \tau^2)}\right)\right),
\end{aligned}
$$

where $\omega_g = c_q r^{2(2l-1)}$ and $\omega_{g'} = c_q r^{2(l-1)}$ in which $c_q = \sqrt{\max_{0\leq i\leq l}\boldsymbol{d}_i}$. The proof is completed. $\square$

### C.2.4 Proofs of Lemma 9

*Proof.* For brevity, let $\boldsymbol{Q}_{ts}$ denote $\nabla_{\boldsymbol{w}_{(t)}}\left(\nabla_{\boldsymbol{w}_{(s)}}f(\boldsymbol{w}, \boldsymbol{x})\right)$. Then, by Lemma 6 we have

$$
\boldsymbol{Q}_{ts} = \begin{cases} \left(\boldsymbol{B}_{l:s+1}^T\boldsymbol{e}\boldsymbol{x}^T\boldsymbol{B}_{t-1:1}^T\right)\otimes\boldsymbol{B}_{s-1:t+1} + \left(\boldsymbol{B}_{s-1:1}\boldsymbol{x}\boldsymbol{x}^T\boldsymbol{B}_{t-1:1}^T\right)\otimes\left(\boldsymbol{B}_{l:s+1}^T\boldsymbol{B}_{l:t+1}\right), & \text{if } s > t, \\ \left(\boldsymbol{B}_{s-1:1}\boldsymbol{x}\boldsymbol{x}^T\boldsymbol{B}_{s-1:1}\right)\otimes\left(\boldsymbol{B}_{l:s+1}^T\boldsymbol{B}_{l:s+1}\right), & \text{if } s = t, \\ \left(\boldsymbol{B}_{t-1:s+1}^T\right)\otimes\left(\boldsymbol{B}_{s-1:1}\boldsymbol{x}\boldsymbol{e}^T\boldsymbol{B}_{l:t+1}^T\right) + \left(\boldsymbol{B}_{s-1:1}\boldsymbol{x}\boldsymbol{x}^T\boldsymbol{B}_{t-1:1}^T\right)\otimes\left(\boldsymbol{B}_{l:s+1}^T\boldsymbol{B}_{l:t+1}\right), & \text{if } s < t. \end{cases}
$$

Then we know that the $(i,k)$-th entry $\boldsymbol{Q}_{ts}^{ik}$ has the form $\boldsymbol{Q}_{ts}^{ik} = \sum_{p,q} z_{pq}^{ik} \boldsymbol{x}_p \boldsymbol{x}_q + \sum_p y_p^{ik} \boldsymbol{x}_p + r^{ik}$ (explained in the following Step 1. I) where $\boldsymbol{x}_p$ denotes the $p$-th entry in $\boldsymbol{x}$. Note that $z_{pq}^{ik}, y_p^{ik}$ and $r^{ik}$ are constant and independent on $\boldsymbol{x}$. For convenience, we let $\boldsymbol{Q}_{ts} = \boldsymbol{H}_{ts} + \boldsymbol{G}_{ts}$, where $\boldsymbol{G}_{ts} = \left(\boldsymbol{B}_{s-1:1} \boldsymbol{x} \boldsymbol{x}^T \boldsymbol{B}_{t-1:1}^T\right) \otimes \left(\boldsymbol{B}_{l:s+1}^T \boldsymbol{B}_{l:t+1}\right)$ and $\boldsymbol{H}_{ts}$ is defined as

$$\boldsymbol{H}_{ts} = \begin{cases} \left(\boldsymbol{B}_{l:s+1}^T \boldsymbol{e} \boldsymbol{x}^T \boldsymbol{B}_{t-1:1}^T\right) \otimes \boldsymbol{B}_{s-1:t+1}, & \text{if } s > t, \\ \boldsymbol{0}, & \text{if } s = t, \\ \left(\boldsymbol{B}_{t-1:s+1}^T\right) \otimes \left(\boldsymbol{B}_{s-1:1} \boldsymbol{x} \boldsymbol{e}^T \boldsymbol{B}_{l:t+1}^T\right), & \text{if } s < t. \end{cases}$$

Let

$$\boldsymbol{E} = \frac{1}{n} \sum_{j=1}^n \left\langle \boldsymbol{\lambda}, \left(\nabla_{\boldsymbol{w}}^2 f(\boldsymbol{w}, \boldsymbol{x}) - \mathbb{E}\nabla_{\boldsymbol{w}}^2 f(\boldsymbol{w}, \boldsymbol{x})\right) \boldsymbol{\lambda}\right\rangle, \ \boldsymbol{E}_h = \frac{1}{n} \sum_{j=1}^n \sum_{t,s} \langle \boldsymbol{\lambda}_t, (\boldsymbol{H}_{ts} - \mathbb{E}(\boldsymbol{H}_{ts})) \boldsymbol{\lambda}_s \rangle,$$

$$\boldsymbol{E}_g = \frac{1}{n} \sum_{j=1}^n \sum_{t,s} \langle \boldsymbol{\lambda}_t, (\boldsymbol{G}_{ts} - \mathbb{E}(\boldsymbol{G}_{ts})) \boldsymbol{\lambda}_s \rangle.$$

Then we divide the event as two events:

$$\mathbb{P}\left(\boldsymbol{E} > t\right) = \mathbb{P}\left(\boldsymbol{E}_h + \boldsymbol{E}_g > t\right) \leq \mathbb{P}\left(\boldsymbol{E}_h > t/2\right) + \mathbb{P}\left(\boldsymbol{E}_g > t/2\right).$$

Now we look each event separately. Similar to $\boldsymbol{Q}_{ts}$, the $(i,k)$-th entry $\boldsymbol{H}_{ts}^{ik}$ has the form $\boldsymbol{H}_{ts}^{ik} = \sum_{p,q} z_{pq}^{ik} \boldsymbol{x}_p \boldsymbol{x}_q + \sum_p y_p^{ik} \boldsymbol{x}_p + r^{ik}$. We divide the unit vector $\boldsymbol{\lambda} \in \mathbb{R}^d$ as $\boldsymbol{\lambda} = (\boldsymbol{\lambda}_1; \cdots; \boldsymbol{\lambda}_l)$ where $\boldsymbol{\lambda}_j \in \mathbb{R}^{\boldsymbol{d}_j \boldsymbol{d}_{j-1}}$. For input vector $\boldsymbol{x}$, let $\sum_{t,s} \langle \boldsymbol{\lambda}_t, (\boldsymbol{H}_{ts} - \mathbb{E}(\boldsymbol{H}_{ts})) \boldsymbol{\lambda}_s \rangle = \boldsymbol{E}_{h1} + \boldsymbol{E}_{h2} + \boldsymbol{E}_{h3}$, where

$$\boldsymbol{E}_{h1} = \sum_{p,q:p \neq q} \left(\sum_{t,s} \sum_{i,k} (\boldsymbol{\lambda}_t^i \boldsymbol{\lambda}_s^k) z_{pq}^{ik}\right) (\boldsymbol{x}_p \boldsymbol{x}_q - \mathbb{E}\boldsymbol{x}_p \boldsymbol{x}_q), \quad \boldsymbol{E}_{h2} = \sum_p \left(\sum_{t,s} \sum_{i,k} (\boldsymbol{\lambda}_t^i \boldsymbol{\lambda}_s^k) z_{pq}^{ik}\right) (\boldsymbol{x}_p^2 - \mathbb{E}\boldsymbol{x}_p^2),$$

$$\boldsymbol{E}_{h3} = \sum_p \left(\sum_{t,s} \sum_{i,k} (\boldsymbol{\lambda}_t^i \boldsymbol{\lambda}_s^k) y_p^{ik}\right) (\boldsymbol{x}_p - \mathbb{E}\boldsymbol{x}_p), \tag{16}$$

where $\boldsymbol{x}_p$ denotes the $p$-th entry in $\boldsymbol{x}$ and $\boldsymbol{\lambda}_j^i$ denotes the $i$-th entry of $\boldsymbol{\lambda}_j$. Let $\boldsymbol{E}_{h_1}^j$, $\boldsymbol{E}_{h_2}^j$, and $\boldsymbol{E}_{h_3}^j$ denote the $\boldsymbol{E}_{h_1}$, $\boldsymbol{E}_{h_2}$, and $\boldsymbol{E}_{h_3}^j$ of the $j$-th sample. Thus, considering $n$ samples, we can further separately divide the two events above as:

$$\mathbb{P}\left(\boldsymbol{E}_h > \frac{t}{2}\right) \leq \mathbb{P}\left(\frac{1}{n} \sum_{j=1}^n \boldsymbol{E}_{h1}^j > \frac{t}{6}\right) + \mathbb{P}\left(\frac{1}{n} \sum_{j=1}^n \boldsymbol{E}_{h2}^j > \frac{t}{6}\right) + \mathbb{P}\left(\frac{1}{n} \sum_{j=1}^n \boldsymbol{E}_{h3}^j > \frac{t}{6}\right).$$

Similarly, we can define $\boldsymbol{E}_{g1}, \boldsymbol{E}_{g2}$ and $\boldsymbol{E}_{g3}$.

$$\mathbb{P}\left(\boldsymbol{E}_g > \frac{t}{2}\right) \leq \mathbb{P}\left(\frac{1}{n} \sum_{j=1}^n \boldsymbol{E}_{g1}^j > \frac{t}{6}\right) + \mathbb{P}\left(\frac{1}{n} \sum_{j=1}^n \boldsymbol{E}_{g2}^j > \frac{t}{6}\right) + \mathbb{P}\left(\frac{1}{n} \sum_{j=1}^n \boldsymbol{E}_{g3}^j > \frac{t}{6}\right).$$

Thus, to prove our conclusion, we can respectively establish the upper bounds of $\mathbb{P}\left(\boldsymbol{E}_h > \frac{t}{2}\right)$ and $\mathbb{P}\left(\boldsymbol{E}_g > \frac{t}{2}\right)$.

**Step 1: Bound $\mathbb{P}\left(\boldsymbol{E}_h > \frac{t}{2}\right)$**

To achieve our goal, for each input sample $\boldsymbol{x}_{(i)}$, we divide its corresponding $\sum_{t,s}(\boldsymbol{H}_{ts} - \mathbb{E}\boldsymbol{H}_{ts})$ as $\boldsymbol{E}_{h1}, \boldsymbol{E}_{h2}$ and $\boldsymbol{E}_{h3}$. Then we bound the three events separately. Before that, we first give two equalities. Since $\boldsymbol{B}_{j:s} = \boldsymbol{W}^{(j)} \boldsymbol{W}^{(j-1)} \cdots \boldsymbol{W}^{(s)}$ ($j \geq s$), by Lemma 14 we have

$$\|\boldsymbol{B}_{j:s}\|_F^2 \leq r^{2(j-s+1)} \ \text{ and } \ \|\boldsymbol{B}_{l:t+1}\|_F^2 \|\boldsymbol{B}_{t-1:s+1}\|_F^2 \|\boldsymbol{B}_{s-1:1}\|_F^2 \leq r^{2(l-2)}, \tag{17}$$

These two inequalities can be obtained by using $\|\boldsymbol{W}^{(i)}\|_F^2 = \|\boldsymbol{w}_{(i)}\|_2^2 \leq r^2$.

**I. Divide $H_{ts} - \mathbb{E}H_{ts}$:** For $t \neq s$, we can write the $(i,k)$-th entry $H_{ts}^{ik}$ as the form $H_{ts}^{ik} = \sum_{p,q} z_{pq}^{ik} x_p x_q + \sum_p y_p^{ik} x_p + r^{ik}$. Now we try to bound $z_{pq}^{ik}$ and $y_p^{ik}$. We first consider the case $s < t$. Note that $e = v^{(l)} - y = B_{l:1}x - y$. Specifically, we have

$$H_{ts} = \left(B_{t-1:s+1}^T\right) \otimes \left(B_{s-1:1}xx^T B_{l:1}^T B_{l:t+1}^T - B_{s-1:1}xy^T B_{l:t+1}^T\right).$$

So the $(i',k')$-th entry in the matrix $B_{s-1:1}xx^T B_{l:1}^T B_{l:t+1}^T$ is $[B_{s-1:1}xx^T B_{l:1}^T B_{l:t+1}^T]_{i'k'} = (B_{s-1:1})(i',:)x(B_{l:1}B_{l:t+1})(k',:)x = x^T((B_{s-1:1})(i',:))^T(B_{l:1}B_{l:t+1})(k',:)x$, where $A(i',:)$ denotes the $i'$-th row of $A$. Let $i_i' = \mathrm{mod}(i, d_s)$, $k_k' = \mathrm{mod}(k, d_{t-1})$, $i_i'' = \lfloor i/d_s \rfloor$ and $k_k'' = \lfloor k/d_{t-1} \rfloor$. In this case, the $(i,k)$-th entry $H_{ts}^{ik} = [B_{t-1:s+1}]_{k_k''i_i''}x^T((B_{s-1:1})(i_i',:))^T(B_{l:1}B_{l:t+1})(k_k',:)x + [B_{t-1:s+1}]_{k_k''i_i''}y^T(B_{l:t+1})(k_k',:)^T(B_{s-1:1})(i_i',:)x$. Therefore, we have

$$\sum_{p,q}(z_{pq}^{ik})^2 = [B_{t-1:s+1}]_{k_k''i_i''}^2 \left\|((B_{s-1:1})(i_i',:))^T(B_{l:1}B_{l:t+1})(k_k',:)\right\|_F^2$$

$$\leq [B_{t-1:s+1}]_{k_k''i_i''}^2 \|(B_{s-1:1})(i_i',:)\|_2^2 \|(B_{l:1}B_{l:t+1})(k_k',:)\|_2^2.$$

Therefore, we can further establish

$$\sum_{i,k}\sum_{p,q}(z_{pq}^{ik})^2 \leq \sum_{i,k}[B_{t-1:s+1}]_{k_k''i_i''}^2 \|(B_{s-1:1})(i_i',:)\|_2^2 \|(B_{l:1}B_{l:t+1})(k_k',:)\|_2^2$$

$$\leq \sum_{i,k}[B_{t-1:s+1}]_{k_k''i_i''}^2 \|(B_{s-1:1})(i_i',:)\|_2^2 \|(B_{l:1}B_{l:t+1})(k_k',:)\|_2^2$$

$$= \sum_k \|(B_{t-1:s+1})(k_k'',:)\|_2^2 \|B_{s-1:1}\|_F^2 \|(B_{l:1}B_{l:t+1})(k_k',:)\|_2^2 \tag{18}$$

$$= \|B_{t-1:s+1}\|_F^2 \|B_{s-1:1}\|_F^2 \|B_{l:1}B_{l:t+1}\|_F^2$$

$$\overset{①}{\leq} r^{4(l-1)} \triangleq \omega.$$

where ① uses Eqn. (17). Similarly, we can bound

$$\sum_p(y_p^{ik})^2 = [B_{t-1:s+1}]_{k_k''i_i''}^2 \left\|y^T(B_{l:t+1})(k_k',:)^T(B_{s-1:1})(i_i',:)\right\|_F^2$$

$$\leq [B_{t-1:s+1}]_{k_k''i_i''}^2 \|y\|_2^2 \|(B_{l:t+1})(k_k',:)\|_2^2 \|(B_{s-1:1})(i_i',:)\|_2^2.$$

So it further yields

$$\sum_{i,k}\sum_p(y_p^{ik})^2 \leq \sum_{i,k}[B_{t-1:s+1}]_{k_k''i_i''}^2 \|y\|_2^2 \|(B_{l:t+1})(k_k',:)\|_2^2 \|(B_{s-1:1})(i_i',:)\|_2^2$$

$$\leq \|y\|_2^2 \|B_{t-1:s+1}\|_F^2 \|B_{l:t+1}\|_F^2 \|B_{s-1:1}\|_F^2 \overset{①}{\leq} \|y\|_2^2 r^{2(l-2)} \triangleq \omega', \tag{19}$$

where ① uses Eqn. (17). Note that for the case $s \geq t$, Eqn. (18) and (19) also holds. Let $\lambda_j^i$ denote the $i$-th entry of $\lambda_j$. Then, by Eqn. (16), we can obtain

$$\sum_{t,s}(\langle \lambda_t, (H_{ts} - \mathbb{E}(H_{ts}))\lambda_s\rangle) = \sum_{p,q:p\neq q} a_{pq}(x_p x_q - \mathbb{E}x_p x_q) + \sum_p a_{pp}(x_p^2 - \mathbb{E}x_p^2) + \sum_p b_p(x_p - \mathbb{E}x_p)$$

$$= E_{h1} + E_{h2} + E_{h3},$$

where $a_{pq}$ and $b_p$ are defined as

$$a_{pq} = \sum_{t,s}\sum_{i,k}(\lambda_t^i \lambda_s^k)z_{pq}^{ik} \quad \text{and} \quad b_p = \sum_{t,s}\sum_{i,k}(\lambda_t^i \lambda_s^k)y_p^{ik}.$$

Then according to Eqn. (18) and $\sum_{t,s}\left(\sum_{i,k}(\lambda_t^i \lambda_s^k)^2\right) = 1$, we can bound $a_{pq}$ as follows:

$$a_{pq}^2 \leq l^2 \sum_{t,s}\left(\sum_{i,k}(\lambda_t^i \lambda_s^k)z_{pq}^{ik}\right)^2 \leq l^2 \sum_{t,s}\left(\sum_{i,k}(\lambda_t^i \lambda_s^k)^2\right)\left(\sum_{i,k}(z_{pq}^{ik})^2\right) \leq \omega l^2 \sum_{t,s}\left(\sum_{i,k}(\lambda_t^i \lambda_s^k)^2\right) \leq \omega l^2.$$

which further yields

$$\sum_{p,q} a_{pq}^2 \le l^2 \sum_{t,s} \left( \sum_{i,k} (\boldsymbol{\lambda}_t^i \boldsymbol{\lambda}_s^k)^2 \right) \left( \sum_{i,k} \sum_{p,q} (z_{pq}^{ik})^2 \right) \le \omega l^2 \sum_{t,s} \left( \sum_{i,k} (\boldsymbol{\lambda}_t^i \boldsymbol{\lambda}_s^k)^2 \right) \le \omega l^2.$$

Similarly, by using Eqn. (19), we have

$$b_p^2 \le l^2 \sum_{t,s} \left( \sum_{i,k} (\boldsymbol{\lambda}_t^i \boldsymbol{\lambda}_s^k) y_p^{ik} \right)^2 \le l^2 \sum_{t,s} \left( \sum_{i,k} (\boldsymbol{\lambda}_t^i \boldsymbol{\lambda}_s^k)^2 \right) \left( \sum_{i,k} (y_p^{ik})^2 \right) \overset{①}{\le} \omega' l^2.$$

Accordingly, we can have

$$\sum_p b_p^2 \le l^2 \sum_{t,s} \left( \sum_{i,k} (\boldsymbol{\lambda}_t^i \boldsymbol{\lambda}_s^k)^2 \right) \left( \sum_{i,k} \sum_p (y_p^{ik})^2 \right) \le \omega' l^2.$$

**II. Bound $\mathbb{P}(\boldsymbol{E}_{h1} > t/6)$, $\mathbb{P}(\boldsymbol{E}_{h2} > t/6)$ and $\mathbb{P}(\boldsymbol{E}_{h3} > t/6)$:** Let $E_{h1}^j$ denotes the $\boldsymbol{E}_{h1}^j$ which corresponds to the $j$-th sample $\boldsymbol{x}_{(i)}$. Therefore, we can bound

$$
\begin{aligned}
\mathbb{P}\left( \frac{1}{n} \sum_{j=1}^n \boldsymbol{E}_{h1}^j > \frac{t}{6} \right) &\le \mathbb{P}\left( s \sum_{j=1}^n \left( \sum_{p,q:p\ne q} a_{pq}^j \left( \boldsymbol{x}_p^j \boldsymbol{x}_q^j - \mathbb{E}\boldsymbol{x}_p^j \boldsymbol{x}_q^j \right) \right) > \frac{snt}{6} \right) \\
&\overset{①}{\le} \exp\left( -\frac{nst}{6} \right) \mathbb{E} \exp\left( s \sum_{j=1}^n \left( \sum_{p,q:p\ne q} a_{pq}^j \left( \boldsymbol{x}_p^j \boldsymbol{x}_q^j - \mathbb{E}\boldsymbol{x}_p^j \boldsymbol{x}_q^j \right) \right) \right) \\
&\overset{②}{\le} \exp\left( -\frac{nst}{6} \right) \prod_{j=1}^n \mathbb{E} \exp\left( s \left( \sum_{p,q:p\ne q} a_{pq}^j \left( \boldsymbol{x}_p^j \boldsymbol{x}_q^j - \mathbb{E}\boldsymbol{x}_p^j \boldsymbol{x}_q^j \right) \right) \right) \\
&\overset{③}{\le} \exp\left( -\frac{nst}{6} \right) \prod_{j=1}^n \exp\left( 2\tau^2 s^2 \sum_{p,q:p\ne q} (a_{pq}^j)^2 \right) \quad |s| \le \frac{1}{2\tau l\sqrt{\omega}} \\
&\le \exp\left( -\frac{nst}{6} \right) \prod_{j=1}^n \exp\left( 2\tau^2 s^2 l^2 \omega \right) \\
&\overset{④}{\le} \exp\left( -c'n \min\left( \frac{t^2}{\omega l^2 \tau^2}, \frac{t}{\sqrt{\omega} l \tau} \right) \right),
\end{aligned}
$$

where ① holds because of Chebyshev's inequality. ② holds since $\boldsymbol{x}_{(i)}$ are independent. ③ is established because of Lemma 12. We have ④ by optimizing $s$. Similarly, we can bound $\mathbb{P}\left( \frac{1}{n} \sum_{j=1}^n \boldsymbol{E}_{h2}^j > \frac{t}{6} \right)$ as follows:

$$
\begin{aligned}
\mathbb{P}\left( \frac{1}{n} \sum_{j=1}^n \boldsymbol{E}_{h2}^j > \frac{t}{6} \right) &\le \exp\left( -\frac{nst}{6} \right) \prod_{j=1}^n \mathbb{E} \exp\left( s \left( \sum_p a_{pp}^j \left( (\boldsymbol{x}_p^j)^2 - \mathbb{E}(\boldsymbol{x}_p^j)^2 \right) \right) \right) \\
&\le \exp\left( -\frac{nst}{6} \right) \prod_{j=1}^n \exp\left( 128\tau^4 s^2 l^2 \omega \right) \quad |s| \le \frac{1}{\tau^2 l\sqrt{\omega}} \\
&\le \exp\left( -c''n \min\left( \frac{t^2}{\omega l^2 \tau^4}, \frac{t}{\sqrt{\omega} l \tau^2} \right) \right).
\end{aligned}
$$

Finally, since $\boldsymbol{x}_{(i)}$ are independent sub-Gaussian, we can use Hoeffding inequality and obtain

$$\mathbb{P}\left( \frac{1}{n} \sum_{j=1}^n \boldsymbol{E}_{h3}^j > \frac{t}{6} \right) = \mathbb{P}\left( \frac{1}{n} \sum_{j=1}^n \left( \sum_p b_p^j \left( \boldsymbol{x}_p^j - \mathbb{E}\boldsymbol{x}_p^j \right) \right) > \frac{t}{6} \right) \le \exp\left( -\frac{c'''nt^2}{\omega' l^2 \tau^2} \right).$$

Since for $s = t$, $\mathbb{P}\left(\frac{1}{n}\sum_{j=1}^{n}\boldsymbol{E}_{h1}^{j} > \frac{t}{6}\right) = \mathbb{P}\left(\frac{1}{n}\sum_{j=1}^{n}\boldsymbol{E}_{h2}^{j} > \frac{t}{6}\right) = \mathbb{P}\left(\frac{1}{n}\sum_{j=1}^{n}\boldsymbol{E}_{h3}^{j} > \frac{t}{6}\right) = 0$, the above upper bounds also hold.

**III: Bound $\mathbb{P}\left(\boldsymbol{E}_h > \frac{t}{2}\right)$** By comparing the values of $\omega$ and $\omega'$, we can obtain

$$\mathbb{P}\left(\boldsymbol{E}_h > \frac{t}{2}\right) \leq \mathbb{P}\left(\frac{1}{n}\sum_{j=1}^{n}\boldsymbol{E}_{h1}^{j} > \frac{t}{6}\right) + \mathbb{P}\left(\frac{1}{n}\sum_{j=1}^{n}\boldsymbol{E}_{h2}^{j} > \frac{t}{6}\right) + \mathbb{P}\left(\frac{1}{n}\sum_{j=1}^{n}\boldsymbol{E}_{h3}^{j} > \frac{t}{6}\right)$$
$$\leq 3\exp\left(-c_2'n\min\left(\frac{t^2}{l^2\max\left(\omega\tau^2, \omega\tau^4, \omega_q\tau^2\right)}, \frac{t}{\sqrt{\omega}l\max\left(\tau, \tau^2\right)}\right)\right),$$

where $\omega_q = r^{2(l-2)}$.

**Step 2: Bound $\mathbb{P}\left(\boldsymbol{E}_g > \frac{t}{2}\right)$** To achieve our goal, for each input sample $\boldsymbol{x}_{(i)}$, we also divide its corresponding $\sum_{t,s}(\boldsymbol{G}_{ts} - \mathbb{E}\boldsymbol{G}_{ts})$ as $\boldsymbol{E}_{h1}$, $\boldsymbol{E}_{h2}$ and $\boldsymbol{E}_{h3}$. Then we bound the three events separately. Before that, we first give several equalities.

**I. Divide $\boldsymbol{G}_{ts} - \mathbb{E}\boldsymbol{G}_{ts}$:** Dividing $\boldsymbol{G}_{ts} - \mathbb{E}\boldsymbol{G}_{ts}$ is more easy than dividing $\boldsymbol{H}_{ts} - \mathbb{E}\boldsymbol{H}_{ts}$ since the later has more complex form. Since $\boldsymbol{G}_{ts} = \left(\boldsymbol{B}_{s-1:1}\boldsymbol{x}\boldsymbol{x}^T\boldsymbol{B}_{t-1:1}^T\right) \otimes \left(\boldsymbol{B}_{l:s+1}^T\boldsymbol{B}_{l:t+1}\right)$. we also can write the $(i,k)$-th entry $\boldsymbol{G}_{ts}^{ik}$ as the form $\boldsymbol{G}_{ts}^{ik} = \sum_{p,q}z_{pq}^{ik}\boldsymbol{x}_p\boldsymbol{x}_q + \sum_p y_p^{ik}\boldsymbol{x}_p + r^{ik}$. But here $y_p^{ik} = 0$.

Then similar to the step in dividing $\boldsymbol{H}_{ts} - \mathbb{E}\boldsymbol{H}_{ts}$, we can bound

$$a_{pq}^2 \leq \omega_g l^2 \quad \text{and} \quad \sum_{p,q}a_{pq}^2 \leq \omega_g l^2 \quad \text{where } \omega_g = r^{4(l-1)}.$$

**II. Bound $\mathbb{P}(\boldsymbol{E}_{g1} > t/6)$, $\mathbb{P}(\boldsymbol{E}_{g2} > t/6)$ and $\mathbb{P}(\boldsymbol{E}_{g3} > t/6)$:** Since $y_p^{ik} = 0$, $\mathbb{P}(\boldsymbol{E}_{h3} > t/6) = 0$. Similar to the above methods, we can bound

$$\mathbb{P}\left(\frac{1}{n}\sum_{j=1}^{n}\boldsymbol{E}_{g1}^{j} > \frac{t}{6}\right) \leq \exp\left(-c_1'n\left(\frac{t^2}{\omega_g l^2\tau^2}, \frac{t}{\sqrt{\omega_g}l\tau}\right)\right),$$

and

$$\mathbb{P}\left(\frac{1}{n}\sum_{j=1}^{n}\boldsymbol{E}_{g2}^{j} > \frac{t}{6}\right) \leq \exp\left(-c_1''n\left(\frac{t^2}{\omega_g l^2\tau^4}, \frac{t}{\sqrt{\omega_g}l\tau^2}\right)\right).$$

**III: Bound $\mathbb{P}\left(\boldsymbol{E}_h > \frac{t}{2}\right)$** We can obtain $\mathbb{P}\left(\boldsymbol{E}_g > \frac{t}{2}\right)$ as follows:

$$\mathbb{P}\left(\boldsymbol{E}_g > \frac{t}{2}\right) \leq \mathbb{P}\left(\frac{1}{n}\sum_{j=1}^{n}\boldsymbol{E}_{g1}^{j} > \frac{t}{6}\right) + \mathbb{P}\left(\frac{1}{n}\sum_{j=1}^{n}\boldsymbol{E}_{g2}^{j} > \frac{t}{6}\right) + \mathbb{P}\left(\frac{1}{n}\sum_{j=1}^{n}\boldsymbol{E}_{g3}^{j} > \frac{t}{6}\right)$$
$$\leq 2\exp\left(-c_2'n\min\left(\frac{t^2}{\omega_g l^2\max\left(\tau^2, \tau^4\right)}, \frac{t}{\sqrt{\omega_g}l\max\left(\tau, \tau^2\right)}\right)\right).$$

**Step 3: Bound $\mathbb{P}(\boldsymbol{E} > t)$** Finally, we combine the above results and obtain

$$\mathbb{P}\left(\boldsymbol{E} > t\right) \leq \mathbb{P}\left(\boldsymbol{E}_h > \frac{t}{2}\right) + \mathbb{P}\left(\boldsymbol{E}_g > \frac{t}{2}\right)$$
$$\leq 5\exp\left(-c_{h'}n\min\left(\frac{t^2}{\tau^2 l^2\max\left(\omega_g, \omega_g\tau^2, \omega_h\right)}, \frac{t}{\sqrt{\omega_g}l\max\left(\tau, \tau^2\right)}\right)\right),$$

where $\omega_g = r^{4(l-1)}$ and $\omega_h = r^{2(l-2)}$. $\qquad\square$

### C.2.5 PROOF OF LEMMA 10

*Proof.* Before proving our conclusion, we first give an inequality:

$$\|\boldsymbol{e}\|_2^2 = \|\boldsymbol{B}_{l:1}\boldsymbol{x} - \boldsymbol{y}\|_2^2 \leq \|\boldsymbol{B}_{l:1}\boldsymbol{x}\|_2^2 + 2\left|\boldsymbol{y}^T\boldsymbol{B}_{l:1}\boldsymbol{x}\right| + \|\boldsymbol{y}\|_2^2 \overset{\text{①}}{\leq} r_x^2\omega_f^2 + 2r_x\omega_f\|\boldsymbol{y}\|_2 + \|\boldsymbol{y}\|_2^2,$$

where $\omega_f = r^l$. Notice, ① holds since by Lemma 14, we have $\|\boldsymbol{B}_{l:1}\|_F^2 \le r^{2l}$.

Then we consider $\nabla_{\boldsymbol{w}} f(\boldsymbol{w}, \boldsymbol{x})$. Firstly, by Lemma 6 we can bound $\|\nabla_{\boldsymbol{w}_{(j)}} f(\boldsymbol{w}, \boldsymbol{x})\|_2^2$ as follows:

$$\|\nabla_{\boldsymbol{w}_{(j)}} f(\boldsymbol{w}, \boldsymbol{x})\|_2^2 = \left\|\left((\boldsymbol{B}_{j-1:1}\boldsymbol{x}) \otimes \boldsymbol{B}_{l:j+1}^T\right) \boldsymbol{e}\right\|_2^2 \le \|\boldsymbol{B}_{j-1:1}\|_2^2 \|\boldsymbol{x}\|_2^2 \|\boldsymbol{B}_{l:j+1}\|_2^2 \|\boldsymbol{e}\|_2^2$$
$$\overset{①}{\le} r_x^2 \omega_{f_1}^2 \left(r_x^2 \omega_f^2 + 2 r_x \omega_f \|\boldsymbol{y}\|_2 + \|\boldsymbol{y}\|_2^2\right),$$

where $\omega_{f_1} = r^{(l-1)}$. ① holds since we have $\|\boldsymbol{B}_{l:j+1}\|_F^2 \|\boldsymbol{B}_{j-1:1}\|_F^2 \le r^{2(l-1)}$ by using $\|\boldsymbol{W}^{(i)}\|_F^2 = \|\boldsymbol{w}_{(i)}\|_2^2 \le r^2$. Therefore, we can further obtain

$$\|\nabla_{\boldsymbol{w}} f(\boldsymbol{w}, \boldsymbol{x})\|_2^2 = \sum_{i=1}^{l} \|\nabla_{\boldsymbol{w}_{(i)}} f(\boldsymbol{w}, \boldsymbol{x})\|_2^2 \le l r_x^2 \omega_{f_1}^2 \left(r_x^2 \omega_f^2 + 2 r_x \omega_f \|\boldsymbol{y}\|_2 + \|\boldsymbol{y}\|_2^2\right).$$

Notice, $\boldsymbol{y}$ is the label of sample and the weight magnitude $r$ is usually lager than 1. Then we have $\|\boldsymbol{y}\|_2 \le r^l$. Also, the values in input data are usually smaller than $r^l$. Thus, we have

$$\|\nabla_{\boldsymbol{w}} f(\boldsymbol{w}, \boldsymbol{x})\|_2^2 \le c_t l r_x^4 r^{4l-2} \triangleq \alpha_g,$$

where $c_t$ is a constant. Then we use the inequality $\left\|\nabla^2 f(\boldsymbol{w}, \boldsymbol{x})\right\|_{\text{op}} \le \left\|\nabla^2 f(\boldsymbol{w}, \boldsymbol{x})\right\|_F$ to bound $\left\|\nabla^2 f(\boldsymbol{w}, \boldsymbol{x})\right\|_{\text{op}}$. Next we only need to give the upper bound of $\left\|\nabla^2 f(\boldsymbol{w}, \boldsymbol{x})\right\|_F$. Let $\omega_{f_2} = r^{l-2}$. We first consider $\boldsymbol{Q}_{st} \triangleq \nabla_{\boldsymbol{w}_{(s)}}\left(\nabla_{\boldsymbol{w}_{(t)}} f(\boldsymbol{w}, \boldsymbol{x})\right)$. By Lemma 6, if $s < t$, we have

$$\|\boldsymbol{Q}_{st}\|_F^2 = \left\|\left(\boldsymbol{B}_{t-1:s+1}^T\right) \otimes \left(\boldsymbol{B}_{s-1:1}\boldsymbol{x}\boldsymbol{e}^T \boldsymbol{B}_{l:t+1}^T\right) + \left(\boldsymbol{B}_{s-1:1}\boldsymbol{x}\boldsymbol{x}^T \boldsymbol{B}_{t-1:1}^T\right) \otimes \left(\boldsymbol{B}_{l:s+1}^T \boldsymbol{B}_{l:t+1}\right)\right\|_F^2$$
$$\le 2 \left(\left\|\left(\boldsymbol{B}_{t-1:s+1}^T\right) \otimes \left(\boldsymbol{B}_{s-1:1}\boldsymbol{x}\boldsymbol{e}^T \boldsymbol{B}_{l:t+1}^T\right)\right\|_F^2 + \left\|\left(\boldsymbol{B}_{s-1:1}\boldsymbol{x}\boldsymbol{x}^T \boldsymbol{B}_{t-1:1}^T\right) \otimes \left(\boldsymbol{B}_{l:s+1}^T \boldsymbol{B}_{l:t+1}\right)\right\|_F^2\right)$$
$$\le 2 \|\boldsymbol{B}_{t-1:s+1}\|_F^2 \|\boldsymbol{B}_{s-1:1}\|_F^2 \|\boldsymbol{x}\|_2^2 \|\boldsymbol{e}\|_2^2 \|\boldsymbol{B}_{l:t+1}\|_F^2$$
$$+ 2 \|\boldsymbol{B}_{s-1:1}\|_F^2 \|\boldsymbol{x}\|_2^2 \|\boldsymbol{x}\|_2^2 \|\boldsymbol{B}_{t-1:1}\|_F^2 \|\boldsymbol{B}_{l:s+1}\|_F^2 \|\boldsymbol{B}_{l:t+1}\|_F^2$$
$$\overset{①}{\le} 2 \omega_{f_2}^2 r_x^2 \left(r_x^2 \omega_f^2 + r_x \omega_f \|\boldsymbol{y}\|_2 + \|\boldsymbol{y}\|_2^2\right) + 2 \omega_{f_1}^4 r_x^4,$$

where ① holds since we use $\|\boldsymbol{B}_{l:t+1}\|_F^2 \|\boldsymbol{B}_{t-1:s+1}\|_F^2 \|\boldsymbol{B}_{s-1:1}\|_F^2 \le \omega_{f_2}^2$ and $\|\boldsymbol{B}_{s-1:1}\|_F^2 \|\boldsymbol{B}_{l:s+1}\|_F^2 \le \omega_{f_1}^2$. Note that when $s \ge t$, the above inequality also holds. Similarly, consider the values in input data and the values in label, we have

$$\|\boldsymbol{Q}_{st}\|_F^2 \le c_{t'} r_x^4 r^{4l-2} \triangleq \alpha_l,$$

where $c_{t'}$ is a constant. Therefore, we can bound

$$\left\|\nabla^2 f(\boldsymbol{w}, \boldsymbol{x})\right\|_{\text{op}} \le \left\|\nabla^2 f(\boldsymbol{w}, \boldsymbol{x})\right\|_F \le \sqrt{\sum_{s=1}^{l} \sum_{t=1}^{l} \|\boldsymbol{Q}_{st}\|_F^2} \le l \sqrt{\alpha_l}.$$

On the other hand, if the activation functions are linear functions, $f(\boldsymbol{w}, \boldsymbol{x})$ is fourth order differentiable when $l \ge 2$. This means that $\nabla_{\boldsymbol{x}} \nabla_{\boldsymbol{w}}^3 f(\boldsymbol{w}, \boldsymbol{x})$ exists. Also since for any input $\boldsymbol{x} \in \mathsf{B}^{d_0}(r_x)$ and $\boldsymbol{w} \in \Omega$, we can always find a universal constant $\alpha_p$ such that

$$\|\nabla_{\boldsymbol{w}}^3 f(\boldsymbol{w}, \boldsymbol{x})\|_{\text{op}} = \sup_{\|\boldsymbol{\lambda}\|_2 \le 1} \left\langle \boldsymbol{\lambda}^{\otimes^3}, \nabla_{\boldsymbol{w}}^3 f(\boldsymbol{w}, \boldsymbol{x})\right\rangle = \sum_{i,j,k} [\nabla_{\boldsymbol{w}}^3 f(\boldsymbol{w}, \boldsymbol{x})]_{ijk} \boldsymbol{\lambda}_i \boldsymbol{\lambda}_j \boldsymbol{\lambda}_k \le \alpha_p < +\infty.$$

We complete the proofs. $\qquad\qquad\qquad\qquad\qquad\qquad\qquad\qquad\qquad\qquad\qquad\qquad\qquad\qquad$ □

### C.2.6 PROOF OF LEMMA 11

*Proof.* Recall that the weight of each layer has magnitude bound separately, *i.e.* $\|\boldsymbol{w}_{(j)}\|_2 \le r$. Assume that $\boldsymbol{w}_{(j)}$ has $\boldsymbol{s}_j$ non-zero entries. Then we have $\sum_{j=1}^{l} \boldsymbol{s}_j = s$. So here we separately assume $\boldsymbol{w}_\epsilon^j = \{\boldsymbol{w}_1^j, \cdots, \boldsymbol{w}_{n_\epsilon^j}^j\}$ is the $\boldsymbol{d}_j \boldsymbol{d}_{j-1} \epsilon / d$-covering net of the ball $\mathsf{B}^{\boldsymbol{d}_j \boldsymbol{d}_{j-1}}(r)$ which corresponds

to the weight $\boldsymbol{w}_{(j)}$ of the $j$-th layer. Let $n_\epsilon{}^j$ be the $\epsilon/l$-covering number. By $\epsilon$-covering theory in (Vershynin, 2012), we can have

$$
n_\epsilon{}^j \leq \left( \begin{array}{c} \boldsymbol{d}_j \boldsymbol{d}_{j-1} \\ \boldsymbol{s}_j \end{array} \right) \left( \frac{3r}{\boldsymbol{d}_j \boldsymbol{d}_{j-1}\epsilon/d} \right)^{\boldsymbol{s}_j} \leq \exp\left( \boldsymbol{s}_j \log\left( \frac{3r\boldsymbol{d}_j\boldsymbol{d}_{j-1}}{\boldsymbol{d}_j\boldsymbol{d}_{j-1}\epsilon/d} \right) \right) = \exp\left( \boldsymbol{s}_j \log\left( \frac{3rd}{\epsilon} \right) \right).
$$

Let $\boldsymbol{w} \in \Omega$ be an arbitrary vector. Since $\boldsymbol{w} = [\boldsymbol{w}_{(1)}, \cdots, \boldsymbol{w}_{(l)}]$ where $\boldsymbol{w}_{(j)}$ is the weight of the $j$-th layer, we can always find a vector $\boldsymbol{w}_{k_j}^j$ in $\boldsymbol{w}_\epsilon^j$ such that $\|\boldsymbol{w}_{(j)} - \boldsymbol{w}_{k_j}^j\|_2 \leq \boldsymbol{d}_j\boldsymbol{d}_{j-1}\epsilon/d$. For brevity, let $j_w \in [n_\epsilon{}^j]$ denote the index of $\boldsymbol{w}_{k_j}^j$ in $\epsilon$-net $\boldsymbol{w}_\epsilon^j$. Then let $\boldsymbol{w}_{k_{\boldsymbol{w}}} = [\boldsymbol{w}_{k_1}^j; \cdots; \boldsymbol{w}_{k_j}^j; \cdots; \boldsymbol{w}_{k_l}^j]$. This means that we can always find a vector $\boldsymbol{w}_{k_{\boldsymbol{w}}}$ such that $\|\boldsymbol{w} - \boldsymbol{w}_{k_{\boldsymbol{w}}}\|_2 \leq \epsilon$. Now we use the decomposition strategy to bound our goal:

$$
\left\| \nabla^2 \hat{\boldsymbol{J}}_n(\boldsymbol{w}) - \nabla^2 \boldsymbol{J}(\boldsymbol{w}) \right\|_{\mathrm{op}}
$$

$$
= \left\| \frac{1}{n} \sum_{i=1}^n \nabla^2 f(\boldsymbol{w}, \boldsymbol{x}_{(i)}) - \mathbb{E}(\nabla^2 f(\boldsymbol{w}, \boldsymbol{x})) \right\|_{\mathrm{op}}
$$

$$
= \left\| \frac{1}{n} \sum_{i=1}^n \left( \nabla^2 f(\boldsymbol{w}, \boldsymbol{x}_{(i)}) - \nabla f(\boldsymbol{w}_{k_{\boldsymbol{w}}}, \boldsymbol{x}_{(i)}) \right) + \frac{1}{n} \sum_{i=1}^n \nabla^2 f(\boldsymbol{w}_{k_{\boldsymbol{w}}}, \boldsymbol{x}_{(i)}) - \mathbb{E}(\nabla^2 f(\boldsymbol{w}_{k_{\boldsymbol{w}}}, \boldsymbol{x})) \right.
$$

$$
\left. + \mathbb{E}(\nabla^2 f(\boldsymbol{w}_{k_{\boldsymbol{w}}}, \boldsymbol{x})) - \mathbb{E}(\nabla^2 f(\boldsymbol{w}, \boldsymbol{x})) \right\|_{\mathrm{op}}
$$

$$
\leq \left\| \frac{1}{n} \sum_{i=1}^n \left( \nabla^2 f(\boldsymbol{w}, \boldsymbol{x}_{(i)}) - \nabla^2 f(\boldsymbol{w}_{k_{\boldsymbol{w}}}, \boldsymbol{x}_{(i)}) \right) \right\|_{\mathrm{op}} + \left\| \frac{1}{n} \sum_{i=1}^n \nabla^2 f(\boldsymbol{w}_{k_{\boldsymbol{w}}}, \boldsymbol{x}_{(i)}) - \mathbb{E}(\nabla^2 f(\boldsymbol{w}_{k_{\boldsymbol{w}}}, \boldsymbol{x})) \right\|_{\mathrm{op}}
$$

$$
+ \left\| \mathbb{E}(\nabla^2 f(\boldsymbol{w}_{k_{\boldsymbol{w}}}, \boldsymbol{x})) - \mathbb{E}(\nabla^2 f(\boldsymbol{w}, \boldsymbol{x})) \right\|_{\mathrm{op}}.
$$

Here we also define four events $\boldsymbol{E}_0$, $\boldsymbol{E}_1$, $\boldsymbol{E}_2$ and $\boldsymbol{E}_3$ as

$$
\boldsymbol{E}_0 = \left\{ \sup_{\boldsymbol{w}\in\Omega} \left\| \nabla^2 \hat{\boldsymbol{J}}_n(\boldsymbol{w}) - \nabla^2 \boldsymbol{J}(\boldsymbol{w}) \right\|_{\mathrm{op}} \geq t \right\},
$$

$$
\boldsymbol{E}_1 = \left\{ \sup_{\boldsymbol{w}\in\Omega} \left\| \frac{1}{n} \sum_{i=1}^n \left( \nabla^2 f(\boldsymbol{w}, \boldsymbol{x}_{(i)}) - \nabla^2 f(\boldsymbol{w}_{k_{\boldsymbol{w}}}, \boldsymbol{x}_{(i)}) \right) \right\|_{\mathrm{op}} \geq \frac{t}{3} \right\},
$$

$$
\boldsymbol{E}_2 = \left\{ \sup_{j_w\in[n_\epsilon{}^j], j=[l]} \left\| \frac{1}{n} \sum_{i=1}^n \nabla^2 f(\boldsymbol{w}_{k_{\boldsymbol{w}}}, \boldsymbol{x}_{(i)}) - \mathbb{E}(\nabla^2 f(\boldsymbol{w}_{k_{\boldsymbol{w}}}, \boldsymbol{x})) \right\|_{\mathrm{op}} \geq \frac{t}{3} \right\},
$$

$$
\boldsymbol{E}_3 = \left\{ \sup_{\boldsymbol{w}\in\Omega} \left\| \mathbb{E}(\nabla^2 f(\boldsymbol{w}_{k_{\boldsymbol{w}}}, \boldsymbol{x})) - \mathbb{E}(\nabla^2 f(\boldsymbol{w}, \boldsymbol{x})) \right\|_{\mathrm{op}} \geq \frac{t}{3} \right\}.
$$

Accordingly, we have

$$
\mathbb{P}\left( \boldsymbol{E}_0 \right) \leq \mathbb{P}\left( \boldsymbol{E}_1 \right) + \mathbb{P}\left( \boldsymbol{E}_2 \right) + \mathbb{P}\left( \boldsymbol{E}_3 \right).
$$

So we can respectively bound $\mathbb{P}\left( \boldsymbol{E}_1 \right)$, $\mathbb{P}\left( \boldsymbol{E}_2 \right)$ and $\mathbb{P}\left( \boldsymbol{E}_3 \right)$ to bound $\mathbb{P}\left( \boldsymbol{E}_0 \right)$.

**Step 1. Bound $\mathbb{P}\left(\boldsymbol{E}_1\right)$:** We first bound $\mathbb{P}\left(\boldsymbol{E}_1\right)$ as follows:

$$
\begin{aligned}
\mathbb{P}\left(\boldsymbol{E}_1\right) =& \mathbb{P}\left(\sup_{\boldsymbol{w}\in\Omega}\left\|\frac{1}{n}\sum_{i=1}^{n}\left(\nabla^2 f(\boldsymbol{w},\boldsymbol{x}_{(i)})-\nabla^2 f(\boldsymbol{w}_{k_{\boldsymbol{w}}},\boldsymbol{x}_{(i)})\right)\right\|_2 \geq \frac{t}{3}\right)\\
\overset{\text{\textcircled{1}}}{\leq}& \frac{3}{t}\mathbb{E}\left(\sup_{\boldsymbol{w}\in\Omega}\left\|\frac{1}{n}\sum_{i=1}^{n}\left(\nabla^2 f(\boldsymbol{w},\boldsymbol{x}_{(i)})-\nabla^2 f(\boldsymbol{w}_{k_{\boldsymbol{w}}},\boldsymbol{x}_{(i)})\right)\right\|_2\right)\\
\leq& \frac{3}{t}\mathbb{E}\left(\sup_{\boldsymbol{w}\in\Omega}\left\|\nabla^2 f(\boldsymbol{w},\boldsymbol{x})-\nabla^2 f(\boldsymbol{w}_{k_{\boldsymbol{w}}},\boldsymbol{x})\right\|_2\right)\\
\leq& \frac{3}{t}\mathbb{E}\left(\sup_{\boldsymbol{w}\in\Omega}\frac{\left|\frac{1}{n}\sum_{i=1}^{n}\left(\nabla^2 f(\boldsymbol{w},\boldsymbol{x}_{(i)})-\nabla^2 f(\boldsymbol{w}_{k_{\boldsymbol{w}}},\boldsymbol{x}_{(i)})\right)\right|}{\|\boldsymbol{w}-\boldsymbol{w}_{k_{\boldsymbol{w}}}\|_2}\sup_{\boldsymbol{w}\in\Omega}\|\boldsymbol{w}-\boldsymbol{w}_{k_{\boldsymbol{w}}}\|_2\right)\\
\overset{\text{\textcircled{2}}}{\leq}& \frac{3\alpha_p\epsilon}{t},
\end{aligned}
$$

where \textcircled{1} holds since by Markov inequality and \textcircled{2} holds because of Lemma 10.

Therefore, we can set

$$
t\geq\frac{6\alpha_p\epsilon}{\varepsilon}.
$$

Then we can bound $\mathbb{P}(\boldsymbol{E}_1)$:

$$
\mathbb{P}(\boldsymbol{E}_1)\leq\frac{\varepsilon}{2}.
$$

**Step 2. Bound $\mathbb{P}\left(\boldsymbol{E}_2\right)$:** By Lemma 2, we know that for any matrix $\boldsymbol{X}\in\mathbb{R}^{d\times d}$, its operator norm can be computed as

$$
\|\boldsymbol{X}\|_{\mathrm{op}}\leq\frac{1}{1-2\epsilon}\sup_{\boldsymbol{\lambda}\in\boldsymbol{\lambda}_\epsilon}|\langle\boldsymbol{\lambda},\boldsymbol{X}\boldsymbol{\lambda}\rangle|.
$$

where $\boldsymbol{\lambda}_\epsilon=\{\boldsymbol{\lambda}_1,\ldots,\boldsymbol{\lambda}_{k_{\boldsymbol{w}}}\}$ be an $\epsilon$-covering net of $\mathsf{B}^d(1)$.

Let $\boldsymbol{\lambda}_{1/4}$ be the $\frac{1}{4}$-covering net of $\mathsf{B}^d(1)$ but it has only $s$ nonzero entries. So the size of its $\epsilon$-net is

$$
\binom{d}{s}\left(\frac{3}{1/4}\right)^s\leq\exp\left(s\log\left(12d\right)\right).
$$

Recall that we use $j_w$ to denote the index of $\boldsymbol{w}_{k_j}^j$ in $\epsilon$-net $\boldsymbol{w}_\epsilon^j$ and we have $j_w\in[n_\epsilon{}^j]$, $(n_\epsilon{}^j\leq\exp\left(s_j\log\left(\frac{3rd}{\epsilon}\right)\right))$. Then we can bound $\mathbb{P}\left(\boldsymbol{E}_2\right)$ as follows:

$$
\begin{aligned}
\mathbb{P}\left(\boldsymbol{E}_2\right) =& \mathbb{P}\left(\sup_{j_w\in[n_\epsilon^j]\,j\in[l]}\left\|\frac{1}{n}\sum_{i=1}^{n}\nabla^2 f(\boldsymbol{w}_{k_{\boldsymbol{w}}},\boldsymbol{x}_{(i)})-\mathbb{E}(\nabla^2 f(\boldsymbol{w}_{k_{\boldsymbol{w}}},\boldsymbol{x}))\right\|_2\geq\frac{t}{3}\right)\\
\leq& \mathbb{P}\left(\sup_{j_w\in[n_\epsilon^j]\,j\in[l],\boldsymbol{\lambda}\in\boldsymbol{\lambda}_{1/4}}2\left|\left\langle\boldsymbol{\lambda},\left(\frac{1}{n}\sum_{i=1}^{n}\nabla^2 f(\boldsymbol{w}_{k_{\boldsymbol{w}}},\boldsymbol{x}_{(i)})-\mathbb{E}\left(\nabla^2 f(\boldsymbol{w}_{k_{\boldsymbol{w}}},\boldsymbol{x})\right)\right)\boldsymbol{\lambda}\right\rangle\right|\geq\frac{t}{3}\right)\\
\leq& \exp\left(s\log\left(12d\right)\right)\exp\left(\sum_{j=1}^{l}\boldsymbol{s}_j\log\left(\frac{3rd}{\epsilon}\right)\right)\sup_{j_w\in[n_\epsilon^j]\,j\in[l],\boldsymbol{\lambda}\in\boldsymbol{\lambda}_{1/4}}\mathbb{P}\left(\left|\frac{1}{n}\sum_{i=1}^{n}\left\langle\boldsymbol{\lambda},\right.\right.\right.\\
& \left.\left.\left.\left(\nabla^2 f(\boldsymbol{w}_{k_{\boldsymbol{w}}},\boldsymbol{x}_{(i)})-\mathbb{E}\left(\nabla^2 f(\boldsymbol{w}_{k_{\boldsymbol{w}}},\boldsymbol{x})\right)\right)\right\rangle\right|\geq\frac{t}{6}\right)\\
\overset{\text{\textcircled{1}}}{\leq}& \exp\left(s\log\left(\frac{36rd^2}{\epsilon}\right)\right)10\exp\left(-c_{h'}n\min\left(\frac{t^2}{36\tau^2 l^2\max\left(\omega_g,\omega_g\tau^2,\omega_h\right)},\frac{t}{6\sqrt{\omega_g}l\max\left(\tau,\tau^2\right)}\right)\right),
\end{aligned}
$$

where \textcircled{1} holds since by Lemma 9, we have

$$
\begin{aligned}
\mathbb{P}\left(\left|\frac{1}{n}\sum_{i=1}^{n}\left(\langle\boldsymbol{\lambda},(\nabla_{\boldsymbol{w}}^2 f(\boldsymbol{w},\boldsymbol{x})-\mathbb{E}\nabla_{\boldsymbol{w}}^2 f(\boldsymbol{w},\boldsymbol{x}))\boldsymbol{\lambda}\rangle\right)\right|>t\right)\\
\leq 10\exp\left(-c_{h'}n\min\left(\frac{t^2}{\tau^2 l^2\max\left(\omega_g,\omega_g\tau^2,\omega_h\right)},\frac{t}{\sqrt{\omega_g}l\max\left(\tau,\tau^2\right)}\right)\right),
\end{aligned}
$$

where $\omega_g = r^{4(l-1)}$ and $\omega_h = r^{2(l-2)}$.

Let $d_\epsilon = s \log(36d^2 r/\epsilon) + \log(20/\varepsilon)$. Thus, if we set

$$t \geq \max\left(\sqrt{\frac{36\tau^2 l^2 \max\left(\omega_g, \omega_g \tau^2, \omega_h\right) d_\epsilon}{c_{h'} n}}, \frac{6\sqrt{\omega_g} l \max\left(\tau, \tau^2\right) d_\epsilon}{c_{h'} n}\right),$$

then we have

$$\mathbb{P}\left(\boldsymbol{E}_2\right) \leq \frac{\varepsilon}{2}.$$

**Step 3. Bound $\mathbb{P}\left(\boldsymbol{E}_3\right)$:** We first bound $\mathbb{P}\left(\boldsymbol{E}_3\right)$ as follows:

$$
\begin{aligned}
\mathbb{P}\left(\boldsymbol{E}_3\right) =& \mathbb{P}\left(\sup_{\boldsymbol{w}\in\Omega} \left\|\mathbb{E}(\nabla^2 f(\boldsymbol{w}_{k_{\boldsymbol{w}}}, \boldsymbol{x})) - \mathbb{E}(\nabla^2 f(\boldsymbol{w}, \boldsymbol{x}))\right\|_2 \geq \frac{t}{3}\right) \\
\leq& \mathbb{P}\left(\mathbb{E}\sup_{\boldsymbol{w}\in\Omega} \left\|(\nabla^2 f(\boldsymbol{w}_{k_{\boldsymbol{w}}}, \boldsymbol{x}) - \nabla^2 f(\boldsymbol{w}, \boldsymbol{x})\right\|_2 \geq \frac{t}{3}\right) \\
\leq& \mathbb{P}\left(\sup_{\boldsymbol{w}\in\Omega} \frac{\left|\frac{1}{n}\sum_{i=1}^{n}\left(\nabla^2 f(\boldsymbol{w}, \boldsymbol{x}_{(i)}) - \nabla^2 f(\boldsymbol{w}_{k_{\boldsymbol{w}}}, \boldsymbol{x}_{(i)})\right)\right|}{\|\boldsymbol{w} - \boldsymbol{w}_{k_{\boldsymbol{w}}}\|_2} \sup_{\boldsymbol{w}\in\Omega} \|\boldsymbol{w} - \boldsymbol{w}_{k_{\boldsymbol{w}}}\|_2 \geq \frac{t}{3}\right) \\
\overset{①}{\leq}& \mathbb{P}\left(\alpha_p \epsilon \geq \frac{t}{3}\right),
\end{aligned}
$$

where ① holds because of Lemma 10. We set $\epsilon$ enough small such that $\alpha_p \epsilon < t/3$ always holds. Then it yields $\mathbb{P}\left(\boldsymbol{E}_3\right) = 0$.

**Step 4. Final result**: For brevity, let $\omega_2 = 36\tau^2 l^2 \max\left(\omega_g, \omega_g \tau^2, \omega_h\right)$ and $\omega_3 = 6\sqrt{\omega_g} l \max\left(\tau, \tau^2\right)$. To ensure $\mathbb{P}(\boldsymbol{E}_0) \leq \varepsilon$, we just set $\epsilon = 36rl/n$ and

$$
\begin{aligned}
t \geq& \max\left(\frac{6\alpha_p \epsilon}{\varepsilon}, \ 3\alpha_p \epsilon, \ \sqrt{\frac{\omega_2(s\log(36d^2 r/\epsilon) + \log(20/\varepsilon))}{c_{h'} n}}, \ \frac{\omega_3(s\log(36d^2 r/\epsilon) + \log(20/\varepsilon))}{c_{h'} n}\right) \\
=& \max\left(\frac{216\alpha_p r}{n\varepsilon}, \ \sqrt{\frac{\omega_2(s\log(d^2 nl) + \log(20/\varepsilon))}{c_{h'} n}}, \ \frac{\omega_3(s\log(36d^2 n/l) + \log(20/\varepsilon))}{c_{h'} n}\right).
\end{aligned}
$$

Thus, there exit two universal constants $c_{h_1}$ and $c_{h_2}$ such that if $n \geq c_{h_2} \max(\frac{\alpha_p^2 r^2}{\tau^2 l^2 \omega_h^2 \varepsilon^2 s \log(d/l)}, s\log(d/l)/(l\tau^2))$, then

$$\sup_{\boldsymbol{w}\in\Omega}\left\|\nabla^2 \hat{\boldsymbol{J}}_n(\boldsymbol{w}) - \nabla^2 \boldsymbol{J}(\boldsymbol{w})\right\|_{\mathrm{op}} \leq c_{h_1}\tau l\omega_h \sqrt{\frac{d\log(nl) + \log(20/\varepsilon)}{n}}$$

holds with probability at least $1 - \varepsilon$, where $\omega_h = \max\left(\tau r^{2(l-1)}, r^{2(l-2)}, r^{l-2}\right)$. The proof is completed. $\qquad\square$

## C.3 PROOFS OF MAIN THEOREMS

### C.3.1 PROOF OF THEOREM 1

*Proof.* Recall that the weight of each layer has magnitude bound separately, *i.e.* $\|\boldsymbol{w}_{(j)}\|_2 \leq r$. Assume that $\boldsymbol{w}_{(j)}$ has $\boldsymbol{s}_j$ non-zero entries. Then we have $\sum_{j=1}^{l} \boldsymbol{s}_j = s$. So here we separately assume $\boldsymbol{w}_\epsilon^j = \{\boldsymbol{w}_1^j, \cdots, \boldsymbol{w}_{n_{\epsilon^j}}^j\}$ is the $\boldsymbol{d}_j \boldsymbol{d}_{j-1} \epsilon/d$-covering net of the ball $\mathsf{B}^{\boldsymbol{d}_j \boldsymbol{d}_{j-1}}(r)$ which corresponds to the weight $\boldsymbol{w}_{(j)}$ of the $j$-th layer. Let $n_{\epsilon^j}$ be the $\epsilon/l$-covering number. By $\epsilon$-covering theory in (Vershynin, 2012), we can have

$$n_{\epsilon^j} \leq \left(\begin{array}{c}\boldsymbol{d}_j \boldsymbol{d}_{j-1} \\ \boldsymbol{s}_j\end{array}\right)\left(\frac{3r}{\boldsymbol{d}_j \boldsymbol{d}_{j-1} \epsilon/d}\right)^{\boldsymbol{s}_j} \leq \exp\left(\boldsymbol{s}_j \log\left(\frac{3r \boldsymbol{d}_j \boldsymbol{d}_{j-1}}{\boldsymbol{d}_j \boldsymbol{d}_{j-1} \epsilon/d}\right)\right) = \exp\left(\boldsymbol{s}_j \log\left(\frac{3rd}{\epsilon}\right)\right).$$

Let $\boldsymbol{w} \in \Omega$ be an arbitrary vector. Since $\boldsymbol{w} = [\boldsymbol{w}_{(1)}, \cdots, \boldsymbol{w}_{(l)}]$ where $\boldsymbol{w}_{(j)}$ is the weight of the $j$-th layer, we can always find a vector $\boldsymbol{w}_{k_j}^j$ in $\boldsymbol{w}_\epsilon^j$ such that $\|\boldsymbol{w}_{(j)} - \boldsymbol{w}_{k_j}^j\|_2 \leq \boldsymbol{d}_j \boldsymbol{d}_{j-1} \epsilon / d$. For brevity, let $j_w \in [n_\epsilon{}^j]$ denote the index of $\boldsymbol{w}_{k_j}^j$ in $\epsilon$-net $\boldsymbol{w}_\epsilon^j$. Then let $\boldsymbol{w}_{k_w} = [\boldsymbol{w}_{k_1}^j; \cdots; \boldsymbol{w}_{k_j}^j; \cdots; \boldsymbol{w}_{k_l}^j]$. This means that we can always find a vector $\boldsymbol{w}_{k_w}$ such that $\|\boldsymbol{w} - \boldsymbol{w}_{k_w}\|_2 \leq \epsilon$. Accordingly, we can decompose $\left\|\nabla \hat{\boldsymbol{J}}_n(\boldsymbol{w}) - \nabla \boldsymbol{J}(\boldsymbol{w})\right\|_2$ as

$$
\begin{aligned}
&\left\|\nabla \hat{\boldsymbol{J}}_n(\boldsymbol{w}) - \nabla \boldsymbol{J}(\boldsymbol{w})\right\|_2 \\
=&\left\|\frac{1}{n}\sum_{i=1}^n \nabla f(\boldsymbol{w}, \boldsymbol{x}_{(i)}) - \mathbb{E}(\nabla f(\boldsymbol{w}, \boldsymbol{x}))\right\|_2 \\
=&\left\|\frac{1}{n}\sum_{i=1}^n \left(\nabla f(\boldsymbol{w}, \boldsymbol{x}_{(i)}) - \nabla f(\boldsymbol{w}_{k_w}, \boldsymbol{x}_{(i)})\right) + \frac{1}{n}\sum_{i=1}^n \nabla f(\boldsymbol{w}_{k_w}, \boldsymbol{x}_{(i)}) - \mathbb{E}(\nabla f(\boldsymbol{w}_{k_w}, \boldsymbol{x}))\right. \\
&\left. + \mathbb{E}(\nabla f(\boldsymbol{w}_{k_w}, \boldsymbol{x})) - \mathbb{E}(\nabla f(\boldsymbol{w}, \boldsymbol{x}))\right\|_2 \\
\leq&\left\|\frac{1}{n}\sum_{i=1}^n \left(\nabla f(\boldsymbol{w}, \boldsymbol{x}_{(i)}) - \nabla f(\boldsymbol{w}_{k_w}, \boldsymbol{x}_{(i)})\right)\right\|_2 + \left\|\frac{1}{n}\sum_{i=1}^n \nabla f(\boldsymbol{w}_{k_w}, \boldsymbol{x}_{(i)}) - \mathbb{E}(\nabla f(\boldsymbol{w}_{k_w}, \boldsymbol{x}))\right\|_2 \\
&+ \left\|\mathbb{E}(\nabla f(\boldsymbol{w}_{k_w}, \boldsymbol{x})) - \mathbb{E}(\nabla f(\boldsymbol{w}, \boldsymbol{x}))\right\|_2.
\end{aligned}
$$

Here we also define four events $\boldsymbol{E}_0$, $\boldsymbol{E}_1$, $\boldsymbol{E}_2$ and $\boldsymbol{E}_3$ as

$$
\begin{aligned}
\boldsymbol{E}_0 &= \left\{\sup_{\boldsymbol{w} \in \Omega}\left\|\nabla \hat{\boldsymbol{J}}_n(\boldsymbol{w}) - \nabla \boldsymbol{J}(\boldsymbol{w})\right\|_2 \geq t\right\}, \\
\boldsymbol{E}_1 &= \left\{\sup_{\boldsymbol{w} \in \Omega}\left\|\frac{1}{n}\sum_{i=1}^n \left(\nabla f(\boldsymbol{w}, \boldsymbol{x}_{(i)}) - \nabla f(\boldsymbol{w}_{k_w}, \boldsymbol{x}_{(i)})\right)\right\|_2 \geq \frac{t}{3}\right\}, \\
\boldsymbol{E}_2 &= \left\{\sup_{j_w \in [n_\epsilon{}^j], j=[l]}\left\|\frac{1}{n}\sum_{i=1}^n \nabla f(\boldsymbol{w}_{k_w}, \boldsymbol{x}_{(i)}) - \mathbb{E}(\nabla f(\boldsymbol{w}_{k_w}, \boldsymbol{x}))\right\|_2 \geq \frac{t}{3}\right\}, \\
\boldsymbol{E}_3 &= \left\{\sup_{\boldsymbol{w} \in \Omega}\left\|\mathbb{E}(\nabla f(\boldsymbol{w}_{k_w}, \boldsymbol{x})) - \mathbb{E}(\nabla f(\boldsymbol{w}, \boldsymbol{x}))\right\|_2 \geq \frac{t}{3}\right\}.
\end{aligned}
$$

Accordingly, we have
$$
\mathbb{P}\left(\boldsymbol{E}_0\right) \leq \mathbb{P}\left(\boldsymbol{E}_1\right) + \mathbb{P}\left(\boldsymbol{E}_2\right) + \mathbb{P}\left(\boldsymbol{E}_3\right).
$$
So we can respectively bound $\mathbb{P}\left(\boldsymbol{E}_1\right)$, $\mathbb{P}\left(\boldsymbol{E}_2\right)$ and $\mathbb{P}\left(\boldsymbol{E}_3\right)$ to bound $\mathbb{P}\left(\boldsymbol{E}_0\right)$.

**Step 1. Bound $\mathbb{P}\left(\boldsymbol{E}_1\right)$:** We first bound $\mathbb{P}\left(\boldsymbol{E}_1\right)$ as follows:

$$
\begin{aligned}
\mathbb{P}\left(\boldsymbol{E}_1\right) =&\mathbb{P}\left(\sup_{\boldsymbol{w} \in \Omega}\left\|\frac{1}{n}\sum_{i=1}^n \left(\nabla f(\boldsymbol{w}, \boldsymbol{x}_{(i)}) - \nabla f(\boldsymbol{w}_{k_w}, \boldsymbol{x}_{(i)})\right)\right\|_2 \geq \frac{t}{3}\right) \\
\overset{①}{\leq}&\frac{3}{t}\mathbb{E}\left(\sup_{\boldsymbol{w} \in \Omega}\left\|\frac{1}{n}\sum_{i=1}^n \left(\nabla f(\boldsymbol{w}, \boldsymbol{x}_{(i)}) - \nabla f(\boldsymbol{w}_{k_w}, \boldsymbol{x}_{(i)})\right)\right\|_2\right) \\
\leq&\frac{3}{t}\mathbb{E}\left(\sup_{\boldsymbol{w} \in \Omega}\frac{\left\|\frac{1}{n}\sum_{i=1}^n \left(\nabla f(\boldsymbol{w}, \boldsymbol{x}_{(i)}) - \nabla f(\boldsymbol{w}_{k_w}, \boldsymbol{x}_{(i)})\right)\right\|_2}{\|\boldsymbol{w} - \boldsymbol{w}_{k_w}\|_2}\sup_{\boldsymbol{w} \in \Omega}\|\boldsymbol{w} - \boldsymbol{w}_{k_w}\|_2\right) \\
\leq&\frac{3\epsilon}{t}\mathbb{E}\left(\sup_{\boldsymbol{w} \in \Omega}\left\|\nabla^2 \hat{\boldsymbol{J}}_n(\boldsymbol{w}, \boldsymbol{x})\right\|_2\right),
\end{aligned}
$$

where ① holds since by Markov inequality, we have that for an arbitrary nonnegative random variable $x$, then $\mathbb{P}(x \geq t) \leq \frac{\mathbb{E}(x)}{t}$.

Now we only need to bound $\mathbb{E}\left(\sup_{\boldsymbol{w}\in\Omega}\left\|\nabla^2\hat{\boldsymbol{J}}_n(\boldsymbol{w},\boldsymbol{x})\right\|_2\right)$. Now we utilize Lemma 10 to achieve this goal:

$$\mathbb{E}\left(\sup_{\boldsymbol{w}\in\Omega}\left\|\nabla^2\hat{\boldsymbol{J}}_n(\boldsymbol{w},\boldsymbol{x})\right\|_2\right)\leq=\mathbb{E}\left(\sup_{\boldsymbol{w}\in\Omega}\left\|\nabla^2f(\boldsymbol{w},\boldsymbol{x})-\nabla^2f(\boldsymbol{w}^*,\boldsymbol{x})\right\|_2\right)\leq l\sqrt{\alpha_l}.$$

where $\alpha_l=c_{t'}r_x^4r^{4l-2}$. Therefore, we have

$$\mathbb{P}\left(\boldsymbol{E}_1\right)\leq\frac{3l\sqrt{\alpha_l}\epsilon}{t}.$$

We further let

$$t\geq\frac{6l\sqrt{\alpha_l}\epsilon}{\varepsilon}.$$

Then we can bound $\mathbb{P}(\boldsymbol{E}_1)$:

$$\mathbb{P}(\boldsymbol{E}_1)\leq\frac{\varepsilon}{2}.$$

**Step 2. Bound** $\mathbb{P}\left(\boldsymbol{E}_2\right)$: By Lemma 1, we know that for any vector $\boldsymbol{x}\in\mathbb{R}^d$, its $\ell_2$-norm can be computed as

$$\|\boldsymbol{x}\|_2\leq\frac{1}{1-\epsilon}\sup_{\boldsymbol{\lambda}\in\boldsymbol{\lambda}_\epsilon}\langle\boldsymbol{\lambda},\boldsymbol{x}\rangle.$$

where $\boldsymbol{\lambda}_\epsilon=\{\boldsymbol{\lambda}_1,\ldots,\boldsymbol{\lambda}_{k_{\boldsymbol{w}}}\}$ be an $\epsilon$-covering net of $\mathsf{B}^d(1)$.

Let $\boldsymbol{\lambda}_{1/2}$ be the $\frac{1}{2}$-covering net of $\mathsf{B}^d(1)$ but it has only $s$ nonzero entries. So the size of its $\epsilon$-net is

$$\binom{d}{s}\left(\frac{3}{1/2}\right)^s\leq\exp\left(s\log\left(6d\right)\right).$$

Recall that we use $j_w$ to denote the index of $\boldsymbol{w}_{k_j}^j$ in $\epsilon$-net $\boldsymbol{w}_\epsilon^j$ and we have $j_w\in[n_\epsilon{}^j]$, $(n_\epsilon{}^j\leq\exp\left(\boldsymbol{s}_j\log\left(\frac{3rd}{\epsilon}\right)\right))$. Then we can bound $\mathbb{P}\left(\boldsymbol{E}_2\right)$ as follows:

$$\mathbb{P}\left(\boldsymbol{E}_2\right)=\mathbb{P}\left(\sup_{j_w\in[n_\epsilon{}^j],j=[l]}\left\|\frac{1}{n}\sum_{i=1}^n\nabla f(\boldsymbol{w}_{k_{\boldsymbol{w}}},\boldsymbol{x}_{(i)})-\mathbb{E}(\nabla f(\boldsymbol{w}_{k_{\boldsymbol{w}}},\boldsymbol{x}))\right\|_2\geq\frac{t}{3}\right)$$

$$=\mathbb{P}\left(\sup_{j_w\in[n_\epsilon{}^j],j=[l],\boldsymbol{\lambda}\in\boldsymbol{\lambda}_{1/2}}2\left\langle\boldsymbol{\lambda},\frac{1}{n}\sum_{i=1}^n\nabla f(\boldsymbol{w}_{k_{\boldsymbol{w}}},\boldsymbol{x}_{(i)})-\mathbb{E}\left(\nabla f(\boldsymbol{w}_{k_{\boldsymbol{w}}},\boldsymbol{x})\right)\right\rangle\geq\frac{t}{3}\right)$$

$$\leq\exp\left(s\log\left(6d\right)\right)\exp\left(\sum_{j=1}^l\boldsymbol{s}_j\log\left(\frac{3rd}{\epsilon}\right)\right)\sup_{j_w\in[n_\epsilon{}^j],j=[l],\boldsymbol{\lambda}\in\boldsymbol{\lambda}_{1/2}}\mathbb{P}\left(\frac{1}{n}\sum_{i=1}^n\left\langle\boldsymbol{\lambda},\right.\right.$$

$$\left.\left.\nabla f(\boldsymbol{w}_{k_{\boldsymbol{w}}},\boldsymbol{x}_{(i)})-\mathbb{E}\left(\nabla f(\boldsymbol{w}_{k_{\boldsymbol{w}}},\boldsymbol{x})\right)\right\rangle\geq\frac{t}{6}\right)$$

$$\overset{①}{\leq}\exp\left(s\log\left(\frac{18rd}{\epsilon}\right)\right)6\exp\left(-c_{g'}n\min\left(\frac{t^2}{36l\max\left(\omega_g\tau^2,\omega_g\tau^4,\omega_{g'}\tau^2\right)},\frac{t}{6\sqrt{l\omega_g}\max\left(\tau,\tau^2\right)}\right)\right),$$

where ① holds since by Lemma 8, we have

$$\mathbb{P}\left(\frac{1}{n}\sum_{i=1}^n\left(\langle\boldsymbol{\lambda},\nabla_{\boldsymbol{w}}f(\boldsymbol{w},\boldsymbol{x}_{(i)})-\mathbb{E}\nabla_{\boldsymbol{w}}f(\boldsymbol{w},\boldsymbol{x}_{(i)})\rangle\right)>t\right)$$

$$\leq 3\exp\left(-c_{g'}n\min\left(\frac{t^2}{l\max\left(\omega_g\tau^2,\omega_g\tau^4,\omega_{g'}\tau^2\right)},\frac{t}{\sqrt{l\omega_g}\max\left(\tau,\tau^2\right)}\right)\right),$$

where $c_{g'}$ is a constant; $\omega_g=c_qr^{2(2l-1)}$ and $\omega_{g'}=c_qr^{2(l-1)}$ in which $c_q=\sqrt{\max_{0\leq i\leq l}\boldsymbol{d}_i}$.

Let $\omega_2=36l\max\left(\omega_g\tau^2,\omega_g\tau^4,\omega_{g'}\tau^2\right)$ and $\omega_3=6\sqrt{l\omega_g}\max\left(\tau,\tau^2\right)$. Thus, if we set

$$t\geq\max\left(\sqrt{\frac{\omega_2(s\log(18dr/\epsilon)+\log(12/\varepsilon))}{c_{g'}n}},\frac{\omega_3(s\log(18dr/\epsilon)+\log(12/\varepsilon))}{c_{g'}n}\right),$$

then we have

$$\mathbb{P}\left(\boldsymbol{E}_2\right) \leq \frac{\varepsilon}{2}.$$

**Step 3. Bound $\mathbb{P}\left(\boldsymbol{E}_3\right)$:** We first bound $\mathbb{P}\left(\boldsymbol{E}_3\right)$ as follows:

$$
\begin{aligned}
\mathbb{P}\left(\boldsymbol{E}_3\right) =& \mathbb{P}\left(\sup_{\boldsymbol{w}\in\Omega} \|\mathbb{E}(f(\boldsymbol{w}_{k_{\boldsymbol{w}}},\boldsymbol{x})) - \mathbb{E}(f(\boldsymbol{w},\boldsymbol{x}))\|_2 \geq \frac{t}{3}\right) \\
=& \mathbb{P}\left(\sup_{\boldsymbol{w}\in\Omega} \frac{\|\mathbb{E}\left(f(\boldsymbol{w}_{k_{\boldsymbol{w}}},\boldsymbol{x}) - f(\boldsymbol{w},\boldsymbol{x})\|_2\right)}{\|\boldsymbol{w} - \boldsymbol{w}_{k_{\boldsymbol{w}}}\|_2} \sup_{\boldsymbol{w}\in\Omega} \|\boldsymbol{w} - \boldsymbol{w}_{k_{\boldsymbol{w}}}\|_2 \geq \frac{t}{3}\right) \\
\leq& \mathbb{P}\left(\epsilon\mathbb{E}\sup_{\boldsymbol{w}\in\Omega} \left\|\nabla^2\hat{\boldsymbol{J}}_n(\boldsymbol{w},\boldsymbol{x})\right\|_2 \geq \frac{t}{3}\right) \\
\leq& \mathbb{P}\left(l\sqrt{\alpha_l}\epsilon \geq \frac{t}{3}\right).
\end{aligned}
$$

We set $\epsilon$ enough small such that $l\sqrt{\alpha_l}\epsilon < t/3$ always holds. Then it yields $\mathbb{P}\left(\boldsymbol{E}_3\right) = 0$.

**Step 4. Final result:** Finally, to ensure $\mathbb{P}(\boldsymbol{E}_0) \leq \varepsilon$, we just set $\epsilon = 18lr/n$ and

$$
\begin{aligned}
t \geq& \max\left(\frac{6l\sqrt{\alpha_l}\epsilon}{\varepsilon},\ 3l\sqrt{\alpha_l}\epsilon,\ \sqrt{\frac{\omega_2(s\log(18dr/\epsilon)+\log(12/\varepsilon))}{c_{g'}n}},\ \frac{\omega_3(s\log(18dr/\epsilon)+\log(12/\varepsilon))}{c_{g'}n}\right) \\
=& \max\left(\frac{108l^2\sqrt{\alpha_l}r}{n\varepsilon},\ \sqrt{\frac{\omega_2(s\log(dn/l)+\log(12/\varepsilon))}{c_{g'}n}},\ \frac{\omega_3(s\log(dn/l)+\log(12/\varepsilon))}{c_{g'}n}\right).
\end{aligned}
$$

Notice, we have $\alpha_l = c_{t'}r_x^4 r^{4l-2}$ where $c_{t'}$ is a constant. Therefore, there exists two universal constants $c_g$ and $c_{g'}$ such that $n \geq c_{g'}\max(\frac{l^3 r^2 r_x^4}{c_q s\log(d/l)\varepsilon^2\tau^4\log(1/\varepsilon)}, s\log(d/l)/(l\tau^2))$, then

$$\sup_{\boldsymbol{w}\in\Omega}\left\|\nabla\hat{\boldsymbol{J}}_n(\boldsymbol{w}) - \nabla\boldsymbol{J}(\boldsymbol{w})\right\|_2 \leq c_g\tau\omega_g\sqrt{lc_q}\sqrt{\frac{s\log(dn/l)+\log(12/\varepsilon)}{n}}$$

holds with probability at least $1-\varepsilon$, where $\omega_g = \max\left(\tau r^{2l-1}, r^{2l-1}, r^{l-1}\right)$. □

### C.3.2 PROOF OF THEOREM 2

*Proof.* Suppose that $\{\boldsymbol{w}^{(1)}, \boldsymbol{w}^{(2)}, \cdots, \boldsymbol{w}^{(m)}\}$ are the non-degenerate critical points of $\boldsymbol{J}(\boldsymbol{w})$. So for any $\boldsymbol{w}^{(k)}$, it obeys

$$\inf_i\left|\lambda_i^k\left(\nabla^2\boldsymbol{J}(\boldsymbol{w}^{(k)})\right)\right| \geq \zeta,$$

where $\lambda_i^k\left(\nabla^2\boldsymbol{J}(\boldsymbol{w}^{(k)})\right)$ denotes the $i$-th eigenvalue of the Hessian $\nabla^2\boldsymbol{J}(\boldsymbol{w}^{(k)})$ and $\zeta$ is a constant. We further define a set $D = \{\boldsymbol{w}\in\mathbb{R}^d \mid \|\nabla\boldsymbol{J}(\boldsymbol{w})\|_2 \leq \epsilon \text{ and } \inf_i|\lambda_i\left(\nabla^2\boldsymbol{J}(\boldsymbol{w}^{(k)})\right)| \geq \zeta\}$. According to Lemma 4, $D = \cup_{k=1}^\infty D_k$ where each $D_k$ is a disjoint component with $\boldsymbol{w}^{(k)}\in D_k$ for $k \leq m$ and $D_k$ does not contain any critical point of $\boldsymbol{J}(\boldsymbol{w})$ for $k \geq m+1$. On the other hand, by the continuity of $\nabla\boldsymbol{J}(\boldsymbol{w})$, it yields $\|\nabla\boldsymbol{J}(\boldsymbol{w})\|_2 = \epsilon$ for $\boldsymbol{w}\in\partial D_k$. Notice, we set the value of $\epsilon$ blow which is actually a function related to $n$.

Then by utilizing Theorem 1, we let sample number $n$ sufficient large such that

$$\sup_{\boldsymbol{w}\in\Omega}\left\|\nabla\hat{\boldsymbol{J}}_n(\boldsymbol{w}) - \nabla\boldsymbol{J}(\boldsymbol{w})\right\|_2 \leq z_g \triangleq \frac{\epsilon}{2}$$

holds with probability at least $1-\varepsilon$, where if $n \geq c_{g'}\max(\frac{l^3 r^2 r_x^4}{c_q s\log(d/l)\varepsilon^2\tau^4\log(1/\varepsilon)}, \frac{s\log(d/l)}{l\tau^2})$, $z_g = c_g\tau\omega_g\sqrt{lc_q}\sqrt{\frac{s\log(dn/l)+\log(12/\varepsilon)}{n}}$.

This further gives that for arbitrary $\boldsymbol{w} \in D_k$, we have

$$
\inf_{\boldsymbol{w} \in D_k} \left\| t \nabla \hat{\boldsymbol{J}}_n(\boldsymbol{w}) + (1-t) \nabla \boldsymbol{J}(\boldsymbol{w}) \right\|_2 = \inf_{\boldsymbol{w} \in D_k} \left\| t \left( \nabla \hat{\boldsymbol{J}}_n(\boldsymbol{w}) - \nabla \boldsymbol{J}(\boldsymbol{w}) \right) + \nabla \boldsymbol{J}(\boldsymbol{w}) \right\|_2
$$

$$
\geq \inf_{\boldsymbol{w} \in D_k} \left\| \nabla \boldsymbol{J}(\boldsymbol{w}) \right\|_2 - \sup_{\boldsymbol{w} \in D_k} t \left\| \nabla \hat{\boldsymbol{J}}_n(\boldsymbol{w}) - \nabla \boldsymbol{J}(\boldsymbol{w}) \right\|_2
$$

$$
\geq \frac{\epsilon}{2}. \tag{20}
$$

Similarly, by utilizing Lemma 11, let $n$ be sufficient large such that

$$
\sup_{\boldsymbol{w} \in \Omega} \left\| \nabla^2 \hat{\boldsymbol{J}}_n(\boldsymbol{w}) - \nabla^2 \boldsymbol{J}(\boldsymbol{w}) \right\|_{\mathrm{op}} \leq z_s \leq \frac{\zeta}{2}
$$

holds with probability at least $1 - \varepsilon$, where if $n \geq c_{h_2} \max(\frac{\alpha_p^2 r^2}{\tau^2 l^2 \omega_h^2 \varepsilon^2 s \log(d/l)}, s \log(d/l)/(l\tau^2))$, $z_s = c_{h_1} \tau l \omega_h \sqrt{\frac{s \log(nl) + \log(20/\varepsilon)}{n}}$.

Assume that $\boldsymbol{b} \in \mathbb{R}^d$ is a vector and satisfies $\boldsymbol{b}^T \boldsymbol{b} = 1$. In this case, we can bound $\lambda_i^k \left( \nabla^2 \hat{\boldsymbol{J}}_n(\boldsymbol{w}) \right)$ for arbitrary $\boldsymbol{w} \in D_k$ as follows:

$$
\inf_{\boldsymbol{w} \in D_k} \left| \lambda_i^k \left( \nabla^2 \hat{\boldsymbol{J}}_n(\boldsymbol{w}) \right) \right| = \inf_{\boldsymbol{w} \in D_k} \min_{\boldsymbol{b}^T \boldsymbol{b} = 1} \left| \boldsymbol{b}^T \nabla^2 \hat{\boldsymbol{J}}_n(\boldsymbol{w}) \boldsymbol{b} \right|
$$

$$
= \inf_{\boldsymbol{w} \in D_k} \min_{\boldsymbol{b}^T \boldsymbol{b} = 1} \left| \boldsymbol{b}^T \left( \nabla^2 \hat{\boldsymbol{J}}_n(\boldsymbol{w}) - \nabla^2 \boldsymbol{J}(\boldsymbol{w}) \right) \boldsymbol{b} + \boldsymbol{b}^T \nabla^2 \boldsymbol{J}(\boldsymbol{w}) \boldsymbol{b} \right|
$$

$$
\geq \inf_{\boldsymbol{w} \in D_k} \min_{\boldsymbol{b}^T \boldsymbol{b} = 1} \left| \boldsymbol{b}^T \nabla^2 \boldsymbol{J}(\boldsymbol{w}) \boldsymbol{b} \right| - \min_{\boldsymbol{b}^T \boldsymbol{b} = 1} \left| \boldsymbol{b}^T \left( \nabla^2 \hat{\boldsymbol{J}}_n(\boldsymbol{w}) - \nabla^2 \boldsymbol{J}(\boldsymbol{w}) \right) \boldsymbol{b} \right|
$$

$$
\geq \inf_{\boldsymbol{w} \in D_k} \min_{\boldsymbol{b}^T \boldsymbol{b} = 1} \left| \boldsymbol{b}^T \nabla^2 \boldsymbol{J}(\boldsymbol{w}) \boldsymbol{b} \right| - \max_{\boldsymbol{b}^T \boldsymbol{b} = 1} \left| \boldsymbol{b}^T \left( \nabla^2 \hat{\boldsymbol{J}}_n(\boldsymbol{w}) - \nabla^2 \boldsymbol{J}(\boldsymbol{w}) \right) \boldsymbol{b} \right|
$$

$$
= \inf_{\boldsymbol{w} \in D_k} \inf_i \left| \lambda_i^k \left( \nabla^2 f(\boldsymbol{w}_{(k)}, \boldsymbol{x}) \right) \right| - \left\| \nabla^2 \hat{\boldsymbol{J}}_n(\boldsymbol{w}) - \nabla^2 \boldsymbol{J}(\boldsymbol{w}) \right\|_{\mathrm{op}}
$$

$$
\geq \frac{\zeta}{2}.
$$

This means that in each set $D_k$, $\nabla^2 \hat{\boldsymbol{J}}_n(\boldsymbol{w})$ has no zero eigenvalues. Then, combine this and Eqn. (20), by Lemma 3 we know that if the population risk $\boldsymbol{J}(\boldsymbol{w})$ has no critical point in $D_k$, then the empirical risk $\hat{\boldsymbol{J}}_n(\boldsymbol{w})$ has also no critical point in $D_k$; otherwise it also holds. By Lemma 3, we can also obtain that in $D_k$, if $\boldsymbol{J}(\boldsymbol{w})$ has a unique critical point $\boldsymbol{w}_{(k)}$ with non-degenerate index $s_k$, then $\hat{\boldsymbol{J}}_n(\boldsymbol{w})$ also has a unique critical point $\boldsymbol{w}_{(k)}^n$ in $D_k$ with the same non-degenerate index $s_k$. The first conclusion is proved.

Now we bound the distance between the corresponding critical points of $\boldsymbol{J}(\boldsymbol{w})$ and $\hat{\boldsymbol{J}}_n(\boldsymbol{w})$. Assume that in $D_k$, $\boldsymbol{J}(\boldsymbol{w})$ has a unique critical point $\boldsymbol{w}^{(k)}$ and $\hat{\boldsymbol{J}}_n(\boldsymbol{w})$ also has a unique critical point $\boldsymbol{w}_n^{(k)}$. Then, there exists $t \in [0,1]$ such that for any $\boldsymbol{z} \in \partial \mathsf{B}^d(1)$, we have

$$
\epsilon \geq \| \nabla \boldsymbol{J}(\boldsymbol{w}_n^{(k)}) \|_2
$$

$$
= \max_{\boldsymbol{z}^T \boldsymbol{z} = 1} \langle \nabla \boldsymbol{J}(\boldsymbol{w}_n^{(k)}), \boldsymbol{z} \rangle
$$

$$
= \max_{\boldsymbol{z}^T \boldsymbol{z} = 1} \langle \nabla \boldsymbol{J}(\boldsymbol{w}^{(k)}), \boldsymbol{z} \rangle + \langle \nabla^2 \boldsymbol{J}(\boldsymbol{w}^{(k)} + t(\boldsymbol{w}_n^{(k)} - \boldsymbol{w}^{(k)}))(\boldsymbol{w}_n^{(k)} - \boldsymbol{w}^{(k)}), \boldsymbol{z} \rangle
$$

$$
\overset{①}{\geq} \left\langle \left( \nabla^2 \boldsymbol{J}(\boldsymbol{w}^{(k)}) \right)^2 (\boldsymbol{w}_n^{(k)} - \boldsymbol{w}^{(k)}), (\boldsymbol{w}_n^{(k)} - \boldsymbol{w}^{(k)}) \right\rangle^{1/2}
$$

$$
\overset{②}{\geq} \zeta \| \boldsymbol{w}_n^{(k)} - \boldsymbol{w}^{(k)} \|_2,
$$

where ① holds since $\nabla \boldsymbol{J}(\boldsymbol{w}^{(k)}) = \boldsymbol{0}$ and ② holds since $\boldsymbol{w}^{(k)} + t(\boldsymbol{w}_n^{(k)} - \boldsymbol{w}^{(k)})$ is in $D_k$ and for any $\boldsymbol{w} \in D_k$ we have $\inf_i |\lambda_i \left( \nabla^2 \boldsymbol{J}(\boldsymbol{w}) \right)| \geq \zeta$. Consider the conditions in Lemma 11 and Theorem 1, we can obtain that if $n \geq c_h \max(\frac{l^3 r^2 r_x^4}{c_q s \log(d/l) \varepsilon^2 \tau^4 \log(1/\varepsilon)}, s \log(d/l)/\zeta^2)$ where $c_h$ is a constant, then

$$
\| \boldsymbol{w}_n^{(k)} - \boldsymbol{w}^{(k)} \|_2 \leq \frac{2 c_g \tau \omega_g}{\zeta} \sqrt{l c_q} \sqrt{\frac{s \log(dn/l) + \log(12/\varepsilon)}{n}}
$$

holds with probability at least $1 - \varepsilon$. □

### C.3.3 Proof of Theorem 3

*Proof.* Recall that the weight of each layer has magnitude bound separately, *i.e.* $\|\boldsymbol{w}_{(j)}\|_2 \leq r$. Assume that $\boldsymbol{w}_{(j)}$ has $\boldsymbol{s}_j$ non-zero entries. Then we have $\sum_{j=1}^{l} \boldsymbol{s}_j = s$. So here we separately assume $\boldsymbol{w}_\epsilon^j = \{\boldsymbol{w}_1^j, \cdots, \boldsymbol{w}_{n_\epsilon^j}^j\}$ is the $\boldsymbol{d}_j \boldsymbol{d}_{j-1} \epsilon / d$-covering net of the ball $\mathsf{B}^{\boldsymbol{d}_j \boldsymbol{d}_{j-1}}(r)$ which corresponds to the weight $\boldsymbol{w}_{(j)}$ of the $j$-th layer. Let $n_\epsilon^j$ be the $\epsilon/l$-covering number. By $\epsilon$-covering theory in (Vershynin, 2012), we can have

$$n_\epsilon^j \leq \left( \begin{array}{c} \boldsymbol{d}_j \boldsymbol{d}_{j-1} \\ \boldsymbol{s}_j \end{array} \right) \left( \frac{3r}{\boldsymbol{d}_j \boldsymbol{d}_{j-1} \epsilon / d} \right)^{\boldsymbol{s}_j} \leq \exp\left( \boldsymbol{s}_j \log\left( \frac{3r \boldsymbol{d}_j \boldsymbol{d}_{j-1}}{\boldsymbol{d}_j \boldsymbol{d}_{j-1} \epsilon / d} \right) \right) = \exp\left( \boldsymbol{s}_j \log\left( \frac{3rd}{\epsilon} \right) \right).$$

Let $\boldsymbol{w} \in \Omega$ be an arbitrary vector. Since $\boldsymbol{w} = [\boldsymbol{w}_{(1)}, \cdots, \boldsymbol{w}_{(l)}]$ where $\boldsymbol{w}_{(j)}$ is the weight of the $j$-th layer, we can always find a vector $\boldsymbol{w}_{k_j}^j$ in $\boldsymbol{w}_\epsilon^j$ such that $\|\boldsymbol{w}_{(j)} - \boldsymbol{w}_{k_j}^j\|_2 \leq \boldsymbol{d}_j \boldsymbol{d}_{j-1} \epsilon / d$. For brevity, let $j_w \in [n_\epsilon^j]$ denote the index of $\boldsymbol{w}_{k_j}^j$ in $\epsilon$-net $\boldsymbol{w}_\epsilon^j$. Then let $\boldsymbol{w}_{k_w} = [\boldsymbol{w}_{k_1}^j; \cdots; \boldsymbol{w}_{k_j}^j; \cdots; \boldsymbol{w}_{k_l}^j]$. This means that we can always find a vector $\boldsymbol{w}_{k_w}$ such that $\|\boldsymbol{w} - \boldsymbol{w}_{k_w}\|_2 \leq \epsilon$. Now we use the decomposition strategy to bound our goal:

$$\left| \hat{\boldsymbol{J}}_n(\boldsymbol{w}) - \boldsymbol{J}(\boldsymbol{w}) \right| = \left| \frac{1}{n} \sum_{i=1}^{n} f(\boldsymbol{w}, \boldsymbol{x}_{(i)}) - \mathbb{E}(f(\boldsymbol{w}, \boldsymbol{x})) \right|$$

$$= \left| \frac{1}{n} \sum_{i=1}^{n} (f(\boldsymbol{w}, \boldsymbol{x}_{(i)}) - f(\boldsymbol{w}_{k_w}, \boldsymbol{x}_{(i)})) + \frac{1}{n} \sum_{i=1}^{n} f(\boldsymbol{w}_{k_w}, \boldsymbol{x}_{(i)}) - \mathbb{E}f(\boldsymbol{w}_{k_w}, \boldsymbol{x}) + \mathbb{E}f(\boldsymbol{w}_{k_w}, \boldsymbol{x}) - \mathbb{E}f(\boldsymbol{w}, \boldsymbol{x}) \right|$$

$$\leq \left| \frac{1}{n} \sum_{i=1}^{n} (f(\boldsymbol{w}, \boldsymbol{x}_{(i)}) - f(\boldsymbol{w}_{k_w}, \boldsymbol{x}_{(i)})) \right| + \left| \frac{1}{n} \sum_{i=1}^{n} f(\boldsymbol{w}_{k_w}, \boldsymbol{x}_{(i)}) - \mathbb{E}f(\boldsymbol{w}_{k_w}, \boldsymbol{x}) \right| + \left| \mathbb{E}f(\boldsymbol{w}_{k_w}, \boldsymbol{x}) - \mathbb{E}f(\boldsymbol{w}, \boldsymbol{x}) \right|.$$

Then, we define four events $\boldsymbol{E}_0$, $\boldsymbol{E}_1$, $\boldsymbol{E}_2$ and $\boldsymbol{E}_3$ as

$$\boldsymbol{E}_0 = \left\{ \sup_{\boldsymbol{w} \in \Omega} \left| \hat{\boldsymbol{J}}_n(\boldsymbol{w}) - \boldsymbol{J}(\boldsymbol{w}) \right| \geq t \right\},$$

$$\boldsymbol{E}_1 = \left\{ \sup_{\boldsymbol{w} \in \Omega} \left| \frac{1}{n} \sum_{i=1}^{n} \left( f(\boldsymbol{w}, \boldsymbol{x}_{(i)}) - f(\boldsymbol{w}_{k_w}, \boldsymbol{x}_{(i)}) \right) \right| \geq \frac{t}{3} \right\},$$

$$\boldsymbol{E}_2 = \left\{ \sup_{j_w \in [n_\epsilon^j], j=[l]} \left| \frac{1}{n} \sum_{i=1}^{n} f(\boldsymbol{w}_{k_w}, \boldsymbol{x}_{(i)}) - \mathbb{E}(f(\boldsymbol{w}_{k_w}, \boldsymbol{x})) \right| \geq \frac{t}{3} \right\},$$

$$\boldsymbol{E}_3 = \left\{ \sup_{\boldsymbol{w} \in \Omega} \left| \mathbb{E}(f(\boldsymbol{w}_{k_w}, \boldsymbol{x})) - \mathbb{E}(f(\boldsymbol{w}, \boldsymbol{x})) \right| \geq \frac{t}{3} \right\}.$$

Accordingly, we have

$$\mathbb{P}\left( \boldsymbol{E}_0 \right) \leq \mathbb{P}\left( \boldsymbol{E}_1 \right) + \mathbb{P}\left( \boldsymbol{E}_2 \right) + \mathbb{P}\left( \boldsymbol{E}_3 \right).$$

So we can respectively bound $\mathbb{P}\left( \boldsymbol{E}_1 \right)$, $\mathbb{P}\left( \boldsymbol{E}_2 \right)$ and $\mathbb{P}\left( \boldsymbol{E}_3 \right)$ to bound $\mathbb{P}\left( \boldsymbol{E}_0 \right)$.

**Step 1. Bound $\mathbb{P}\left( \boldsymbol{E}_1 \right)$:** We first bound $\mathbb{P}\left( \boldsymbol{E}_1 \right)$ as follows:

$$\mathbb{P}\left( \boldsymbol{E}_1 \right) = \mathbb{P}\left( \sup_{\boldsymbol{w} \in \Omega} \left| \frac{1}{n} \sum_{i=1}^{n} \left( f(\boldsymbol{w}, \boldsymbol{x}_{(i)}) - f(\boldsymbol{w}_{k_w}, \boldsymbol{x}_{(i)}) \right) \right| \geq \frac{t}{3} \right)$$

$$\overset{①}{\leq} \frac{3}{t} \mathbb{E}\left( \sup_{\boldsymbol{w} \in \Omega} \left| \frac{1}{n} \sum_{i=1}^{n} \left( f(\boldsymbol{w}, \boldsymbol{x}_{(i)}) - f(\boldsymbol{w}_{k_w}, \boldsymbol{x}_{(i)}) \right) \right| \right)$$

$$\leq \frac{3}{t} \mathbb{E}\left( \sup_{\boldsymbol{w} \in \Omega} \frac{\left| \frac{1}{n} \sum_{i=1}^{n} \left( f(\boldsymbol{w}, \boldsymbol{x}_{(i)}) - f(\boldsymbol{w}_{k_w}, \boldsymbol{x}_{(i)}) \right) \right|}{\|\boldsymbol{w} - \boldsymbol{w}_{k_w}\|_2} \sup_{\boldsymbol{w} \in \Omega} \|\boldsymbol{w} - \boldsymbol{w}_{k_w}\|_2 \right)$$

$$\leq \frac{3\epsilon}{t} \mathbb{E}\left( \sup_{\boldsymbol{w} \in \Omega} \left\| \nabla \hat{\boldsymbol{J}}_n(\boldsymbol{w}, \boldsymbol{x}) \right\|_2 \right),$$

where ① holds since by Markov inequality, we have that for an arbitrary nonnegative random variable $x$, then

$$\mathbb{P}(x \geq t) \leq \frac{\mathbb{E}(x)}{t}.$$

Now we only need to bound $\mathbb{E}\left(\sup_{\boldsymbol{w}\in\Omega}\left\|\nabla\hat{\boldsymbol{J}}_n(\boldsymbol{w},\boldsymbol{x})\right\|_2\right)$. Therefore, by Lemma 10, we have

$$\mathbb{E}\left(\sup_{\boldsymbol{w}\in\Omega}\left\|\nabla\hat{\boldsymbol{J}}_n(\boldsymbol{w},\boldsymbol{x})\right\|_2\right) = \mathbb{E}\left(\sup_{\boldsymbol{w}\in\Omega}\left\|\frac{1}{n}\sum_{i=1}^{n}\nabla f(\boldsymbol{w},\boldsymbol{x}_{(i)})\right\|_2\right) = \mathbb{E}\left(\sup_{\boldsymbol{w}\in\Omega}\|\nabla f(\boldsymbol{w},\boldsymbol{x})\|_2\right) \leq \sqrt{\alpha_g}.$$

where $\alpha_g = c_t l r_x^4 r^{4l-2}$. Therefore, we have

$$\mathbb{P}\left(\boldsymbol{E}_1\right) \leq \frac{3\epsilon\sqrt{\alpha_g}}{t}.$$

We further let

$$t \geq \frac{6\epsilon\sqrt{\alpha_g}}{\varepsilon}.$$

Then we can bound $\mathbb{P}(\boldsymbol{E}_1)$:

$$\mathbb{P}(\boldsymbol{E}_1) \leq \frac{\varepsilon}{2}.$$

**Step 2. Bound $\mathbb{P}\left(\boldsymbol{E}_2\right)$:** Recall that we use $j_w$ to denote the index of $\boldsymbol{w}_{k_j}^j$ in $\epsilon$-net $\boldsymbol{w}_\epsilon^j$ and we have $j_w \in [n_\epsilon{}^j]$, $(n_\epsilon{}^j \leq \exp\left(\boldsymbol{s}_j \log\left(\frac{3rd}{\epsilon}\right)\right))$. We can bound $\mathbb{P}\left(\boldsymbol{E}_2\right)$ as follows:

$$\mathbb{P}\left(\boldsymbol{E}_2\right) = \mathbb{P}\left(\sup_{j_w\in[n_\epsilon^j]\,j\in[l]}\left|\frac{1}{n}\sum_{i=1}^{n}f(\boldsymbol{w}_{k_w},\boldsymbol{x}_{(i)}) - \mathbb{E}(f(\boldsymbol{w}_{k_w},\boldsymbol{x}))\right| \geq \frac{t}{3}\right)$$

$$\leq \exp\left(\sum_{j=1}^{l}\boldsymbol{s}_j\log\left(\frac{3rd}{\epsilon}\right)\right)\sup_{j_w\in[n_\epsilon^j]\,j\in[l]}\mathbb{P}\left(\left|\frac{1}{n}\sum_{i=1}^{n}f(\boldsymbol{w}_{k_w},\boldsymbol{x}_{(i)}) - \mathbb{E}(f(\boldsymbol{w}_{k_w},\boldsymbol{x}))\right| \geq \frac{t}{3}\right)$$

$$\overset{①}{\leq} 4\left(\frac{3dr}{\epsilon}\right)^s\exp\left(-c_{f'}n\min\left(\frac{t^2}{9\omega_f^2\max\left(\boldsymbol{d}_l\omega_f^2\tau^4,\tau^2\right)},\frac{t}{3\omega_f^2\tau^2}\right)\right),$$

where ① holds because in Lemma 7, we have

$$\mathbb{P}\left(\frac{1}{n}\sum_{i=1}^{n}\left(f(\boldsymbol{w},\boldsymbol{x}_{(i)}) - \mathbb{E}(f(\boldsymbol{w},\boldsymbol{x}_{(i)}))\right) > t\right) \leq 2\exp\left(-c_{f'}n\min\left(\frac{t^2}{\omega_f^2\max\left(\boldsymbol{d}_l\omega_f^2\tau^4,\tau^2\right)},\frac{t}{\omega_f^2\tau^2}\right)\right),$$

where $c_{f'}$ is a positive constant and $\omega_f = r^l$. Thus, if we set

$$t \geq \max\left(\sqrt{\frac{9\omega_f^2(s\log(3rd/\epsilon) + \log(8/\varepsilon))\max\left(\boldsymbol{d}_l\omega_f^2\tau^4,\tau^2\right)}{c_{f'}n}},\frac{3\omega_f^2\tau^2(s\log(3rd/\epsilon) + \log(8/\varepsilon))}{c_{f'}n}\right),$$

then we have

$$\mathbb{P}\left(\boldsymbol{E}_2\right) \leq \frac{\varepsilon}{2}.$$

**Step 3. Bound $\mathbb{P}\left(\boldsymbol{E}_3\right)$:** We first bound $\mathbb{P}\left(\boldsymbol{E}_3\right)$ as follows:

$$\mathbb{P}\left(\boldsymbol{E}_3\right) = \mathbb{P}\left(\sup_{\boldsymbol{w}\in\Omega}\|\mathbb{E}(f(\boldsymbol{w}_{k_w},\boldsymbol{x})) - \mathbb{E}(f(\boldsymbol{w},\boldsymbol{x}))\|_2 \geq \frac{t}{3}\right)$$

$$= \mathbb{P}\left(\sup_{\boldsymbol{w}\in\Omega}\frac{\|\mathbb{E}\left(f(\boldsymbol{w}_{k_w},\boldsymbol{x}) - f(\boldsymbol{w},\boldsymbol{x}\right)\|_2)}{\|\boldsymbol{w} - \boldsymbol{w}_{k_w}\|_2}\sup_{\boldsymbol{w}\in\Omega}\|\boldsymbol{w} - \boldsymbol{w}_{k_w}\|_2 \geq \frac{t}{3}\right)$$

$$\leq \mathbb{P}\left(\epsilon\mathbb{E}\sup_{\boldsymbol{w}\in\Omega}\|\nabla\boldsymbol{J}_{\boldsymbol{w}}(\boldsymbol{w},\boldsymbol{x})\|_2 \geq \frac{t}{3}\right)$$

$$\overset{①}{\leq} \mathbb{P}\left(\sqrt{\alpha_g}\epsilon \geq \frac{t}{3}\right),$$

where ① holds since we utilize Lemma 10. We set $\epsilon$ enough small such that $\sqrt{\alpha_g}\epsilon < t/3$ always holds. Then it yields $\mathbb{P}(\boldsymbol{E}_3) = 0$.

**Step 4. Final result**: To ensure $\mathbb{P}(\boldsymbol{E}_0) \leq \varepsilon$, we just set $\epsilon = 3rl/n$. Note that $\frac{6\sqrt{\alpha_g}\epsilon}{\varepsilon} > 3\sqrt{\alpha_g}\epsilon$. Thus we can obtain

$$
t \geq \max\left(\frac{6\sqrt{\alpha_g}\epsilon}{\varepsilon}, \sqrt{\frac{9\omega_f^2(s\log(3rd/\epsilon)+\log(8/\varepsilon))\max\left(\boldsymbol{d}_l\omega_f^2\tau^4, \tau^2\right)}{c_{f'}n}}, \frac{3\omega_f^2\tau^2(s\log(3rd/\epsilon)+\log(8/\varepsilon))}{c_{f'}n}\right)
$$

$$
= \max\left(\frac{18l\sqrt{\alpha_g}r}{n\varepsilon}, \sqrt{\frac{9\omega_f^2(s\log(dn/l)+\log(8/\varepsilon))\max\left(\boldsymbol{d}_l\omega_f^2\tau^4, \tau^2\right)}{c_{f'}n}}, \frac{3\omega_f^2\tau^2(s\log(dn/l)+\log(8/\varepsilon))}{c_{f'}n}\right).
$$

Note that we have $\alpha_g = c_t l r_x^4 r^{4l-2}$ where $c_t$ is a constant. Then Then there exist four universal constants $c_f$ and $c_{f'}$ such that if $n \geq c_{f'}\max\left(\frac{l^3 r_x^4}{\boldsymbol{d}_l s\log(d)\varepsilon^2\tau^4\log(1/\varepsilon)}, s\log(d)/(\tau^2\boldsymbol{d}_l)\right)$, then

$$
\sup_{\boldsymbol{w}\in\Omega}\left\|\hat{\boldsymbol{J}}_n(\boldsymbol{w}) - \boldsymbol{J}(\boldsymbol{w})\right\|_2 \leq c_f\omega_f\tau\max\left(\sqrt{\boldsymbol{d}_l}\omega_f\tau, 1\right)\sqrt{\frac{s\log(dn/l)+\log(8/\varepsilon)}{n}}
$$

holds with probability at least $1 - \varepsilon$. □

### C.3.4 PROOF OF COROLLARY 1

*Proof.* By Lemma 5, we know $\epsilon_s = \epsilon_g$. Thus, the remaining work is to bound $\epsilon_s$. Actually, we can have

$$
\left|\mathbb{E}_{\boldsymbol{S}\sim\mathcal{D},\boldsymbol{A},(\boldsymbol{x}'_{(j)},\boldsymbol{y}'_{(j)})\sim\mathcal{D}}\frac{1}{n}\sum_{j=1}^{n}\Big(f_j(\boldsymbol{w}_*^j;\boldsymbol{x}'_{(j)},\boldsymbol{y}'_{(j)})-f_j(\boldsymbol{w}^n;\boldsymbol{x}'_{(j)},\boldsymbol{y}'_{(j)})\Big)\right| \leq \mathbb{E}_{\boldsymbol{S}\sim\mathcal{D}}\left(\sup_{\boldsymbol{w}\in\Omega}\left|\hat{\boldsymbol{J}}_n(\boldsymbol{w}) - \boldsymbol{J}(\boldsymbol{w})\right|\right)
$$

$$
\leq \sup_{\boldsymbol{w}\in\Omega}\left|\hat{\boldsymbol{J}}_n(\boldsymbol{w}) - \boldsymbol{J}(\boldsymbol{w})\right|
$$

$$
\leq \epsilon_f.
$$

Thus, we have $\epsilon_g = \epsilon_s \leq \epsilon_f$. The proof is completed. □

### C.4 PROOF OF OTHER LEMMAS

### C.4.1 PROOF OF LEMMA 13

**Lemma 15.** *(Rigollet, 2015) Suppose a random variable $x$ is $\tau^2$-sub-Gaussian, then the random variable $x^2 - \mathbb{E}x^2$ is sub-exponential and obeys:*

$$
\mathbb{E}\left(\exp\lambda\left(x^2 - \mathbb{E}x^2\right)\right) \leq \exp\left(\frac{256\lambda^2\tau^4}{2}\right), \quad |\lambda| \leq \frac{1}{16\tau^2}. \tag{21}
$$

*Proof.* Here we utilize Lemma 15 to prove our conclusion. We have

$$
\mathbb{E}\exp\left(\lambda\left(\sum_{i=1}^{k}\boldsymbol{a}_i\boldsymbol{x}_i^2 - \mathbb{E}\left(\sum_{i=1}^{k}\boldsymbol{a}_i\boldsymbol{x}_i^2\right)\right)\right) \overset{①}{=} \prod_{i=1}^{k}\mathbb{E}\exp\left(\lambda\boldsymbol{a}_i\left(\boldsymbol{x}_i^2 - \mathbb{E}\boldsymbol{x}_i^2\right)\right)
$$

$$
\overset{②}{\leq} \prod_{i=1}^{k}\mathbb{E}\exp\left(128\lambda^2\boldsymbol{a}_i^2\tau_i^4\right), \quad |\lambda| \leq \frac{1}{\max_i\boldsymbol{a}_i\tau^2}
$$

$$
\leq \mathbb{E}\exp\left(128\lambda^2\tau^4\left(\sum_{i=1}^{k}\boldsymbol{a}_i^2\right)\right),
$$

where ① holds since $\boldsymbol{x}_i$ are independent and ② holds because of Lemma 15. □

### C.4.2 Proof of Lemma 14

*Proof.* Since the $\ell_2$-norm of each $\boldsymbol{w}_{(j)}$ is bounded, *i.e.* $\|\boldsymbol{w}_{(j)}\|_2 \le r \, (1 \le j \le l)$, we can obtain

$$\|\boldsymbol{B}_{s:t}\|_F^2 \le \left\|\boldsymbol{W}^{(s)}\right\|_F^2 \left\|\boldsymbol{W}^{(s-1)}\right\|_F^2 \cdots \left\|\boldsymbol{W}^{(t)}\right\|_F^2 \le r^{2(t-s+1)} \triangleq \omega_r^2 \overset{①}{\le} \max\left(r^2, r^{2l}\right),$$

where ① holds since the function $r^{2x}$ obtains its maximum at two endpoints $x = 1$ and $x = l$ for case $r < 1$ and $r \ge 1$, respectively. On the other hand, we have $\|\boldsymbol{B}_{s:t}\|_{\text{op}} \le \|\boldsymbol{B}_{s:t}\|_F \le \omega_r$. Specifically, we have $\|\boldsymbol{B}_{l:1}\|_F^2 \le r^{2l} \triangleq \omega_f^2$. $\qquad\square$

## D  Proofs for Deep Nonlinear Neural Networks

In this section, we first present the technical lemmas in Sec. D.1. Then in Sec. D.2 we give the proofs of these lemmas. Next, we utilize these technical lemmas to prove the results in Theorems $4 \sim 6$ and Corollary 2 in Sec. D.3. Finally, we give the proofs of other lemmas in Sec. D.4.

### D.1  Technical Lemmas

Here we present the key lemmas and theorems for proving our desired results. For brevity, we define an operation $\mathsf{G}$ which maps an arbitrary vector $\boldsymbol{z} \in \mathbb{R}^k$ into a diagonal matrix $\mathsf{G}(\boldsymbol{z}) \in \mathbb{R}^{k \times k}$ with its $i$-th diagonal entry equal to $\sigma(\boldsymbol{z}_i)(1 - \sigma(\boldsymbol{z}_i))$ in which $\boldsymbol{z}_i$ denotes the $i$-th entry of $\boldsymbol{z}$. We further define $\boldsymbol{A}_i \in \mathbb{R}^{d_{i-1} \times d_i}$ as follows:

$$\boldsymbol{A}_i = (\boldsymbol{W}^{(i)})^T \mathsf{G}(\boldsymbol{u}^{(i)}) \in \mathbb{R}^{d_{i-1} \times d_i} \quad (i = 1, \cdots, l), \tag{22}$$

where $\boldsymbol{W}^{(i)}$ is the weight matrix in the $i$-th layer and $\boldsymbol{u}^{(i)}$ is the linear output of the $i$-th layer. In this section, we define

$$\boldsymbol{B}_{s:t} = \boldsymbol{A}_s \boldsymbol{A}_{s+1} \cdots \boldsymbol{A}_t \in \mathbb{R}^{d_{s-1} \times d_t}, \, (s \le t) \quad \text{and} \quad \boldsymbol{B}_{s:t} = \boldsymbol{I}, \, (s > t). \tag{23}$$

**Lemma 16.** *Suppose that the activation function in deep neural network are sigmoid functions. Then the gradient of $f(\boldsymbol{w}, \boldsymbol{x})$ with respect to $\boldsymbol{w}_{(j)}$ can be formulated as*

$$\nabla_{\boldsymbol{w}_{(j)}} f(\boldsymbol{w}, \boldsymbol{x}) = \mathsf{vec}\left(\left(\mathsf{G}(\boldsymbol{u}^{(j)})\boldsymbol{B}_{j+1:l}(\boldsymbol{v}^{(l)} - \boldsymbol{y})\right)(\boldsymbol{v}^{(j-1)})^T\right), \, (j = 1, \cdots, l-1),$$

*and*

$$\nabla_{\boldsymbol{w}_{(l)}} f(\boldsymbol{w}, \boldsymbol{x}) = \mathsf{vec}\left(\left(\mathsf{G}(\boldsymbol{u}^{(l)})(\boldsymbol{v}^{(l)} - \boldsymbol{y})\right)(\boldsymbol{v}^{(l-1)})^T\right).$$

*Besides, the loss $f(\boldsymbol{w}, \boldsymbol{x})$ is $\alpha$-Lipschitz,*

$$\|\nabla_{\boldsymbol{w}} f(\boldsymbol{w}, \boldsymbol{x})\|_2 \le \alpha,$$

*where $\alpha = \sqrt{\frac{1}{16} c_y c_d (1 + c_r(l-1))}$ in which $c_y$, $c_d$ and $c_r$ are defined as*

$$\|\boldsymbol{v}^{(l)} - \boldsymbol{y}\|_2^2 \le c_y < +\infty, \quad c_d = \max(\boldsymbol{d}_0, \boldsymbol{d}_1, \cdots, \boldsymbol{d}_l) \quad \text{and} \quad c_r = \max\left(\frac{r^2}{16}, \left(\frac{r^2}{16}\right)^{l-1}\right).$$

**Lemma 17.** *Suppose that the activation functions in deep neural network are sigmoid functions. Then there exists two universal constants $c_{s_1}$ and $c_{s_2}$ such that*

$$\left\|\nabla_{\boldsymbol{w}}^2 f(\boldsymbol{w}, \boldsymbol{x})\right\|_{op} \le \left\|\nabla_{\boldsymbol{w}}^2 f(\boldsymbol{w}, \boldsymbol{x})\right\|_F \le \varsigma,$$

*where $\varsigma = \sqrt{c_{s_1} c_r c_d^2 l^4}$ in which $c_d = \max_i \boldsymbol{d}_i$ and $c_r = \max\left(\frac{r^2}{16}, \left(\frac{r^2}{16}\right)^{l-1}\right)$. Moreover, the gradient $\nabla_{\boldsymbol{w}} f(\boldsymbol{w}, \boldsymbol{x})$ is $\varsigma$-Lipschitz, i.e.*

$$\|\nabla_{\boldsymbol{w}} f(\boldsymbol{w}_1, \boldsymbol{x}) - \nabla_{\boldsymbol{w}} f(\boldsymbol{w}_2, \boldsymbol{x})\|_2 \le \varsigma \|\boldsymbol{w}_1 - \boldsymbol{w}_2\|_2.$$

*Similarly, there also exist a universal constant $\xi$ such that*

$$\left\|\nabla_{\boldsymbol{w}}^3 f(\boldsymbol{w}, \boldsymbol{x})\right\|_{op} \le \left\|\nabla_{\boldsymbol{w}}^3 f(\boldsymbol{w}, \boldsymbol{x})\right\|_F \le \xi.$$

**Lemma 18.** *Suppose that the activation function in deep neural network are sigmoid functions. Then we have*

$$\|\nabla_{\boldsymbol{w}}\nabla_{\boldsymbol{x}}f(\boldsymbol{w},\boldsymbol{x})\|_{op} \leq \|\nabla_{\boldsymbol{w}}\nabla_{\boldsymbol{x}}f(\boldsymbol{w},\boldsymbol{x})\|_F \leq \beta,$$

*where $\beta = \sqrt{\frac{2^6}{3^8}l(l+2)c_y c_r c_d\left(lc_r+1\right)}$ in which $c_y$, $c_d$ and $c_r$ are defined in Lemma 16.*

**Lemma 19.** *Suppose that the input sample $\boldsymbol{x}$ obeys Assumption 2 and the activation functions in deep neural network are sigmoid functions. The gradient of the loss is $8\beta^2\tau^2$-sub-Gaussian. Specifically, for any $\boldsymbol{\lambda} \in \mathbb{R}^d$, we have*

$$\mathbb{E}\left(\langle \boldsymbol{\lambda}, \nabla_{\boldsymbol{w}}f(\boldsymbol{w},\boldsymbol{x}) - \mathbb{E}\nabla_{\boldsymbol{w}}f(\boldsymbol{w},\boldsymbol{x})\rangle\right) \leq \exp\left(\frac{8\beta^2\tau^2\|\boldsymbol{\lambda}\|_2^2}{2}\right),$$

*where $\beta = \sqrt{\frac{2^6}{3^8}l(l+2)c_y c_r c_d\left(lc_r+1\right)}$ in which $c_y$, $c_d$ and $c_r$ are defined in Lemma 16.*

**Lemma 20.** *Suppose that the input sample $\boldsymbol{x}$ obeys Assumption 2 and the activation functions in deep neural network are sigmoid functions. The Hessian of the loss, evaluated on a unit vector, is sub-Gaussian. Specifically, for any unit $\boldsymbol{\lambda} \in \mathbb{S}^{d-1}$ (i.e. $\|\boldsymbol{\lambda}\|_2 = 1$), there exist universal constant $\gamma$ such that*

$$\mathbb{E}\left(t\left\langle \boldsymbol{\lambda}, \left(\nabla_{\boldsymbol{w}}^2 f(\boldsymbol{w},\boldsymbol{x}) - \mathbb{E}\nabla_{\boldsymbol{w}}^2 f(\boldsymbol{w},\boldsymbol{x})\right)\boldsymbol{\lambda}\right\rangle\right) \leq \exp\left(\frac{8t^2\gamma^2\tau^2}{2}\right).$$

*Notice, $\gamma$ obeys $\gamma \geq \|\nabla_{\boldsymbol{x}}\nabla_{\boldsymbol{w}}^2 f(\boldsymbol{w},\boldsymbol{x})\|_{op}$.*

**Lemma 21.** *Assume that the input sample $\boldsymbol{x}$ obeys Assumption 2 and the activation functions in deep neural network are sigmoid functions. Then the sample Hessian uniformly converges to the population Hessian in operator norm. That is, there exists such two universal constants $c_{m'}$ and $c_m$ such that if $n \geq \frac{c_{m'}\xi^2 l^2 r^2}{\gamma^2\tau^2\varepsilon^2 s\log(d)\log(1/\varepsilon)}$, then*

$$\sup_{\boldsymbol{w}\in\Omega}\left\|\nabla^2\hat{\boldsymbol{J}}_n(\boldsymbol{w}) - \nabla^2\boldsymbol{J}(\boldsymbol{w})\right\|_{op} \leq c_m\gamma\tau\sqrt{\frac{s\log(dn/l)+\log(4/\varepsilon)}{n}}$$

*holds with probability at least $1-\varepsilon$. Here $\gamma$ is the same parameter in Lemma 20.*

### D.2 PROOFS OF TECHNICAL LEMMAS

For brevity, we also define

$$\boldsymbol{D}_{s:t} = \|\boldsymbol{W}^{(s)}\|_F^2 \cdots \|\boldsymbol{W}^{(t)}\|_F^2 \ (s \leq t) \quad \text{and} \quad \boldsymbol{D}_{s:t} = 1, \ (s > t).$$

We define a matrix $\boldsymbol{P}_k \in \mathbb{R}^{d_k^2 \times d_k}$ whose $((s-1)\boldsymbol{d}_k+s,s)$ $(s = 1,\cdots,\boldsymbol{d}_k)$ entry equal to $\sigma(\boldsymbol{u}_s^{(k)})(1-\sigma(\boldsymbol{u}_s^{(k)}))(1-2\sigma(\boldsymbol{u}_s^{(k)}))$ and rest entries are all 0. On the other hand, since the values in $\boldsymbol{v}^{(l)}$ belong to the range $[0,1]$ and $\boldsymbol{y}$ is the label, $\|\boldsymbol{v}^{(l)}-\boldsymbol{y}\|_2^2$ can be bounded:

$$\|\boldsymbol{v}^{(l)}-\boldsymbol{y}\|_2^2 \leq c_y < +\infty,$$

where $c_y$ is a universal constant. We further define $c_d = \max(\boldsymbol{d}_0,\boldsymbol{d}_1,\cdots,\boldsymbol{d}_l)$.

Then we give a lemma to summarize the properties of $\mathrm{G}(\boldsymbol{u}^{(i)})$ defined in Eqn. (22), $\boldsymbol{B}_{s:t}$ defined in Eqn. (23), $\boldsymbol{D}_{s:t}$ and $\boldsymbol{P}_k$.

**Lemma 22.** *For $\mathrm{G}(\boldsymbol{u}^{(i)})$ defined in Eqn. (22), $\boldsymbol{B}_{s:t}$ defined in Eqn. (23), $\boldsymbol{D}_{s:t}$ and $\boldsymbol{P}_k$, we have the following properties:*

*(1) For arbitrary matrices $\boldsymbol{M}$ and $\boldsymbol{N}$ of proper sizes, we have*

$$\|\mathrm{G}(\boldsymbol{u}^{(i)})\boldsymbol{M}\|_F^2 \leq \frac{1}{16}\|\boldsymbol{M}\|_F^2 \quad \text{and} \quad \|\boldsymbol{N}\mathrm{G}(\boldsymbol{u}^{(i)})\|_F^2 \leq \frac{1}{16}\|\boldsymbol{N}\|_F^2.$$

*(2) For arbitrary matrices $\boldsymbol{M}$ and $\boldsymbol{N}$ of proper sizes, we have*

$$\|\boldsymbol{P}_k\boldsymbol{M}\|_F^2 \leq \frac{2^6}{3^8}\|\boldsymbol{M}\|_F^2 \quad \text{and} \quad \|\boldsymbol{N}\boldsymbol{P}_k\|_F^2 \leq \frac{2^6}{3^8}\|\boldsymbol{N}\|_F^2.$$

*(3) For arbitrary matrices $\boldsymbol{M}$ and $\boldsymbol{N}$ of proper sizes, we have*

$$\|\boldsymbol{B}_{s:t}\|_F^2 \le \frac{1}{16^{t-s+1}}\boldsymbol{D}_{s:t} \quad and \quad \frac{1}{16^{t-s+1}}\boldsymbol{D}_{s:t} \le c_{st} \le c_r,$$

*where $c_{st} = \left(\frac{r}{4}\right)^{2(t-s+1)}$ and $c_r = \max\left(\frac{r^2}{16}, \left(\frac{r^2}{16}\right)^{l-1}\right)$.*

*(4) For arbitrary matrices $\boldsymbol{M}$, $\boldsymbol{N}$ and $\boldsymbol{I}$ of proper sizes, let $\boldsymbol{m} = \text{vec}\,(\boldsymbol{M})$. Then we have*

$$\|(\boldsymbol{N}\otimes\boldsymbol{I})\boldsymbol{m}\|_F^2 \le \|\boldsymbol{M}\|_F^2\,\|\boldsymbol{N}\|_F^2 \quad and \quad \|(\boldsymbol{I}\otimes\boldsymbol{N})\boldsymbol{m}\|_F^2 \le \|\boldsymbol{M}\|_F^2\,\|\boldsymbol{N}\|_F^2\,.$$

It should be pointed out that we defer the proof of Lemma 22 to Sec. D.4.

### D.2.1 PROOF OF LEMMA 16

*Proof.* We use chain rule to compute the gradient of $f(\boldsymbol{w},\boldsymbol{x})$ with respect to $\boldsymbol{w}_{(j)}$. We first compute several basis gradient. According to the relationship between $\boldsymbol{u}^{(j)}, \boldsymbol{v}^{(j)}, \boldsymbol{W}^{(j)}$ and $\boldsymbol{f}(\boldsymbol{w},\boldsymbol{x})$, we have

$$
\begin{aligned}
\nabla_{\boldsymbol{v}^{(l)}} f(\boldsymbol{w},\boldsymbol{x}) &= \boldsymbol{v}^{(l)} - \boldsymbol{y},\\
\nabla_{\boldsymbol{v}^{(i)}} f(\boldsymbol{w},\boldsymbol{x}) &= \frac{\partial \boldsymbol{u}^{(i+1)}}{\partial \boldsymbol{v}^{(i)}}\frac{\partial f(\boldsymbol{w},\boldsymbol{x})}{\partial \boldsymbol{u}^{(i+1)}} = (\boldsymbol{W}^{(i+1)})^T \frac{\partial f(\boldsymbol{w},\boldsymbol{x})}{\partial \boldsymbol{u}^{(i+1)}}, & (i=1,\cdots,l-1),\\
\nabla_{\boldsymbol{u}^{(i)}} f(\boldsymbol{w},\boldsymbol{x}) &= \frac{\partial \boldsymbol{v}^{(i)}}{\partial \boldsymbol{u}^{(i)}}\frac{\partial f(\boldsymbol{w},\boldsymbol{x})}{\partial \boldsymbol{v}^{(i)}} = \mathsf{G}(\boldsymbol{u}^{(i)})\frac{\partial f(\boldsymbol{w},\boldsymbol{x})}{\partial \boldsymbol{v}^{(i)}}, & (i=1,\cdots,l),\\
\nabla_{\boldsymbol{W}^{(i)}} f(\boldsymbol{w},\boldsymbol{x}) &= \frac{\partial \boldsymbol{u}^{(i)}}{\partial \boldsymbol{w}_{(i)}}\left(\frac{\partial f(\boldsymbol{w},\boldsymbol{x})}{\partial \boldsymbol{u}^{(i)}}\right)^T = \boldsymbol{v}^{(i-1)}\left(\frac{\partial f(\boldsymbol{w},\boldsymbol{x})}{\partial \boldsymbol{u}^{(i)}}\right)^T, & (i=1,\cdots,l).
\end{aligned}
\tag{24}
$$

Then by chain rule, we can easily compute the gradient of $f(\boldsymbol{w},\boldsymbol{x})$ with respect to $\boldsymbol{w}_{(j)}$ which is formulated as

$$\nabla_{\boldsymbol{w}_{(j)}} f(\boldsymbol{w},\boldsymbol{x}) = \text{vec}\left(\boldsymbol{v}^{(j-1)}\left(\mathsf{G}(\boldsymbol{u}^{(j)})\boldsymbol{A}_{j+1}\boldsymbol{A}_{j+2}\cdots\boldsymbol{A}_l(\boldsymbol{v}^{(l)} - \boldsymbol{y})\right)^T\right), \; (j=1,\cdots,l-1),$$

and

$$\nabla_{\boldsymbol{w}_{(l)}} f(\boldsymbol{w},\boldsymbol{x}) = \text{vec}\left(\boldsymbol{v}^{(l-1)}\left(\mathsf{G}(\boldsymbol{u}^{(l)})(\boldsymbol{v}^{(l)} - \boldsymbol{y})\right)^T\right).$$

Besides, since the values in $\boldsymbol{v}^{(l)}$ belong to the range $[0,1]$. Combine with Lemma 22, we can bound $\|\nabla_{\boldsymbol{w}} f(w,x)\|_2$ as follows:

$$
\begin{aligned}
\|\nabla_{\boldsymbol{w}} f(\boldsymbol{w},\boldsymbol{x})\|_2^2 &= \sum_{j=1}^{l}\left\|\nabla_{\boldsymbol{w}_{(j)}} f(\boldsymbol{w},\boldsymbol{x})\right\|_2^2 \\
&= \left\|\boldsymbol{v}^{(l-1)}\left(\mathsf{G}(\boldsymbol{u}^{(l)})(\boldsymbol{v}^{(l)} - \boldsymbol{y})\right)^T\right\|_F^2 + \sum_{j=1}^{l-1}\left\|\boldsymbol{v}^{(j-1)}\left(\mathsf{G}(\boldsymbol{u}^{(j)})\boldsymbol{B}_{j+1:l}(\boldsymbol{v}^{(l)} - \boldsymbol{y})\right)^T\right\|_F^2 \\
&\le \frac{1}{16}\boldsymbol{d}_{l-1}\left\|\boldsymbol{v}^{(l)} - \boldsymbol{y}\right\|_2^2 + \frac{1}{16}\left\|\boldsymbol{v}^{(l)} - \boldsymbol{y}\right\|_2^2\sum_{j=1}^{l-1}\boldsymbol{d}_{j-1}\|\boldsymbol{B}_{j+1:l}\|_F^2 \\
&\overset{\textcircled{1}}{\le} \frac{1}{16}c_y c_d + \frac{1}{16}c_y c_d c_r(l-1),
\end{aligned}
$$

where $c_y, c_d, c_r$ are defined as

$$\|\boldsymbol{v}^{(l)} - \boldsymbol{y}\|_2^2 \le c_y, \quad c_d = \max(\boldsymbol{d}_0, \boldsymbol{d}_1, \cdots, \boldsymbol{d}_l) \quad \text{and} \quad c_r = \max\left(\frac{r^2}{16}, \left(\frac{r^2}{16}\right)^{l-1}\right).$$

Notice, $\textcircled{1}$ holds since in Lemma 22, we have

$$\|\boldsymbol{B}_{s:t}\|_F^2 \le \left(\frac{r}{4}\right)^{2(t-s+1)} \le \max\left(\frac{r^2}{16}, \left(\frac{r^2}{16}\right)^{l-1}\right).$$

Thus, we can obtain

$$\|\nabla_{\boldsymbol{w}} f(w, x)\|_2 \leq \sqrt{\frac{1}{16} c_y c_d \left(1 + c_r(l-1)\right)} \triangleq \alpha.$$

The proof is completed. $\qquad\square$

### D.2.2  Proof of Lemma 17

For convenience, we first give the computation of some gradients.

**Lemma 23.** *Assume the activation functions in deep neural network are sigmoid functions. Then the following properties hold:*

*(1) We can compute the gradients $\frac{\partial f(\boldsymbol{w},\boldsymbol{x})}{\partial \boldsymbol{u}^{(i)}}$ and $\frac{\partial f(\boldsymbol{w},\boldsymbol{x})}{\partial \boldsymbol{v}^{(i)}}$ as*

$$\frac{\partial f(\boldsymbol{w}, \boldsymbol{x})}{\partial \boldsymbol{u}^{(i)}} = \mathsf{G}(\boldsymbol{u}^{(i)}) \boldsymbol{B}_{i+1:l}(\boldsymbol{v}^{(l)} - \boldsymbol{y}) \quad and \quad \frac{\partial f(\boldsymbol{w}, \boldsymbol{x})}{\partial \boldsymbol{v}^{(i)}} = \boldsymbol{B}_{i+1:l}(\boldsymbol{v}^{(l)} - \boldsymbol{y}).$$

*(2) We can compute the gradient $\frac{\partial \boldsymbol{u}^{(i)}}{\partial \boldsymbol{w}_{(j)}}$ as*

$$\frac{\partial \boldsymbol{u}^{(i)}}{\partial \boldsymbol{w}_{(j)}} = (\boldsymbol{v}^{(j-1)})^T \otimes \left( \mathsf{G}(\boldsymbol{u}^{(j)}) \boldsymbol{B}_{j+1:i-1} (\boldsymbol{W}^{(i)})^T \right)^T \in \mathbb{R}^{d_i \times d_j d_{j-1}}, \ (i > j).$$

$$\frac{\partial \boldsymbol{u}^{(i)}}{\partial \boldsymbol{w}_{(i)}} = (\boldsymbol{v}^{(i-1)})^T \otimes \boldsymbol{I}_{d_i} \in \mathbb{R}^{d_i \times d_i d_{i-1}}, \ (i = j).$$

*(3) We can compute the gradient $\frac{\partial \boldsymbol{v}^{(i)}}{\partial \boldsymbol{w}_{(j)}}$ as*

$$\frac{\partial \boldsymbol{v}^{(i)}}{\partial \boldsymbol{w}_{(j)}} = (\boldsymbol{v}^{(j-1)})^T \otimes \left( \mathsf{G}(\boldsymbol{u}^{(j)}) \boldsymbol{B}_{j+1:i} \right)^T \in \mathbb{R}^{d_i \times d_j d_{j-1}}, \ (i \geq j).$$

It should be pointed out that the proof of Lemma 23 can be founded Sec. D.4.

*Proof.* To prove our conclusion, we have two steps: computing the Hessian and bounding its operation norm.

**Step 1. Compute the Hessian**: We first consider the computation of $\frac{\partial^2 f(\boldsymbol{w},\boldsymbol{x})}{\partial \boldsymbol{w}_{(i)}^T \partial \boldsymbol{w}_{(j)}}$:

$$\frac{\partial^2 f(\boldsymbol{w}, \boldsymbol{x})}{\partial \boldsymbol{w}_{(i)}^T \partial \boldsymbol{w}_{(j)}} = \frac{\partial \left( \mathsf{vec} \left( \left( \mathsf{G}(\boldsymbol{u}^{(j)}) \boldsymbol{A}_{j+1} \boldsymbol{A}_{j+2} \cdots \boldsymbol{A}_l (\boldsymbol{v}^{(l)} - \boldsymbol{y}) \right) (\boldsymbol{v}^{(j-1)})^T \right) \right)}{\partial \boldsymbol{w}_{(i)}^T}.$$

Recall that we define

$$\boldsymbol{B}_{s:t} = \boldsymbol{A}_s \boldsymbol{A}_{s+1} \cdots \boldsymbol{A}_t \in \mathbb{R}^{d_{s-1} \times d_t}, \ (s \leq t) \quad and \quad \boldsymbol{B}_{s:t} = \boldsymbol{I}, \ (s > t).$$

Then we have

$$\frac{\partial^2 f(\boldsymbol{w}, \boldsymbol{x})}{\partial \boldsymbol{w}_{(i)}^T \partial \boldsymbol{w}_{(j)}} = \left( \boldsymbol{v}^{(j-1)} (\boldsymbol{v}^{(l)} - \boldsymbol{y})^T \boldsymbol{B}_{j+1:l}^T \right) \otimes \left( \boldsymbol{I}_{d_j} \right) \frac{\partial \mathsf{vec}\left( \mathsf{G}(\boldsymbol{u}^{(j)}) \right)}{\partial \boldsymbol{w}_{(i)}^T} (\triangleq \boldsymbol{Q}_1^{ij})$$

$$+ \sum_{k=j+1}^{l} \left( \boldsymbol{v}^{(j-1)} (\boldsymbol{v}^{(l)} - \boldsymbol{y})^T \boldsymbol{B}_{k+1:l}^T \right) \otimes \left( \mathsf{G}(\boldsymbol{u}^{(j)}) \boldsymbol{B}_{j+1:k-1} \boldsymbol{W}_k^T \right) \frac{\partial \mathsf{vec}\left( \mathsf{G}(\boldsymbol{u}^{(k)}) \right)}{\partial \boldsymbol{w}_{(i)}^T} (\triangleq \boldsymbol{Q}_2^{ij})$$

$$+ \left( \boldsymbol{v}^{(j-1)} (\boldsymbol{v}^{(l)} - \boldsymbol{y})^T \boldsymbol{B}_{i+1:l}^T \mathsf{G}(\boldsymbol{u}^{(i)}) \right) \otimes \left( \mathsf{G}(\boldsymbol{u}^{(j)}) \boldsymbol{B}_{j+1:i-1} \right) \frac{\partial \mathsf{vec}\left( \boldsymbol{W}_i^T \right)}{\partial \boldsymbol{w}_{(i)}^T} (\triangleq \boldsymbol{Q}_3^{ij})$$

$$+ \boldsymbol{v}^{(j-1)} \otimes \left( \mathsf{G}(\boldsymbol{u}^{(j)}) \boldsymbol{B}_{j+1:l} \right) \frac{\partial (\boldsymbol{v}^{(l)} - \boldsymbol{y})}{\partial \boldsymbol{w}_{(i)}^T} (\triangleq \boldsymbol{Q}_4^{ij})$$

$$+ \boldsymbol{I}_{d_{j-1}} \otimes \left( \mathsf{G}(\boldsymbol{u}^{(j)}) \boldsymbol{B}_{j+1:l} (\boldsymbol{v}^{(l)} - \boldsymbol{y}) \right) \frac{\partial \boldsymbol{v}^{(j-1)}}{\partial \boldsymbol{w}_{(i)}^T} (\triangleq \boldsymbol{Q}_5^{ij})$$

**Case I:** $i > j$. We first consider the case that $i > j$. In this is case, $\boldsymbol{Q}_1^{ij} = \boldsymbol{0}$ since $\frac{\partial \mathrm{vec}\big(\mathsf{G}(\boldsymbol{u}^{(j)})\big)}{\partial \boldsymbol{w}_{(i)}^T} = \boldsymbol{0}$.

Computing $\boldsymbol{Q}_2^{ij}$ needs more efforts. By utilizing the computation of $\frac{\partial \boldsymbol{u}^{(k)}}{\partial \boldsymbol{w}_{(i)}}$ in Lemma 23, we have

$$\frac{\partial \mathrm{vec}\big(\mathsf{G}(\boldsymbol{u}^{(k)})\big)}{\partial \boldsymbol{w}_{(i)}} = \frac{\partial \mathrm{vec}\big(\mathsf{G}(\boldsymbol{u}^{(k)})\big)}{\partial \boldsymbol{u}^{(k)}} \frac{\partial \boldsymbol{u}^{(k)}}{\partial \boldsymbol{w}_{(i)}} = \boldsymbol{P}_k \left( \boldsymbol{v}^{(i-1)T} \otimes \Big( \mathsf{G}(\boldsymbol{u}^{(i)}) \boldsymbol{B}_{i+1:k-1} (\boldsymbol{W}^{(k)})^T \Big)^T \right), (k > i)$$

where $\boldsymbol{P}_k$ is a matrix of size $\boldsymbol{d}_k^2 \times \boldsymbol{d}_k$ whose $((s-1)\boldsymbol{d}_k + s, s)$ $(s = 1, \cdots, \boldsymbol{d}_k)$ entry equal to $\sigma(\boldsymbol{u}_s^{(k)})(1 - \sigma(\boldsymbol{u}_s^{(k)}))(1 - 2\sigma(\boldsymbol{u}_s^{(k)}))$ and rest entries are all 0. When $k = i$,

$$\frac{\partial \mathrm{vec}\big(\mathsf{G}(\boldsymbol{u}^{(k)})\big)}{\partial \boldsymbol{w}_{(k)}} = \frac{\partial \mathrm{vec}\big(\mathsf{G}(\boldsymbol{u}^{(k)})\big)}{\partial \boldsymbol{u}^{(k)}} \frac{\partial \boldsymbol{u}^{(k)}}{\partial \boldsymbol{w}_{(k)}} = \boldsymbol{P}_k \left( (\boldsymbol{v}^{(k-1)})^T \otimes \boldsymbol{I}_{\boldsymbol{d}_k} \right) \in \mathbb{R}^{\boldsymbol{d}_k^2 \times \boldsymbol{d}_k \boldsymbol{d}_{k-1}}.$$

Note that for $k < i$, we have $\frac{\partial \mathsf{G}(\boldsymbol{u}^{(k)})}{\partial \boldsymbol{w}_{(i)}} = \boldsymbol{0}$. For brevity, let

$$\boldsymbol{D}_k \triangleq \left( \Big( \boldsymbol{v}^{(j-1)} (\boldsymbol{v}^{(l)} - \boldsymbol{y})^T \boldsymbol{B}_{k+1:l}^T \Big) \otimes \Big( \mathsf{G}(\boldsymbol{u}^{(j)}) \boldsymbol{B}_{j+1:k-1} \boldsymbol{W}_k^T \Big) \right) \ (k = i, \cdots, l). \tag{25}$$

Therefore, we have

$$\boldsymbol{Q}_2^{ij} = \boldsymbol{D}_i \boldsymbol{P}_i \left( (\boldsymbol{v}^{(i-1)})^T \otimes \boldsymbol{I}_{\boldsymbol{d}_i} \right) + \sum_{k=i+1}^{l} \boldsymbol{D}_k \boldsymbol{P}_k \left( (\boldsymbol{v}^{(i-1)})^T \otimes \Big( \mathsf{G}(\boldsymbol{u}^{(i)}) \boldsymbol{B}_{i+1:k-1} (\boldsymbol{W}^{(k)})^T \Big)^T \right).$$

Then we consider $\boldsymbol{Q}_3^{ij}$.

$$\boldsymbol{Q}_3^{ij} = \left( \boldsymbol{v}^{(j-1)} (\boldsymbol{v}^{(l)} - \boldsymbol{y})^T \boldsymbol{B}_{i+1:l}^T \mathsf{G}(\boldsymbol{u}^{(i)}) \right) \otimes \left( \mathsf{G}(\boldsymbol{u}^{(j)}) \boldsymbol{B}_{j+1:i-1} \right).$$

Also we can use the computation of $\frac{\partial \boldsymbol{v}^{(l)}}{\partial \boldsymbol{w}_{(i)}}$ in Lemma 23 and compute $\boldsymbol{Q}_4^{ij}$ as follows:

$$\begin{aligned} \boldsymbol{Q}_4^{ij} &= \boldsymbol{v}^{(j-1)} \otimes \left( \mathsf{G}(\boldsymbol{u}^{(j)}) \boldsymbol{B}_{j+1:l} \right) \frac{\partial (\boldsymbol{v}^{(l)} - \boldsymbol{y})}{\partial \boldsymbol{w}_{(i)}^T} \\ &= \left( \boldsymbol{v}^{(j-1)} \otimes \left( \mathsf{G}(\boldsymbol{u}^{(j)}) \boldsymbol{B}_{j+1:l} \right) \right) \left( (\boldsymbol{v}^{(i-1)})^T \otimes \left( \mathsf{G}(\boldsymbol{u}^{(i)}) \boldsymbol{B}_{i+1:l} \right)^T \right). \end{aligned}$$

Finally, since $i > j$, we can compute $\boldsymbol{Q}_5^{ij} = \boldsymbol{0}$.

**Case II:** $i = j$. We first consider $\frac{\partial \mathsf{G}(\boldsymbol{u}^{(k)})}{\partial \boldsymbol{w}_{(k)}}$:

$$\frac{\partial \mathrm{vec}\big(\mathsf{G}(\boldsymbol{u}^{(k)})\big)}{\partial \boldsymbol{w}_{(k)}^T} = \frac{\partial \mathrm{vec}\big(\mathsf{G}(\boldsymbol{u}^{(k)})\big)}{\partial \boldsymbol{u}^{(k)}} \frac{\partial \boldsymbol{u}^{(k)}}{\partial \boldsymbol{w}_{(k)}^T} = \boldsymbol{P}_k \left( (\boldsymbol{v}^{(k-1)})^T \otimes \boldsymbol{I}_{\boldsymbol{d}_k} \right) \in \mathbb{R}^{\boldsymbol{d}_k^2 \times \boldsymbol{d}_k \boldsymbol{d}_{k-1}},$$

where $\boldsymbol{P}_k$ is a matrix of size $\boldsymbol{d}_k^2 \times \boldsymbol{d}_k$ whose $(s, (s-1)\boldsymbol{d}_k + s)$ entry equal to $\sigma(\boldsymbol{u}_s^{(k)})(1 - \sigma(\boldsymbol{u}_s^{(k)}))(1 - 2\sigma(\boldsymbol{u}_s^{(k)}))$ and rest entries are all 0. $\boldsymbol{Q}_1^{jj}$ can be computed as

$$\begin{aligned} \boldsymbol{Q}_1^{jj} &= \left( \boldsymbol{v}^{(j-1)} (\boldsymbol{v}^{(l)} - \boldsymbol{y})^T \boldsymbol{B}_{j+1:l}^T \right) \otimes \left( \boldsymbol{I}_{\boldsymbol{d}_j} \right) \frac{\partial \mathrm{vec}\big(\mathsf{G}(\boldsymbol{u}^{(j)})\big)}{\partial \boldsymbol{w}_{(j)}^T} \\ &= \left( \left( \boldsymbol{v}^{(j-1)} (\boldsymbol{v}^{(l)} - \boldsymbol{y})^T \boldsymbol{B}_{j+1:l}^T \right) \otimes \left( \boldsymbol{I}_{\boldsymbol{d}_j} \right) \right) \left( \boldsymbol{P}_j \left( (\boldsymbol{v}^{(j-1)})^T \otimes \boldsymbol{I}_{\boldsymbol{d}_j} \right) \right). \end{aligned}$$

As for $\boldsymbol{Q}_2^{jj}$, by Eqn. (25) we have

$$\boldsymbol{Q}_2^{jj} = \sum_{k=j+1}^{l} \boldsymbol{D}_k \boldsymbol{P}_k \left( \boldsymbol{v}^{(j-1)T} \otimes \Big( \mathsf{G}(\boldsymbol{u}^{(j)}) \boldsymbol{B}_{j+1:k-1} (\boldsymbol{W}^{(k)})^T \Big)^T \right).$$

Since $i = j$, $\boldsymbol{Q}_3^{jj}$ does not exist. For convenience, we just set $\boldsymbol{Q}_3^{jj} = \boldsymbol{0}$.

Now we consider $\boldsymbol{Q}_4^{jj}$ which can be computed as follows:

$$
\begin{aligned}
\boldsymbol{Q}_4^{jj} =& \boldsymbol{v}^{(j-1)} \otimes \left( \mathsf{G}(\boldsymbol{u}^{(j)}) \boldsymbol{B}_{j+1:l} \right) \frac{\partial(\boldsymbol{v}^{(l)} - \boldsymbol{y})}{\partial \boldsymbol{w}_{(j)}^T} \\
=& \left( \boldsymbol{v}^{(j-1)} \otimes \left( \mathsf{G}(\boldsymbol{u}^{(j)}) \boldsymbol{B}_{j+1:l} \right) \right) \left( (\boldsymbol{v}^{(j-1)})^T \otimes \left( \mathsf{G}(\boldsymbol{u}^{(j)}) \boldsymbol{B}_{j+1:l} \right)^T \right).
\end{aligned}
$$

Finally, since $i = j$, we can compute $\boldsymbol{Q}_5^{jj} = \boldsymbol{0}$.

**Case III:** $i < j$. Since $\frac{\partial^2 f(\boldsymbol{w}, \boldsymbol{x})}{\partial \boldsymbol{w} \partial \boldsymbol{w}^T}$ is symmetrical, we have $\boldsymbol{Q}_k^{ij} = \boldsymbol{Q}_k^{ji}$ $(k = 1, \cdots, 5)$.

**Step 2. Bound the operation norm of Hessian:** We mainly use Lemma 22 to achieve this goal. From Lemma 22, we have

(1) For arbitrary matrices $\boldsymbol{M}$ and $\boldsymbol{N}$ of proper size, we have

$$
\|\mathsf{G}(\boldsymbol{u}^{(i)})\boldsymbol{M}\|_F^2 \le \frac{1}{16}\|\boldsymbol{M}\|_F^2 \quad \text{and} \quad \|\boldsymbol{N}\mathsf{G}(\boldsymbol{u}^{(i)})\|_F^2 \le \frac{1}{16}\|\boldsymbol{N}\|_F^2.
$$

(2) For arbitrary matrices $\boldsymbol{M}$ and $\boldsymbol{N}$ of proper size, we have

$$
\|\boldsymbol{P}_k \boldsymbol{M}\|_F^2 \le \frac{2^6}{3^8}\|\boldsymbol{M}\|_F^2 \quad \text{and} \quad \|\boldsymbol{N}\boldsymbol{P}_k\|_F^2 \le \frac{2^6}{3^8}\|\boldsymbol{N}\|_F^2.
$$

(3) For $\boldsymbol{B}_{s:t}$ and $\boldsymbol{D}_{s:t}$, we have

$$
\|\boldsymbol{B}_{s:t}\|_F^2 \le \frac{1}{16^{t-s+1}}\boldsymbol{D}_{s:t} \quad \text{and} \quad \frac{1}{16^{t-s+1}}\boldsymbol{D}_{s:t} \le c_r,
$$

where $c_r = \max\left( \frac{r^2}{16}, \left(\frac{r^2}{16}\right)^l \right)$.

(4) For arbitrary matrices $\boldsymbol{M}$, $\boldsymbol{N}$ and $\boldsymbol{I}$ of proper sizes, let $\boldsymbol{m} = \mathsf{vec}\,(\boldsymbol{M})$. Then we have

$$
\|(\boldsymbol{N} \otimes \boldsymbol{I})\boldsymbol{m}\|_F^2 \le \|\boldsymbol{M}\|_F^2 \|\boldsymbol{N}\|_F^2 \quad \text{and} \quad \|(\boldsymbol{I} \otimes \boldsymbol{N})\boldsymbol{m}\|_F^2 \le \|\boldsymbol{M}\|_F^2 \|\boldsymbol{N}\|_F^2.
$$

The values of entries in $\boldsymbol{v}^{(h)}$ are bounded by $0 \le \sigma(\boldsymbol{u}_h^{(i)}) \le 1$ which leads to $\left\|\boldsymbol{v}^{(h)}\right\|_F^2 \le d_h \le c_d$, where $c_d = \max_i \boldsymbol{d}_i$. On the other hand, since the values in $\boldsymbol{v}^{(l)}$ belong to the range $[0, 1]$ and $\boldsymbol{y}$ is the label, $\|\boldsymbol{v}^{(l)} - \boldsymbol{y}\|_2^2$ can be bounded:

$$
\|\boldsymbol{v}^{(l)} - \boldsymbol{y}\|_2^2 \le c_y < +\infty,
$$

where $c_y$ is a universal constant.

We first define

$$
\boldsymbol{C}_k^{ij} = \boldsymbol{D}_k \boldsymbol{P}_k \left( \boldsymbol{v}^{(i-1)} \right)^T \otimes \left( \mathsf{G}(\boldsymbol{u}^{(i)}) \boldsymbol{B}_{i+1:k-1} (\boldsymbol{W}^{(k)})^T \right)^T \right)
$$

and

$$
\begin{aligned}
\boldsymbol{C}^{ij} =& \boldsymbol{D}_i \boldsymbol{P}_i \left( (\boldsymbol{v}^{(i-1)})^T \otimes \boldsymbol{I}_{d_i} \right) = \left( \left( \boldsymbol{v}^{(i-1)} \otimes \boldsymbol{I}_{d_i} \right) (\boldsymbol{D}_i \boldsymbol{P}_i)^T \right)^T \overset{\text{①}}{=} \left( \boldsymbol{v}^{(i-1)} \otimes (\boldsymbol{D}_i \boldsymbol{P}_i)^T \right)^T \\
=& (\boldsymbol{v}^{(i-1)})^T \otimes (\boldsymbol{D}_i \boldsymbol{P}_i),
\end{aligned}
$$

where $\boldsymbol{D}_k$ is defined in Eqn. (25). ① holds since for an arbitrary vector $\boldsymbol{u} \in \mathbb{R}^k$ and an arbitrary matrix $\boldsymbol{M} \in \mathbb{R}^{k \times k}$, we have $(\boldsymbol{u} \otimes \boldsymbol{I}_k) \boldsymbol{M} = \boldsymbol{u} \otimes \boldsymbol{M}$.

**Case I:** $i > j$. According to the definition of $\boldsymbol{C}^{ij}$ and $\boldsymbol{C}_k^{ij}$, we have $\boldsymbol{Q}_2^{ij} = \boldsymbol{C}^{ij} + \sum_{k=i+1}^l \boldsymbol{C}_k^{ij}$. So we have

$$
\begin{aligned}
\left\| \frac{\partial^2 f(\boldsymbol{w}, \boldsymbol{x})}{\partial \boldsymbol{w}_{(i)}^T \partial \boldsymbol{w}_{(j)}} \right\|_F^2 &= \left\| \boldsymbol{Q}_1^{ij} + \boldsymbol{Q}_2^{ij} + \boldsymbol{Q}_3^{ij} + \boldsymbol{Q}_4^{ij} + \boldsymbol{Q}_5^{ij} \right\|_F^2 \\
&= \left\| \boldsymbol{C}^{ij} + \sum_{k=i+1}^l \boldsymbol{C}_k^{ij} + \boldsymbol{Q}_3^{ij} + \boldsymbol{Q}_4^{ij} \right\|_F^2 \\
&= (l - i + 3) \left( \left\| \boldsymbol{C}^{ij} \right\|_F^2 + \sum_{k=i+1}^l \left\| \boldsymbol{C}_k^{ij} \right\|_F^2 + \left\| \boldsymbol{Q}_3^{ij} \right\|_F^2 + \left\| \boldsymbol{Q}_4^{ij} \right\|_F^2 \right).
\end{aligned}
$$

Here we bound each term separately:

$$\left\|\boldsymbol{C}^{ij}\right\|_F^2 \leq \left\|\boldsymbol{v}^{(j-1)}\right\|_F^2 \left\|\boldsymbol{v}^{(l)}-\boldsymbol{y}\right\|_F^2 \|\boldsymbol{B}_{i+1:l}\|_F^2 \frac{1}{16} \left\|\boldsymbol{B}_{j+1:i-1}\boldsymbol{W}_i^T\right\|_F^2 \frac{2^6}{3^8} \left\|\boldsymbol{v}^{(i-1)}\right\|_F^2$$

$$\leq \frac{2^6}{3^8} c_y \boldsymbol{d}_{j-1} \boldsymbol{d}_{i-1} \frac{1}{16^{l-i}} \boldsymbol{D}_{i+1:l} \frac{1}{16^{i-j}} \boldsymbol{D}_{j+1:i}$$

$$\leq \frac{2^6}{3^8} c_y \boldsymbol{d}_{j-1} \boldsymbol{d}_{i-1} \frac{1}{16^{l-j}} \boldsymbol{D}_{j+1:l}$$

$$\leq \frac{2^6}{3^8} c_y \boldsymbol{d}_{j-1} \boldsymbol{d}_{i-1} c_r.$$

Similarly, we can bound $\|\boldsymbol{C}_k^{ij}\|_F^2$ as follows:

$$\left\|\boldsymbol{C}_k^{ij}\right\|_F^2$$

$$\leq \left\|\boldsymbol{v}^{(j-1)}\right\|_F^2 \left\|\boldsymbol{v}^{(l)}-\boldsymbol{y}\right\|_F^2 \|\boldsymbol{B}_{k+1:l}\|_F^2 \frac{1}{16} \left\|\boldsymbol{B}_{j+1:k-1}\boldsymbol{W}_k^T\right\|_F^2 \frac{2^6}{3^8} \left\|\boldsymbol{v}^{(i-1)}\right\|_F^2 \frac{1}{16} \left\|\boldsymbol{B}_{i+1:k-1}(\boldsymbol{W}^{(k)})^T\right\|_F^2$$

$$\leq \frac{2^6}{3^8} c_y \boldsymbol{d}_{j-1} \boldsymbol{d}_{i-1} \frac{1}{16^{l-k}} \boldsymbol{D}_{k+1:l} \frac{1}{16^{k-j-1}} \boldsymbol{D}_{j+1:k} \frac{1}{16^{k-i-1}} \boldsymbol{D}_{i+1:k}$$

$$= \frac{2^6}{3^8} c_y \boldsymbol{d}_{j-1} \boldsymbol{d}_{i-1} \frac{1}{16^{l-j-1}} \boldsymbol{D}_{j+1:l} \frac{1}{16^{k-i-1}} \boldsymbol{D}_{i+1:k}$$

$$\leq \frac{2^{14}}{3^8} c_y \boldsymbol{d}_{j-1} \boldsymbol{d}_{i-1} c_r^2.$$

We also bound $\left\|\boldsymbol{Q}_3^{ij}\right\|_F^2$ as

$$\left\|\boldsymbol{Q}_3^{ij}\right\|_F^2 \leq \left\|\boldsymbol{v}^{(j-1)}\right\|_F^2 \left\|\boldsymbol{v}^{(l)}-\boldsymbol{y}\right\|_F^2 \frac{1}{16} \|\boldsymbol{B}_{i+1:l}\|_F^2 \frac{1}{16} \|\boldsymbol{B}_{j+1:i-1}\|_F^2 \leq \frac{1}{2^8} c_y \boldsymbol{d}_{j-1} c_r.$$

Finally, we bound $\left\|\boldsymbol{Q}_4^{ij}\right\|_F^2$ as follows:

$$\left\|\boldsymbol{Q}_4^{ij}\right\|_F^2 \leq \left\|\boldsymbol{v}^{(j-1)}\right\|_F^2 \frac{1}{16} \|\boldsymbol{B}_{j+1:l}\|_F^2 \left\|\boldsymbol{v}^{(i-1)}\right\|_F^2 \frac{1}{16} \|\boldsymbol{B}_{i+1:l}\|_F^2 \leq \frac{1}{2^8} \boldsymbol{d}_{j-1} \boldsymbol{d}_{i-1} c_r^2.$$

Note that $\boldsymbol{d}_i \leq c_d$. Thus, we can bound $\left\|\frac{\partial^2 f(\boldsymbol{w},\boldsymbol{x})}{\partial \boldsymbol{w}_{(j)} \partial \boldsymbol{w}_{(i)}^T}\right\|_F^2$ as

$$\left\|\frac{\partial^2 f(\boldsymbol{w},\boldsymbol{x})}{\partial \boldsymbol{w}_{(i)}^T \partial \boldsymbol{w}_{(j)}}\right\|_F^2$$

$$\leq (l-i+3)\left(\frac{2^6}{3^8} c_y \boldsymbol{d}_{j-1} \boldsymbol{d}_{i-1} c_r + \sum_{k=i+1}^{l} \frac{2^{14}}{3^8} c_y \boldsymbol{d}_{j-1} \boldsymbol{d}_{i-1} c_r^2 + \frac{1}{2^8} c_y \boldsymbol{d}_{j-1} c_r + \frac{1}{2^8} \boldsymbol{d}_{j-1} \boldsymbol{d}_{i-1} c_r^2\right)$$

$$\leq (l+1)\left(\frac{64}{6561} c_y c_d^2 c_r + \frac{4096}{6561} c_y (l-2) c_d^2 c_r^2 + \frac{1}{256} c_y c_d c_r + \frac{1}{256} c_d c_r^2\right).$$

**Case II:** $i = j$. According to the definition of $\boldsymbol{C}^{ij}$ and $\boldsymbol{C}_k^{ij}$, we have $\boldsymbol{Q}_2^{jj} = \sum_{k=j+1}^{l} \boldsymbol{C}_k^{jj}$.

Similarly, we have

$$\left\|\frac{\partial^2 f(\boldsymbol{w},\boldsymbol{x})}{\partial \boldsymbol{w}_{(i)}^T \partial \boldsymbol{w}_{(j)}}\right\|_F^2 = \left\|\boldsymbol{Q}_1^{jj} + \boldsymbol{Q}_2^{jj} + \boldsymbol{Q}_3^{jj} + \boldsymbol{Q}_4^{jj} + \boldsymbol{Q}_5^{jj}\right\|_F^2 = \left\|\boldsymbol{Q}_1^{jj} + \sum_{k=j+1}^{l} \boldsymbol{C}_k^{jj} + \boldsymbol{Q}_4^{ij}\right\|_F^2$$

$$\leq (l-j+2)\left(\left\|\boldsymbol{Q}_1^{jj}\right\|_F^2 + \sum_{k=j+1}^{l} \left\|\boldsymbol{C}_k^{jj}\right\|_F^2 + \left\|\boldsymbol{Q}_4^{jj}\right\|_F^2\right).$$

Thus, we can bound $\left\|\boldsymbol{Q}_1^{jj}\right\|_F^2$ first:

$$\left\|\boldsymbol{Q}_1^{jj}\right\|_F^2 \le \left\|\boldsymbol{v}^{(j-1)}\right\|_F^2 \left\|\boldsymbol{v}^{(l)} - \boldsymbol{y}\right\|_F^2 \|\boldsymbol{B}_{j+1:l}\|_F^2 \frac{2^6}{3^8} \left\|\boldsymbol{v}^{(j-1)}\right\|_F^2 \le \frac{2^6}{3^8} c_y \boldsymbol{d}_{j-1}^2 c_r.$$

As for $\boldsymbol{Q}_2^{jj}$, we have

$$\left\|\boldsymbol{C}_k^{ij}\right\|_F^2$$

$$\le \left\|\boldsymbol{v}^{(j-1)}\right\|_F^2 \left\|\boldsymbol{v}^{(l)} - \boldsymbol{y}\right\|_F^2 \|\boldsymbol{B}_{k+1:l}\|_F^2 \frac{1}{16} \|\boldsymbol{B}_{j+1:k-1} \boldsymbol{W}_k^T\|_F^2 \frac{2^6}{3^8} \left\|\boldsymbol{v}^{(j-1)}\right\|_F^2 \frac{1}{16} \left\|\boldsymbol{B}_{j+1:k-1} (\boldsymbol{W}^{(k)})^T\right\|_F^2$$

$$= \frac{2^6}{3^8} c_y \boldsymbol{d}_{j-1}^2 \frac{1}{16^{l-k}} \boldsymbol{D}_{k+1:l} \frac{1}{16^{k-j-1}} \boldsymbol{D}_{j+1:k} \frac{1}{16^{k-j-1}} \boldsymbol{D}_{j+1:k}$$

$$\le \frac{2^{14}}{3^8} c_y \boldsymbol{d}_{j-1}^2 c_r^2.$$

Then we bound $\|\boldsymbol{Q}_4^{jj}\|_F^2$:

$$\left\|\boldsymbol{Q}_4^{jj}\right\|_F^2 \le \left\|\boldsymbol{v}^{(j-1)}\right\|_F^2 \frac{1}{16} \|\boldsymbol{B}_{j+1:l}\|_F^2 \left\|\boldsymbol{v}^{(j-1)}\right\|_F^2 \frac{1}{16} \|\boldsymbol{B}_{j+1:l}\|_F^2 \le \frac{1}{2^8} \boldsymbol{d}_{j-1}^2 c_r^2.$$

Note that for any input, we have $c_v = \max_j \left\|\boldsymbol{v}^{(j-1)}(\boldsymbol{v}^{(l)} - \boldsymbol{y})^T\right\|_F^2 \le \max_j \|\boldsymbol{v}^{(j-1)}\|_F^2 \left\|(\boldsymbol{v}^{(l)} - \boldsymbol{y}\right\|_F^2 \le c_y c_d$, where $\left\|\boldsymbol{v}^{(l)} - \boldsymbol{y}\right\|_F^2$ can be bounded by a constant $c_y$. Thus, we can bound $\left\|\frac{\partial^2 f(\boldsymbol{w},\boldsymbol{x})}{\partial \boldsymbol{w}_{(i)}^T \partial \boldsymbol{w}_{(j)}}\right\|_F^2$ as

$$\left\|\frac{\partial^2 f(\boldsymbol{w},\boldsymbol{x})}{\partial \boldsymbol{w}_{(i)}^T \partial \boldsymbol{w}_{(j)}}\right\|_F^2 \le (l-i+3)\left(\frac{2^6}{3^8} c_y \boldsymbol{d}_{j-1}^2 c_r + \sum_{k=i+1}^{l} \frac{2^{14}}{3^8} c_y \boldsymbol{d}_{j-1}^2 c_r^2 + \frac{1}{2^8} \boldsymbol{d}_{j-1}^2 c_r^2\right)$$

$$\le (l+2)\left(\frac{64}{6561} c_y c_d^2 c_r + \frac{4096}{6561} c_y (l-1) c_d^2 c_r^2 + \frac{1}{256} c_d^2 c_r^2\right).$$

**Case III:** $i < j$. Since $\frac{\partial^2 f(\boldsymbol{w},\boldsymbol{x})}{\partial \boldsymbol{w} \partial \boldsymbol{w}^T}$ is symmetrical, we have $\boldsymbol{Q}_k^{ij} = \boldsymbol{Q}_k^{ji}$ ($k = 1, \cdots, 5$). Thus, it yields

$$\left\|\frac{\partial^2 f(\boldsymbol{w},\boldsymbol{x})}{\partial \boldsymbol{w}_{(i)}^T \partial \boldsymbol{w}_{(j)}}\right\|_F^2 \le (l+1)\left(\frac{64}{6561} c_y c_d^2 c_r + \frac{4096}{6561} c_y (l-2) c_d^2 c_r^2 + \frac{1}{256} c_y c_d c_r + \frac{1}{256} c_d c_r^2\right).$$

**Final result**: Thus we can bound

$$\left\|\nabla_{\boldsymbol{w}}^2 f(\boldsymbol{w},\boldsymbol{x})\right\|_{\text{op}} \le \left\|\nabla_{\boldsymbol{w}}^2 f(\boldsymbol{w},\boldsymbol{x})\right\|_F$$

$$\le \sqrt{(l-1)l \max_{i,j:i\neq j} \left\|\frac{\partial^2 f(\boldsymbol{w},\boldsymbol{x})}{\partial \boldsymbol{w}_{(j)} \partial \boldsymbol{w}_{(i)}^T}\right\|_F^2 + \sum_{j=1}^{l} \left\|\frac{\partial^2 f(\boldsymbol{w},\boldsymbol{x})}{\partial \boldsymbol{w}_{(j)} \partial \boldsymbol{w}_{(i)}^T}\right\|_F^2}$$

$$\le \left((l-1)l(l+1)\left(\frac{64}{6561} c_y c_d^2 c_r + \frac{4096}{6561} c_y (l-2) c_d^2 c_r^2 + \frac{1}{256} c_y c_d c_r + \frac{1}{256} c_d c_r^2\right)\right.$$

$$\left. + (l+2)\left(\frac{64}{6561} c_y c_d^2 c_r + \frac{4096}{6561} c_y (l-1) l c_d^2 c_r^2 + \frac{1}{256} l c_d^2 c_r^2\right)\right)^{\frac{1}{2}}$$

$$\le \sqrt{c_{s_1} c_r c_d^2 l^4},$$

where $c_{s_1}$ and $c_{s_2}$ are two constants.

Since $\left\|\nabla_{\boldsymbol{w}}^2 f(\boldsymbol{w},\boldsymbol{x})\right\|_{\text{op}} \le \left\|\nabla_{\boldsymbol{w}}^2 f(\boldsymbol{w},\boldsymbol{x})\right\|_F$, we know that the gradient $\nabla_{\boldsymbol{w}} f(\boldsymbol{w},\boldsymbol{x})$ is $\varsigma$-Lipschitz, where $\varsigma = \sqrt{c_{s_1} c_r c_d^2 l^4}$.

On the other hand, since for any input $\boldsymbol{x}$, $\sigma(\boldsymbol{x})$ belongs to $[0, 1]$, the values of the entries of $\nabla_{\boldsymbol{w}}^3 f(\boldsymbol{w}, \boldsymbol{x})$ can be bounded. Thus, we can bound

$$\|\nabla_{\boldsymbol{w}}^3 f(\boldsymbol{w}, \boldsymbol{x})\|_{\text{op}} = \sup_{\|\boldsymbol{\lambda}\|_2 \leq 1} \left\langle \boldsymbol{\lambda}^{\otimes^3}, \nabla_{\boldsymbol{w}}^3 f(\boldsymbol{w}, \boldsymbol{x}) \right\rangle = [\nabla_{\boldsymbol{w}}^3 f(\boldsymbol{w}, \boldsymbol{x})]_{ijk} \boldsymbol{\lambda}_i \boldsymbol{\lambda}_j \boldsymbol{\lambda}_k \leq \xi < +\infty.$$

We complete the proof. $\qquad\qquad\qquad\qquad\qquad\qquad\qquad\qquad\qquad\qquad\qquad\qquad\qquad\square$

### D.2.3    PROOF OF LEMMA 18

For convenience, we first give the computation of some gradients.

**Lemma 24.** *Assume the activation functions in deep neural network are sigmoid functions. Then we can compute the gradients $\frac{\partial \boldsymbol{u}^{(j)}}{\partial \boldsymbol{u}^{(1)}}$ and $\frac{\partial \boldsymbol{v}^{(j)}}{\partial \boldsymbol{u}^{(1)}}$ as*

$$\frac{\partial \boldsymbol{u}^{(j)}}{\partial \boldsymbol{u}^{(1)}} = \left( \mathsf{G}(\boldsymbol{u}^{(1)}) \boldsymbol{A}_2 \cdots \boldsymbol{A}_{j-1} (\boldsymbol{W}^j)^T \right)^T \in \mathbb{R}^{d_j \times d_1}, \; (j > 1).$$

$$\frac{\partial \boldsymbol{v}^{(j)}}{\partial \boldsymbol{u}^{(1)}} = \left( \mathsf{G}(\boldsymbol{u}^{(1)}) \boldsymbol{A}_2 \cdots \boldsymbol{A}_j \right)^T \in \mathbb{R}^{d_j \times d_1}, \; (j > 1).$$

It should be pointed out that the proof of Lemma 24 can be founded Sec. D.4.

*Proof.* To prove our conclusion, we have two steps: computing $\nabla_{\boldsymbol{x}} \nabla_{\boldsymbol{w}} f(\boldsymbol{w}, \boldsymbol{x})$ and bounding its operation norm.

**Step 1. Compute $\nabla_{\boldsymbol{x}} \nabla_{\boldsymbol{w}} f(\boldsymbol{w}, \boldsymbol{x})$:**

We first consider the computation of $\frac{\partial^2 f(\boldsymbol{w}, \boldsymbol{x})}{\partial \boldsymbol{x}^T \partial \boldsymbol{w}_{(j)}}$:

$$\frac{\partial^2 f(\boldsymbol{w}, \boldsymbol{x})}{\partial \boldsymbol{x}^T \partial \boldsymbol{w}_{(j)}} = \frac{\partial \left( \text{vec} \left( \left( \mathsf{G}(\boldsymbol{u}^{(j)}) \boldsymbol{A}_{j+1} \boldsymbol{A}_{j+2} \cdots \boldsymbol{A}_l (\boldsymbol{v}^{(l)} - \boldsymbol{y}) \right) (\boldsymbol{v}^{(j-1)})^T \right) \right)}{\partial \boldsymbol{x}^T}.$$

Recall that we define

$$\boldsymbol{A}_i = (\boldsymbol{W}^{(i)})^T \mathsf{G}(\boldsymbol{u}^{(i)}) \in \mathbb{R}^{d_{i-1} \times d_i}.$$

$$\boldsymbol{B}_{s:t} = \boldsymbol{A}_s \boldsymbol{A}_{s+1} \cdots \boldsymbol{A}_t \in \mathbb{R}^{d_{s-1} \times d_t}, \; (s \leq t) \quad \text{and} \quad \boldsymbol{B}_{s:t} = \boldsymbol{I}, \; (s > t).$$

Then we have

$$\frac{\partial^2 f(\boldsymbol{w}, \boldsymbol{x})}{\partial \boldsymbol{x}^T \partial \boldsymbol{w}_{(j)}} = \left( \boldsymbol{v}^{(j-1)} (\boldsymbol{v}^{(l)} - \boldsymbol{y})^T \boldsymbol{B}_{j+1:l}^T \right) \otimes \left( \boldsymbol{I}_{d_j} \right) \frac{\partial \text{vec} \left( \mathsf{G}(\boldsymbol{u}^{(j)}) \right)}{\partial \boldsymbol{x}^T} (\triangleq \boldsymbol{Q}_1^j)$$

$$+ \sum_{k=j+1}^{l} \left( \boldsymbol{v}^{(j-1)} (\boldsymbol{v}^{(l)} - \boldsymbol{y})^T \boldsymbol{B}_{k+1:l}^T \right) \otimes \left( \mathsf{G}(\boldsymbol{u}^{(j)}) \boldsymbol{B}_{j+1:k-1} \boldsymbol{W}_k^T \right) \frac{\partial \text{vec} \left( \mathsf{G}(\boldsymbol{u}^{(k)}) \right)}{\partial \boldsymbol{x}^T} (\triangleq \boldsymbol{Q}_2^j)$$

$$+ \boldsymbol{v}^{(j-1)} \otimes \left( \mathsf{G}(\boldsymbol{u}^{(j)}) \boldsymbol{B}_{j+1:l} \right) \frac{\partial (\boldsymbol{v}^{(l)} - \boldsymbol{y})}{\partial \boldsymbol{x}^T} (\triangleq \boldsymbol{Q}_3^j)$$

$$+ \boldsymbol{I}_{d_{j-1}} \otimes \left( \mathsf{G}(\boldsymbol{u}^{(j)}) \boldsymbol{B}_{j+1:l} (\boldsymbol{v}^{(l)} - \boldsymbol{y}) \right) \frac{\partial \boldsymbol{v}^{(j-1)}}{\partial \boldsymbol{x}^T} (\triangleq \boldsymbol{Q}_4^j)$$

By using Lemma 24, we can compute $\boldsymbol{Q}_1^{ij}$ as

$$\frac{\partial \text{vec} \left( \mathsf{G}(\boldsymbol{u}^{(k)}) \right)}{\partial \boldsymbol{x}^T} = \frac{\partial \text{vec} \left( \mathsf{G}(\boldsymbol{u}^{(k)}) \right)}{\partial \boldsymbol{u}^{(k)}} \frac{\partial \boldsymbol{u}^{(k)}}{\partial \boldsymbol{x}^T} = \boldsymbol{P}_k \left( \mathsf{G}(\boldsymbol{u}^{(1)}) \boldsymbol{B}_{2:k-1} (\boldsymbol{W}^k)^T \right)^T.$$

Thus, we have

$$\boldsymbol{Q}_1^j = \left( \boldsymbol{v}^{(j-1)} (\boldsymbol{v}^{(l)} - \boldsymbol{y})^T \boldsymbol{B}_{j+1:l}^T \right) \otimes \boldsymbol{I}_{d_j} \frac{\partial \text{vec} \left( \mathsf{G}(\boldsymbol{u}^{(j)}) \right)}{\partial \boldsymbol{x}^T}$$

$$= \left( \left( \boldsymbol{v}^{(j-1)} (\boldsymbol{v}^{(l)} - \boldsymbol{y})^T \boldsymbol{B}_{j+1:l}^T \right) \otimes \boldsymbol{I}_{d_j} \right) \boldsymbol{P}_k \left( \mathsf{G}(\boldsymbol{u}^{(1)}) \boldsymbol{B}_{2:k-1} (\boldsymbol{W}^k)^T \right)^T.$$

As for $\boldsymbol{Q}_2^j$, we also can utilize Lemma 24 to compute it:

$$
\begin{aligned}
\boldsymbol{Q}_2^j &= \sum_{k=j+1}^{l} \left( \boldsymbol{v}^{(j-1)} (\boldsymbol{v}^{(l)} - \boldsymbol{y})^T \boldsymbol{B}_{k+1:l}^T \right) \otimes \left( \text{G}(\boldsymbol{u}^{(j)}) \boldsymbol{B}_{j+1:k-1} \boldsymbol{W}_k^T \right) \frac{\partial \text{vec} \left( \text{G}(\boldsymbol{u}^{(k)}) \right)}{\partial \boldsymbol{x}^T} \\
&= \sum_{k=i+1}^{l} \left( \left( \boldsymbol{v}^{(j-1)} (\boldsymbol{v}^{(l)} - \boldsymbol{y})^T \boldsymbol{B}_{k+1:l}^T \right) \otimes \left( \text{G}(\boldsymbol{u}^{(j)}) \boldsymbol{B}_{j+1:k-1} \boldsymbol{W}_k^T \right) \right) \boldsymbol{P}_k \left( \text{G}(\boldsymbol{u}^{(1)}) \boldsymbol{B}_{2:k-1} (\boldsymbol{W}^k)^T \right)^T .
\end{aligned}
$$

Then we consider $\boldsymbol{Q}_3^{ij}$.

$$
\boldsymbol{Q}_3^j = \boldsymbol{v}^{(j-1)} \otimes \left( \text{G}(\boldsymbol{u}^{(j)}) \boldsymbol{B}_{j+1:l} \right) \frac{\partial (\boldsymbol{v}^{(l)} - \boldsymbol{y})}{\partial \boldsymbol{x}^T} = \left( \boldsymbol{v}^{(j-1)} \otimes \left( \text{G}(\boldsymbol{u}^{(j)}) \boldsymbol{B}_{j+1:l} \right) \right) \left( \text{G}(\boldsymbol{u}^{(1)}) \boldsymbol{B}_{2:l} \right)^T .
$$

$\boldsymbol{Q}_4^j$ can be computed as follows:

$$
\boldsymbol{Q}_4^j = \boldsymbol{I}_{d_{j-1}} \otimes \left( \text{G}(\boldsymbol{u}^{(j)}) \boldsymbol{B}_{j+1:l} (\boldsymbol{v}^{(l)} - \boldsymbol{y}) \right) \frac{\partial \boldsymbol{v}^{(j-1)}}{\partial \boldsymbol{x}^T} = \left( \boldsymbol{I}_{d_{j-1}} \otimes \left( \text{G}(\boldsymbol{u}^{(j)}) \boldsymbol{B}_{j+1:l} (\boldsymbol{v}^{(l)} - \boldsymbol{y}) \right) \right) \left( \text{G}(\boldsymbol{u}^{(1)}) \boldsymbol{B}_{2:j} \right)^T .
$$

**Step 2. Bound the operation norm of Hessian**: We mainly use Lemma 22 to achieve this goal. From Lemma 22, we have

(1) For arbitrary matrices $\boldsymbol{M}$ and $\boldsymbol{N}$ of proper size, we have

$$
\| \text{G}(\boldsymbol{u}^{(i)}) \boldsymbol{M} \|_F^2 \leq \frac{1}{16} \| \boldsymbol{M} \|_F^2 \quad \text{and} \quad \| \boldsymbol{N} \text{G}(\boldsymbol{u}^{(i)}) \|_F^2 \leq \frac{1}{16} \| \boldsymbol{N} \|_F^2 .
$$

(2) For arbitrary matrices $\boldsymbol{M}$ and $\boldsymbol{N}$ of proper size, we have

$$
\| \boldsymbol{P}_k \boldsymbol{M} \|_F^2 \leq \frac{2^6}{3^8} \| \boldsymbol{M} \|_F^2 \quad \text{and} \quad \| \boldsymbol{N} \boldsymbol{P}_k \|_F^2 \leq \frac{2^6}{3^8} \| \boldsymbol{N} \|_F^2 .
$$

(3) For $\boldsymbol{B}_{s:t}$ and $\boldsymbol{D}_{s:t}$, we have

$$
\| \boldsymbol{B}_{s:t} \|_F^2 \leq \frac{1}{16^{t-s+1}} \boldsymbol{D}_{s:t} \quad \text{and} \quad \frac{1}{16^{t-s+1}} \boldsymbol{D}_{s:t} \leq c_r ,
$$

where $c_r = \max \left( \frac{r^2}{4}, \left( \frac{r^2}{16} \right)^{l-1} \right)$.

(4) For arbitrary matrices $\boldsymbol{M}$, $\boldsymbol{N}$ and $\boldsymbol{I}$ of proper sizes, let $\boldsymbol{m} = \text{vec}(\boldsymbol{M})$. Then we have

$$
\| (\boldsymbol{N} \otimes \boldsymbol{I}) \boldsymbol{m} \|_F^2 \leq \| \boldsymbol{M} \|_F^2 \| \boldsymbol{N} \|_F^2 \quad \text{and} \quad \| (\boldsymbol{I} \otimes \boldsymbol{N}) \boldsymbol{m} \|_F^2 \leq \| \boldsymbol{M} \|_F^2 \| \boldsymbol{N} \|_F^2 .
$$

The values of entries in $\boldsymbol{v}^{(h)}$ are bounded by $0 \leq \sigma(\boldsymbol{u}_h^{(i)}) \leq 1$ which leads to $\| \boldsymbol{v}^{(h)} \|_F^2 \leq \boldsymbol{d}_h \leq c_d$, where $c_d = \max_i \boldsymbol{d}_i$. On the other hand, since the values in $\boldsymbol{v}^{(l)}$ belong to the range $[0, 1]$ and $\boldsymbol{y}$ is the label, $\| \boldsymbol{v}^{(l)} - \boldsymbol{y} \|_2^2$ can be bounded:

$$
\| \boldsymbol{v}^{(l)} - \boldsymbol{y} \|_2^2 \leq c_y < +\infty ,
$$

where $c_y$ is a universal constant.

We first define

$$
\boldsymbol{C}_k^j = \left( \left( \boldsymbol{v}^{(j-1)} (\boldsymbol{v}^{(l)} - \boldsymbol{y})^T \boldsymbol{B}_{k+1:l}^T \right) \otimes \left( \text{G}(\boldsymbol{u}^{(j)}) \boldsymbol{B}_{j+1:k-1} \boldsymbol{W}_k^T \right) \right) \boldsymbol{P}_k \left( \text{G}(\boldsymbol{u}^{(1)}) \boldsymbol{B}_{2:k-1} (\boldsymbol{W}^k)^T \right)^T .
$$

Then we have $\boldsymbol{Q}_2^j = \sum_{k=j+1}^{l} \boldsymbol{C}_k^j$. So we have

$$
\begin{aligned}
\left\| \frac{\partial^2 f(\boldsymbol{w}, \boldsymbol{x})}{\partial \boldsymbol{x}^T \partial \boldsymbol{w}_{(j)}} \right\|_F^2 &= \left\| \boldsymbol{Q}_1^j + \boldsymbol{Q}_2^j + \boldsymbol{Q}_3^j + \boldsymbol{Q}_4^j \right\|_F^2 = \left\| \boldsymbol{Q}_1^j + \sum_{k=j+1}^{l} \boldsymbol{C}_k^j + \boldsymbol{Q}_3^j + \boldsymbol{Q}_4^j \right\|_F^2 \\
&= (l - j + 3) \left( \left\| \boldsymbol{Q}_1^j \right\|_F^2 + \sum_{k=j+1}^{l} \left\| \boldsymbol{C}_k^j \right\|_F^2 + \left\| \boldsymbol{Q}_3^j \right\|_F^2 + \left\| \boldsymbol{Q}_4^j \right\|_F^2 \right) .
\end{aligned}
$$

Then we bound each term separately:

$$\left\|\boldsymbol{Q}_1^j\right\|_F^2 \leq \left\|\boldsymbol{v}^{(j-1)}\right\|_F^2 \left\|\boldsymbol{v}^{(l)} - \boldsymbol{y}\right\|_F^2 \|\boldsymbol{B}_{j+1:l}\|_F^2 \frac{2^6}{3^8}\frac{1}{16}\left\|\boldsymbol{B}_{2:k-1}(\boldsymbol{W}^k)^T\right\|_F^2 \leq \frac{2^6}{3^8}c_y\boldsymbol{d}_{j-1}c_r^2.$$

Similarly, we bound $\left\|\boldsymbol{C}_k^j\right\|_F^2$:

$$\left\|\boldsymbol{C}_k^j\right\|_F^2 = \left\|\boldsymbol{v}^{(j-1)}\right\|_F^2 \left\|\boldsymbol{v}^{(l)} - \boldsymbol{y}\right\|_F^2 \|\boldsymbol{B}_{k+1:l}\|_F^2 \frac{1}{16}\left\|\boldsymbol{B}_{j+1:k-1}\boldsymbol{W}_k^T\right\|_F^2 \frac{2^6}{3^8}\frac{1}{16}\left\|\boldsymbol{B}_{2:k-1}(\boldsymbol{W}^{(k)})^T\right\|_F^2$$

$$= \frac{2^6}{3^8}c_y\boldsymbol{d}_{j-1}\frac{1}{16^{l-k}}\boldsymbol{D}_{k+1:l}\frac{1}{16^{k-j-1}}\boldsymbol{D}_{j+1:k}\frac{1}{16^{k-1}}\boldsymbol{D}_{2:k}$$

$$\leq \frac{2^6}{3^8}c_y\boldsymbol{d}_{j-1}c_r^2.$$

We also bound $\left\|\boldsymbol{Q}_3^{ij}\right\|_F^2$ as

$$\left\|\boldsymbol{Q}_3^{ij}\right\|_F^2 \leq \left\|\boldsymbol{v}^{(j-1)}\right\|_2^2 \frac{1}{16}\|\boldsymbol{B}_{j+1:l}\|_F^2 \frac{1}{16}\|\boldsymbol{B}_{2:l}\|_F^2 \leq \frac{1}{2^8}\boldsymbol{d}_{j-1}c_r^2.$$

Finally, we bound $\left\|\boldsymbol{Q}_4^j\right\|_F^2$ as follows:

$$\left\|\boldsymbol{Q}_4^j\right\|_F^2 = \frac{1}{16}\|\boldsymbol{B}_{j+1:l}\|_F^2 \left\|\boldsymbol{v}^{(l)} - \boldsymbol{y}\right\|_F^2 \frac{1}{16}\|\boldsymbol{B}_{2:j}\|_F^2 \leq \frac{1}{2^8}c_yc_r.$$

Since $c_d = \max_i \boldsymbol{d}_i$, we can bound $\left\|\frac{\partial^2 f(\boldsymbol{w},\boldsymbol{x})}{\partial\boldsymbol{w}_{(j)}\partial\boldsymbol{x}^T}\right\|_F^2$ as

$$\left\|\frac{\partial^2 f(\boldsymbol{w},\boldsymbol{x})}{\partial\boldsymbol{x}^T\partial\boldsymbol{w}_{(j)}}\right\|_F^2 \leq (l-j+3)\left(\frac{2^6}{3^8}c_y\boldsymbol{d}_{j-1}c_r^2 + \sum_{k=j+1}^l \frac{2^6}{3^8}c_y\boldsymbol{d}_{j-1}c_r^2 + \frac{1}{2^8}c_y\boldsymbol{d}_{j-1}c_r + \frac{1}{2^8}c_yc_r\right)$$

$$\leq (l+2)\left(\frac{2^6}{3^8}c_y\boldsymbol{d}_{j-1}c_r^2 + \sum_{k=j+1}^l \frac{2^6}{3^8}c_y\boldsymbol{d}_{j-1}c_r^2 + \frac{1}{2^8}c_y\boldsymbol{d}_{j-1}c_r + \frac{1}{2^8}c_yc_r\right).$$

**Final result**: Thus we can bound

$$\|\nabla_{\boldsymbol{w}}\nabla_{\boldsymbol{x}}f(\boldsymbol{w},\boldsymbol{x})\|_{\mathrm{op}} \leq \|\nabla_{\boldsymbol{w}}\nabla_{\boldsymbol{x}}f(\boldsymbol{w},\boldsymbol{x})\|_F \leq \sqrt{\sum_{j=1}^l\left\|\frac{\partial^2 f(\boldsymbol{w},\boldsymbol{x})}{\partial\boldsymbol{w}_{(j)}\partial\boldsymbol{x}^T}\right\|_F^2}$$

$$\leq \sqrt{\sum_{j=1}^l(l+2)\left(\frac{2^6}{3^8}c_y\boldsymbol{d}_{j-1}c_r^2 + \sum_{k=j+1}^l \frac{2^6}{3^8}c_y\boldsymbol{d}_{j-1}c_r^2 + \frac{1}{2^8}c_y\boldsymbol{d}_{j-1}c_r + \frac{1}{2^8}c_yc_r\right)}$$

$$\leq \sqrt{\frac{2^6}{3^8}l(l+2)c_yc_rc_d\left(lc_r + 1\right)},$$

where $c_d = \max_j \boldsymbol{d}_j$. The proof is completed. $\qquad\square$

### D.2.4 PROOF OF LEMMAS 19 AND 20

**Lemma 25.** *(Alessandro, 2016; Rigollet, 2015) Let $(\boldsymbol{x}_1, \cdots, \boldsymbol{x}_k)$ be a vector of i.i.d. Gaussian variables from $\mathcal{N}(0, \tau^2)$ and let $f : \mathbb{R}^{\boldsymbol{d}_0} \to \mathbb{R}$ be $L$-Lipschitz. Then the variable $f(\boldsymbol{x}) - \mathbb{E}f(\boldsymbol{x})$ is sub-Gaussian. That is, we have*

$$\mathbb{P}\left(f(\boldsymbol{x}) - \mathbb{E}f(\boldsymbol{x}) > t\right) \leq \exp\left(-\frac{t^2}{2L^2\tau^2}\right), \quad (\forall t \geq 0),$$

*or*

$$\mathbb{E}\left(\lambda(f(\boldsymbol{x}) - \mathbb{E}f(\boldsymbol{x}))\right) \leq \exp\left(4\lambda^2 L^2\tau^2\right), \quad (\forall\lambda \geq 0).$$

*Remarkably, this is a dimension free inequality.*

*Proof of Lemma 19.* We first define a function $g(\boldsymbol{x}) = \boldsymbol{z}^T \nabla_{\boldsymbol{w}} f(\boldsymbol{w}, \boldsymbol{x})$ where $\boldsymbol{z} \in \mathbb{R}^d$ is a constant vector. Then we have $\nabla_{\boldsymbol{x}} g(\boldsymbol{x}) = \nabla_{\boldsymbol{x}} \left( \boldsymbol{z}^T \nabla_{\boldsymbol{w}} f(\boldsymbol{w}, \boldsymbol{x}) \right) = \nabla_{\boldsymbol{x}} \nabla_{\boldsymbol{w}} f(\boldsymbol{w}, \boldsymbol{x}) \boldsymbol{z}$. Then by Lemma 18, we can obtain $\|\nabla_{\boldsymbol{x}} g(\boldsymbol{x})\|_2 = \|\nabla_{\boldsymbol{x}} \nabla_{\boldsymbol{w}} f(\boldsymbol{w}, \boldsymbol{x}) \boldsymbol{z}\|_2 \leq \beta \|\boldsymbol{z}\|_2$, where $\beta = \sqrt{\frac{2^6}{3^8} l(l+2) c_y c_r c_d (l c_r + 1)}$ in which $c_y$, $c_d$ and $c_r$ are defined in Lemma 18. This means $g(\boldsymbol{x})$ is $\beta \|\boldsymbol{z}\|_2$-Lipschitz. Thus, by Lemma 25, we have

$$\mathbb{E} \left( t \left\langle \boldsymbol{z}, \nabla_{\boldsymbol{w}} f(\boldsymbol{w}, \boldsymbol{x}) - \mathbb{E} \nabla_{\boldsymbol{w}} f(\boldsymbol{w}, \boldsymbol{x}) \right\rangle \right) = \mathbb{E} \left( t \left( g(\boldsymbol{x}) - \mathbb{E} g(\boldsymbol{x}) \right) \right) \leq \exp \left( 4 t^2 \beta^2 \|\boldsymbol{z}\|_2^2 \tau^2 \right).$$

Let $\boldsymbol{\lambda} = t \boldsymbol{z}$. This further gives

$$\mathbb{E} \left( \left\langle \boldsymbol{\lambda}, \nabla_{\boldsymbol{w}} f(\boldsymbol{w}, \boldsymbol{x}) - \mathbb{E} \nabla_{\boldsymbol{w}} f(\boldsymbol{w}, \boldsymbol{x}) \right\rangle \right) \leq \exp \left( 4 \beta^2 \tau^2 \|\boldsymbol{\lambda}\|_2^2 \right),$$

which means $\left\langle \boldsymbol{\lambda}, \nabla_{\boldsymbol{w}} f(\boldsymbol{w}, \boldsymbol{x}) - \mathbb{E} \nabla_{\boldsymbol{w}} f(\boldsymbol{w}, \boldsymbol{x}) \right\rangle$ is $8 \beta^2 \tau^2$-sub-Gaussian. $\qquad \square$

*Proof of Lemma 20.* We first define a function $h(\boldsymbol{x}) = \boldsymbol{z}^T \nabla_{\boldsymbol{w}}^2 f(\boldsymbol{w}, \boldsymbol{x}) \boldsymbol{z}$ where $\boldsymbol{z} \in \mathbb{S}^{d-1}$, *i.e.* $\|\boldsymbol{z}\|_2 = 1$. Then $h(\boldsymbol{w})$ is a $\gamma$-Lipschitz function, where $\gamma = \|\nabla_{\boldsymbol{x}} \nabla_{\boldsymbol{w}}^2 f(\boldsymbol{w}, \boldsymbol{x})\|_{op}$. Note that since the sigmoid function is infinitely differentiable function, $\nabla_{\boldsymbol{x}} \nabla_{\boldsymbol{w}}^2 f(\boldsymbol{w}, \boldsymbol{x})$ exists. Also since for any input $x$, $\sigma(x)$ belongs to $[0, 1]$. Thus, the values of the entries in $\nabla_{\boldsymbol{x}} \nabla_{\boldsymbol{w}}^2 f(\boldsymbol{w}, \boldsymbol{x})$ can be bounded. So according to the definition of the operation norm of a 3-way tensor, the operation norm of $\nabla_{\boldsymbol{x}} \nabla_{\boldsymbol{w}}^2 f(\boldsymbol{w}, \boldsymbol{x})$ can be bounded by a constant. Without loss of generality, let $\|\nabla_{\boldsymbol{x}} \nabla_{\boldsymbol{w}}^2 f(\boldsymbol{w}, \boldsymbol{x})\|_{op} \leq \gamma < +\infty$. Thus, by Lemma 25, we have

$$\mathbb{E} \left( t \left\langle \boldsymbol{z}, \left( \nabla_{\boldsymbol{w}}^2 f(\boldsymbol{w}, \boldsymbol{x}) - \mathbb{E} \nabla_{\boldsymbol{w}}^2 f(\boldsymbol{w}, \boldsymbol{x}) \right) \boldsymbol{z} \right\rangle \right) = \mathbb{E} \left( t \left( h(\boldsymbol{x}) - \mathbb{E} h(\boldsymbol{x}) \right) \right) \leq \exp \left( \frac{8 t^2 \gamma^2 \tau^2}{2} \right).$$

This means that the hessian of the loss evaluated on a unit vector is $8 \gamma^2 \tau^2$-sub-Gaussian. $\qquad \square$

### D.2.5 PROOF OF LEMMA 21

*Proof.* Recall that the weight of each layer has magnitude bound separately, *i.e.* $\|\boldsymbol{w}_{(j)}\|_2 \leq r$. Assume that $\boldsymbol{w}_{(j)}$ has $\boldsymbol{s}_j$ non-zero entries. Then we have $\sum_{j=1}^l \boldsymbol{s}_j = s$. So here we separately assume $\boldsymbol{w}_\epsilon^j = \{\boldsymbol{w}_1^j, \cdots, \boldsymbol{w}_{n_\epsilon^j}^j\}$ is the $\boldsymbol{d}_j \boldsymbol{d}_{j-1} \epsilon / d$-covering net of the ball $\mathrm{B}^{\boldsymbol{d}_j \boldsymbol{d}_{j-1}}(r)$ which corresponds to the weight $\boldsymbol{w}_{(j)}$ of the $j$-th layer. Let $n_\epsilon^j$ be the $\epsilon / l$-covering number. By $\epsilon$-covering theory in (Vershynin, 2012), we can have

$$n_\epsilon^j \leq \binom{\boldsymbol{d}_j \boldsymbol{d}_{j-1}}{\boldsymbol{s}_j} \left( \frac{3r}{\boldsymbol{d}_j \boldsymbol{d}_{j-1} \epsilon / (ld)} \right)^{\boldsymbol{s}_j} \leq \exp \left( \boldsymbol{s}_j \log \left( \frac{3r \boldsymbol{d}_j \boldsymbol{d}_{j-1}}{\boldsymbol{d}_j \boldsymbol{d}_{j-1} \epsilon / d} \right) \right) = \exp \left( \boldsymbol{s}_j \log \left( \frac{3rd}{\epsilon} \right) \right).$$

Let $\boldsymbol{w} \in \Omega$ be an arbitrary vector. Since $\boldsymbol{w} = [\boldsymbol{w}_{(1)}, \cdots, \boldsymbol{w}_{(l)}]$ where $\boldsymbol{w}_{(j)}$ is the weight of the $j$-th layer, we can always find a vector $\boldsymbol{w}_{k_j}^j$ in $\boldsymbol{w}_\epsilon^j$ such that $\|\boldsymbol{w}_{(j)} - \boldsymbol{w}_{k_j}^j\|_2 \leq \boldsymbol{d}_j \boldsymbol{d}_{j-1} \epsilon / d$. For brevity, let $j_w \in [n_\epsilon^j]$ denote the index of $\boldsymbol{w}_{k_j}^j$ in $\epsilon$-net $\boldsymbol{w}_\epsilon^j$. Then let $\boldsymbol{w}_{k_w} = [\boldsymbol{w}_{k_1}^j; \cdots; \boldsymbol{w}_{k_j}^j; \cdots; \boldsymbol{w}_{k_l}^j]$. This means that we can always find a vector $\boldsymbol{w}_{k_w}$ such that $\|\boldsymbol{w} - \boldsymbol{w}_{k_w}\|_2 \leq \epsilon$. Accordingly, we can

decompose $\left\|\nabla^2 \hat{\boldsymbol{J}}_n(\boldsymbol{w}) - \nabla^2 \boldsymbol{J}(\boldsymbol{w})\right\|_{\text{op}}$ as follows:

$$\left\|\nabla^2 \hat{\boldsymbol{J}}_n(\boldsymbol{w}) - \nabla^2 \boldsymbol{J}(\boldsymbol{w})\right\|_{\text{op}}$$

$$= \left\|\frac{1}{n}\sum_{i=1}^{n} \nabla^2 f(\boldsymbol{w}, \boldsymbol{x}_{(i)}) - \mathbb{E}(\nabla^2 f(\boldsymbol{w}, \boldsymbol{x}))\right\|_{\text{op}}$$

$$= \left\|\frac{1}{n}\sum_{i=1}^{n} \left(\nabla^2 f(\boldsymbol{w}, \boldsymbol{x}_{(i)}) - \nabla f(\boldsymbol{w}_{k_{\boldsymbol{w}}}, \boldsymbol{x}_{(i)})\right) + \frac{1}{n}\sum_{i=1}^{n} \nabla^2 f(\boldsymbol{w}_{k_{\boldsymbol{w}}}, \boldsymbol{x}_{(i)}) - \mathbb{E}(\nabla^2 f(\boldsymbol{w}_{k_{\boldsymbol{w}}}, \boldsymbol{x}))\right.$$

$$\left. + \mathbb{E}(\nabla^2 f(\boldsymbol{w}_{k_{\boldsymbol{w}}}, \boldsymbol{x})) - \mathbb{E}(\nabla^2 f(\boldsymbol{w}, \boldsymbol{x}))\right\|_{\text{op}}$$

$$\leq \left\|\frac{1}{n}\sum_{i=1}^{n} \left(\nabla^2 f(\boldsymbol{w}, \boldsymbol{x}_{(i)}) - \nabla^2 f(\boldsymbol{w}_{k_{\boldsymbol{w}}}, \boldsymbol{x}_{(i)})\right)\right\|_{\text{op}} + \left\|\frac{1}{n}\sum_{i=1}^{n} \nabla^2 f(\boldsymbol{w}_{k_{\boldsymbol{w}}}, \boldsymbol{x}_{(i)}) - \mathbb{E}(\nabla^2 f(\boldsymbol{w}_{k_{\boldsymbol{w}}}, \boldsymbol{x}))\right\|_{\text{op}}$$

$$+ \left\|\mathbb{E}(\nabla^2 f(\boldsymbol{w}_{k_{\boldsymbol{w}}}, \boldsymbol{x})) - \mathbb{E}(\nabla^2 f(\boldsymbol{w}, \boldsymbol{x}))\right\|_{\text{op}}.$$

Here we also define four events $\boldsymbol{E}_0$, $\boldsymbol{E}_1$, $\boldsymbol{E}_2$ and $\boldsymbol{E}_3$ as

$$\boldsymbol{E}_0 = \left\{\sup_{\boldsymbol{w}\in\Omega} \left\|\nabla^2 \hat{\boldsymbol{J}}_n(\boldsymbol{w}) - \nabla^2 \boldsymbol{J}(\boldsymbol{w})\right\|_{\text{op}} \geq t\right\},$$

$$\boldsymbol{E}_1 = \left\{\sup_{\boldsymbol{w}\in\Omega} \left\|\frac{1}{n}\sum_{i=1}^{n} \left(\nabla^2 f(\boldsymbol{w}, \boldsymbol{x}_{(i)}) - \nabla^2 f(\boldsymbol{w}_{k_{\boldsymbol{w}}}, \boldsymbol{x}_{(i)})\right)\right\|_{\text{op}} \geq \frac{t}{3}\right\},$$

$$\boldsymbol{E}_2 = \left\{\sup_{j_{\boldsymbol{w}}\in[n_{\epsilon}{}^j], j=[l]} \left\|\frac{1}{n}\sum_{i=1}^{n} \nabla^2 f(\boldsymbol{w}_{k_{\boldsymbol{w}}}, \boldsymbol{x}_{(i)}) - \mathbb{E}(\nabla^2 f(\boldsymbol{w}_{k_{\boldsymbol{w}}}, \boldsymbol{x}))\right\|_{\text{op}} \geq \frac{t}{3}\right\},$$

$$\boldsymbol{E}_3 = \left\{\sup_{\boldsymbol{w}\in\Omega} \left\|\mathbb{E}(\nabla^2 f(\boldsymbol{w}_{k_{\boldsymbol{w}}}, \boldsymbol{x})) - \mathbb{E}(\nabla^2 f(\boldsymbol{w}, \boldsymbol{x}))\right\|_{\text{op}} \geq \frac{t}{3}\right\}.$$

Accordingly, we have

$$\mathbb{P}\left(\boldsymbol{E}_0\right) \leq \mathbb{P}\left(\boldsymbol{E}_1\right) + \mathbb{P}\left(\boldsymbol{E}_2\right) + \mathbb{P}\left(\boldsymbol{E}_3\right).$$

So we can respectively bound $\mathbb{P}\left(\boldsymbol{E}_1\right)$, $\mathbb{P}\left(\boldsymbol{E}_2\right)$ and $\mathbb{P}\left(\boldsymbol{E}_3\right)$ to bound $\mathbb{P}\left(\boldsymbol{E}_0\right)$.

**Step 1. Bound $\mathbb{P}\left(\boldsymbol{E}_1\right)$:** We first bound $\mathbb{P}\left(\boldsymbol{E}_1\right)$ as follows:

$$\mathbb{P}\left(\boldsymbol{E}_1\right) = \mathbb{P}\left(\sup_{\boldsymbol{w}\in\Omega} \left\|\frac{1}{n}\sum_{i=1}^{n} \left(\nabla^2 f(\boldsymbol{w}, \boldsymbol{x}_{(i)}) - \nabla^2 f(\boldsymbol{w}_{k_{\boldsymbol{w}}}, \boldsymbol{x}_{(i)})\right)\right\|_2 \geq \frac{t}{3}\right)$$

$$\overset{\text{①}}{\leq} \frac{3}{t}\mathbb{E}\left(\sup_{\boldsymbol{w}\in\Omega} \left\|\frac{1}{n}\sum_{i=1}^{n} \left(\nabla^2 f(\boldsymbol{w}, \boldsymbol{x}_{(i)}) - \nabla^2 f(\boldsymbol{w}_{k_{\boldsymbol{w}}}, \boldsymbol{x}_{(i)})\right)\right\|_2\right)$$

$$\leq \frac{3}{t}\mathbb{E}\left(\sup_{\boldsymbol{w}\in\Omega} \frac{\left\|\frac{1}{n}\sum_{i=1}^{n} \left(\nabla^2 f(\boldsymbol{w}, \boldsymbol{x}_{(i)}) - \nabla^2 f(\boldsymbol{w}_{k_{\boldsymbol{w}}}, \boldsymbol{x}_{(i)})\right)\right\|_2}{\|\boldsymbol{w} - \boldsymbol{w}_{k_{\boldsymbol{w}}}\|_2} \sup_{\boldsymbol{w}\in\Omega} \|\boldsymbol{w} - \boldsymbol{w}_{k_{\boldsymbol{w}}}\|_2\right)$$

$$\leq \frac{3\epsilon}{t}\mathbb{E}\left(\sup_{\boldsymbol{w}\in\Omega} \left\|\frac{1}{n}\sum_{i=1}^{n} \nabla^3 f(\boldsymbol{w}, \boldsymbol{x}_{(i)})\right\|_{\text{op}}\right)$$

$$\overset{\text{②}}{\leq} \frac{3\xi\epsilon}{t},$$

where ① holds since by Markov inequality and ② holds because of Lemma 17.

Therefore, we can set

$$t \geq \frac{6\xi\epsilon}{\varepsilon}.$$

Then we can bound $\mathbb{P}(\boldsymbol{E}_1)$:

$$\mathbb{P}(\boldsymbol{E}_1) \leq \frac{\varepsilon}{2}.$$

**Step 2. Bound** $\mathbb{P}(\boldsymbol{E}_2)$: By Lemma 2, we know that for any matrix $\boldsymbol{X} \in \mathbb{R}^{d \times d}$, its operator norm can be computed as

$$\|\boldsymbol{X}\|_{\text{op}} \leq \frac{1}{1 - 2\epsilon} \sup_{\boldsymbol{\lambda} \in \boldsymbol{\lambda}_\epsilon} |\langle \boldsymbol{\lambda}, \boldsymbol{X}\boldsymbol{\lambda} \rangle|.$$

where $\boldsymbol{\lambda}_\epsilon = \{\boldsymbol{\lambda}_1, \ldots, \boldsymbol{\lambda}_{k_w}\}$ be an $\epsilon$-covering net of $\mathsf{B}^d(1)$.

Let $\boldsymbol{\lambda}_{1/4}$ be the $\frac{1}{4}$-covering net of $\mathsf{B}^d(1)$ but it has only $s$ nonzero entries. So the size of its $\epsilon$-net is

$$\binom{d}{s} \left(\frac{3}{1/4}\right)^s \leq \exp\left(s \log(12d)\right).$$

Recall that we use $j_w$ to denote the index of $\boldsymbol{w}_{k_j}^j$ in $\epsilon$-net $\boldsymbol{w}_\epsilon^j$ and we have $j_w \in [n_\epsilon{}^j]$, $(n_\epsilon{}^j \leq \exp\left(\boldsymbol{s}_j \log\left(\frac{3rd}{\epsilon}\right)\right))$. Then we can bound $\mathbb{P}(\boldsymbol{E}_2)$ as follows:

$$\mathbb{P}(\boldsymbol{E}_2) = \mathbb{P}\left(\sup_{j_w \in [n_\epsilon{}^j], j=[l]} \left\|\frac{1}{n} \sum_{i=1}^n \nabla^2 f(\boldsymbol{w}_{k_w}, \boldsymbol{x}_{(i)}) - \mathbb{E}(\nabla^2 f(\boldsymbol{w}_{k_w}, \boldsymbol{x}))\right\|_2 \geq \frac{t}{3}\right)$$

$$= \mathbb{P}\left(\sup_{j_w \in [n_\epsilon{}^j], j=[l], \boldsymbol{\lambda} \in \boldsymbol{\lambda}_{1/4}} 2\left|\left\langle \boldsymbol{\lambda}, \left(\frac{1}{n} \sum_{i=1}^n \nabla^2 f(\boldsymbol{w}_{k_w}, \boldsymbol{x}_{(i)}) - \mathbb{E}\left(\nabla^2 f(\boldsymbol{w}_{k_w}, \boldsymbol{x})\right)\right) \boldsymbol{\lambda}\right\rangle\right| \geq \frac{t}{3}\right)$$

$$\leq \exp\left(s \log(12d)\right) \exp\left(\sum_{j=1}^l \boldsymbol{s}_j \log\left(\frac{3rd}{\epsilon}\right)\right) \sup_{j_w \in [n_\epsilon{}^j], j=[l], \boldsymbol{\lambda} \in \boldsymbol{\lambda}_{1/4}} \mathbb{P}\left(\left|\frac{1}{n} \sum_{i=1}^n \left\langle \boldsymbol{\lambda}, \right.\right.\right.$$

$$\left.\left.\left.\left(\nabla^2 f(\boldsymbol{w}_{k_w}, \boldsymbol{x}_{(i)}) - \mathbb{E}\left(\nabla^2 f(\boldsymbol{w}_{k_w}, \boldsymbol{x})\right)\right) \boldsymbol{\lambda}\right\rangle\right| \geq \frac{t}{6}\right).$$

Since by Lemma 20, $\left\langle \boldsymbol{\lambda}, \left(\nabla_{\boldsymbol{w}}^2 f(\boldsymbol{w}, \boldsymbol{x}) - \mathbb{E}\nabla_{\boldsymbol{w}}^2 f(\boldsymbol{w}, \boldsymbol{x})\right) \boldsymbol{\lambda}\right\rangle$ where $\boldsymbol{\lambda} \in \mathsf{B}^d(1)$ is $8\gamma^2\tau^2$-sub-Gaussian, *i.e.*

$$\mathbb{E}\left(t\left\langle \boldsymbol{\lambda}, \left(\nabla_{\boldsymbol{w}}^2 f(\boldsymbol{w}, \boldsymbol{x}) - \mathbb{E}\nabla_{\boldsymbol{w}}^2 f(\boldsymbol{w}, \boldsymbol{x})\right) \boldsymbol{\lambda}\right\rangle\right) \leq \exp\left(\frac{8t^2\gamma^2\tau^2}{2}\right).$$

Thus, $\frac{1}{n}\sum_{i=1}^n \left\langle \boldsymbol{\lambda}, \left(\nabla_{\boldsymbol{w}}^2 f(\boldsymbol{w}, \boldsymbol{x}) - \mathbb{E}\nabla_{\boldsymbol{w}}^2 f(\boldsymbol{w}, \boldsymbol{x})\right) \boldsymbol{\lambda}\right\rangle$ is $8\gamma^2\tau^2/n$-sub-Gaussian random variable. So we can obtain

$$\mathbb{P}\left(\left|\frac{1}{n} \sum_{i=1}^n \left\langle \boldsymbol{y}, \left(\nabla_{\boldsymbol{w}}^2 f(\boldsymbol{w}, \boldsymbol{x}) - \mathbb{E}\nabla_{\boldsymbol{w}}^2 f(\boldsymbol{w}, \boldsymbol{x})\right) \boldsymbol{y}\right\rangle\right| \geq \frac{t}{6}\right) \leq 2\exp\left(-\frac{nt^2}{72\gamma^2\tau^2}\right).$$

Note $d = \sum_j \boldsymbol{d}_j\boldsymbol{d}_{j-1}$. Then the probability of $\boldsymbol{E}_2$ is upper bounded as

$$\mathbb{P}(\boldsymbol{E}_2) \leq 2\exp\left(-\frac{nt^2}{72\gamma^2\tau^2} + s \log\left(\frac{36d^2r}{\epsilon}\right)\right).$$

Thus, if we set

$$t \geq \gamma\tau\sqrt{\frac{72\left(s \log(36d^2r/\epsilon) + \log(4/\varepsilon)\right)}{n}},$$

then we have

$$\mathbb{P}(\boldsymbol{E}_2) \leq \frac{\varepsilon}{2}.$$

**Step 3. Bound** $\mathbb{P}\left(\boldsymbol{E}_3\right)$: We first bound $\mathbb{P}\left(\boldsymbol{E}_3\right)$ as follows:

$$
\begin{aligned}
\mathbb{P}\left(\boldsymbol{E}_3\right) =& \mathbb{P}\left( \sup_{\boldsymbol{w}\in\Omega} \left\| \mathbb{E}(\nabla^2 f(\boldsymbol{w}_{k_{\boldsymbol{w}}}, \boldsymbol{x})) - \mathbb{E}(\nabla^2 f(\boldsymbol{w}, \boldsymbol{x})) \right\|_2 \geq \frac{t}{3} \right) \\
\leq& \mathbb{P}\left( \mathbb{E}\sup_{\boldsymbol{w}\in\Omega} \left\| (\nabla^2 f(\boldsymbol{w}_{k_{\boldsymbol{w}}}, \boldsymbol{x}) - \nabla^2 f(\boldsymbol{w}, \boldsymbol{x}) \right\|_2 \geq \frac{t}{3} \right) \\
=& \mathbb{P}\left( \mathbb{E}\sup_{\boldsymbol{w}\in\Omega} \frac{\left\| \left(\nabla^2 f(\boldsymbol{w}, \boldsymbol{x}) - \nabla^2 f(\boldsymbol{w}_{k_{\boldsymbol{w}}}, \boldsymbol{x})\right) \right\|_2}{\|\boldsymbol{w} - \boldsymbol{w}_{k_{\boldsymbol{w}}}\|_2} \sup_{\boldsymbol{w}\in\Omega} \|\boldsymbol{w} - \boldsymbol{w}_{k_{\boldsymbol{w}}}\|_2 \geq \frac{t}{3} \right) \\
\leq& \mathbb{P}\left( \mathbb{E}\sup_{\boldsymbol{w}\in\Omega} \left\| \nabla^3 f(\boldsymbol{w}, \boldsymbol{x}) \right\|_{\text{op}} \geq \frac{t}{3} \right) \\
\leq& \mathbb{P}\left( \xi\epsilon \geq \frac{t}{3} \right).
\end{aligned}
$$

We set $\epsilon$ enough small such that $\xi\epsilon < t/3$ always holds. Then it yields $\mathbb{P}\left(\boldsymbol{E}_3\right) = 0$.

**Step 4. Final result**: To ensure $\mathbb{P}(\boldsymbol{E}_0) \leq \varepsilon$, we just set $\epsilon = 36rl^2/n$ and

$$
t \geq \max\left( \frac{6\xi\epsilon}{\varepsilon}, \gamma\tau\sqrt{\frac{72\left(s\log(36rd^2/\epsilon) + \log(4/\varepsilon)\right)}{n}} \right) = \max\left( \frac{108\xi r}{n\varepsilon}, c_4'\gamma\tau\sqrt{\frac{d\log(nl) + \log(4/\varepsilon)}{n}} \right).
$$

Therefore, there exists such two universal constants $c_{m'}$ and $c_m$ such that if $n \geq \frac{c_{m'}\xi^2 l^2 r^2}{\gamma^2 \tau^2 \varepsilon^2 s\log(d)\log(1/\varepsilon)}$, then

$$
\sup_{\boldsymbol{w}\in\Omega} \left\| \nabla^2 \hat{\boldsymbol{J}}_n(\boldsymbol{w}) - \nabla^2 \boldsymbol{J}(\boldsymbol{w}) \right\|_{\text{op}} \leq c_m\gamma\tau\sqrt{\frac{s\log(dn/l) + \log(4/\varepsilon)}{n}}
$$

holds with probability at least $1 - \varepsilon$. $\qquad\square$

### D.3 PROOFS OF MAIN THEORIES

#### D.3.1 PROOF OF THEOREM 4

*Proof.* Recall that the weight of each layer has magnitude bound separately, *i.e.* $\|\boldsymbol{w}_{(j)}\|_2 \leq r$. Assume that $\boldsymbol{w}_{(j)}$ has $\boldsymbol{s}_j$ non-zero entries. Then we have $\sum_{j=1}^{l} \boldsymbol{s}_j = s$. So here we separately assume $\boldsymbol{w}_\epsilon^j = \{\boldsymbol{w}_1^j, \cdots, \boldsymbol{w}_{n_{\epsilon^j}}^j\}$ is the $\boldsymbol{d}_j\boldsymbol{d}_{j-1}\epsilon/d$-covering net of the ball $\mathrm{B}^{\boldsymbol{d}_j\boldsymbol{d}_{j-1}}(r)$ which corresponds to the weight $\boldsymbol{w}_{(j)}$ of the $j$-th layer. Let $n_{\epsilon^j}$ be the $\epsilon/l$-covering number. By $\epsilon$-covering theory in (Vershynin, 2012), we can have

$$
n_\epsilon^j \leq \binom{\boldsymbol{d}_j\boldsymbol{d}_{j-1}}{\boldsymbol{s}_j} \left( \frac{3r}{\boldsymbol{d}_j\boldsymbol{d}_{j-1}\epsilon/(ld)} \right)^{\boldsymbol{s}_j} \leq \exp\left( \boldsymbol{s}_j\log\left( \frac{3r\boldsymbol{d}_j\boldsymbol{d}_{j-1}}{\boldsymbol{d}_j\boldsymbol{d}_{j-1}\epsilon/d} \right) \right) = \exp\left( \boldsymbol{s}_j\log\left( \frac{3rd}{\epsilon} \right) \right).
$$

Let $\boldsymbol{w}\in\Omega$ be an arbitrary vector. Since $\boldsymbol{w} = [\boldsymbol{w}_{(1)}, \cdots, \boldsymbol{w}_{(l)}]$ where $\boldsymbol{w}_{(j)}$ is the weight of the $j$-th layer, we can always find a vector $\boldsymbol{w}_{k_j}^j$ in $\boldsymbol{w}_\epsilon^j$ such that $\|\boldsymbol{w}_{(j)} - \boldsymbol{w}_{k_j}^j\|_2 \leq \boldsymbol{d}_j\boldsymbol{d}_{j-1}\epsilon/d$. For brevity, let $j_w \in [n_\epsilon^j]$ denote the index of $\boldsymbol{w}_{k_j}^j$ in $\epsilon$-net $\boldsymbol{w}_\epsilon^j$. Then let $\boldsymbol{w}_{k_{\boldsymbol{w}}} = [\boldsymbol{w}_{k_1}^j; \cdots; \boldsymbol{w}_{k_j}^j; \cdots; \boldsymbol{w}_{k_l}^j]$. This means that we can always find a vector $\boldsymbol{w}_{k_{\boldsymbol{w}}}$ such that $\|\boldsymbol{w} - \boldsymbol{w}_{k_{\boldsymbol{w}}}\|_2 \leq \epsilon$. Accordingly, we can

decompose $\left\|\nabla \hat{\boldsymbol{J}}_n(\boldsymbol{w}) - \nabla \boldsymbol{J}(\boldsymbol{w})\right\|_2$ as follows:

$$
\left\|\nabla \hat{\boldsymbol{J}}_n(\boldsymbol{w}) - \nabla \boldsymbol{J}(\boldsymbol{w})\right\|_2
$$

$$
= \left\|\frac{1}{n} \sum_{i=1}^n \nabla f(\boldsymbol{w}, \boldsymbol{x}_{(i)}) - \mathbb{E}(\nabla f(\boldsymbol{w}, \boldsymbol{x}))\right\|_2
$$

$$
= \left\|\frac{1}{n} \sum_{i=1}^n \left(\nabla f(\boldsymbol{w}, \boldsymbol{x}_{(i)}) - \nabla f(\boldsymbol{w}_{k_{\boldsymbol{w}}}, \boldsymbol{x}_{(i)})\right) + \frac{1}{n} \sum_{i=1}^n \nabla f(\boldsymbol{w}_{k_{\boldsymbol{w}}}, \boldsymbol{x}_{(i)}) - \mathbb{E}(\nabla f(\boldsymbol{w}_{k_{\boldsymbol{w}}}, \boldsymbol{x})) \right.
$$

$$
\left. + \mathbb{E}(\nabla f(\boldsymbol{w}_{k_{\boldsymbol{w}}}, \boldsymbol{x})) - \mathbb{E}(\nabla f(\boldsymbol{w}, \boldsymbol{x}))\right\|_2
$$

$$
\leq \left\|\frac{1}{n} \sum_{i=1}^n \left(\nabla f(\boldsymbol{w}, \boldsymbol{x}_{(i)}) - \nabla f(\boldsymbol{w}_{k_{\boldsymbol{w}}}, \boldsymbol{x}_{(i)})\right)\right\|_2 + \left\|\frac{1}{n} \sum_{i=1}^n \nabla f(\boldsymbol{w}_{k_{\boldsymbol{w}}}, \boldsymbol{x}_{(i)}) - \mathbb{E}(\nabla f(\boldsymbol{w}_{k_{\boldsymbol{w}}}, \boldsymbol{x}))\right\|_2
$$

$$
+ \left\|\mathbb{E}(\nabla f(\boldsymbol{w}_{k_{\boldsymbol{w}}}, \boldsymbol{x})) - \mathbb{E}(\nabla f(\boldsymbol{w}, \boldsymbol{x}))\right\|_2 .
$$

Here we also define four events $\boldsymbol{E}_0$, $\boldsymbol{E}_1$, $\boldsymbol{E}_2$ and $\boldsymbol{E}_3$ as

$$
\boldsymbol{E}_0 = \left\{\sup_{\boldsymbol{w} \in \Omega} \left\|\nabla \hat{\boldsymbol{J}}_n(\boldsymbol{w}) - \nabla \boldsymbol{J}(\boldsymbol{w})\right\|_2 \geq t\right\},
$$

$$
\boldsymbol{E}_1 = \left\{\sup_{\boldsymbol{w} \in \Omega} \left\|\frac{1}{n} \sum_{i=1}^n \left(\nabla f(\boldsymbol{w}, \boldsymbol{x}_{(i)}) - \nabla f(\boldsymbol{w}_{k_{\boldsymbol{w}}}, \boldsymbol{x}_{(i)})\right)\right\|_2 \geq \frac{t}{3}\right\},
$$

$$
\boldsymbol{E}_2 = \left\{\sup_{j_w \in [n_\epsilon{}^j], j=[l]} \left\|\frac{1}{n} \sum_{i=1}^n \nabla f(\boldsymbol{w}_{k_{\boldsymbol{w}}}, \boldsymbol{x}_{(i)}) - \mathbb{E}(\nabla f(\boldsymbol{w}_{k_{\boldsymbol{w}}}, \boldsymbol{x}))\right\|_2 \geq \frac{t}{3}\right\},
$$

$$
\boldsymbol{E}_3 = \left\{\sup_{\boldsymbol{w} \in \Omega} \left\|\mathbb{E}(\nabla f(\boldsymbol{w}_{k_{\boldsymbol{w}}}, \boldsymbol{x})) - \mathbb{E}(\nabla f(\boldsymbol{w}, \boldsymbol{x}))\right\|_2 \geq \frac{t}{3}\right\} .
$$

Accordingly, we have

$$
\mathbb{P}\left(\boldsymbol{E}_0\right) \leq \mathbb{P}\left(\boldsymbol{E}_1\right) + \mathbb{P}\left(\boldsymbol{E}_2\right) + \mathbb{P}\left(\boldsymbol{E}_3\right).
$$

So we can respectively bound $\mathbb{P}\left(\boldsymbol{E}_1\right)$, $\mathbb{P}\left(\boldsymbol{E}_2\right)$ and $\mathbb{P}\left(\boldsymbol{E}_3\right)$ to bound $\mathbb{P}\left(\boldsymbol{E}_0\right)$.

**Step 1. Bound $\mathbb{P}\left(\boldsymbol{E}_1\right)$:** We first bound $\mathbb{P}\left(\boldsymbol{E}_1\right)$ as follows:

$$
\mathbb{P}\left(\boldsymbol{E}_1\right) = \mathbb{P}\left(\sup_{\boldsymbol{w} \in \Omega} \left\|\frac{1}{n} \sum_{i=1}^n \left(\nabla f(\boldsymbol{w}, \boldsymbol{x}_{(i)}) - \nabla f(\boldsymbol{w}_{k_{\boldsymbol{w}}}, \boldsymbol{x}_{(i)})\right)\right\|_2 \geq \frac{t}{3}\right)
$$

$$
\overset{\text{①}}{\leq} \frac{3}{t} \mathbb{E}\left(\sup_{\boldsymbol{w} \in \Omega} \left\|\frac{1}{n} \sum_{i=1}^n \left(\nabla f(\boldsymbol{w}, \boldsymbol{x}_{(i)}) - \nabla f(\boldsymbol{w}_{k_{\boldsymbol{w}}}, \boldsymbol{x}_{(i)})\right)\right\|_2\right)
$$

$$
\leq \frac{3}{t} \mathbb{E}\left(\sup_{\boldsymbol{w} \in \Omega} \frac{\left\|\frac{1}{n} \sum_{i=1}^n \left(\nabla f(\boldsymbol{w}, \boldsymbol{x}_{(i)}) - \nabla f(\boldsymbol{w}_{k_{\boldsymbol{w}}}, \boldsymbol{x}_{(i)})\right)\right\|_2}{\|\boldsymbol{w} - \boldsymbol{w}_{k_{\boldsymbol{w}}}\|_2} \sup_{\boldsymbol{w} \in \Omega} \|\boldsymbol{w} - \boldsymbol{w}_{k_{\boldsymbol{w}}}\|_2\right)
$$

$$
\leq \frac{3\epsilon}{t} \mathbb{E}\left(\sup_{\boldsymbol{w} \in \Omega} \left\|\nabla^2 \hat{\boldsymbol{J}}_n(\boldsymbol{w}, \boldsymbol{x})\right\|_2\right),
$$

where ① holds because of Markov inequality. Then, we bound $\mathbb{E}\left(\sup_{\boldsymbol{w} \in \Omega} \left\|\nabla^2 \hat{\boldsymbol{J}}_n(\boldsymbol{w}, \boldsymbol{x})\right\|_2\right)$ as follows:

$$
\mathbb{E}\left(\sup_{\boldsymbol{w} \in \Omega} \left\|\nabla^2 \hat{\boldsymbol{J}}_n(\boldsymbol{w}, \boldsymbol{x})\right\|_2\right) \leq \mathbb{E}\left(\sup_{\boldsymbol{w} \in \Omega} \left\|\frac{1}{n} \sum_{i=1}^n \nabla^2 f(\boldsymbol{w}, \boldsymbol{x})\right\|_2\right) = \mathbb{E}\left(\sup_{\boldsymbol{w} \in \Omega} \left\|\nabla^2 f(\boldsymbol{w}, \boldsymbol{x})\right\|_2\right) \overset{\text{①}}{\leq} \varsigma,
$$

where ① holds since by Lemma 17, we have

$$
\left\|\nabla_{\boldsymbol{w}}^2 f(\boldsymbol{w}, \boldsymbol{x})\right\|_{\text{op}} \leq \left\|\nabla_{\boldsymbol{w}}^2 f(\boldsymbol{w}, \boldsymbol{x})\right\|_F \leq \varsigma,
$$

where $\varsigma = \sqrt{c_{s_1} c_r c_d^2 l^4}$ in which $c_d = \max_i \boldsymbol{d}_i$ and $c_r = \max\left(\frac{r^2}{16}, \left(\frac{r^2}{16}\right)^{l-1}\right)$. Therefore, we have

$$\mathbb{P}\left(\boldsymbol{E}_1\right) \leq \frac{3\varsigma\epsilon}{t}.$$

We further let

$$t \geq \frac{6\varsigma\epsilon}{\varepsilon}.$$

Then we can bound $\mathbb{P}(\boldsymbol{E}_1)$:

$$\mathbb{P}(\boldsymbol{E}_1) \leq \frac{\varepsilon}{2}.$$

**Step 2. Bound** $\mathbb{P}\left(\boldsymbol{E}_2\right)$: By Lemma 1, we know that for any vector $\boldsymbol{x} \in \mathbb{R}^d$, its $\ell_2$-norm can be computed as

$$\|\boldsymbol{x}\|_2 \leq \frac{1}{1-\epsilon} \sup_{\boldsymbol{\lambda} \in \boldsymbol{\lambda}_\epsilon} \langle \boldsymbol{\lambda}, \boldsymbol{x} \rangle.$$

where $\boldsymbol{\lambda}_\epsilon = \{\boldsymbol{\lambda}_1, \ldots, \boldsymbol{\lambda}_{k_w}\}$ be an $\epsilon$-covering net of $\mathsf{B}^d(1)$.

Let $\boldsymbol{\lambda}_{1/2}$ be the $\frac{1}{2}$-covering net of $\mathsf{B}^d(1)$ but it has only $s$ nonzero entries. So the size of its $\epsilon$-net is

$$\binom{d}{s} \left(\frac{3}{1/2}\right)^s \leq \exp\left(s \log\left(6d\right)\right).$$

Recall that we use $j_w$ to denote the index of $\boldsymbol{w}_{k_j}^j$ in $\epsilon$-net $\boldsymbol{w}_\epsilon^j$ and we have $j_w \in [n_\epsilon{}^j]$, $(n_\epsilon{}^j \leq \exp\left(\boldsymbol{s}_j \log\left(\frac{3rd}{\epsilon}\right)\right)$. Then we can bound $\mathbb{P}\left(\boldsymbol{E}_2\right)$ as follows:

$$\mathbb{P}\left(\boldsymbol{E}_2\right) = \mathbb{P}\left(\sup_{j_w \in [n_\epsilon{}^j], j=[l]} \left\|\frac{1}{n}\sum_{i=1}^n \nabla f(\boldsymbol{w}_{k_w}, \boldsymbol{x}_{(i)}) - \mathbb{E}(\nabla f(\boldsymbol{w}_{k_w}, \boldsymbol{x}))\right\|_2 \geq \frac{t}{3}\right)$$

$$= \mathbb{P}\left(\sup_{j_w \in [n_\epsilon{}^j], j=[l], \boldsymbol{\lambda} \in \boldsymbol{\lambda}_{1/2}} 2\left\langle \boldsymbol{\lambda}, \frac{1}{n}\sum_{i=1}^n \nabla f(\boldsymbol{w}_{k_w}, \boldsymbol{x}_{(i)}) - \mathbb{E}\left(\nabla f(\boldsymbol{w}_{k_w}, \boldsymbol{x})\right)\right\rangle \geq \frac{t}{3}\right)$$

$$\leq \exp\left(s\log(6d)\right) \exp\left(\sum_{j=1}^l \boldsymbol{s}_j \log\left(\frac{3rd}{\epsilon}\right)\right) \sup_{j_w \in [n_\epsilon{}^j], j=[l], \boldsymbol{\lambda} \in \boldsymbol{\lambda}_{1/2}} \mathbb{P}\left(\frac{1}{n}\sum_{i=1}^n \left\langle \boldsymbol{\lambda}, \right.\right.$$

$$\left.\left. \nabla f(\boldsymbol{w}_{k_w}, \boldsymbol{x}_{(i)}) - \mathbb{E}\left(\nabla f(\boldsymbol{w}_{k_w}, \boldsymbol{x})\right) \right\rangle \geq \frac{t}{6}\right).$$

Since by Lemma 19, $\langle \boldsymbol{y}, \nabla f(\boldsymbol{w}, \boldsymbol{x}) \rangle$ is $8\beta^2\tau^2$-sub-Gaussian, *i.e.*

$$\mathbb{E}\left(\langle \boldsymbol{\lambda}, \nabla_{\boldsymbol{w}} f(\boldsymbol{w}, \boldsymbol{x}) - \mathbb{E}\nabla_{\boldsymbol{w}} f(\boldsymbol{w}, \boldsymbol{x}) \rangle\right) \leq \exp\left(\frac{8\beta^2\tau^2\|\boldsymbol{\lambda}\|_2^2}{2}\right),$$

where $\beta = \sqrt{\frac{2^6}{3^8} l(l+2) c_y c_r c_d \left(l c_r + 1\right)}$ in which $c_y$, $c_d$ and $c_r$ are defined in Lemma 16. Thus, $\frac{1}{n}\sum_{i=1}^n \langle \boldsymbol{y}, \nabla f(\boldsymbol{w}, \boldsymbol{x}) \rangle$ is $8\beta^2\tau^2/n$-sub-Gaussian random variable. Thus, we can obtain

$$\mathbb{P}\left(\frac{1}{n}\sum_{i=1}^n \langle \boldsymbol{y}, \nabla f(\boldsymbol{w}_{k_w}, \boldsymbol{x}_{(i)}) - \mathbb{E}\left(\nabla f(\boldsymbol{w}_{k_w}, \boldsymbol{x})\right) \rangle \geq \frac{t}{6}\right) \leq \exp\left(-\frac{nt^2}{72\beta^2\tau^2}\right).$$

Notice, $\sum_j \boldsymbol{d}_j \boldsymbol{d}_{j-1} = d$. In this case, the probability of $\boldsymbol{E}_2$ is upper bounded as

$$\mathbb{P}\left(\boldsymbol{E}_2\right) \leq \exp\left(-\frac{nt^2}{72\beta^2\tau^2} + d\log\left(\frac{18r}{\epsilon}\right)\right).$$

Thus, if we set

$$t \geq \beta\tau\sqrt{\frac{72\left(s\log(18d^2r/\epsilon) + \log(4/\varepsilon)\right)}{n}},$$

then we have

$$\mathbb{P}\left(\boldsymbol{E}_2\right) \leq \frac{\varepsilon}{2}.$$

**Step 3. Bound $\mathbb{P}\left(\boldsymbol{E}_3\right)$:** We first bound $\mathbb{P}\left(\boldsymbol{E}_3\right)$ as follows:

$$
\begin{aligned}
\mathbb{P}\left(\boldsymbol{E}_3\right) =& \mathbb{P}\left(\sup_{\boldsymbol{w}\in\Omega}\|\mathbb{E}(\nabla f(\boldsymbol{w}_{k_{\boldsymbol{w}}},\boldsymbol{x})) - \mathbb{E}(\nabla f(\boldsymbol{w},\boldsymbol{x}))\|_2 \geq \frac{t}{3}\right) \\
=& \mathbb{P}\left(\sup_{\boldsymbol{w}\in\Omega}\frac{\|\mathbb{E}\left(\nabla f(\boldsymbol{w}_{k_{\boldsymbol{w}}},\boldsymbol{x}) - \nabla f(\boldsymbol{w},\boldsymbol{x})\|_2\right)}{\|\boldsymbol{w}-\boldsymbol{w}_{k_{\boldsymbol{w}}}\|_2} \sup_{\boldsymbol{w}\in\Omega}\|\boldsymbol{w}-\boldsymbol{w}_{k_{\boldsymbol{w}}}\|_2 \geq \frac{t}{3}\right) \\
\leq& \mathbb{P}\left(\epsilon\mathbb{E}\sup_{\boldsymbol{w}\in\Omega}\left\|\nabla^2\hat{\boldsymbol{J}}_n(\boldsymbol{w},\boldsymbol{x})\right\|_2 \geq \frac{t}{3}\right) \\
\overset{\textcircled{1}}{\leq}& \mathbb{P}\left(\varsigma\epsilon \geq \frac{t}{3}\right).
\end{aligned}
$$

where ① holds since by Lemma 17. We set $\epsilon$ enough small such that $\varsigma\epsilon < t/3$ always holds. Then it yields $\mathbb{P}\left(\boldsymbol{E}_3\right) = 0$.

**Step 4. Final result:** To ensure $\mathbb{P}(\boldsymbol{E}_0) \leq \varepsilon$, we just set $\epsilon = 18r/n$ and

$$
\begin{aligned}
t \geq& \max\left(\frac{6\varsigma\epsilon}{\varepsilon}, \ \beta\tau\sqrt{\frac{72\left(s\log(18d^2r/\epsilon) + \log(4/\varepsilon)\right)}{n}}\right) \\
=& \max\left(\frac{108\varsigma r}{n\varepsilon}, \ \beta\tau\sqrt{\frac{72\left(s\log(nl) + \log(4/\varepsilon)\right)}{n}}\right).
\end{aligned}
$$

Note that $\varsigma = \mathcal{O}(\sqrt{lc_d}\beta)$. Therefore, there exists a universal constant $c_{y'}$ such that if $n \geq c_{y'}c_d l^3 r^2/(s\log(d)\tau^2\varepsilon^2\log(1/\varepsilon))$, then

$$\sup_{\boldsymbol{w}\in\Omega}\left\|\nabla\hat{\boldsymbol{J}}_n(\boldsymbol{w}) - \nabla\boldsymbol{J}(\boldsymbol{w})\right\|_2 \leq \tau\sqrt{\frac{512}{729}c_y l(l+2)\left(lc_r+1\right)c_r c_d}\sqrt{\frac{s\log(dn/l)+\log(4/\varepsilon)}{n}}$$

holds with probability at least $1 - \varepsilon$, where $c_y$, $c_d$ and $c_r$ are defined in Lemma 16. $\qquad\square$

### D.3.2 PROOF OF THEOREM 5

*Proof.* Suppose that $\{\boldsymbol{w}^{(1)}, \boldsymbol{w}^{(2)}, \cdots, \boldsymbol{w}^{(m)}\}$ are the non-degenerate critical points of $\boldsymbol{J}(\boldsymbol{w})$. So for any $\boldsymbol{w}^{(k)}$, it obeys

$$\inf_i\left|\lambda_i^k\left(\nabla^2\boldsymbol{J}(\boldsymbol{w}^{(k)})\right)\right| \geq \zeta,$$

where $\lambda_i^k\left(\nabla^2\boldsymbol{J}(\boldsymbol{w}^{(k)})\right)$ denotes the $i$-th eigenvalue of the Hessian $\nabla^2\boldsymbol{J}(\boldsymbol{w}^{(k)})$ and $\zeta$ is a constant. We further define a set $D = \{\boldsymbol{w}\in\mathbb{R}^d\,|\,\|\nabla\boldsymbol{J}(\boldsymbol{w})\|_2 \leq \epsilon$ and $\inf_i|\lambda_i\left(\nabla^2\boldsymbol{J}(\boldsymbol{w}^{(k)})\right)| \geq \zeta\}$. According to Lemma 4, $D = \cup_{k=1}^{\infty}D_k$ where each $D_k$ is a disjoint component with $\boldsymbol{w}^{(k)}\in D_k$ for $k \leq m$ and $D_k$ does not contain any critical point of $\boldsymbol{J}(\boldsymbol{w})$ for $k \geq m+1$. On the other hand, by the continuity of $\nabla\boldsymbol{J}(\boldsymbol{w})$, it yields $\|\nabla\boldsymbol{J}(\boldsymbol{w})\|_2 = \epsilon$ for $\boldsymbol{w}\in\partial D_k$. Notice, we set the value of $\epsilon$ blow which is actually a function related $n$.

Then by utilizing Theorem 4, we let sample number $n$ sufficient large such that

$$\sup_{\boldsymbol{w}\in\Omega}\left\|\nabla\hat{\boldsymbol{J}}_n(\boldsymbol{w}) - \nabla\boldsymbol{J}(\boldsymbol{w})\right\|_2 \leq \beta \triangleq \frac{\epsilon}{2}$$

holds with probability at least $1 - \varepsilon$, where $\beta = \tau\sqrt{\frac{512}{729}c_y l(l+2)\left(lc_r+1\right)c_r c_d}\sqrt{\frac{s\log(dn/l)+\log(4/\varepsilon)}{n}}$. This further gives that for arbitrary $\boldsymbol{w}\in D_k$, we have

$$
\begin{aligned}
\inf_{\boldsymbol{w}\in D_k}\left\|t\nabla\hat{\boldsymbol{J}}_n(\boldsymbol{w}) + (1-t)\nabla\boldsymbol{J}(\boldsymbol{w})\right\|_2 =& \inf_{\boldsymbol{w}\in D_k}\left\|t\left(\nabla\hat{\boldsymbol{J}}_n(\boldsymbol{w}) - \nabla\boldsymbol{J}(\boldsymbol{w})\right) + \nabla\boldsymbol{J}(\boldsymbol{w})\right\|_2 \\
\geq& \inf_{\boldsymbol{w}\in D_k}\|\nabla\boldsymbol{J}(\boldsymbol{w})\|_2 - \sup_{\boldsymbol{w}\in D_k}t\left\|\nabla\hat{\boldsymbol{J}}_n(\boldsymbol{w}) - \nabla\boldsymbol{J}(\boldsymbol{w})\right\|_2 \\
\geq& \frac{\epsilon}{2}.
\end{aligned}
\tag{26}
$$

Similarly, by utilizing Lemma 21, let $n$ be sufficient large such that

$$\sup_{\boldsymbol{w}\in\Omega}\left\|\nabla^2\hat{\boldsymbol{J}}_n(\boldsymbol{w})-\nabla^2\boldsymbol{J}(\boldsymbol{w})\right\|_{\text{op}}\leq c_m\gamma\tau\sqrt{\frac{s\log(dn/l)+\log(4/\varepsilon)}{n}}\leq\frac{\zeta}{2}$$

holds with probability at least $1-\varepsilon$. Assume that $\boldsymbol{b}\in\mathbb{R}^d$ is a vector and satisfies $\boldsymbol{b}^T\boldsymbol{b}=1$. In this case, we can bound $\lambda_i^k\left(\nabla^2\hat{\boldsymbol{J}}_n(\boldsymbol{w})\right)$ for arbitrary $\boldsymbol{w}\in D_k$ as follows:

$$
\begin{aligned}
\inf_{\boldsymbol{w}\in D_k}\left|\lambda_i^k\left(\nabla^2\hat{\boldsymbol{J}}_n(\boldsymbol{w})\right)\right|&=\inf_{\boldsymbol{w}\in D_k}\min_{\boldsymbol{b}^T\boldsymbol{b}=1}\left|\boldsymbol{b}^T\nabla^2\hat{\boldsymbol{J}}_n(\boldsymbol{w})\boldsymbol{b}\right|\\
&=\inf_{\boldsymbol{w}\in D_k}\min_{\boldsymbol{b}^T\boldsymbol{b}=1}\left|\boldsymbol{b}^T\left(\nabla^2\hat{\boldsymbol{J}}_n(\boldsymbol{w})-\nabla^2\boldsymbol{J}(\boldsymbol{w})\right)\boldsymbol{b}+\boldsymbol{b}^T\nabla^2\boldsymbol{J}(\boldsymbol{w})\boldsymbol{b}\right|\\
&\geq\inf_{\boldsymbol{w}\in D_k}\min_{\boldsymbol{b}^T\boldsymbol{b}=1}\left|\boldsymbol{b}^T\nabla^2\boldsymbol{J}(\boldsymbol{w})\boldsymbol{b}\right|-\min_{\boldsymbol{b}^T\boldsymbol{b}=1}\left|\boldsymbol{b}^T\left(\nabla^2\hat{\boldsymbol{J}}_n(\boldsymbol{w})-\nabla^2\boldsymbol{J}(\boldsymbol{w})\right)\boldsymbol{b}\right|\\
&\geq\inf_{\boldsymbol{w}\in D_k}\min_{\boldsymbol{b}^T\boldsymbol{b}=1}\left|\boldsymbol{b}^T\nabla^2\boldsymbol{J}(\boldsymbol{w})\boldsymbol{b}\right|-\max_{\boldsymbol{b}^T\boldsymbol{b}=1}\left|\boldsymbol{b}^T\left(\nabla^2\hat{\boldsymbol{J}}_n(\boldsymbol{w})-\nabla^2\boldsymbol{J}(\boldsymbol{w})\right)\boldsymbol{b}\right|\\
&=\inf_{\boldsymbol{w}\in D_k}\inf_i\left|\lambda_i^k\left(\nabla^2f(\boldsymbol{w}_{(k)},\boldsymbol{x})\right)-\left\|\nabla^2\hat{\boldsymbol{J}}_n(\boldsymbol{w})-\nabla^2\boldsymbol{J}(\boldsymbol{w})\right\|_{\text{op}}\right.\\
&\geq\frac{\zeta}{2}.
\end{aligned}
$$

This means that in each set $D_k$, $\nabla^2\hat{\boldsymbol{J}}_n(\boldsymbol{w})$ has no zero eigenvalues. Then, combining this and Eqn. (26), by Lemma 3 we know that if the population risk $\boldsymbol{J}(\boldsymbol{w})$ has no critical point in $D_k$, then the empirical risk $\hat{\boldsymbol{J}}_n(\boldsymbol{w})$ has also no critical point in $D_k$; otherwise it also holds. By Lemma 3, we can also obtain that in $D_k$, if $\boldsymbol{J}(\boldsymbol{w})$ has a unique critical point $\boldsymbol{w}^{(k)}$ with non-degenerate index $s_k$, then $\hat{\boldsymbol{J}}_n(\boldsymbol{w})$ also has a unique critical point $\boldsymbol{w}_n^{(k)}$ in $D_k$ with the same non-degenerate index $s_k$. The first conclusion is proved.

Now we bound the distance between the corresponding critical points of $\boldsymbol{J}(\boldsymbol{w})$ and $\hat{\boldsymbol{J}}_n(\boldsymbol{w})$. Assume that in $D_k$, $\boldsymbol{J}(\boldsymbol{w})$ has a unique critical point $\boldsymbol{w}^{(k)}$ and $\hat{\boldsymbol{J}}_n(\boldsymbol{w})$ also has a unique critical point $\boldsymbol{w}_n^{(k)}$. Then, there exists $t\in[0,1]$ such that for any $\boldsymbol{z}\in\partial\mathsf{B}^d(1)$, we have

$$
\begin{aligned}
\epsilon&\geq\|\nabla\boldsymbol{J}(\boldsymbol{w}_n^{(k)})\|_2\\
&=\max_{\boldsymbol{z}^T\boldsymbol{z}=1}\langle\nabla\boldsymbol{J}(\boldsymbol{w}_n^{(k)}),\boldsymbol{z}\rangle\\
&=\max_{\boldsymbol{z}^T\boldsymbol{z}=1}\langle\nabla\boldsymbol{J}(\boldsymbol{w}^{(k)}),\boldsymbol{z}\rangle+\langle\nabla^2\boldsymbol{J}(\boldsymbol{w}^{(k)}+t(\boldsymbol{w}_n^{(k)}-\boldsymbol{w}^{(k)}))(\boldsymbol{w}_n^{(k)}-\boldsymbol{w}^{(k)}),\boldsymbol{z}\rangle\\
&\overset{\text{①}}{\geq}\left\langle\left(\nabla^2\boldsymbol{J}(\boldsymbol{w}^{(k)})\right)^2(\boldsymbol{w}_n^{(k)}-\boldsymbol{w}^{(k)}),(\boldsymbol{w}_n^{(k)}-\boldsymbol{w}^{(k)})\right\rangle^{1/2}\\
&\overset{\text{②}}{\geq}\zeta\|\boldsymbol{w}_n^{(k)}-\boldsymbol{w}^{(k)}\|_2,
\end{aligned}
$$

where ① holds since $\nabla\boldsymbol{J}(\boldsymbol{w}^{(k)})=\boldsymbol{0}$ and ② holds since $\boldsymbol{w}^{(k)}+t(\boldsymbol{w}_n^{(k)}-\boldsymbol{w}^{(k)})$ is in $D_k$ and for any $\boldsymbol{w}\in D_k$ we have $\inf_i|\lambda_i\left(\nabla^2\boldsymbol{J}(\boldsymbol{w})\right)|\geq\zeta$. Then if $n\geq c_s\max\left(c_dl^3r^2/(s\log(d)\tau^2\varepsilon^2\log(1/\varepsilon)),s\log(d/l)/\zeta^2\right)$ where $c_s$ is a constant, then

$$\|\boldsymbol{w}_n^{(k)}-\boldsymbol{w}^{(k)}\|_2\leq\frac{2\tau}{\zeta}\sqrt{\frac{512}{729}c_yl(l+2)\left(lc_r+1\right)c_rc_d}\sqrt{\frac{s\log(dn/l)+\log(4/\varepsilon)}{n}}$$

holds with probability at least $1-\varepsilon$. The proof is completed. $\qquad\square$

### D.3.3 PROOF OF THEOREM 6

*Proof.* Recall that the weight of each layer has magnitude bound separately, *i.e.* $\|\boldsymbol{w}_{(j)}\|_2\leq r$. Assume that $\boldsymbol{w}_{(j)}$ has $\boldsymbol{s}_j$ non-zero entries. Then we have $\sum_{j=1}^l\boldsymbol{s}_j=s$. So here we separately assume $\boldsymbol{w}_\epsilon^j=\{\boldsymbol{w}_1^j,\cdots,\boldsymbol{w}_{n_{\epsilon^j}}^j\}$ is the $\boldsymbol{d}_j\boldsymbol{d}_{j-1}\epsilon/d$-covering net of the ball $\mathsf{B}^{\boldsymbol{d}_j\boldsymbol{d}_{j-1}}(r)$ which corresponds

to the weight $\boldsymbol{w}_{(j)}$ of the $j$-th layer. Let $n_\epsilon{}^j$ be the $\epsilon/l$-covering number. By $\epsilon$-covering theory in (Vershynin, 2012), we can have

$$n_\epsilon{}^j \leq \binom{\boldsymbol{d}_j \boldsymbol{d}_{j-1}}{\boldsymbol{s}_j} \left( \frac{3r}{\boldsymbol{d}_j \boldsymbol{d}_{j-1} \epsilon/(ld)} \right)^{\boldsymbol{s}_j} \leq \exp\left( \boldsymbol{s}_j \log\left( \frac{3r \boldsymbol{d}_j \boldsymbol{d}_{j-1}}{\boldsymbol{d}_j \boldsymbol{d}_{j-1} \epsilon/d} \right) \right) = \exp\left( \boldsymbol{s}_j \log\left( \frac{3rd}{\epsilon} \right) \right).$$

Let $\boldsymbol{w} \in \Omega$ be an arbitrary vector. Since $\boldsymbol{w} = [\boldsymbol{w}_{(1)}, \cdots, \boldsymbol{w}_{(l)}]$ where $\boldsymbol{w}_{(j)}$ is the weight of the $j$-th layer, we can always find a vector $\boldsymbol{w}_{k_j}^j$ in $\boldsymbol{w}_\epsilon^j$ such that $\|\boldsymbol{w}_{(j)} - \boldsymbol{w}_{k_j}^j\|_2 \leq \boldsymbol{d}_j \boldsymbol{d}_{j-1} \epsilon/d$. For brevity, let $j_w \in [n_\epsilon{}^j]$ denote the index of $\boldsymbol{w}_{k_j}^j$ in $\epsilon$-net $\boldsymbol{w}_\epsilon^j$. Then let $\boldsymbol{w}_{k_w} = [\boldsymbol{w}_{k_1}^j; \cdots; \boldsymbol{w}_{k_j}^j; \cdots; \boldsymbol{w}_{k_l}^j]$. This means that we can always find a vector $\boldsymbol{w}_{k_w}$ such that $\|\boldsymbol{w} - \boldsymbol{w}_{k_w}\|_2 \leq \epsilon$. Accordingly, we can decompose $\left| \hat{\boldsymbol{J}}_n(\boldsymbol{w}) - \boldsymbol{J}(\boldsymbol{w}) \right|$ as

$$\left| \hat{\boldsymbol{J}}_n(\boldsymbol{w}) - \boldsymbol{J}(\boldsymbol{w}) \right| = \left| \frac{1}{n} \sum_{i=1}^n f(\boldsymbol{w}, \boldsymbol{x}_{(i)}) - \mathbb{E}(f(\boldsymbol{w}, \boldsymbol{x})) \right|$$

$$= \left| \frac{1}{n} \sum_{i=1}^n (f(\boldsymbol{w}, \boldsymbol{x}_{(i)}) - f(\boldsymbol{w}_{k_w}, \boldsymbol{x}_{(i)})) + \frac{1}{n} \sum_{i=1}^n f(\boldsymbol{w}_{k_w}, \boldsymbol{x}_{(i)}) - \mathbb{E}f(\boldsymbol{w}_{k_w}, \boldsymbol{x}) + \mathbb{E}f(\boldsymbol{w}_{k_w}, \boldsymbol{x}) - \mathbb{E}f(\boldsymbol{w}, \boldsymbol{x}) \right|$$

$$\leq \left| \frac{1}{n} \sum_{i=1}^n (f(\boldsymbol{w}, \boldsymbol{x}_{(i)}) - f(\boldsymbol{w}_{k_w}, \boldsymbol{x}_{(i)})) \right| + \left| \frac{1}{n} \sum_{i=1}^n f(\boldsymbol{w}_{k_w}, \boldsymbol{x}_{(i)}) - \mathbb{E}f(\boldsymbol{w}_{k_w}, \boldsymbol{x}) \right| + \left| \mathbb{E}f(\boldsymbol{w}_{k_w}, \boldsymbol{x}) - \mathbb{E}f(\boldsymbol{w}, \boldsymbol{x}) \right|.$$

Then, we define four events $\boldsymbol{E}_0$, $\boldsymbol{E}_1$, $\boldsymbol{E}_2$ and $\boldsymbol{E}_3$ as

$$\boldsymbol{E}_0 = \left\{ \sup_{\boldsymbol{w} \in \Omega} \left| \hat{\boldsymbol{J}}_n(\boldsymbol{w}) - \boldsymbol{J}(\boldsymbol{w}) \right| \geq t \right\},$$

$$\boldsymbol{E}_1 = \left\{ \sup_{\boldsymbol{w} \in \Omega} \left| \frac{1}{n} \sum_{i=1}^n \left( f(\boldsymbol{w}, \boldsymbol{x}_{(i)}) - f(\boldsymbol{w}_{k_w}, \boldsymbol{x}_{(i)}) \right) \right| \geq \frac{t}{3} \right\},$$

$$\boldsymbol{E}_2 = \left\{ \sup_{j_w \in [n_\epsilon{}^j], j=[l]} \left| \frac{1}{n} \sum_{i=1}^n f(\boldsymbol{w}_{k_w}, \boldsymbol{x}_{(i)}) - \mathbb{E}(f(\boldsymbol{w}_{k_w}, \boldsymbol{x})) \right| \geq \frac{t}{3} \right\},$$

$$\boldsymbol{E}_3 = \left\{ \sup_{\boldsymbol{w} \in \Omega} \left| \mathbb{E}(f(\boldsymbol{w}_{k_w}, \boldsymbol{x})) - \mathbb{E}(f(\boldsymbol{w}, \boldsymbol{x})) \right| \geq \frac{t}{3} \right\}.$$

Accordingly, we have
$$\mathbb{P}(\boldsymbol{E}_0) \leq \mathbb{P}(\boldsymbol{E}_1) + \mathbb{P}(\boldsymbol{E}_2) + \mathbb{P}(\boldsymbol{E}_3).$$
So we can respectively bound $\mathbb{P}(\boldsymbol{E}_1)$, $\mathbb{P}(\boldsymbol{E}_2)$ and $\mathbb{P}(\boldsymbol{E}_3)$ to bound $\mathbb{P}(\boldsymbol{E}_0)$.

**Step 1. Bound $\mathbb{P}(\boldsymbol{E}_1)$**: We first bound $\mathbb{P}(\boldsymbol{E}_1)$ as follows:

$$\mathbb{P}(\boldsymbol{E}_1) = \mathbb{P}\left( \sup_{\boldsymbol{w} \in \Omega} \left| \frac{1}{n} \sum_{i=1}^n \left( f(\boldsymbol{w}, \boldsymbol{x}_{(i)}) - f(\boldsymbol{w}_{k_w}, \boldsymbol{x}_{(i)}) \right) \right| \geq \frac{t}{3} \right)$$

$$\overset{\text{①}}{\leq} \frac{3}{t} \mathbb{E}\left( \sup_{\boldsymbol{w} \in \Omega} \left| \frac{1}{n} \sum_{i=1}^n \left( f(\boldsymbol{w}, \boldsymbol{x}_{(i)}) - f(\boldsymbol{w}_{k_w}, \boldsymbol{x}_{(i)}) \right) \right| \right)$$

$$\leq \frac{3}{t} \mathbb{E}\left( \sup_{\boldsymbol{w} \in \Omega} \frac{\left| \frac{1}{n} \sum_{i=1}^n \left( f(\boldsymbol{w}, \boldsymbol{x}_{(i)}) - f(\boldsymbol{w}_{k_w}, \boldsymbol{x}_{(i)}) \right) \right|}{\|\boldsymbol{w} - \boldsymbol{w}_{k_w}\|_2} \sup_{\boldsymbol{w} \in \Omega} \|\boldsymbol{w} - \boldsymbol{w}_{k_w}\|_2 \right)$$

$$\leq \frac{3\epsilon}{t} \mathbb{E}\left( \sup_{\boldsymbol{w} \in \Omega} \left\| \nabla \hat{\boldsymbol{J}}_n(\boldsymbol{w}, \boldsymbol{x}) \right\|_2 \right),$$

where ① holds since by Markov inequality, for an arbitrary nonnegative random variable $x$, then we have
$$\mathbb{P}(x \geq t) \leq \frac{\mathbb{E}(x)}{t}.$$

Now we only need to bound $\mathbb{E}\left(\sup_{\boldsymbol{w} \in \Omega} \left\|\nabla \hat{\boldsymbol{J}}_n(\boldsymbol{w}, \boldsymbol{x})\right\|_2\right)$. Then by Lemma 16, we can bound it as follows:

$$\mathbb{E}\left(\sup_{\boldsymbol{w} \in \Omega} \left\|\nabla \hat{\boldsymbol{J}}_n(\boldsymbol{w}, \boldsymbol{x})\right\|_2\right) \leq \mathbb{E}\left(\sup_{\boldsymbol{w} \in \Omega} \left\|\frac{1}{n} \sum_{i=1}^n \nabla f(\boldsymbol{w}, \boldsymbol{x}_{(i)})\right\|_2\right) \leq \alpha,$$

where $\alpha = \sqrt{\frac{1}{16} c_y c_d \left(1 + c_r(l-1)\right)}$ in which $c_y$, $c_d$ and $c_r$ are defined in Lemma 16.

Therefore, we have

$$\mathbb{P}\left(\boldsymbol{E}_1\right) \leq \frac{3\alpha\epsilon}{t}.$$

We further let

$$t \geq \frac{6\alpha\epsilon}{\varepsilon}.$$

Then we can bound $\mathbb{P}(\boldsymbol{E}_1)$:

$$\mathbb{P}(\boldsymbol{E}_1) \leq \frac{\varepsilon}{2}.$$

**Step 2. Bound $\mathbb{P}\left(\boldsymbol{E}_2\right)$**: Recall that we use $j_w$ to denote the index of $\boldsymbol{w}_{k_j}^j$ in $\epsilon$-net $\boldsymbol{w}_\epsilon^j$ and we have $j_w \in [n_\epsilon{}^j]$, $(n_\epsilon{}^j \leq \exp\left(\boldsymbol{s}_j \log\left(\frac{3rd}{\epsilon}\right)\right))$. We can bound $\mathbb{P}\left(\boldsymbol{E}_2\right)$ as follows:

$$\mathbb{P}\left(\boldsymbol{E}_2\right) = \mathbb{P}\left(\sup_{j_w \in [n_\epsilon{}^j], j=[l]} \left|\frac{1}{n} \sum_{i=1}^n f(\boldsymbol{w}_{k_w}, \boldsymbol{x}_{(i)}) - \mathbb{E}(f(\boldsymbol{w}_{k_w}, \boldsymbol{x}))\right| \geq \frac{t}{3}\right)$$

$$\leq \exp\left(\sum_{j=1}^l \boldsymbol{s}_j \log\left(\frac{3rd}{\epsilon}\right)\right) \sup_{j_w \in [n_\epsilon{}^j], j=[l]} \mathbb{P}\left(\left|\frac{1}{n} \sum_{i=1}^n f(\boldsymbol{w}_j, \boldsymbol{x}_{(i)}) - \mathbb{E}(f(\boldsymbol{w}_j, \boldsymbol{x}))\right| \geq \frac{t}{3}\right).$$

Since when the activation functions are sigmoid functions, the loss $f(\boldsymbol{w}, \boldsymbol{x})$ is $\alpha$-Lipschitz. Besides, we assume $\boldsymbol{x}$ to be a vector of *i.i.d.* Gaussian variables from $\mathcal{N}(0, \tau^2)$. Then by Lemma 25, we know that the variable $f(\boldsymbol{x}) - \mathbb{E}f(\boldsymbol{x})$ is $8\alpha^2\tau^2$-sub-Gaussian. Thus, we have

$$\mathbb{P}\left(|f(\boldsymbol{x}) - \mathbb{E}f(\boldsymbol{x})| > t\right) \leq 2\exp\left(-\frac{t^2}{2\alpha^2\tau^2}\right), \quad (\forall t \geq 0),$$

where $\alpha = \sqrt{\frac{1}{16} c_y c_d \left(1 + c_r(l-1)\right)}$ in which $c_y$, $c_d$ and $c_r$ are defined in Lemma 16. Thus, $\frac{1}{n} \sum_{i=1}^n f(\boldsymbol{w}_j, \boldsymbol{x}_{(i)}) - \mathbb{E}(f(\boldsymbol{w}_j, \boldsymbol{x}))$ is $8\alpha^2\tau^2/n$-sub-Gaussian random variable. Thus, we can obtain

$$\mathbb{P}\left(\left|\frac{1}{n} \sum_{i=1}^n f(\boldsymbol{w}_j, \boldsymbol{x}_{(i)}) - \mathbb{E}(f(\boldsymbol{w}_j, \boldsymbol{x}))\right| \geq \frac{t}{3}\right) \leq 2\exp\left(-\frac{nt^2}{18\alpha^2\tau^2}\right).$$

Notice $\sum_{j=1}^l \boldsymbol{s}_j = s$. In this case, the probability of $\boldsymbol{E}_2$ is upper bounded as

$$\mathbb{P}\left(\boldsymbol{E}_2\right) \leq 2\exp\left(-\frac{nt^2}{18\alpha^2\tau^2} + s \log\left(\frac{3dr}{\epsilon}\right)\right).$$

Thus, if we set

$$t \geq \alpha\tau \sqrt{\frac{18\left(s \log(3dr/\epsilon) + \log(4/\varepsilon)\right)}{n}},$$

then we have

$$\mathbb{P}\left(\boldsymbol{E}_2\right) \leq \frac{\varepsilon}{2}.$$

**Step 3. Bound $\mathbb{P}\left(\boldsymbol{E}_3\right)$**: We first bound $\mathbb{P}\left(\boldsymbol{E}_3\right)$ as follows:

$$\mathbb{P}\left(\boldsymbol{E}_3\right) = \mathbb{P}\left(\sup_{\boldsymbol{w} \in \Omega} |\mathbb{E}(f(\boldsymbol{w}_{k_w}, \boldsymbol{x})) - \mathbb{E}(f(\boldsymbol{w}, \boldsymbol{x}))| \geq \frac{t}{3}\right)$$

$$= \mathbb{P}\left(\sup_{\boldsymbol{w} \in \Omega} \frac{|\mathbb{E}\left(f(\boldsymbol{w}_{k_w}, \boldsymbol{x}) - f(\boldsymbol{w}, \boldsymbol{x})\right)|}{\|\boldsymbol{w} - \boldsymbol{w}_{k_w}\|_2} \sup_{\boldsymbol{w} \in \Omega} \|\boldsymbol{w} - \boldsymbol{w}_{k_w}\|_2 \geq \frac{t}{3}\right)$$

$$\leq \mathbb{P}\left(\epsilon \mathbb{E} \sup_{\boldsymbol{w} \in \Omega} \|\nabla \boldsymbol{J}_{\boldsymbol{w}}(\boldsymbol{w}, \boldsymbol{x})\|_2 \geq \frac{t}{3}\right)$$

$$\overset{①}{\leq} \mathbb{P}\left(\alpha\epsilon \geq \frac{t}{3}\right),$$

where ① holds since by Lemma 16, for arbitrary $\boldsymbol{x}$ and $\boldsymbol{w} \in \Omega$, we have $\|\nabla_{\boldsymbol{w}} f(\boldsymbol{w}, \boldsymbol{x})\|_2 \leq \alpha$. We set $\epsilon$ enough small such that $\alpha\epsilon < t/3$ always holds. Then it yields $\mathbb{P}(\boldsymbol{E}_3) = 0$.

**Step 4. Final result**: Notice, we have $\frac{6\alpha\epsilon}{\varepsilon} \geq 3\alpha\epsilon$. To ensure $\mathbb{P}(\boldsymbol{E}_0) \leq \varepsilon$, we just set $\epsilon = 3r/n$ and

$$t \geq \max\left(\frac{6\alpha\epsilon}{\varepsilon}, \alpha\tau\sqrt{\frac{18\left(s\log(3dr/\epsilon) + \log(4/\varepsilon)\right)}{n}}\right) = \max\left(\frac{18\alpha r}{n\varepsilon}, \alpha\tau\sqrt{\frac{18\left(s\log(nd) + \log(4/\varepsilon)\right)}{n}}\right).$$

Therefore, if $n \geq 18l^2r^2/(s\log(d)\tau^2\varepsilon^2\log(1/\varepsilon))$, then

$$\sup_{\boldsymbol{w}\in\Omega}\left|\hat{\boldsymbol{J}}_n(\boldsymbol{w}) - \boldsymbol{J}(\boldsymbol{w})\right| \leq \tau\sqrt{\frac{9}{8}c_y c_d\left(1 + c_r(l-1)\right)}\sqrt{\frac{s\log(nd/l) + \log(4/\varepsilon)}{n}}$$

holds with probability at least $1 - \varepsilon$, where $c_y$, $c_d$, and $c_r$ are defined as

$$\|\boldsymbol{v}^{(l)} - \boldsymbol{y}\|_2^2 \leq c_y < +\infty, \quad c_d = \max(\boldsymbol{d}_0, \boldsymbol{d}_1, \cdots, \boldsymbol{d}_l) \quad \text{and} \quad c_r = \max\left(\frac{r^2}{16}, \left(\frac{r^2}{16}\right)^{l-1}\right).$$

The proof is completed. $\qquad\square$

### D.3.4 PROOF OF COROLLARY 2

*Proof.* By Lemma 5, we know $\epsilon_s = \epsilon_g$. Thus, the remaining work is to bound $\epsilon_s$. Actually, we can have

$$\left|\mathbb{E}_{\boldsymbol{S}\sim\boldsymbol{D},\boldsymbol{A},(\boldsymbol{x}'_{(j)},\boldsymbol{y}'_{(j)})\sim\boldsymbol{D}}\frac{1}{n}\sum_{j=1}^{n}\left(f_j(\boldsymbol{w}_*^j;\boldsymbol{x}'_{(j)},\boldsymbol{y}'_{(j)}) - f_j(\boldsymbol{w}^n;\boldsymbol{x}'_{(j)},\boldsymbol{y}'_{(j)})\right)\right| \leq \mathbb{E}_{\boldsymbol{S}\sim\boldsymbol{D}}\left(\sup_{\boldsymbol{w}\in\Omega}\left|\hat{\boldsymbol{J}}_n(\boldsymbol{w}) - \boldsymbol{J}(\boldsymbol{w})\right|\right)$$

$$\leq \sup_{\boldsymbol{w}\in\Omega}\left|\hat{\boldsymbol{J}}_n(\boldsymbol{w}) - \boldsymbol{J}(\boldsymbol{w})\right|$$

$$\leq \epsilon_n.$$

Thus, we have $\epsilon_g = \epsilon_s \leq \epsilon_n$. The proof is completed. $\qquad\square$

## D.4 PROOF OF OTHER LEMMAS

### D.4.1 PROOF OF LEMMA 22

*Proof.* Since $\mathsf{G}(\boldsymbol{u}^{(i)})$ is a diagonal matrix and its diagonal values are upper bounded by $\sigma(\boldsymbol{u}_h^{(i)})(1 - \sigma(\boldsymbol{u}_h^{(i)})) \leq 1/4$ where $\boldsymbol{u}_h^{(i)}$ denotes the $h$-th entry of $\boldsymbol{u}^{(i)}$, we can conclude

$$\|\mathsf{G}(\boldsymbol{u}^{(i)})\boldsymbol{M}\|_F^2 \leq \frac{1}{16}\|\boldsymbol{M}\|_F^2 \quad \text{and} \quad \|\boldsymbol{N}\mathsf{G}(\boldsymbol{u}^{(i)})\|_F^2 \leq \frac{1}{16}\|\boldsymbol{N}\|_F^2.$$

Note that $\boldsymbol{P}_k$ is a matrix of size $\boldsymbol{d}_k^2 \times \boldsymbol{d}_k$ whose $((s-1)\boldsymbol{d}_k + s, s)$ $(s = 1, \cdots, \boldsymbol{d}_k)$ entry equal to $\sigma(\boldsymbol{u}_s^{(k)})(1 - \sigma(\boldsymbol{u}_s^{(k)}))(1 - 2\sigma(\boldsymbol{u}_s^{(k)}))$ and rest entries are all 0. This gives

$$\sigma(\boldsymbol{u}_s^{(k)})(1 - \sigma(\boldsymbol{u}_s^{(k)}))(1 - 2\sigma(\boldsymbol{u}_s^{(k)})) = \frac{1}{3}(3\sigma(\boldsymbol{u}_s^{(k)}))(1 - \sigma(\boldsymbol{u}_s^{(k)}))(1 - 2\sigma(\boldsymbol{u}_s^{(k)}))$$

$$\leq \frac{1}{3}\left(\frac{3\sigma(\boldsymbol{u}_s^{(k)}) + 1 - \sigma(\boldsymbol{u}_s^{(k)}) + 1 - 2\sigma(\boldsymbol{u}_s^{(k)})}{3}\right)^3$$

$$\leq \frac{2^3}{3^4}.$$

This means the maximal value in $\boldsymbol{P}_k$ is at most $\frac{2^3}{3^4}$. Consider the structure in $\boldsymbol{P}_k$, we can obtain

$$\|\boldsymbol{P}_k\boldsymbol{M}\|_F^2 \leq \frac{2^6}{3^8}\|\boldsymbol{M}\|_F^2 \quad \text{and} \quad \|\boldsymbol{N}\boldsymbol{P}_k\|_F^2 \leq \frac{2^6}{3^8}\|\boldsymbol{N}\|_F^2.$$

As for $\boldsymbol{B}_{s:t}$, we have

$$\|\boldsymbol{B}_{s:t}\|_F^2 \leq \|\boldsymbol{A}_s\|_F^2 \|\boldsymbol{A}_{s+1}\|_F^2 \cdots \|\boldsymbol{A}_t\|_F^2$$
$$= \left\|(\boldsymbol{W}^s)^T \mathsf{G}(\boldsymbol{u}^{(s)})\right\|_F^2 \left\|(\boldsymbol{W}^{(s+1)})^T \mathsf{G}(\boldsymbol{u}^{(s+1)})\right\|_F^2 \cdots \left\|(\boldsymbol{W}^{(t)})^T \mathsf{G}(\boldsymbol{u}^{(t)})\right\|_F^2$$
$$\leq \frac{1}{16^{t-s+1}} \left\|\boldsymbol{W}^{(s)}\right\|_F^2 \left\|\boldsymbol{W}^{(s+1)}\right\|_F^2 \cdots \left\|\boldsymbol{W}^{(t)}\right\|_F^2$$
$$= \frac{1}{16^{t-s+1}} \boldsymbol{D}_{s:t}.$$

Since the $\ell_2$-norm of each $\boldsymbol{w}_{(j)}$ is bounded, *i.e.* $\|\boldsymbol{w}_{(j)}\|_2 \leq r$, we can obtain

$$\frac{1}{16^{t-s+1}} \boldsymbol{D}_{s:t} \leq \frac{1}{16^{t-s+1}} r^{2(t-s+1)} = \left(\frac{r}{4}\right)^{2(t-s+1)} \triangleq c_{st}.$$

Now we prove the final result. According to the property of Kronecker product that for any matrices $\boldsymbol{A}$, $\boldsymbol{B}$ and $\boldsymbol{X}$ of proper sizes, $\mathsf{vec}\,(\boldsymbol{A}\boldsymbol{X}\boldsymbol{B}) = (\boldsymbol{B}^T \otimes \boldsymbol{A})\mathsf{vec}\,(\boldsymbol{X})$, we have

$$\mathsf{vec}\,(\boldsymbol{M}\boldsymbol{N}^T) = (\boldsymbol{N} \otimes \boldsymbol{I})\mathsf{vec}\,(\boldsymbol{M}) = (\boldsymbol{N} \otimes \boldsymbol{I})\boldsymbol{m}.$$

This further yields

$$\|(\boldsymbol{N} \otimes \boldsymbol{I})\boldsymbol{m}\|_F^2 = \|\mathsf{vec}\,(\boldsymbol{M}\boldsymbol{N}^T)\|_F^2 = \|\boldsymbol{M}\boldsymbol{N}^T\|_F^2 \leq \|\boldsymbol{M}\|_F^2 \|\boldsymbol{N}\|_F^2.$$

By similar way, we can obtain

$$\|(\boldsymbol{I} \otimes \boldsymbol{N})\boldsymbol{m}\|_F^2 \leq \|\boldsymbol{M}\|_F^2 \|\boldsymbol{N}\|_F^2.$$

The proof is completed. $\qquad\square$

### D.4.2 PROOF OF LEMMA 23

*Proof.* By utilizing the chain rule in Eqn. (24) in Sec. D.2.1, we can easily compute $\frac{\partial f(\boldsymbol{w},\boldsymbol{x})}{\partial \boldsymbol{u}^{(i)}}$ and $\frac{\partial f(\boldsymbol{w},\boldsymbol{x})}{\partial \boldsymbol{v}^{(i)}}$ as follows:

$$\frac{\partial f(\boldsymbol{w},\boldsymbol{x})}{\partial \boldsymbol{u}^{(i)}} = \mathsf{G}(\boldsymbol{u}^{(i)})\boldsymbol{A}_{i+1}\cdots\boldsymbol{A}_l(\boldsymbol{v}^{(l)} - \boldsymbol{y}) = \mathsf{G}(\boldsymbol{u}^{(i)})\boldsymbol{B}_{i+1:l}(\boldsymbol{v}^{(l)} - \boldsymbol{y})$$

and

$$\frac{\partial f(\boldsymbol{w},\boldsymbol{x})}{\partial \boldsymbol{v}^{(i)}} = \boldsymbol{A}_{i+1}\cdots\boldsymbol{A}_l(\boldsymbol{v}^{(l)} - \boldsymbol{y}) = \boldsymbol{B}_{i+1:l}(\boldsymbol{v}^{(l)} - \boldsymbol{y}).$$

Therefore, we can further obtain

$$\frac{\partial f(\boldsymbol{w},\boldsymbol{x})}{\partial \boldsymbol{w}_{(j)}}$$
$$= \mathsf{vec}\left(\left(\mathsf{G}(\boldsymbol{u}^{(j)})\boldsymbol{A}_{j+1}\boldsymbol{A}_{j+2}\cdots\boldsymbol{A}_l(\boldsymbol{v}^{(l)} - \boldsymbol{y})\right)(\boldsymbol{v}^{(j-1)})^T\right)$$
$$= \mathsf{vec}\left(\left(\mathsf{G}(\boldsymbol{u}^{(j)})\boldsymbol{A}_{j+1}\boldsymbol{A}_{j+2}\cdots\boldsymbol{A}_{i-1}(\boldsymbol{W}^{(i)})^T\right)\left(\mathsf{G}(\boldsymbol{u}^{(i)})\boldsymbol{A}^{i+1}\cdots\boldsymbol{A}_l(\boldsymbol{v}^{(l)} - \boldsymbol{y})\right)(\boldsymbol{v}^{(j-1)})^T\right)$$
$$= \left(\boldsymbol{v}^{(j-1)} \otimes \left(\mathsf{G}(\boldsymbol{u}^{(j)})\boldsymbol{A}_{j+1}\boldsymbol{A}_{j+2}\cdots\boldsymbol{A}_{i-1}(\boldsymbol{W}^{(i)})^T\right)\right)\mathsf{vec}\left(\mathsf{G}(\boldsymbol{u}^{(i)})\boldsymbol{A}_{i+1}\cdots\boldsymbol{A}_l(\boldsymbol{v}^{(l)} - \boldsymbol{y})\right)$$
$$= \left(\boldsymbol{v}^{(j-1)} \otimes \left(\mathsf{G}(\boldsymbol{u}^{(j)})\boldsymbol{A}_{j+1}\boldsymbol{A}_{j+2}\cdots\boldsymbol{A}_{i-1}(\boldsymbol{W}^{(i)})^T\right)\right)\left(\frac{\partial f(\boldsymbol{w},\boldsymbol{x})}{\partial \boldsymbol{u}^{(i)}}\right).$$

Note that we have $\frac{\partial f(\boldsymbol{w},\boldsymbol{x})}{\partial \boldsymbol{w}_{(j)}} = \frac{\partial \boldsymbol{u}^{(i)}}{\partial \boldsymbol{w}_{(j)}}\left(\frac{\partial f(\boldsymbol{w},\boldsymbol{x})}{\partial \boldsymbol{u}^{(i)}}\right)$. This gives

$$\frac{\partial \boldsymbol{u}^{(i)}}{\partial \boldsymbol{w}_{(j)}} = (\boldsymbol{v}^{(j-1)})^T \otimes \left(\mathsf{G}(\boldsymbol{u}^{(j)})\boldsymbol{B}_{j+1:i-1}(\boldsymbol{W}^{(i)})^T\right)^T \in \mathbb{R}^{\boldsymbol{d}_i \times \boldsymbol{d}_j \boldsymbol{d}_{j-1}} \; (i > j).$$

When $i = j$, we have

$$\frac{\partial \boldsymbol{u}^{(i)}}{\partial \boldsymbol{w}_{(i)}} = (\boldsymbol{v}^{(i-1)})^T \otimes \boldsymbol{I}_{\boldsymbol{d}_i} \in \mathbb{R}^{\boldsymbol{d}_i \times \boldsymbol{d}_i \boldsymbol{d}_{i-1}}.$$

Similarly, we can obtain

$$\frac{\partial \boldsymbol{v}^{(i)}}{\partial \boldsymbol{w}_{(j)}} = (\boldsymbol{v}^{(j-1)})^T \otimes \left(\mathsf{G}(\boldsymbol{u}^{(j)})\boldsymbol{A}_{j+1}\boldsymbol{A}_{j+2}\cdots\boldsymbol{A}_i\right)^T = (\boldsymbol{v}^{(j-1)})^T \otimes \left(\mathsf{G}(\boldsymbol{u}^{(j)})\boldsymbol{B}_{j+1:i}\right)^T \in \mathbb{R}^{\boldsymbol{d}_i \times \boldsymbol{d}_j \boldsymbol{d}_{j-1}} \; (i \geq j).$$

The proof is completed. $\qquad\square$

### D.4.3 PROOF OF LEMMA 24

*Proof.* By Lemma 23, we have

$$\frac{\partial f(\boldsymbol{w}, \boldsymbol{x})}{\partial \boldsymbol{u}^{(i)}} = \text{G}(\boldsymbol{u}^{(i)}) \boldsymbol{B}_{i+1:l}(\boldsymbol{v}^{(l)} - \boldsymbol{y}) \quad \text{and} \quad \frac{\partial f(\boldsymbol{w}, \boldsymbol{x})}{\partial \boldsymbol{v}^{(i)}} = \boldsymbol{B}_{i+1:l}(\boldsymbol{v}^{(l)} - \boldsymbol{y}).$$

Therefore, we can further obtain

$$\begin{aligned}
\frac{\partial f(\boldsymbol{w}, \boldsymbol{x})}{\partial \boldsymbol{u}^{(1)}} &= \text{G}(\boldsymbol{u}^{(1)}) \boldsymbol{A}_2 \cdots \boldsymbol{A}_l (\boldsymbol{v}^{(l)} - \boldsymbol{y}) \\
&= \text{G}(\boldsymbol{u}^{(1)}) \boldsymbol{A}_2 \cdots \boldsymbol{A}_{j-1} (\boldsymbol{W}^j)^T \text{G}(\boldsymbol{u}^{(j)}) \boldsymbol{A}_{j+1} \cdots \boldsymbol{A}_l (\boldsymbol{v}^{(l)} - \boldsymbol{y}) \\
&= \left( \text{G}(\boldsymbol{u}^{(1)}) \boldsymbol{A}_2 \cdots \boldsymbol{A}_{j-1} (\boldsymbol{W}^j)^T \right) \left( \frac{\partial f(\boldsymbol{w}, \boldsymbol{x})}{\partial \boldsymbol{u}^{(j)}} \right).
\end{aligned}$$

Note that we have $\frac{\partial f(\boldsymbol{w}, \boldsymbol{x})}{\partial \boldsymbol{u}^{(1)}} = \left( \frac{\partial \boldsymbol{u}^{(j)}}{\partial \boldsymbol{u}^{(1)}} \right)^T \left( \frac{\partial f(\boldsymbol{w}, \boldsymbol{x})}{\partial \boldsymbol{u}^{(j)}} \right)$. This gives

$$\frac{\partial \boldsymbol{u}^{(j)}}{\partial \boldsymbol{u}^{(1)}} = \left( \text{G}(\boldsymbol{u}^{(1)}) \boldsymbol{A}_2 \cdots \boldsymbol{A}_{j-1} (\boldsymbol{W}^j)^T \right)^T = \left( \text{G}(\boldsymbol{u}^{(1)}) \boldsymbol{B}_{2:j-1} (\boldsymbol{W}^j)^T \right)^T \in \mathbb{R}^{\boldsymbol{d}_j \times \boldsymbol{d}_1} \ (j > 1).$$

Similarly, we can obtain

$$\frac{\partial \boldsymbol{v}^{(j)}}{\partial \boldsymbol{u}^{(1)}} = \left( \text{G}(\boldsymbol{u}^{(1)}) \boldsymbol{A}_2 \cdots \boldsymbol{A}_j \right)^T = \left( \text{G}(\boldsymbol{u}^{(1)}) \boldsymbol{B}_{2:j} \right)^T \in \mathbb{R}^{\boldsymbol{d}_j \times \boldsymbol{d}_1} \ (j > 1).$$

The proof is completed. $\qquad\square$

