# OpenReview forum: "Empirical Risk Landscape Analysis for Understanding Deep Neural Networks"
_ICLR.cc/2018/Conference — Accept (Poster)_

### Official Review · AnonReviewer1 · 2017-11-27
**This theoretical paper lacks rigor and references.**

**Rating:** 3
**Confidence:** 3

**Review:**

This paper studies empirical risk in deep neural networks. Results are provided in Section 4 for linear networks and in Section 5 for nonlinear networks.
Results for deep linear neural networks are puzzling. Whatever the number of layers, a deep linear NN is simply a matrix multiplication and minimizing the MSE is simply a linear regression. So results in Section 4 are just results for linear regression and I do not understand why the number of layers come into play?
Also this is never explicitly mentioned in the paper, I guess the authors make an assumption that the samples (x_i,y_i) are drawn i.i.d. from a given distribution D. In such a case, I am sure results on the population risk minimization can be found for linear regression and should be compare to results in Section 4.

---

> ### Public Comment · ~Pan_Zhou1 · 2017-12-21
> **Reply to Reviewer 1**
>
> (1) Results in Sec. 4 are for linear regression (LR) and why the number of layers come into play?
>
> Reply: The issue w.r.t. the number of layers comes from the assumption that each weight matrix W_i is individually bounded, i.e., ||W_i||_F<=r, which gives the multiplied matrices are bounded by ||W_1*…*W_l||_F<=r^l. This bound is commonly used in our proof and unavoidable, since both our proof and the analysis of a LR model need such a bound of ||W_1*…*W_l||_F on the regression matrix.
>
> Besides, our analysis does not transform the multi-layer linear network into an LR model (explained in 2nd question) for investigating the effect of network parameters upon model performance. Thus we need to consider the number of layers. E.g., to bound the risk sup_w | Jn(w)-J(w) |, we need the Lipschitz constant of Jn(w) which is the upper bound on the norm of the gradient of Jn(w) and is at the order of O(l r^{l-1}).
>
> (2) Assumption that samples drawn i.i.d. from distribution D is not mentioned. In such a case, results on the population risk minimization can be found for linear regression (LR) and should be compared.
>
> Reply: As explained when introducing the problem Eqn. (1), the data are i.i.d. drawn from D.
>
> We first explain the difference with the results of linear regression. First, the multi-layer linear network is not equivalent to an LR due to an additional rank constraint on the regression parameter. Let W’=W_1*W_2*…*W_l. We have Y=W’*X s.t. rank(W’)<=min{rank(W_1),…,rank(W_l)}. Thus relevant results on LR are not applicable.
> Besides, our results give a similar convergence rate as the one for LR models (at least in n).
> (1) Our convergence rate for non-degenerate stationary points matches that of LR models, which are both O(1/sqrt(n)). E.g., Loh and Wainwright (JMLR 2015) proved that for sparse LR, the distance between the optimization solution and the optimum shrinks at O(sqrt(h s \log(d)/n )). Here s and d respectively denote the nonzero entry number and the dimension of parameter. h denotes the upper bound on the magnitude of the gradient of Jn(w) and is at the order of O(r^l) as explained above. Negahban et al. (NIPS 2011) provided similar results. These results accord with ours in n. The difference lies in d and s as we consider different solution space W=[W_1,…W_l] in our results from the solution space W=W_1*…*W_l in the linear regression (see reply to 1st question of Reviewer2). But the results from Loh et al. and Negahban et al. require restricted strong convexity (RSC) condition which is hard to verify and  the noise in the data to be i.i.d. standard Gaussian. In contrast, our analysis does not use the RSC condition and only assume that the input datum x is sub-Gaussian and has bounded magnitude.
> (2) As for the convergence rate of empirical risk, to our best knowledge there are no similar results. Applying the Rademacher complexity (RC) based approach  (commonly used to analyze the uniform convergence rate and usually gives tighter bounds than VC-dimension) gives the RC of linear regression at the order of O(h/sqrt{n}) (Ofer Dekel’s lecture notes, CSE522, 2011) and hence the convergence rate is |sup_f Jn(W’)- J(W’)| <= O(h/sqrt{n}). Here f denotes the linear hypothesis class, and h denotes the upper bound on the F-norm of W’ and is at the order of O(r^l) when each \|W_i\|_F is bound by r as in our analysis. This bound accords with our provided convergence rate O(1/sqrt{n}). The extra parameters s and d are involved in our results as we consider the whole parameter space rather than the function hypothesis f, which gives more transparent explanations on the roles of various network parameters.
> (3) As for the convergence rate of the gradient, we did not find any related works or existing directly applicable technique. We are the first to provide such results.
>
> The most important reason why we do not transform to LR is that analyzing LR model cannot provide useful results for further analyzing deep ReLU networks which are the focus of this work and most popular models in practice. So our proof considers the multi-layer architecture. Specifically,
> (1) Our analysis technique may benefit other property analysis of deep neural networks. Meanwhile, we cannot transform the deep nonlinear network (e.g. deep ReLU networks) into a linear regression model, since each layer involves the ReLU function. So resorting to the analysis of linear regression model cannot really benefit the analysis of deep nonlinear networks, which is not expected by us.
> (2) Avoiding transforming the deep linear networks to linear regression puts the analysis of deep linear networks in a similar framework as the deep ReLU networks. Thus, we derive consistent and neat results for both deep linear and ReLU networks which both include parameters about the multi-layer architecture (like depth l) and layer-wise weight matrix (like dimensions and magnitude bound).
>
> In the updated version, we have added the explanations at the end of Sec. 4.2.

---

### Official Review · AnonReviewer2 · 2017-11-27
**An interesting attempt on uniform convergence of general DNNs**

**Rating:** 7
**Confidence:** 3

**Review:**

This paper provides the analysis of empirical risk landscape for GENERAL deep neural networks (DNNs). Assumptions are comparable to existing results for OVERSIMPLIFED shallow neural networks. The main results analyzed: 1) Correspondence of non-degenerate stationary points between empirical risk and the population counterparts. 2) Uniform convergence of the empirical risk to population risk. 3) Generalization bound based on stability. The theory is first developed for linear DNNs and then generalized to nonlinear DNNs with sigmoid activations.

Here are two detailed comments:

1) For deep linear networks with squared loss, Kawaguchi 2016 has shown that the global optima are the only non-degerenate stationary points. Thus, the obtained non-degerenate stationary deep linear network should be equivalent to the linear regression model Y=XW. Should the risk bound only depends on the dimensions of the matrix W?

2) The comparison with Bartlett & Maass’s (BM) work is a bit unfair, because their result holds for polynomial activations while this paper handles linear activations. Thus, the authors need to refine BM's result for comparison.

---

> ### Public Comment · ~Pan_Zhou1 · 2017-12-21
> **Reply to Reviewer 2**
>
> (1)	For deep linear networks with squared loss, Kawaguchi 2016 has shown that the global optima are the only non-degerenate stationary points. Thus, the obtained non-degerenate stationary of a deep linear network should be equivalent to the linear regression model Y=XW. Should the risk bound only depends on the dimensions of the matrix W?
>
> Reply: Thanks for your comments. Some existing works (e.g., Loh and Wainwright, JMLR 2015; Negahban et al. NIPS 2011) proved that the risk bound of linear regression only depend on the dimension of the regression matrix W. However, as explained in the reply to the second question of Reviewer 1, the multi-layer linear network is not equivalent to a linear regression due to an additional rank constraint on the regression parameter and the multi-factor form in our analysis: let W’=W_1W_2…W_l and we have Y=W’X subject to rank(W’)<=min{rank(W_1),…,rank(W_l)}. Thus, relevant results on linear regression are not applicable here.
>
> Our risk bound of multi-layer linear networks involves the dimensions of each weight matrix. This is because in the proof we consider the essential multi-layer architecture of the deep linear network. In order to derive the uniform convergence sup_{W} |Jn(W)-J(W)| we need to consider the solution space of W=[W_1,W_2,...,W_l] instead of the solution space of W=W_1*W_2*…*W_l in the linear regression. We explained the reason why we do not transform the deep linear network into a linear regression in the reply to the second question of Reviewer 1. Please refer to it. The main reason is that we aim to build consistent analysis techniques for both deep linear and ReLU networks. This work is mainly devoted to analyzing the multi-layer architecture of deep linear networks which can pave a way for analyzing deep ReLU networks. Besides, the obtained results are more consistent with the results for deep nonlinear networks (see Sec. 5). About this we also explained in the updated version. Please refer to the last paragraph of the reply to the second question of Reviewer 1.
>
> (2)	The comparison with Bartlett & Maass’s (BM) work is a bit unfair, because their result holds for polynomial activations while this paper handles linear activations. Thus, the authors need to refine BM's result for comparison.
>
> Reply: Thanks for your suggestion. Since the linear activation function is a case of polynomial activations, we compared our results (Theorem 3) on the deep linear networks with Bartlett & Maass’s result. Actually, our risk bound (Theorem 6) on the nonlinear networks is also tighter than theirs. Our bound is sup_w |\hat{Jn(w)-J(w)}| <= O(\sqrt{[s log(dn/l) + log(1/epsilon)](l-1)/ n} ), while in Bartlett & Maass’s work, their result is |\hat{Jn(w)- inf_f J(w) }| <= O(\sqrt{[\gamma log(n)^2 + log(1/epsilon)] / n} ) where \gamma is at the order of O(ld log(d) + l^2d). In the updated version, we added comparison between our results in Theorem 6 on the deep nonlinear networks and Bartlett & Maass’s results at the end of the third paragraph in Sec. 5.2.
>
> “Similar to linear networks, our risk convergence rate is also tighter than the convergence rate on the networks with polynomial activation functions and one-dimensional output in (Bartlett & Maass, 2003) since ours is at the order of O(\sqrt{ (l-1)(s \log(dn/l)+\log(1/\varepsilon))/n), while the later one is $O(\sqrt{(\gamma \log^2(n)+\log(1/\varepsilon))/n}) where \gamma is at the order of O(ld \log(d)+l^2 d) (Bartlett & Maass, 2003).”

---

### Official Review · AnonReviewer4 · 2017-12-13
**Review of empirical landscape**

**Rating:** 7
**Confidence:** 3

**Review:**

Overall, this work seems like a reasonable attempt to answer the question of how the empirical loss landscape relates to the true population loss landscape.  The analysis answers:

1) When empirical gradients are close to true gradients
2) When empirical isolated saddle points are close to true isolated saddle points
3) When the empirical risk is close to the true risk.

The answers are all of the form that if the number of training examples exceeds a quantity that grows with the number of layers, width and the exponential of the norm of the weights with respect to depth, then empirical quantities will be close to true quantities.  I have not verified the proofs in this paper (given short notice to review) but the scaling laws in the upper bounds found seem reasonably correct.

Another reviewer's worry about why depth plays a role in the convergence of empirical to true values in deep linear networks is a reasonable worry, but I suspect that depth will necessarily play a role even in deep linear nets because the backpropagation of gradients in linear nets can still lead to exponential propagation of errors between empirical and true quantities due to finite training data.  Moreover the loss surface of deep linear networks depends on depth even though the expressive capacity does not.   An analysis of dynamics on this loss surface was presented in Saxe et. al. ICLR 2014 which could be cited to address that reviewer's concern.  However, the reviewer's suggestion that the results be compared to what is known more exactly for simple linear regression is a nice one.

Overall, I believe this paper is a nice contribution to the deep learning theory literature. However,  it would even better to help the reader with more intuitive statements about the implications of their results for practice, and the gap between their upper bounds and practice, especially given the intense interest in the generalization error problem.   Because their upper bounds look similar to those based on Rademacher complexity or VC dimension (although they claim theirs are a little tighter) - they should put numbers in to their upper bounds taken from trained neural networks, and see what the numerical evaluation of their upper bounds turn out to be in situations of practical interest where deep networks show good generalization performance despite having significantly less training data than number of parameters.  I suspect their upper bounds will be loose, but still  - it would be an excellent contribution to the literature to quantitatively compare theory and practice with bounds that are claimed to be slightly tigher than previous bounds.  Even if they are loose - identifying the degree of looseness could inspire interesting future work.

---

> ### Public Comment · ~Pan_Zhou1 · 2017-12-21
> **Reply to Reviewer 4**
>
> (1)	Another reviewer's suggestion that the results be compared to what is known more exactly for simple linear regression is a nice one.
>
> Reply: Thanks for the suggestion. Please refer to the reply to the second question of Reviewer 1 for our explanations.
>
> (2)	It would better to help the reader with more intuitive statements about the implications of their results for practice, and the gap between their upper bounds and practice, especially given the intense interest in the generalization error problem. Because their upper bounds look similar to those based on Rademacher complexity or VC dimension (although they claim theirs are a little tighter) - they should put numbers into their upper bounds taken from trained neural networks, and see what the numerical evaluation of their upper bounds turn out to be in situations of practical interest where deep networks show good generalization performance despite having significantly less training data than number of parameters.  I suspect their upper bounds will be loose, but still - it would be an excellent contribution to the literature to quantitatively compare theory and practice with bounds that are claimed to be slightly tighter than previous bounds.  Even if they are loose - identifying the degree of looseness could inspire interesting future work.
>
> Reply: Thanks for your suggestion. Similar to other works which focus on the analysis of generalization error (Bartlett & Maass 2003; Neyshabur et al., COLT 2015), stability and robustness (Xu & Mannor, Machine learning 2012), and Rademacher complexity of networks (Sun et al., AAAI 2016; Xie et al., Axriv 2015), the derived theoretical bounds are relatively looser than the computed empirical bound estimated on a specific dataset (Kawaguchi et al., Axriv 2017). This is because the property of the computed solution is unknown when employing general optimization algorithms. Therefore, in the analysis, we have to consider the worst case (worst solution or worst function in a hypothesis class) to bound the error. In particular, we need to bound the generalization error |E_{S~D, A} (Jn(w^n)-J(w^n))| where w^n is a computed optimization solution by an algorithm A. In this case, the distance between the expected empirical loss E_{S~D} Jn(w^n) and the expected population loss  $E_{S~D} J(w^n) is small. In contrast, due to the unknown property of the solution w^n which relies on the concrete optimization  algorithm, testing dataset, etc., we have to use the worst case (namely consider the whole solution space) to bound this error. As mentioned in the manuscript, the generalization error and stability error are directly derived by Theorems 3 and 6. So the comparison between the empirical bound and the derived bound is not meaningful. This is also why other works do not present the comparison either. In the future we will focus on a specific algorithm and utilize the property of its computed optimization solution to bound the generalization error which may give a more practical error bound.
>
> It is worth mentioning that the results in Theorems 3 and 6 are tighter than or comparable to the similar results in existing works (e.g. Bartlett & Maass  2003; Neyshabur et al., COLT 2015). Also the uniform convergence of the gradient and the non-degenerate stationary points of the empirical risk are new which benefit the theoretical understanding on how the neural network parameters determine the neural network landscape.

---

### Decision · Program_Chairs · 2018-01-29
**ICLR 2018 Conference Acceptance Decision**

**Decision:**

Accept (Poster)

**Comment:**

Based on the positive reviews, I recommend acceptance. The paper analyzes when empirical risk is close to the population version, when empirical saddle points are close to the population version and empirical gradients are close to the population version.